# A Refined Generalization Analysis for Extreme Multi-class Supervised Contrastive Representation Learning

**Nong Minh Hieu** [1]   **Antoine Ledent** [1]

## Abstract

Contrastive Representation Learning (CRL) has achieved strong empirical success in multiple machine learning disciplines, yet its theoretical sample complexity remains poorly understood. Existing analyses usually assume that input tuples are identically and independently distributed, an assumption violated in most practical settings where contrastive tuples are constructed from a finite pool of labeled data, inducing dependencies among tuples. While one recent work analyzed this learning setting using U-Statistics to estimate the population risk, the techniques used therein require the risk of each class to concentrate uniformly, making excess risk bounds scale in the order of $\rho_{\min}^{-1/2}$ where $\rho_{\min}$ denotes the probability of the rarest class. Such a dependency can be overly pessimistic in the extreme multiclass settings where there are many tail classes which contribute minimally to the overall population risk. Our contributions are two-fold. Firstly, we improve upon the previous work and prove a bound with a sample complexity of the same order as the number of classes $R$, regardless of the distribution over classes. Furthermore, we formulate a different estimator that captures the concentration of the risk *across classes*, enabling sharper bounds in extreme multi-class learning scenarios, especially where class distributions are long-tailed. Under mild assumptions on the class distributions, the resulting sample complexity is $\mathcal{O}(k)$ where $k$ is the number of samples per tuple.

[1]School of Computing and Information Systems, Singapore Management University. Correspondence to: Nong Minh Hieu <mh.nong.2024@phdcs.smu.edu.sg>.

## 1. Introduction

The performance of machine learning models depends critically on data representation. In multi-class classification, effective representations typically promote intra-class compactness and inter-class separability. Contrastive Representation Learning (CRL) has therefore become a popular approach for learning structured representations from raw data prior to downstream tasks. Despite its empirical success, theoretical understanding of CRL is still in the early stage. Existing analyses of excess risk bounds, such as Arora (2019) and Lei et al. (2023), commonly assume that unlabeled input $(k + 2)$-tuples are drawn i.i.d. from an unknown distribution. While analytically convenient, this assumption may not reflect practical CRL pipelines. In many real-world implementations, contrastive tuples are constructed from a finite pool of **labeled examples** (Sohn, 2016; Khosla et al., 2020; Zhang et al., 2021), leading to repeated use of data points across tuples, thereby violating the independence assumption.

To explicitly model tuple dependencies in supervised CRL, Hieu & Ledent (2025) introduced a U-statistic based estimator that decomposes the population risk into a sum of class-wise risks then estimates each component independently (cf. Eqn. (4)). While this formulation is natural and effective when the number of classes is moderate, it implicitly ties the overall sample complexity to the worst class-wise estimation error. In particular, their analysis requires all class-wise risk estimators to concentrate simultaneously, leading to a sample complexity that scales with $\rho_{\min}^{-1}$ where $\rho_{\min}$ denotes the rarest class probability. As a consequence, the bound becomes overly pessimistic in regimes with many classes, where $\rho_{\min}$ is necessarily small, and reduces to a dependence on the number of classes $R$ only in the restrictive case of perfectly balanced class distributions.

Our first contribution relaxes this worst-case requirement by allowing heterogeneous accuracy across class-wise estimators. Intuitively, classes with negligible contribution to the population risk need not be estimated with the same precision as dominant ones. This eliminates the parasitic dependence on $\rho_{\min}$ and yields bounds that depend on $R$ irrespective of the class distribution. Nevertheless, even this scale remains pessimistic in extreme multi-class set-

tings where $R$ is very large, and it stands in contrast with classical results on excess risk bounds for $k$-tuple learning where sample complexity depends only on the tuple size $k$ (Clémençon et al., 2008; Clémençon et al., 2016). This mismatch suggests that controlling class-wise estimators, even in aggregate, is fundamentally sub-optimal. We therefore introduce an **entirely different** U-Statistic formulation (cf. Eqn. (13)) that enforces joint concentration across classes. The resulting sample complexity scales as $k(1-\tau)^2$, where $\tau$ is the overall class collision probability (see, for example, Ash et al. (2022); Awasthi et al. (2022) or Arora (2019, Section 4.2)). Specifically, when negatives are sampled without excluding positive classes, $\tau$ represents the overall likelihood that there exists at least a positive-class sample among the selected negatives (cf. Eqn. (12)). In extreme multi-class regime, $\tau \to 0$, recovering a dependency on $k$ alone and aligning with established results on tuple-learning.

## 2. Related Work

**Empirical Minimization of U-Statistics:** U-Statistics were introduced by Hoeffding (1948) for parameters estimation using symmetric kernels. Later, the concentration properties of U-Statistics were analyzed rigorously by Gine & Zinn (1984) and Arcones & Gine (1993), laying the foundation for works in statistical learning theory where losses rely on two or more samples. Concentration inequalities for U-Statistics have been utilized in multiple machine learning disciplines, including ranking (Clémençon et al., 2008), metric learning (Cao et al., 2016; Jin et al., 2009), pairwise learning (Lei et al., 2018; 2020), clustering (Li & Liu, 2021) and self-supervised learning (HaoChen et al., 2021).

**Generalization under the i.i.d. Tuples Regime:** The i.i.d. tuple framework was introduced by Arora (2019), who formalized unsupervised risk for contrastive learning. Ignoring the dependency on the function class, their excess risk bounds scale as $\mathcal{O}(\sqrt{k/\mathrm{N}})$ where $k$ denotes the number of negative samples and N is the number of i.i.d. *tuples* available to the learner. Using tools such as worst-case Rademacher complexity and fat-shattering dimension, Lei et al. (2023) subsequently improved this dependence to $\widetilde{\mathcal{O}}(\sqrt{1/\mathrm{N}})$. We note that in these works, 'sample complexity' is expressed in terms of the number N of required $(k+2)$-*tuples* (with each tuple being one 'sample'), while in our work, it refers to the number $N$ of required *labeled data points* which are subsequently used to form collision-free tuples. Hence a sample complexity of $\widetilde{\mathcal{O}}(k)$ in the supervised CRL formulation is comparable to a sample complexity of $\widetilde{\mathcal{O}}(1)$ in i.i.d. tuples regime. Indeed, this is what can be expected from a Hoeffding-type argument in the pure tuple-wise learning (without constraints on collisions), since $\mathcal{O}(N/k)$ tuples can be formed with $N$ samples. Subsequently, the work of Arora (2019) was extended to

other CRL settings such as adversarial CRL (Zou & Liu, 2023; Ghanooni et al., 2024), PAC-Bayes analysis and deep CRL (Nozawa et al., 2020; Nozawa & Sato, 2021; Alves & Ledent, 2024; Hieu et al., 2024). Apart from generalization, prior works inspired by the i.i.d. regime also investigates other angles of contrastive learning theory such as the role of negative samples (Ash et al., 2022; Awasthi et al., 2022; Bao et al., 2022), mitigating spurious features in CRL (Ghanooni et al., 2025), and the effect of the unsupervised risk on downstream tasks (Arora, 2019; Chuang et al., 2020; Li & Liu, 2021; Bao et al., 2022; Huang et al., 2023; Cui et al., 2025b;a).

**Theoretical Analysis for SS-CRL:** A strong body of work inspired by Arora (2019) explores generalization and other theoretical aspects of self-supervised CRL. For example, Wang et al. (2022) proposed augmentation overlap theory, suggesting that aggressive data augmentation can implicitly improve intra-class compactness. Along similar lines, Cui et al. (2025a) study how augmentation affects downstream performance by relating the supervised risk to the empirical self-supervised risk (over a set of i.i.d. tuples) and the expected same-class or same-instance augmentation distance. Cui et al. (2025b) provide both theoretical and practical insights on a variant of supervised CRL with non-collided i.i.d. tuples with label noise. Huang et al. (2023) studied the effect of the self-supervised risk on classification by bounding the error rate of nearest-neighbor classifiers built on top of representation functions using misalignment probability. Nozawa & Sato (2021) provided a theoretical explanation for why using many negative samples improves downstream performance. In HaoChen et al. (2021), the authors established guarantees for the excess contrastive risk for loss functions in *two arguments* constructed from fixed pool of *unlabeled* samples via independent augmentation of both samples. Whilst this involves a second order U-statistic, this work makes no constraint on the valid pairs/tuples in terms of class-collision, making the concentration of the unsupervised risk straightforward: the core contribution instead lies in the relationship between the contrastive risk and the classification risk of a linear probe under structural assumptions on the informativeness of the data-augmentation generation. Their framework was later extended to other generalization analyses for variants of SS-CRL including domain adaptation (Shen et al., 2022), weakly-supervised learning (Cui et al., 2023), multi-modal learning (Zhang et al., 2023).

**Generalization in Supervised CRL:** Hieu & Ledent (2025) explored the CRL setting where input tuples are selected from a finite pool of labeled samples. In this regime, the authors proposed a U-Statistic formulation that estimates each class-wise contrastive risk separately. In their analysis, all class-wise estimators are required to concentrate with equal accuracy, leading to a dependency on $\rho_{\min}$, the smallest class probability. For their estimator, we achieve a

dependency of $R$ instead. We also achieve a faster rate of $k$ by constructing a radically different estimator and analysis.

## 3. Preliminaries

### 3.1. Definitions and Notations

Let $\mathcal{X}$ be the input space and $[R] = \{1, \dots, R\}$ be a labels set. We are given a (labeled) dataset $S = \{X_j\}_{j=1}^N \sim \bar{\mathcal{D}}^{\otimes N}$ where $\bar{\mathcal{D}} := \sum_{r=1}^R \rho_r \mathcal{D}_r$ is a mixture of distributions with $\rho_1, \dots, \rho_R$ representing class probabilities and $\mathcal{D}_r$ being the class-conditional distributions over $\mathcal{X}$. In supervised CRL, we aim to estimate a risk associated to some contrastive loss function using the available labeled data points. Specifically, for a certain representation function $f \in \mathcal{F}$ where $\mathcal{F}$ is a pre-fixed class of functions (e.g. neural networks, linear maps, etc.), we define $\ell_{\phi,f} : \mathcal{X}^{k+2} \to \mathbb{R}_+$ as a tuple-wise loss evaluated on the representations produced by $f$ as follows:

$$\ell_{\phi,f}\left(X, X^+, \{X_i^-\}_{i=1}^k\right) = \tag{1}$$
$$\phi\Big(\Big\{f(X)^\top\left[f(X^+) - f(X_i^-)\right]\Big\}_{i=1}^k\Big),$$

where $k \geqslant 1$ is the number of negative samples and $\phi : \mathbb{R}^k \to \mathbb{R}_+$ is a vector function often referred to as the contrastive loss. For example, if we use the logistic loss (Sohn, 2016) for training, then for all $k$-dimensional vectors $\mathbf{v} = (v_1, \dots, v_k)^\top$, we have $\phi(\mathbf{v}) := \ln\left(1 + \sum_{i=1}^k e^{-v_i}\right)$. In this paper, we are interested in bounding the **uniform excess risk** (over a class of representation functions) associated to the contrastive risk defined as follows.

**Definition 3.1** (Population Contrastive Risk). Let $f \in \mathcal{F}$ be a representation function and $\phi : \mathbb{R}^k \to \mathbb{R}_+$ be a contrastive loss (where $k \geqslant 1$ is the number of negative samples). Then, the contrastive risk of $f$, denoted $\mathrm{L}_\phi(f)$, is defined as:

$$\mathrm{L}_\phi(f) = \tag{2}$$
$$\mathbb{E}_{\substack{r \sim \rho}} \mathbb{E}_{\substack{X, X^+ \sim \mathcal{D}_r^{\otimes 2} \\ \{X_i^-\}_{i=1}^k \sim \bar{\mathcal{D}}_r^{\otimes k}}} \left[\ell_{\phi,f}\left(X, X^+, \{X_i^-\}_{i=1}^k\right)\right],$$

where $\rho$ is the discrete distribution over the classes that assigns probability $\rho_r$ to class $r$ for all $r \in [R]$ and $\bar{\mathcal{D}}_r$ is a distribution over $\mathcal{X}$ defined by the following mixture:

$$\bar{\mathcal{D}}_r := \frac{1}{1 - \rho_r} \sum_{q \neq r} \rho_q \mathcal{D}_q, \tag{3}$$

which is the re-normalized mixture of class-conditional distributions excluding class $r$, i.e., the distribution of negative samples when anchor-positive pairs come from class $r$.

**Remark 3.2.** The definition in Eqn. (2) differs from the definition of negative distribution used in Arora (2019). Our

formulation forbids negative samples $\{X_i^-\}_{i=1}^k$ from sharing classes with the anchor–positive pair $(X, X^+)$, whereas Arora (2019) allow such class collisions by drawing negatives i.i.d. from the full mixture $\bar{\mathcal{D}}$ rather than $\bar{\mathcal{D}}_r$. In the self-supervised case (Chen et al., 2020; He et al., 2020; Caron et al., 2020; Cui et al., 2025a), class-collision is often unavoidable since the labels are not available to prevent it. On the other hand, in supervised CRL, it is natural to avoid class-collision, as it has been shown to improve performance (Khosla et al., 2020; Chuang et al., 2020; Robinson et al., 2021; Chen et al., 2022; Huynh et al., 2022). Therefore, it is the primary setting we investigate. However, we also provide excess risk guarantee (Thm. E.7) for the collision-allowed risk in Arora (2019) although this result mostly serves as a intermediary step to provide excess risk guarantees for the desired risk $\mathrm{L}_\phi$.

**Notations**: Throughout this manuscript, we use $\mathbb{E}_{X \sim \mu}[\cdot]$ to denote the population expectation over some distribution $\mu$ or simply $\mathbb{E}[\cdot]$ when the distribution is obvious in given context. Additionally, we use $\widehat{\mathbb{E}}_V[\cdot]$ or $\widehat{\mathbb{E}}_{v \sim V}[\cdot]$ to denote the empirical average over elements $v$ of some set $V$. For example, $\widehat{\mathbb{E}}_V[g] = \widehat{\mathbb{E}}_{v \sim V}[g(v)] := \frac{1}{|V|} \sum_{v \in V} g(v)$. For each $r \in [R]$, we also denote $S_r \subseteq S$ as the set of labeled data points belonging to class $r$, $N_r := |S_r|$ and $\widehat{\rho}_r := \frac{N_r}{N}$. Finally, we assume that the contrastive loss is **uniformly bounded**, i.e., $|\ell_{\phi,f}| \leqslant \mathcal{B}$ for all $f \in \mathcal{F}$ for some $\mathcal{B} \in \mathbb{R}_+$. For the reader's convenience, we provide an exhaustive list of notations in the appendix (Table 4).

### 3.2. Contributions

In this work, we are interested in formulating estimators for the class-collision free population risk defined in Eqn. (2) and bounding the corresponding uniform excess risk. Our contributions can be summarized as follows:

(i) We provide a refined excess risk guarantee for the U-Statistic estimator proposed in Hieu & Ledent (2025) by completely removing the dependency on $\rho_{\min}^{-1}$, the rarest class probability, from the sample complexity. By imposing non-uniform accuracy for the concentration of each class-wise U-Statistic, we prove that the sample complexity of their estimator scales linearly with $R$, the number of classes, regardless of the class distribution's structure.

(ii) We propose a different U-Statistic formula inspired by de-biasing the population unsupervised risk used by previous works in the i.i.d. tuples regime (Arora, 2019; Lei et al., 2023) in which class-collision is arbitrarily allowed. Specifically, we decompose the desired collision-free risk into a combination of risks conditioned on at least one and zero collided negative samples (Eqn. (11)). Then, we approximate each

component separately to formulate the final estimator.

(iii) We prove that the proposed U-Statistic in this work yields a sample complexity that scales with $k(1-\tau)^2$ where $\tau$ is the population class-collision probability (Eqn. (12)). In typical extreme multi-class scenarios where all class probabilities are small, the collision probability $\tau$ is approximately zero and the sample complexity recovers the $\mathcal{O}(k)$ rate. Therefore, this type of bound is particularly useful for large multi-class representation learning problems where $R$ is significantly larger than $k$. Furthermore, for certain long-tailed imbalanced class distributions, we have $1-\tau \in \Omega(1)$ with high probability. Therefore, the dominant dependency on $k$ remains unchanged.

An outline of our main results (summarized in sample complexities) with comparison to prior works is provided in Table 1. Additionally, we also provide a more thorough summary of the literature in Appendix B. Furthermore, in Appendix C.2, we provide a brief discussion on the differences between our analysis and previous works, as well as the additional technical difficulties involved.

## 4. Main Results

### 4.1. Refined Analysis of Prior Work

For a given function $f \in \mathcal{F}$, Hieu & Ledent (2025) estimates the collision-free population contrastive risk in Eqn. (2) using a U-Statistic formulation $U_N^{\mathrm{hl}}$ defined as follows:

$$U_N^{\mathrm{hl}}(f) := \sum_{r=1}^{R} \widehat{\rho}_r U_{\Theta_r}(f). \tag{4}$$

Here, for all $r \in [R]$, $U_{\Theta_r}(f) := \widehat{\mathbb{E}}_{\Theta_r}[\ell_{\phi,f}]$ and $\Theta_r$ is a collection of $(k+2)$-tuples, defined as:

$$\Theta_r := \left\{ \left( X, X^+, \{X_i^-\}_{i=1}^k \right) : \right. \tag{5}$$
$$\left. X, X^+ \in S_r, \{X_i^-\}_{i=1}^k \subseteq S \setminus S_r \right\}.$$

Essentially, their proposed estimator is a weighted sum of class-wise U-Statistics where each class' estimator averages over the set of class-collision free tuples with anchor-positive pairs belonging to that class. In their analysis, the uniform excess risk bound is proved to scale roughly as:

$$\mathcal{O}\left[ \sum_{r=1}^{R} \frac{\rho_r \mathfrak{C}_N(\mathcal{H})}{\sqrt{\min\left(\frac{N_r}{2}, \frac{N-N_r}{k}\right)}} + \mathcal{B}\sqrt{\frac{\ln(R/\Delta)}{N\rho_{\min}}} \right], \tag{6}$$

with probability of at least $1-\Delta$ for any $\Delta \in (0,1)$ where $\mathcal{H}$ is the class of loss functions:

$$\mathcal{H} := \left\{ \left( X, X^+, \{X_i^-\}_{i=1}^k \right) \mapsto \right. \tag{7}$$
$$\left. \ell_{\phi,f}\left( X, X^+, \{X_i^-\}_{i=1}^k \right) : f \in \mathcal{F} \right\},$$

and $\mathfrak{C}_N(\mathcal{H})$ measures the worst-case Dudley entropy integral of $\mathcal{H}$ over the collections of $(k+2)$-tuples with size $N$ (cf. Eqn. (100)). For common function spaces (linear maps, neural networks, etc), $\mathfrak{C}_N(\mathcal{H})$ grows at most in polylogarithmic order of $N$. For instance, a parameter-counting argument (Long & Sedghi, 2020; Graf et al., 2022) yields $\mathfrak{C}_N(\mathcal{H}) \in \widetilde{\mathcal{O}}(\sqrt{WL})$ for an $L$-layered neural network with $W$ parameters, where the $\widetilde{\mathcal{O}}$ notation hides logarithmic factors in weight norms. In Hieu & Ledent (2025), the authors used a high-probability argument to show that the following lower bound holds simultaneously for all $r \in [R]$:

$$\min\left( \frac{N_r}{2}, \frac{N-N_r}{k} \right) \geqslant \min\left( \frac{N\rho_{\min}}{2}, \frac{N[1-\rho_{\max}]}{k} \right),$$

resulting in a sample complexity that scales in the order of $\mathcal{O}\left( \max\left( \frac{1}{\rho_{\min}}, \frac{k}{1-\rho_{\max}} \right) \right)$. While this estimation of the overall sample complexity is valid, it is vastly pessimistic. Specifically, it requires every class in the distribution to concentrate uniformly well around their respective class-wise risk. However, this is not necessary since tail classes often contribute minimally to the overall population risk. Therefore, the estimation of these small classes need not to be as accurate as major classes. Using this intuition, we improved upon previous proof techniques to derive a sample complexity that scales in the order of at most $\mathcal{O}\left( \max\left(R, k\right) \right)$.

**Theorem 4.1** (cf. Theorem H.5, Table 1). *Let $\mathcal{F}$ be a class of representation functions and let $U_N^{\mathrm{hl}}(f)$ be defined for each $f \in \mathcal{F}$ as in Eqn. (4). Suppose $\rho_r \leqslant \frac{1}{2}$ for all $r \in [R]$. Then, for $\Delta \in (0,1)$, as long as $N \geqslant \widetilde{\mathcal{O}}\left(k^2\right)$, we have:*

$$\sup_{f \in \mathcal{F}} \left| U_N^{\mathrm{hl}}(f) - \mathrm{L}_\phi(f) \right| \leqslant \tag{8}$$
$$\mathcal{O}\left( \mathfrak{C}_N(\mathcal{H})\sqrt{\frac{\max(k,R)}{N}} + \mathcal{B}\sqrt{\frac{R\ln(R/\Delta)}{N}} \right),$$

*with probability of at least $1-\Delta$. The complexity $\mathfrak{C}_N(\mathcal{H})$ of $\mathcal{H}$ is defined in Eqn. (100) or Definition 4.4 and the $\widetilde{\mathcal{O}}$ hides logarithmic orders of $R, \Delta$.*

In the above, we presented a simplified result where the bound scales in worst-case order of $\mathcal{O}(\sqrt{R/N})$. As previously claimed, the result is independent of the class distribution. In the precise version of the theorem (Theorem H.5), the order of the excess risk interpolates between square-root dependencies on both $k$ and $R$. Specifically, when the total mass of small probabilities (cf. Eqn. (20)) isn't negligible, the reliance on the number of classes dominates the dependency on tuple size. For instance, we demonstrate in Theorem H.3 that the order of the bound collapses to depending on $\sqrt{R}$ entirely when $\rho_r \lesssim 1/k, \forall r \in [R]$. In the next section, we formulate a fundamentally different estimator that encourages $\mathcal{O}(\sqrt{k})$ dependency when the class distribution consists of mostly small probabilities, making it suitable for extreme multi-class scenarios.

*Table 1.* Comparison of our work with the previous result in terms of sample complexity. For simplicity, we assume that the loss upper bound $\mathcal{B} = 1$. The notation ignores polylogarithmic factors of all relevant quantities ($\mathfrak{C}_N^2$, $R$, $k$, $1/(1-\tau)$ and the failure probability $\Delta$). For $\mathcal{L}$-Lipschitz parameterized function classes $\mathcal{H}$, the complexity term $\mathfrak{C}_N(\mathcal{H})$ grows at most as $\widetilde{\mathcal{O}}[(W)^{\frac{1}{2}}]$ where $W$ denotes the number of parameters and the $\widetilde{\mathcal{O}}$ notation hides polylogarithmic factors of $\mathcal{L}$ and $W$. The probability $\gamma_k$ is defined in Eqn. (20).

| Reference | Est. | Default | $\gamma_k \in \Omega(1)$ | Relevant quantity |
|---|---|---|---|---|
| Hieu & Ledent (2025) | $U_N^{\mathrm{hl}}$ | $[\mathfrak{C}_N^2(\mathcal{H}) + 1] \max\left[\rho_{\min}^{-1}, (1-\rho_{\max})^{-1}\right]$ | | $\rho_{\min} := \min_{r \in [R]} \rho_r$ $\rho_{\max} := \max_{r \in [R]} \rho_r$ |
| This Work | $U_N^{\mathrm{hl}}$ | $\mathfrak{C}_N^2(\mathcal{H})[\widehat{\theta}_{k+2}R + (1-\widehat{\theta}_{k+2})^2 k] +$ $[R + k^2]$ (Theorem 4.1/H.5) | $\mathfrak{C}_N^2(\mathcal{H})R +$ $[R + k^2]$ (Theorem H.3) | $\widehat{\theta}_{k+2} := \mathbb{P}_\rho\left[\rho_r \leqslant \frac{2}{k+2}\right]$ |
| This Work | $U_N$ $\bar{U}_N$ | $\mathfrak{C}_N^2(\mathcal{H})(1-\tau)^{-2}k +$ $[R(1-\tau)^{-4} + \max(Rk, k^2 + (1-\tau)^{-1})]$ (Theorem 4.5/G.5) | $\mathfrak{C}_N^2(\mathcal{H})k +$ $[R + Rk]$ (Theorem G.4) | $\tau := 1 - \sum_{r=1}^{R} \rho_r(1-\rho_r)^k$ |

## 4.2. Refined U-Statistics Formulation

**U-Statistics Formulation**

As seen in the previous section, the desired collision-free risk can be estimated by combining the class-wise U-Statistics, each weighted by the empirical class probability. However, we found that faster concentration can be achieved with an indirect approach involving the estimation of risks that **allow class-collision** to various extents. Specifically, we introduce the following population risks:

**Definition 4.2** (Contrastive Risks with Class-collision). Let $f \in \mathcal{F}$ be a representation function and $\phi : \mathbb{R}^k \to \mathbb{R}_+$ be a contrastive loss. We define $\mathrm{L}_\Omega(f)$ and $\mathrm{L}_\Lambda(f)$ as follows:

$$\mathrm{L}_\Omega(f) := \tag{9}$$
$$\mathop{\mathbb{E}}_{\substack{r \sim \rho \\ }} \mathop{\mathbb{E}}_{\substack{X,X^+ \sim \mathcal{D}_r^{\otimes 2} \\ \{X_i^-\}_{i=1}^k \sim \bar{\mathcal{D}}^{\otimes k}}} \left[\ell_{\phi,f}\left(X, X^+, \left\{X_i^-\right\}_{i=1}^k\right)\right],$$

$$\mathrm{L}_\Lambda(f) := \tag{10}$$
$$\mathop{\mathbb{E}}_{\substack{r \sim \rho \\ }} \mathop{\mathbb{E}}_{\substack{X,X^+,\bar{X} \sim \mathcal{D}_r^{\otimes 3} \\ \{X_i^-\}_{i=1}^{k-1} \sim \bar{\mathcal{D}}^{\otimes k-1}}} \left[\ell_{\phi,f}\left(X, X^+, \bar{X}, \left\{X_i^-\right\}_{i=1}^{k-1}\right)\right].$$

Here, $\mathrm{L}_\Omega(f)$ denotes the risk of $f$ when negative samples either collide with the anchor-positive pairs or not. On the other hand, $\mathrm{L}_\Lambda(f)$ denotes the risk where negative samples must collide **at least once** with the anchor-positive pairs.

Crucially, we can then express the collision-free contrastive risk $\mathrm{L}_\phi$ (Eqn. (2)) in terms of $\mathrm{L}_\Omega$ and $\mathrm{L}_\Lambda$ using a debiased formula as follows:

$$\mathrm{L}_\phi(f) := \frac{1}{1-\tau}\mathrm{L}_\Omega(f) - \frac{\tau}{1-\tau}\mathrm{L}_\Lambda(f). \tag{11}$$

$$\tau := 1 - \sum_{r=1}^{R} \rho_r\left[1-\rho_r\right]^k. \tag{12}$$

Here, $\tau$ denotes the overall class-collision probability. Using the above re-formulation, instead of directly estimating $\mathrm{L}_\phi$

with collision-free tuples, we can approach this indirectly by estimating $\mathrm{L}_\Omega$, $\mathrm{L}_\Lambda$ and $\tau$ separately. Specifically, we can formulate a U-Statistic to estimate $\mathrm{L}_\phi$ as follows:

$$U_N(f) = \frac{1}{1-\widehat{\tau}}U_\Omega(f) - \frac{\widehat{\tau}}{1-\widehat{\tau}}U_\Lambda(f). \tag{13}$$

Where $\widehat{\tau}$ is an estimator of $\tau$, $U_\Omega(f) := \sum_{r=1}^{R} \widehat{\rho}_r \widehat{\mathbb{E}}_{\Omega_r}[\ell_{\phi,f}]$ and $U_\Lambda(f) := \sum_{r=1}^{R} \widehat{\rho}_r \widehat{\mathbb{E}}_{\Lambda_r}[\ell_{\phi,f}]$ where for all $r \in [R]$, the collections $\Omega_r$ and $\Lambda_r$ of $(k+2)$-tuples are defined as:

$$\Omega_r := \left\{\left(X, X^+, \left\{X_i^-\right\}_{i=1}^k\right) : \tag{14}\right.$$
$$\left. X, X^+ \in S_r, \left\{X_i^-\right\}_{i=1}^k \subseteq S \setminus \left\{X, X^+\right\}\right\},$$

$$\Lambda_r := \left\{\left(X, X^+, \bar{X}, \left\{X_i^-\right\}_{i=1}^{k-1}\right) : \tag{15}\right.$$
$$\left. X, X^+, \bar{X} \in S_r, \left\{X_i^-\right\}_{i=1}^{k-1} \subseteq S \setminus \left\{X, X^+, \bar{X}\right\}\right\}.$$

Intuitively, when $U_\Omega(f), U_\Lambda(f)$ estimates the intermediary risks $\mathrm{L}_\Omega(f), \mathrm{L}_\Lambda(f)$ accurately, then $U_N(f)$ becomes a good estimator for the desired risk $\mathrm{L}_\phi(f)$ (depending also on the accuracy of $\widehat{\tau}$). Unfortunately, both $U_\Omega(f)$ and $U_\Lambda(f)$ are **biased estimators**, making it difficult to conduct any concentration analysis. To be more specific, for every tuple in each collection $\Omega_r$, the selection of negative samples $\left\{X_i^-\right\}_{i=1}^k$ is dependent on the previously selected anchor-positive pair $(X, X^+)$, making these negative samples no longer distributed according to the full mixture of class-conditional distributions $\bar{\mathcal{D}}$. Similarly, the same dependency structure exists for every tuple in each collection $\Lambda_r$.

To address the biased-ness caused by dependent tuples, we construct an **auxiliary estimator** $\bar{U}_N$ defined as follows:

$$\bar{U}_N(f) := \frac{1}{1-\widehat{\tau}}\bar{U}_\Omega(f) - \frac{\widehat{\tau}}{1-\widehat{\tau}}\bar{U}_\Lambda(f), \tag{16}$$

which is structurally similar to the definition of the proposed estimator $U_N$ in Eqn. (13) except $U_\Omega$ and $U_\Lambda$ are replaced by

$\bar{U}_\Omega$ and $\bar{U}_\Lambda$, which are (asymptotically) unbiased estimators of the intermediary population risks $L_\Omega$ and $L_\Lambda$. Particularly, in the construction of $\bar{U}_\Omega$ (and $\bar{U}_\Lambda$), we disassociate the selection of negative samples from anchor-positive pairs by first splitting the labeled dataset $S$ into non-overlapping partitions. Then, anchor-positives are only selected from one partition while negative samples are selected from the other, making all elements of each tuple independent. To preserve the overall flow, we defer the specific construction of both $\bar{U}_\Omega$ and $\bar{U}_\Lambda$ to Appendix C. Additionally, we highlight the difference between the construction of our estimator and that of the previous work in Figure 3. Furthermore, we also prove that when the sample size $N$ is large, the differences between $\bar{U}_\Omega$ and $U_\Omega$ as well as between $\bar{U}_\Lambda$ and $U_\Lambda$ can be relatively well-controlled (cf. Proposition D.6). Therefore, we can focus primarily on concentration properties of the proposed auxiliary estimator $\bar{U}_N$.

**Remark 4.3.** The idea of debiasing the empirical risk by decomposing the population risk based on class-collision has been explored in Arora (2019, Section 4.2). This decomposition has also been employed by later works to bound the surrogate gap between supervised and unsupervised risk. Recently, the same idea was utilized by Chuang et al. (2020) to mitigate negative sampling bias, although for a much simpler variant of contrastive learning (PU-learning) in **semi-supervised settings**. Therefore, we do not claim novelty for the empirical performance of the proposed U-Statistic $U_N$ in Eqn. (13). The primary focus of this work is providing sharpened excess risk bounds.

## Concentration of Auxiliary Estimators

In this section, we will focus on the concentration properties of the auxiliary estimator defined in Eqn. (16). Firstly, we introduce the following important definition.

**Definition 4.4** (Worst-case Dudley Entropy). Let $\mathcal{G}$ be a class of functions $g : \mathcal{Z} \to \mathbb{R}$ such that $|g| \leqslant \mathcal{B}$ for all $g \in \mathcal{G}$ and $n \geqslant 1$ be an integer. Then, the worst-case Dudley entropy of $\mathcal{G}$ over samples of size $n$ is defined as:

$$\mathfrak{C}_n(\mathcal{G}) := \sup_{S \in \mathcal{Z}^n} \int_{\frac{1}{n}}^{\mathcal{B}} \sqrt{\ln \mathcal{N}\left(\mathcal{G}, \varepsilon, \mathrm{L}_2(S)\right)} d\varepsilon, \quad (17)$$

where $\mathcal{N}(\mathcal{G}, \varepsilon, \mathrm{L}_2(S))$ denotes the $\varepsilon$-covering number of $\mathcal{G}$ with respect to the $\mathrm{L}_2(S)$ norm restricted to the dataset $S = \{\mathbf{z}_j\}_{j=1}^n$, defined for any $g, \bar{g} \in \mathcal{G}$ as follows:

$$\|g - \bar{g}\|_{\mathrm{L}_2(S)}^2 := \frac{1}{n} \sum_{j=1}^n |g(\mathbf{z}_j) - \bar{g}(\mathbf{z}_j)|^2. \quad (18)$$

In the following result, for a tuple-wise loss class $\mathcal{H}$, we express our generalization bounds in terms of the quantity $\mathfrak{C}_N(\mathcal{H})$, which is an upper bound on the worst-case Rademacher complexity (Lei et al., 2019; Srebro

et al., 2010) of the loss class $\mathcal{H}$ over any set of $N$ tuples $Z_1, \ldots, Z_N \in \mathcal{X}^{k+2}$. For a parametric model, $\mathfrak{C}_N(\mathcal{G})$ will be dominated by the the number of parameters in the model, up to polylogarithmic factors (Long & Sedghi, 2020; Graf et al., 2022), and the argument can be extended to $\mathfrak{C}_N(\mathcal{H})$ with arguments such as those in (Hieu et al., 2024) at the cost of polylogarithmic factors in all quantities including the number of negative samples $k$. For a class of linear maps $A \in \mathbb{R}^{m \times d}$ with norm bounded by $s$, the complexity will scale like $\widetilde{\mathcal{O}}(md)$ where the $\widetilde{\mathcal{O}}$ notation includes polylogarithmic factors of both $s$ and $k$, for a neural network, the complexity will scale as $\widetilde{\mathcal{O}}(\mathcal{D}L)$ where $L$ is the depth of the network and $\mathcal{D}$ is the number of parameters (Anthony & Bartlett, 1999; Graf et al., 2022; Ledent et al., 2025).

**Theorem 4.5** (cf. Theorem G.5, Table 1). *Let $\mathcal{F}$ be a class of representation functions and $\bar{U}_N(f)$ be defined for each $f \in \mathcal{F}$ as in Eqn. (16). Suppose that $R \geqslant k$. Then, for any $\Delta \in (0, 1)$, with probability of at least $1 - \Delta$, we have:*

$$\sup_{f \in \mathcal{F}} \left|\bar{U}_N(f) - \mathrm{L}_\phi(f)\right| \leqslant \quad (19)$$

$$\mathcal{O}\left(\frac{\mathfrak{C}_N(\mathcal{H})}{1 - \tau} \sqrt{\frac{k}{N}} + \frac{\mathcal{B}\sqrt{R}}{(1 - \tau)^2} \sqrt{\frac{\ln(R/\Delta)}{N}}\right),$$

*as long as $N \geqslant \widetilde{\mathcal{O}}\left(k \cdot \max\left\{R, k + \frac{1}{k(1-\tau)}\right\}\right)$. Here, the $\widetilde{\mathcal{O}}$ notation hides logarithmic orders of $R, \Delta$.*

First, we note that the above result and the corresponding sample complexity in Table 1 also applies to the natural estimator $U_N$ (cf. Remark B.1). For a fixed $\tau$ and sufficiently large $N$, the dominant term is $\mathfrak{C}_N(\mathcal{H})\sqrt{k/N}$, which indicates that the required number of samples grows like $k$. This is exactly what one would expect from a pure Hoeffding-type argument (Clémençon et al., 2008) over tuples in the case where collisions are ignored. We first note that even though a square-root dependency of $R$ shows up in the second term, it is completely detached from the *function class complexity factor* $\mathfrak{C}_N(\mathcal{H})$, which is a dominant quantity. For instance, if the number of parameters is $W$ (for a Lipschitz parametrized function class), Theorem 4.5 corresponds to a sample complexity of $\widetilde{\mathcal{O}}(Wk(1-\tau)^{-2} + R(1-\tau)^{-4})$: the additive factor of $R$ is only relevant if $W \leqslant R/k$. This is also largely unavoidable: if the number of samples $N$ is less than $\mathcal{O}(R)$, there are simply no valid tuples to be formed due to the inability to select *any* (positive, anchor positive) pair with distinct elements. Next, the bound also depends on the inverse of the non-collision probability $1 - \tau$. However, it should be noted that in extreme multi-class scenarios, this probability is typically well-controlled (i.e., constant) given that the "total mass of small class probabilities" (cf. Eqn. (20)) is sufficiently large, which certainly occurs in the approximately uniform extreme multi-class scenario: in this case we have $1 - \tau \simeq 1$. To provide better insights, we present several illustrative examples.

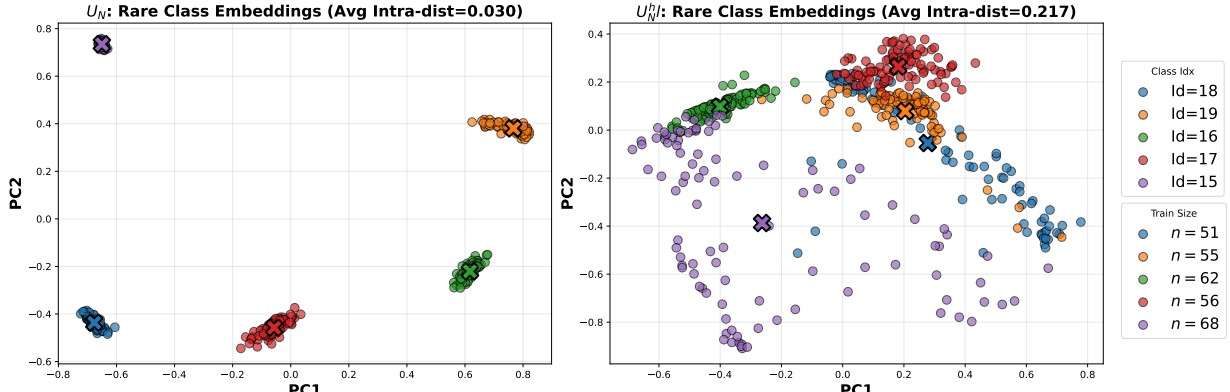

*Figure 1.* Synthetic data experiment results for representation learning on an imbalanced dataset. **Left**: PCA reduced representations of the model trained with sub-sampled estimation of $U_N$ as empirical risk. **Right**: PCA reduced representations of model trained with sub-sampled estimation of $U_N^{\mathrm{hl}}$ as empirical risk. Both plots show representations of 100 data points from top five rarest classes in the synthesized dataset. The legends show class indices as well as the number of data points from each class in the labeled training dataset.

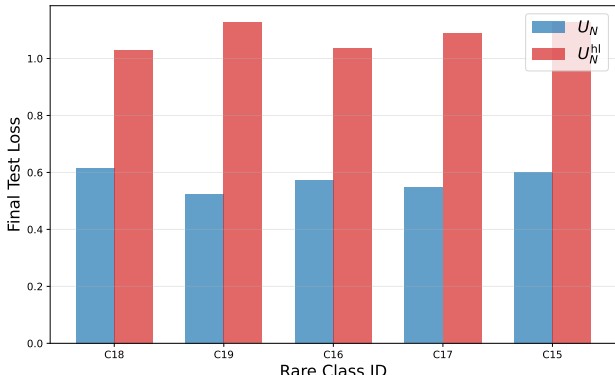

*Figure 2.* Final contrastive test loss for the model trained using sub-sampled estimations of two U-Statistics formulations, $U_N$ (blue) and $U_N^{\mathrm{hl}}$, as empirical risks (red) on synthetic dataset.

**Example 4.1** (Typical Extreme Scenario)**.** Suppose that $\rho_r \leqslant \frac{1}{k+1}$ for all $r \in [R]$. Then, for triplet learning ($k = 1$), we have $1 - \tau = 1 - \sum_{r=1}^{R} \rho_r^2 \geqslant 1 - \frac{1}{2}\sum_{r=1}^{R}\rho_r = \frac{1}{2}$. On the other hand, for $k \geqslant 2$, we have:

$$1 - \tau = \sum_{r=1}^{R} \rho_r(1 - \rho_r)^k \geqslant \left(1 - \frac{1}{k+1}\right)^k \geqslant \frac{1}{e}.$$

Essentially, when $R$ is large and all classes are relatively well-spread we have $1 - \tau \in \Omega(1)$.

**Example 4.2** (General Extreme Scenario)**.** Let $\gamma_k$ denote the total mass of small probabilities bounded by $1/k$. Formally, we define $\gamma_k$ as follows:

$$\gamma_k := \mathbb{P}_{r \sim \rho}(\rho_r \leqslant 1/k) = \sum_{r=1}^{R} \rho_r \mathbb{1}_{\{\rho_r \leqslant 1/k\}}. \quad (20)$$

Suppose that $\gamma_k \in \Omega(1)$, i.e., the total mass of small probabilities is not vanishingly small. Then, for $k \geqslant 2$, we have

the following lower bound on $1 - \tau$:

$$1 - \tau \geqslant \sum_{r:\rho_r \leqslant 1/k} \rho_r(1 - \rho_r)^k \geqslant \gamma_k\left(1 - \frac{1}{k}\right)^k \geqslant \frac{\gamma_k}{4}.$$

Therefore, we also have $1 - \tau \in \Omega(1)$ in this case. In other words, as long as $\gamma_k$ is not infinitesimally small, the main result (Theorem 4.5) yields a sample complexity of $k \cdot \mathfrak{C}_N(\mathcal{H})$ with probability of at least $1 - \Delta$ as long as $N \geqslant \widetilde{\mathcal{O}}(\max(Rk, k^2)) = \widetilde{\mathcal{O}}(Rk)$.

**Example 4.3** (Imbalanced Distribution)**.** Another interesting case where our result will be tighter is in imbalanced distributions favoring a few major classes (e.g., anomaly detection, medical imaging problems). As a simple example, let $\{\rho_1, \ldots, \rho_R\}$ be a class distribution with $\rho_{\max} = \frac{1}{2}$ and $\rho_r \leqslant \frac{1}{k+1}, \forall \rho_r \neq \rho_{\max}$. Since $[(1 - 1/(1+x))^x]' < 0$:

$$1 - \tau \geqslant \sum_{r:\rho_r < \rho_{\max}} \rho_r(1 - \rho_r)^k \geqslant \frac{1}{2}\left(1 - \frac{1}{k+1}\right)^k$$

$$\geqslant \frac{1}{2}\lim_{k \to \infty}\left(1 - \frac{1}{k+1}\right)^k = \frac{1}{2e}.$$

For clarity, this is a notable special case of the previous example with sufficiently large total mass of small probabilities (where $\gamma_k = 1/2$ in this example). In the Appendix, we also provide some dedicated results (Theorem G.3, G.4) for the scenario where $\gamma_k \in \Omega(1)$. In the following section, we will present empirical results to verify that the proposed estimator performs well under class imbalance.

## 5. Numerical Experiments

### 5.1. Experiment on Synthetic Imbalanced Dataset

We hypothesize that the estimator $U_N(f)$ in Eqn. (13) generalizes better than the formula $U_N^{\mathrm{hl}}(f)$ used by prior work

*Table 2.* Average precisions, recalls and F1-scores of a simple deep neural network trained using sub-sampled estimations of $U_N$ and $U_N^{\mathrm{hl}}$ as empirical risks. The one with higher score is highlighted in bold font.

| No. Negatives | Metric | MNIST | | FashionMNIST | | CIFAR10 | |
|---|---|---|---|---|---|---|---|
| | | $U_N$ | $U_N^{\mathrm{hl}}$ | $U_N$ | $U_N^{\mathrm{hl}}$ | $U_N$ | $U_N^{\mathrm{hl}}$ |
| | Avg. Precision | **0.9473** | 0.8243 | **0.7729** | 0.6369 | 0.2516 | **0.2916** |
| $k = 3$ | Avg. Recall | **0.8965** | 0.7650 | **0.7280** | 0.5545 | **0.1745** | 0.1240 |
| | Avg. F1-Score | **0.9208** | 0.7830 | **0.7473** | 0.5595 | **0.1994** | 0.1311 |
| | Avg. Precision | **0.9460** | 0.6119 | **0.4891** | 0.3361 | **0.2817** | 0.2668 |
| $k = 5$ | Avg. Recall | **0.8930** | 0.4860 | **0.4501** | 0.2139 | **0.1465** | 0.1455 |
| | Avg. F1-Score | **0.9186** | 0.4867 | **0.4123** | 0.1783 | **0.1795** | 0.1746 |
| | Avg. Precision | **0.9446** | 0.3928 | **0.4195** | 0.2822 | 0.2616 | **0.2968** |
| $k = 7$ | Avg. Recall | **0.9180** | 0.1970 | **0.3710** | 0.1870 | **0.1821** | 0.1455 |
| | Avg. F1-Score | **0.9311** | 0.2507 | **0.3637** | 0.1471 | **0.2111** | 0.1708 |

in certain long-tailed scenarios, especially for minority classes. For example, when a single class is overly dominant while others are extremely small and we have $k \ll R$. To verify this hypothesis, we synthesized an imbalanced dataset using a pre-defined mixture of Gaussian distributions $\bar{\mathcal{D}} = \sum_{r=1}^{R} \rho_r \mathcal{N}(\cdot | \mu_r, \sigma_r^2)$ where $R = 20$, $\sigma_r^2 = 1$ for all $r \in [R]$ and $\rho_{\max} = 0.5$ (one class dominating half of the dataset). Then, we trained the same model architecture (shallow neural network with ReLU activations) on the synthesized dataset with the **sub-sampled estimators** (cf. Section 5.4) corresponding to $U_N$ and $U_N^{\mathrm{hl}}$ as empirical risks until convergence. After training, we evaluate the performance of each estimator on the test set. In Figure 2, we report the final contrastive loss computed on test tuples whose anchor–positive pairs belong to the top five rarest classes. From the result, it can be seen that the model trained with the sub-sampled estimation of $U_N$ outperforms that trained using $U_N^{\mathrm{hl}}$ on every rare class. Additionally, we visualize the PCA transformed representations of each model on test data points of the rarest classes in Figure 1 and compare the intra-class compactness of each model's output. From the experiment results, we observe that the model trained with sub-sampled $U_N$ as the empirical risk achieved much better generalization performances on every rare class. Furthermore, the representations produced by this model are also much more compact compared to the representations produced by the model trained with the old estimator $U_N^{\mathrm{hl}}$. In Appendix I, we provide a more thorough description of experiment settings as well as detailed algorithms for calculating sub-sampled estimations of the two U-Statistics formulas $U_N$ and $U_N^{\mathrm{hl}}$.

### 5.2. Experiment on Real Imbalanced Dataset

We follow the same representation learning procedure on synthetic data for real datasets. For the real-data experiments, we extract subsets from MNIST, FashionMNIST, and CIFAR-10 to simulate a long-tailed class distribution. Specifically, we choose samples from the full datasets such

that one class is vastly dominant (occupying roughly half of the extracted subset) while the sizes of the remaining classes decay at an exponential rate. Using the representations learned by each model, one trained with sub-sampled $U_N$ and the other with sub-sampled $U_N^{\mathrm{hl}}$, we then train a simple linear classifier on top. Finally, we report the average precisions, recalls, and F1-scores of both models evaluated on the five rarest classes in each test set of the corresponding dataset. These metrics are reported for different values of negative samples $k$ in Table 2. Overall, all performance metric deteriorates as the number of negatives increases. This observation is consistent with the assessment of several empirical works in contrastive learning for long-tailed classification problems (Khosla et al., 2020; Wang et al., 2021; Li et al., 2022; Zhu et al., 2022; Hou et al., 2023). Intuitively, standard supervised contrastive learning typically performs poorly on long-tailed data because the overwhelming number of negative samples from major classes distorts the feature space and harms tail-class generalization. Interestingly, we can see that, despite the long-tailed condition, the model trained on the sub-sampled version of $U_N$ outperforms the one trained with sub-sampled $U_N^{\mathrm{hl}}$ on almost every metric across considered benchmarks. This empirical result further confirms our initial hypothesis about the tail-class generalization behavior of both U-Statistics.

### 5.3. Typical Extreme Multi-class Scenarios

In order to further verify that the proposed estimator $U_N$ outperforms the class-wise estimator $U_N^{\mathrm{hl}}$ in typical extreme multi-class scenarios, we conducted further experiments on MNIST, FashionMNIST, CIFAR10 and CIFAR100 with convolutional neural networks (CNNs). For MNIST, FashionMNIST and CIFAR10, we simulate the class-imbalanced distribution illustrated in Figure 4 as originally done for deep neural networks in Section 5.2. For CIFAR100, we use the entire balanced dataset without modifying the class distribution. The CNN architecture used in this experiment comprises of three (Conv $\rightarrow$ BN $\rightarrow$ ReLU $\rightarrow$ MaxPool)

*Table 3.* Average (macro) precisions, recalls and F1-scores of a simple *convolutional* neural network trained using sub-sampled estimations of $U_N$, $U_N^{\text{hl}}$ and SupCon as empirical risks. The one with higher score between the two estimators $U_N$ and $U_N^{\text{hl}}$ is highlighted in bold font. The metrics of SupCon is underlined if it outperforms both $U_N$, $U_N^{\text{hl}}$.

| Metr. | ♯Neg. | MNIST (imbl) | | | FashionMNIST (imbl) | | | CIFAR10 (imbl) | | | CIFAR100 | | |
| --- | --- | --- | --- | --- | --- | --- | --- | --- | --- | --- | --- | --- | --- |
| | | $U_N$ | $U_N^{\text{hl}}$ | SupCon | $U_N$ | $U_N^{\text{hl}}$ | SupCon | $U_N$ | $U_N^{\text{hl}}$ | SupCon | $U_N$ | $U_N^{\text{hl}}$ | SupCon |
| Prec. | $k=3$ | **0.9835** | 0.9238 | | **0.4146** | 0.3487 | | **0.8128** | 0.7794 | | **0.3490** | 0.2836 | |
| | $k=5$ | **0.9898** | 0.5031 | 0.9874 | **0.9212** | 0.3560 | 0.9382 | **0.8078** | 0.8045 | 0.8057 | **0.3239** | 0.3196 | 0.2772 |
| | $k=7$ | **0.9903** | 0.2845 | | **0.9253** | 0.3765 | | 0.7977 | **0.8033** | | **0.3173** | 0.2778 | |
| Rec. | $k=3$ | **0.9809** | 0.7599 | | **0.4998** | 0.4076 | | **0.5649** | 0.5404 | | **0.3298** | 0.2675 | |
| | $k=5$ | **0.9869** | 0.4315 | 0.9801 | **0.8629** | 0.4071 | 0.8435 | **0.5507** | 0.5324 | 0.6045 | **0.3422** | 0.2710 | 0.3095 |
| | $k=7$ | **0.9791** | 0.3711 | | **0.8451** | 0.3966 | | **0.5623** | 0.5515 | | **0.2895** | 0.2544 | |
| F1 | $k=3$ | **0.9821** | 0.7771 | | **0.4195** | 0.3222 | | **0.6592** | 0.6328 | | **0.3335** | 0.2723 | |
| | $k=5$ | **0.9883** | 0.4209 | 0.9837 | **0.8862** | 0.3212 | 0.8745 | **0.6406** | 0.6366 | 0.6813 | **0.3306** | 0.2929 | 0.2919 |
| | $k=7$ | **0.9846** | 0.3141 | | **0.8744** | 0.3111 | | 0.6481 | **0.6505** | | **0.2997** | 0.2640 | |

blocks with $32 \to 64 \to 128$ channels, respectively. Aside from the comparison to $U_N^{\text{hl}}$, we also ran this experiment on SupCon (Khosla et al., 2020) to provide a more comprehensive comparative study. Overall, the debiased estimator $U_N$ proposed in this work outperforms the class-wise estimator proposed in Hieu & Ledent (2025), as expected, with only a few exceptions on CIFAR10. Notably, in the typical extreme multi-class scenario reflected in CIFAR100, the performance of $U_N$ is better than both $U_N^{\text{hl}}$ and SupCon.

### 5.4. Estimation of U-Statistics with Sub-sampling

Direct computation of either $U_N$ or $U_N^{\text{hl}}$ requires contrastive loss evaluation over massive collections of valid tuples. Therefore, in the experiments described in Sections 5.1, 5.2 and 5.3, we used the **sub-sampled versions** of the U-Statistics estimators instead of the full averages. In this section, we provide a brief overview of sub-sampled estimators and a more detailed description along with experiment details are provided in Appendix I. Let $\{\mathbf{z}_j\}_{j=1}^K \subset \mathcal{Z}$ be a finite population of elements in some input space $\mathcal{Z}$ and $K$ is an extremely large number[1]. Let $h : \mathcal{Z} \to \mathbb{R}$ and $A_K^w(h) := \sum_{j=1}^K w(\mathbf{z}_j) h(\mathbf{z}_j)$ where $\forall j \in [K], w(\mathbf{z}_j) \in \mathbb{R}$ and $\sum_{j=1}^K w(\mathbf{z}_j) = 1$. Let $q$ be a distribution over the full population $\{\mathbf{z}_j\}_{j=1}^K$ and let $\{\mathbf{z}_\ell^*\}_{\ell=1}^M \subseteq \{\mathbf{z}_j\}_{j=1}^K$ be drawn i.i.d. from $q$. We define the sub-sampled average:

$$\widehat{A}_M^q(h) = \frac{1}{M} \sum_{\ell=1}^M \frac{w(\mathbf{z}_\ell^*)}{q(\mathbf{z}_\ell^*)} h(\mathbf{z}_\ell^*). \tag{21}$$

Trivially, we can show that $\mathbb{E}_q\left[\widehat{A}_M^q(h)\right] = A_K^w(h)$. Hence, by the law of large numbers, we have the convergence $\widehat{A}_M^q(h) \to A_K^w(h)$ with probability one as $M \to \infty$. In fact, one can easily show $|A_K^w(h) - \widehat{A}_M^q(h)| \in \mathcal{O}(1/\sqrt{M})$ with high-probability (with respect to draws from $q$) using simple concentration inequalities (e.g., see Hieu & Ledent (2025, Theorem 5.2)). Therefore, if the weight function $w$

is given, we can always construct a sub-sampled estimator that is close to the weighted sum over the full population using a proposed sub-sampling distribution $q$. Indeed, both U-Statistics in Eqn. (4) and Eqn. (13) can be written in the form of $A_K^w(h)$, which means that we can construct sub-sampled estimators corresponding to both formulations to conduct computationally feasible experiments.

## 6. Conclusion

In this work, we conduct a refined generalization analysis for the supervised contrastive representation learning framework. On the one hand, we provide a much tighter excess risk bound for the combined class-wise U-Statistic proposed by the previous work of Hieu & Ledent (2025). Specifically, we proved that the U-Statistic in Eqn. (4) yields a sample complexity that scales in the worst case with $R$, the number of classes, a rate only achievable by the previous work under the perfectly balanced classes assumption. On the other hand, we propose an entirely different estimator based on separately estimating two auxiliary contrastive risks, one with arbitrary number of class collisions and the other with at least one collision. Then, we combine both estimators via the overall class-collision probability. Under a mild assumption on the class distributions, we show that this estimator achieves a sample complexity that scales with $k$, the tuple size, in extreme multi-class scenarios and atypical conditions such as class imbalance. Using both theoretical examples and empirical experiments, we show that the U-Statistic estimator in Eqn. (13) reflects better generalization performance in a typical extreme multi-class scenario (cf. Section 5.3, CIFAR100). Furthermore, in certain long-tailed class distributions, our experiments (Section 5.2) also demonstrate that the proposed U-Statistic achieves better generalization performance compared to the class-wise formulation in prior work, especially for tail classes.

---

[1]In our case, $K$ refers to the size of valid tuples collection.

## Impact Statement

This paper is primarily theoretical in nature and we cannot foresee any negative societal impact.

## Acknowledgement

This research is supported by the National Research Foundation, Singapore under its AI Singapore Programme (AISG Award No: AISG3-PhD-2025-08-066T).

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

# A. Table of Notations

*Table 4.* Summary of key notations.

| Ntn. | Description | |
|---|---|---|
| **Frequently-used Notations** | | |
| $\mathcal{X}$ | Input data space | |
| $R$ | The number of classes | |
| $k$ | The number of negative samples | |
| $\rho_r$ | The occurrence probability of class $r \in [R]$ | |
| $\mathcal{D}_r$ | The distribution of data points coming from class $r \in [R]$ | |
| $\bar{\mathcal{D}}$ | $\bar{\mathcal{D}} := \sum_{r=1}^{R} \rho_r \mathcal{D}_r$, i.e., the full mixture of all class-conditional distributions | |
| $\bar{\mathcal{D}}_r$ | $\bar{\mathcal{D}}_r := \frac{1}{1-\rho_r} \sum_{q \neq r} \rho_q \mathcal{D}_q$, i.e., re-normalized mixture that excludes class $r \in [R]$ | |
| $S$ | $S = \{X_j\}_{j=1}^{N} \sim \bar{\mathcal{D}}^{\otimes N}$, i.e., the given dataset of $N$ labeled data points | |
| $S_r$ | The set of data points in $S$ that belongs to class $r \in [R]$ | |
| $\bar{S}_r$ | The set of data points in $S$ that **do not** belong to class $r \in [R]$ | |
| $N_r$ | The cardinality of $S_r$ | |
| $\widehat{\rho}_r$ | $\widehat{\rho}_r := \frac{N_r}{N}$, i.e., the empirical occurrence probability of class $r \in [R]$ | |
| $\rho_{\min}$ | $\rho_{\min} := \min_{r \in [R]} \rho_r$, i.e., the probability of the rarest class | |
| $\rho_{\max}$ | $\rho_{\max} := \max_{r \in [R]} \rho_r$, i.e., the probability of the most dominant class | |
| $\tau$ | $\tau := 1 - \sum_{r=1}^{R} \rho_r (1 - \rho_r)^k$, i.e., the population collision probability | |
| $\widehat{\tau}$ | $\widehat{\tau} := 1 - \sum_{r=1}^{R} \widehat{\rho}_r (1 - \widehat{\rho}_r)^k$, i.e., the plug-in estimator of $\tau$ | |
| $\phi$ | The $k$-dimensional vector-valued contrastive loss | (e.g., see Sohn (2016)) |
| $\ell_{\phi,f}$ | $\ell_{\phi,f} : \left( X, X^+, \{X_i^-\}_{i=1}^{k} \right) \mapsto \phi \left( \{ f(X)^\top [f(X^+) - f(X_i^-)] \}_{i=1}^{k} \right)$ | |
| $\mathcal{B}$ | The upper-bound of $\ell_{\phi,f}$, i.e., $|\ell_{\phi,f}| \leqslant \mathcal{B}$ for all $f \in \mathcal{F}$ | |
| $\Pi$ | The set of all bijections (i.e., permutations) $\pi : [N] \to [N]$ | |
| **Operators** | | |
| $\mathbb{E}[\cdot]$ | The population expectation over the distribution $\mu$ , written as $\mathbb{E}_\mu[\cdot]$ or $\underset{X \sim \mu}{\mathbb{E}}[\cdot]$ to specify the underlying distribution ($\mu$) in given contexts | |
| $\widehat{\mathbb{E}}_V[\cdot]$ | $\widehat{\mathbb{E}}_V[g] := \widehat{\mathbb{E}}_{v \sim V}[g(v)] = \frac{1}{|V|} \sum_{v \in V} g(v)$, i.e., empirical average over $V$ | |
| $\|\cdot\|_{\mathcal{G}}$ | Given a class of functions $\mathcal{G}$ and $F : \mathcal{G} \to \mathbb{R}$, $G : \mathcal{G} \to \mathbb{R}$ be two functionals, we have $\|F - G\|_{\mathcal{G}} := \sup_{g \in \mathcal{G}} |F(g) - G(g)|$, i.e., the uniform difference over $\mathcal{G}$ | |
| **Key Population Risks** | | |
| $\mathrm{L}_\phi$ | Population contrastive risk with no class-collision | (cf. Eqn. (2)) |
| $\mathrm{L}_\Omega$ | Population contrastive risk with zero or more class-collisions | (cf. Eqn. (9)) |
| $\mathrm{L}_\Lambda$ | Population contrastive risk with one or more class-collisions | (cf. Eqn. (10)) |
| $\mathrm{L}_\phi^r$ | Population class-wise risk with no class-collision | (cf. Eqn. (96)) |
| $\mathrm{L}_\Omega^r$ | Population class-wise risk with zero or more class-collisions | (cf. Eqn. (28)) |
| $\mathrm{L}_\Lambda^r$ | Population class-wise risk with one or more class-collisions | (cf. Eqn. (29)) |
| **Key Collections of Tuples** | | |
| The following are collections of tuples constructed from $S$ whose anchor-positive pairs are of class $r \in [R]$ | | |
| $\Theta_r$ | The negative samples in each tuple are not allowed to be of class $r$ | |
| $\Omega_r$ | The negative samples in each tuple include **zero or more** sample from class $r$ | |
| $\Lambda_r$ | The negative samples in each tuple include **one or more** sample from class $r$ | |
| **Key Estimators used in Main Text** | | |
| $U_{\Theta_r}$ | $U_{\Theta_r}(f) := \widehat{\mathbb{E}}_{\Theta_r}[\ell_{\phi,f}]$, i.e., the (asymptotically) unbiased estimator for $\mathrm{L}_\phi^r(f)$ | |
| $U_{\Omega_r}$ | $U_{\Omega_r}(f) := \widehat{\mathbb{E}}_{\Omega_r}[\ell_{\phi,f}]$, i.e., the "natural estimator" of $\mathrm{L}_\Omega^r(f)$ | |

| | |
|---|---|
| $U_{\Lambda_r}$ | $U_{\Lambda_r}(f) := \widehat{\mathbb{E}}_{\Lambda_r}[\ell_{\phi,f}]$, i.e., the "natural estimator" of $L_\Lambda^r(f)$ |
| $U_\Omega$ | $U_\Omega(f) := \sum_{r=1}^R \widehat{\rho}_r U_{\Omega_r}(f)$, i.e., the "natural estimator" of $L_\Omega(f)$ (Zero or more collisions) |
| $U_\Lambda$ | $U_\Lambda(f) := \sum_{r=1}^R \widehat{\rho}_r U_{\Lambda_r}(f)$, i.e., the "natural estimator" of $L_\Lambda(f)$ (One or more collisions) |
| $U_N^{hl}$ | $U_N^{hl}(f) := \sum_{r=1}^R \widehat{\rho}_r U_{\Theta_r}(f)$, i.e., the U-Statistic in Hieu & Ledent (2025) |
| $U_N$ | $U_N(f) := \frac{1}{1-\widehat{\tau}} U_\Omega(f) - \frac{\widehat{\tau}}{1-\widehat{\tau}} U_\Lambda(f)$, i.e., the proposed U-Statistic estimator |

## Construction of Auxiliary Estimators

For the following notations, we refer to $\Pi$ as the set of all bijections $\pi : [N] \to [N]$

Furthermore, we let $n = 2\lfloor N/(k+2) \rfloor$ and $m = 3\lfloor N/(k+2) \rfloor$

| | |
|---|---|
| $S_\pi$ | $S_\pi := \{X_{\pi(j)}\}_{j=1}^N$, i.e., the dataset $S$ permuted by $\pi$, meaning for each position $j \in [N]$, the data point $X_j$ is swapped with $X_{\pi(j)}$ |
| $S^{(n)}$ | $S^{(n)} := \{X_j\}_{j=1}^n$, i.e., the set of the first $n$ data points in $S$ |
| $\bar{S}^{(n)}$ | $\bar{S}^{(n)} := \{X_j\}_{j=n+1}^{N-n}$, i.e., the set of the last $N-n$ data points in $S$ |
| $S_\pi^{(n)}$ | $S_\pi^{(n)} := \{X_{\pi(j)}\}_{j=1}^n$, i.e., the set of the first $n$ data points of $S_\pi$ |
| $\bar{S}_\pi^{(n)}$ | $\bar{S}_\pi^{(n)} := \{X_{\pi(j)}\}_{j=n+1}^{N-n}$, i.e., the set of the last $N-n$ data points of $S_\pi$ |

The above notations extend to $S^{(m)}, \bar{S}^{(m)}, S_\pi^{(m)}$ and $\bar{S}_\pi^{(m)}$

| | |
|---|---|
| id | $\text{id} \in \Pi$ is the identity permutation, i.e., $\pi(j) = j, \forall j \in [N]$ |
| $n_r^\pi$ | $n_r^\pi := \lvert S_\pi^{(n)} \cap S_r \rvert$, i.e., number of data points in class $r$ from $S_\pi^{(n)}$ |
| $m_r^\pi$ | $m_r^\pi := \lvert S_\pi^{(m)} \cap S_r \rvert$, i.e., number of data points in class $r$ from $S_\pi^{(m)}$ |
| $\omega_r^\pi$ | $\omega_r^\pi := \lfloor n_r^\pi/2 \rfloor \big/ \left[ \sum_{q=1}^R \lfloor n_q^\pi/2 \rfloor \right]$, i.e., the fraction of independent 2-blocks from class $r$ in $S_\pi^{(n)}$ |
| $\lambda_r^\pi$ | $\lambda_r^\pi := \lfloor m_r^\pi/3 \rfloor \big/ \left[ \sum_{q=1}^R \lfloor m_q^\pi/3 \rfloor \right]$, i.e., the fraction of independent 3-blocks from class $r$ in $S_\pi^{(m)}$ |

We also define $n_r = n_r^{id}, m_r = m_r^{id}, \omega_r = \omega_r^{id}$ and $\lambda_r = \lambda_r^{id}$,

i.e., variants of $n_r^\pi, m_r^\pi, \omega_r^\pi, \lambda_r^\pi$ when no permutation takes place ($\pi = \text{id}$)

| | |
|---|---|
| $\Omega_r^\pi$ | The set of tuples with **zero or more** collisions whose anchor-positive pairs come from $S_\pi^{(n)}$ and negative samples come from $\bar{S}_\pi^{(n)}$ |
| $\Lambda_r^\pi$ | The set of tuples with **one or more** collisions whose anchor-positive-collided triplets come from $S_\pi^{(m)}$ and negative samples come from $\bar{S}_\pi^{(m)}$ |
| $U_{\Omega_r^\pi}$ | $U_{\Omega_r^\pi}(f) := \widehat{\mathbb{E}}_{\Omega_r^\pi}[\ell_{\phi,f}]$, i.e., an (asymptotically) unbiased estimator for $L_\Omega^r(f)$ |
| $U_{\Lambda_r^\pi}$ | $U_{\Lambda_r^\pi}(f) := \widehat{\mathbb{E}}_{\Lambda_r^\pi}[\ell_{\phi,f}]$, i.e., an (asymptotically) unbiased estimator for $L_\Lambda^r(f)$ |
| $\bar{U}_\Omega$ | $\bar{U}_\Omega(f) := \widehat{\mathbb{E}}_\Pi \left[ \sum_{r=1}^R \omega_r^\pi U_{\Omega_r^\pi}(f) \right]$, i.e., the auxiliary estimator for $U_\Omega(f)$ (Zero or more collisions) |
| $\bar{U}_\Lambda$ | $\bar{U}_\Lambda(f) := \widehat{\mathbb{E}}_\Pi \left[ \sum_{r=1}^R \lambda_r^\pi U_{\Lambda_r^\pi}(f) \right]$, i.e., the auxiliary estimator for $U_\Lambda(f)$ (One or more collisions) |
| $\bar{U}_N$ | $\bar{U}_N(f) := \frac{1}{1-\widehat{\tau}} \bar{U}_\Omega(f) - \frac{\widehat{\tau}}{1-\widehat{\tau}} \bar{U}_\Lambda(f)$, i.e., the auxiliary estimator for $U_N(f)$ |

## Auxiliary Risks

The following are auxiliary risks that we define for concentration analysis

(Meaning they will be important for understanding the main proofs)

| | |
|---|---|
| $\overline{\omega}_r$ | $\overline{\omega}_r := \mathbb{E}[\omega_r]$, i.e., the **population expectation** of the fraction $\omega_r$ |
| | Note that $\mathbb{E}[\omega_r] = \mathbb{E}[\omega_r^\pi]$ for any permutation $\pi \in \Pi$ |
| $\omega_r^*$ | $\omega_r^* := \widehat{\mathbb{E}}_\Pi[\omega_r^\pi]$, i.e., the **empirical average** of fractions $\omega_r^\pi$ over all permutations |
| $L_\Omega^*$ | $L_\Omega^*(f) := \sum_{r=1}^R \omega_r^* L_\Omega^r(f)$, i.e., the mixture of class-wise risks weighted by $\{\omega_r^*\}_{r=1}^R$ |
| $\overline{L}_\Omega$ | $\overline{L}_\Omega(f) := \sum_{r=1}^R \overline{\omega}_r L_\Omega^r(f)$, i.e., the mixture of class-wise risks weighted by $\{\overline{\omega}_r\}_{r=1}^R$ |
| $\widehat{L}_\Omega$ | $\widehat{L}_\Omega(f\lvert\pi) := \sum_{r=1}^R \omega_r^\pi L_\Omega^r(f)$, i.e., the mixture of class-wise risks weighted by $\{\omega_r^\pi\}_{r=1}^R$ |
| | Note that we also define $\widehat{L}_\Omega(f) := \widehat{L}_\Omega(f\lvert\text{id}) = \sum_{r=1}^R \omega_r L_\Omega^r(f)$ |

| | |
|---|---|
| $\overline{\lambda}_r$ | $\overline{\lambda}_r := \mathbb{E}[\lambda_r]$, i.e., the **population expectation** of the fraction $\lambda_r$ |

Note that $\mathbb{E}[\lambda_r] = \mathbb{E}[\lambda_r^\pi]$ for any permutation $\pi \in \Pi$

$\lambda_r^*$     $\lambda_r^* := \widehat{\mathbb{E}}_\Pi[\lambda_r^\pi]$, i.e., the **empirical average** of fractions $\lambda_r^\pi$ over all permutations

$L_\Lambda^*$     $L_\Lambda^*(f) := \sum_{r=1}^R \lambda_r^* L_\Lambda^r(f)$, i.e., the mixture of class-wise risks weighted by $\{\lambda_r^*\}_{r=1}^R$

$\overline{L}_\Lambda$     $\overline{L}_\Lambda(f) := \sum_{r=1}^R \overline{\lambda}_r L_\Lambda^r(f)$, i.e., the mixture of class-wise risks weighted by $\{\overline{\lambda}_r\}_{r=1}^R$

$\widehat{L}_\Lambda$     $\widehat{L}_\Lambda(f|\pi) := \sum_{r=1}^R \lambda_r^\pi L_\Lambda^r(f)$, i.e., the mixture of class-wise risks weighted by $\{\lambda_r^\pi\}_{r=1}^R$

Note that we also define $\widehat{L}_\Lambda(f) := \widehat{L}_\Lambda(f|\mathrm{id}) = \sum_{r=1}^R \lambda_r L_\Lambda^r(f)$

## Other Special Quantities

$\gamma_\alpha$     $\gamma_\alpha := \mathbb{P}_\rho(\rho_r \leqslant \alpha^{-1}) = \sum_{r:\rho_r \leqslant \alpha^{-1}} \rho_r$, i.e., total mass of small class probabilities

$\widehat{\gamma}_\alpha$     $\widehat{\gamma}_\alpha := \mathbb{P}_{\widehat{\rho}}(\widehat{\rho}_r \leqslant \alpha^{-1}) = \sum_{r:\widehat{\rho}_r \leqslant \alpha^{-1}} \widehat{\rho}_r$, i.e., total mass of small empirical probabilities

$\widehat{\theta}_{k+2}$     $\widehat{\theta}_{k+2} := \mathbb{P}_{\widehat{\rho}}(\widehat{\rho}_r \leqslant 2/(k+2))$, used in the refined concentration analysis of $U_N^{\mathrm{hl}}$

*Table 5.* Function classes and complexity measures.

| Ntn. | Description |
|------|-------------|
| **Function Classes** | |
| $\mathcal{F}$ | The class of representation functions |
| $\mathcal{H}$ | $\mathcal{H} := \left\{ \left( X, X^+, \{X_i^-\}_{i=1}^k \right) \mapsto \ell_{\phi,f} \left( X, X^+, \{X_i^-\}_{i=1}^k \right) : f \in \mathcal{F} \right\}$ |

**Complexity Measures**

Given an input space $\mathcal{Z}$, a class $\mathcal{G}$ of real-valued functions $g : \mathcal{Z} \to \mathbb{R}$, a distribution $\mu$
over $\mathcal{Z}^m$ and let $\mathbf{\Sigma}_m = \{\sigma_j\}_{j=1}^m$ be a sequence of independent Rademacher variables
Finally, let $S = \{\mathbf{z}_j\}_{j=1}^m \sim \mu$ be a dataset drawn from $\mu$ and suppose $|g| \leqslant \mathcal{B}$ for all $g \in \mathcal{G}$

| | |
|---|---|
| $\mathcal{N}(\mathcal{G}, \epsilon, \mathrm{L}_p(S))$ | The size of the smallest cover $\mathcal{C} \subseteq \mathcal{G}$ such that for all $g \in \mathcal{G}$, there exists $\bar{g} \in \mathcal{C}$ such that $\frac{1}{m} \sum_{j=1}^m |g(\mathbf{z}_j) - \bar{g}(\mathbf{z}_j)|^p \leqslant \epsilon^p$ |
| $\mathfrak{C}_m(\mathcal{G})$ | $\mathfrak{C}_m(\mathcal{G}) := \sup_{S^* \in \mathcal{Z}^m} \int_{1/m}^{\mathcal{B}} \sqrt{\ln 2\mathcal{N}(\mathcal{G}, \epsilon, \mathrm{L}_2(S^*))} d\epsilon$, i.e., the worst-case Dudley integral over datasets of size $m$ |
| $\widehat{\mathfrak{R}}_S(\mathcal{G})$ | $\widehat{\mathfrak{R}}_S(\mathcal{G}) := \mathbb{E}_{\mathbf{\Sigma}_m} \left[ \sup_{g \in \mathcal{G}} \left| \frac{1}{m} \sum_{j=1}^m g(\mathbf{z}_j) \right| \right]$, i.e., the empirical Rademacher complexity of $\mathcal{G}$ given the dataset $S$ |
| $\mathfrak{R}_\mu(\mathcal{G})$ | $\mathfrak{R}_\mu(\mathcal{G}) = \mathbb{E}_{S \sim \mu}[\mathfrak{R}_S(\mathcal{G})]$, i.e., the expected Rademacher complexity where expectation is taken over the distribution $\mu$ over $\mathcal{Z}^m$ |
| $\mathfrak{R}_m^{\mathrm{wc}}(\mathcal{G})$ | $\mathfrak{R}_m^{\mathrm{wc}}(\mathcal{G}) := \sup_{S^* \in \mathcal{Z}^m} \widehat{\mathfrak{R}}_{S^*}(\mathcal{G})$, i.e., the worst-case Rademacher complexity, over datasets of size $m$ |

# B. Summary of the Literature on the Concentration of the Unsupervised Risk

*Table 6.* Review of current works on *generalization bounds* (i.e., finite-sample concentration of the contrastive risk) for various regimes of CRL. Note that all results are expressed as sample complexity in terms of total number of labeled data points $N$: for the references studying the i.i.d. tuples regime (Arora, 2019; Lei et al., 2023), $N$ scales as $\mathcal{O}(N_{\text{tup}}k)$, which explains the rate of $k^2$ in Arora (2019). Indeed Arora (2019) requires $N_{\text{tup}} \in \widetilde{\mathcal{O}}(k)$ tuples, which corresponds to $N \in \widetilde{\mathcal{O}}(k^2)$ individual samples. Similarly, Lei et al. (2023) requires $N_{\text{tup}} \in \widetilde{\mathcal{O}}(1)$ tuples, which corresponds to $\widetilde{\mathcal{O}}(k)$ individual samples.

| Reference | Est. | Default | $\gamma_k \in \mathcal{O}(1)$ |
|---|---|---|---|
| **Unsupervised CRL** (i.i.d. tuples) | | | |
| Arora (2019) | I.I.D-ERM | $\mathfrak{C}_N^2(\mathcal{H})k^2$ | |
| Lei et al. (2023) | I.I.D-ERM | $\mathfrak{C}_N^2(\mathcal{H})k$ | |
| **Supervised CRL** (tuples constructed from pool of i.i.d. samples) | | | |
| Hieu & Ledent (2025) | $U_N^{\text{hl}}$ | $[\mathfrak{C}_N^2(\mathcal{H}) + 1] \max\left[\rho_{\min}^{-1}, (1 - \rho_{\max})^{-1}\right]$ | |
| This Work | $U_N^{\text{hl}}$ | $\mathfrak{C}_N^2(\mathcal{H})[\widehat{\theta}_{k+2}R + (1 - \widehat{\theta}_{k+2})^2 k] +$ $[R + k^2]$ (Theorem 4.1/H.5) | $\mathfrak{C}_N^2(\mathcal{H})R +$ $[R + k^2]$ (Theorem H.3) |
| This Work | $\bar{U}_N/U_N$ | $\mathfrak{C}_N^2(\mathcal{H})(1 - \tau)^{-2}k +$ $[R(1 - \tau)^{-4} + \max(Rk, k^2 + (1 - \tau)^{-1})]$ (Theorem 4.5/G.5) | $\mathfrak{C}_N^2(\mathcal{H})k +$ $[R + Rk]$ (Theorem G.4) |

**Remark B.1** (**on Theorems 4.5/G.5**). : We note that the sample complexity of $\bar{U}_N$ in Theorem 4.5 and Theorem G.5 also applies for the natural proposed estimator $U_N$. Specifically, the bias of $U_N$ incurred a cost of at most $\mathcal{O}(Rk)$ in sample complexity (cf. Proposition D.6), which is already present in the result for $\bar{U}_N$.

**Remark B.2** (**on Difference between Theorems 4.5/G.5 and Lei et al. (2023)**). At first glance, Lei et al. (2023) seems to yield a better sample complexity (independent of $\tau$). However, we note that the sample complexity of their work is for estimating the much easier collision-allowed risk, which we denote as $L_\Omega$. In the i.i.d.-sample regime investigated by this work, the result for $L_\Omega$ is also provided in Theorem E.7 (which yields $\widetilde{\mathcal{O}}(\mathfrak{C}_N^2(\mathcal{H})k + Rk)$ in sample complexity). However, this only serves as an intermediary step towards the result for the more interesting collision-free contrastive risk.

**Remark B.3** (**on Difference between i.i.d.-*sample* and i.i.d.-*tuple* Regimes**). As explained in the main text, the dominant component in the above results is the one which includes a factor of the complexity term $\mathfrak{C}_N^2(\mathcal{H})$: the other additive terms (such as $R + k^2$ in Theorem H.3) arise from the need for the empirical pool of samples to be non-degenerate w.r.t. the construction of valid tuples, and are *independent of the hypothesis class*.

These results only include contributions towards the speed of *concentration of the contrastive risk*: as explained in the main text, there are many works (Arora, 2019; Chuang et al., 2020; Li & Liu, 2021; Bao et al., 2022; Huang et al., 2023) which analyze the relationship between the minimization of the population-level (collided) unsupervised risk and the performance at the downstream classification task under various augmentation strategies. Whilst extremely valuable, this direction is an orthogonal consideration to our work, which studies the concentration of the collision-free contrastive risk in the supervised case. These works often consider different generation and augmentation regimes, and the concentration of the unsupervised risk component of the error usually *follows variants of the i.i.d. tuples regime*. For instance, Cui et al. (2025a) assumes that the positive and anchor positive samples are both generated with an augmentation strategy from the same sample (which elegantly matches practical applications). The negative samples are independently generated, similarly to the i.i.d. tuple regime. Similarly, in Cui et al. (2025b), study a variant of supervised CRL with non-collided i.i.d. tuples (as in Arora (2019) and Lei et al. (2023)) under both augmentation and label noise.

We also note that in the i.i.d. tuples regime, the *collision-free* contrastive risk can only be estimated if the learner can generate i.i.d. tuples with positive and negative samples following the correct class distribution, which essentially requires access to supervised label information. On the other hand, the collided contrastive risk can often be estimated under credible (class-agnostic) i.i.d. tuples generation assumptions. The finite-sample concentration of both collision-free and collided contrastive risks is also much easier to prove in the i.i.d. tuples regime as there is no need for Hoeffding-type arguments. Accordingly, most works in this branch of the literature focus on other considerations such as downstream classification performance and augmentation strategy.

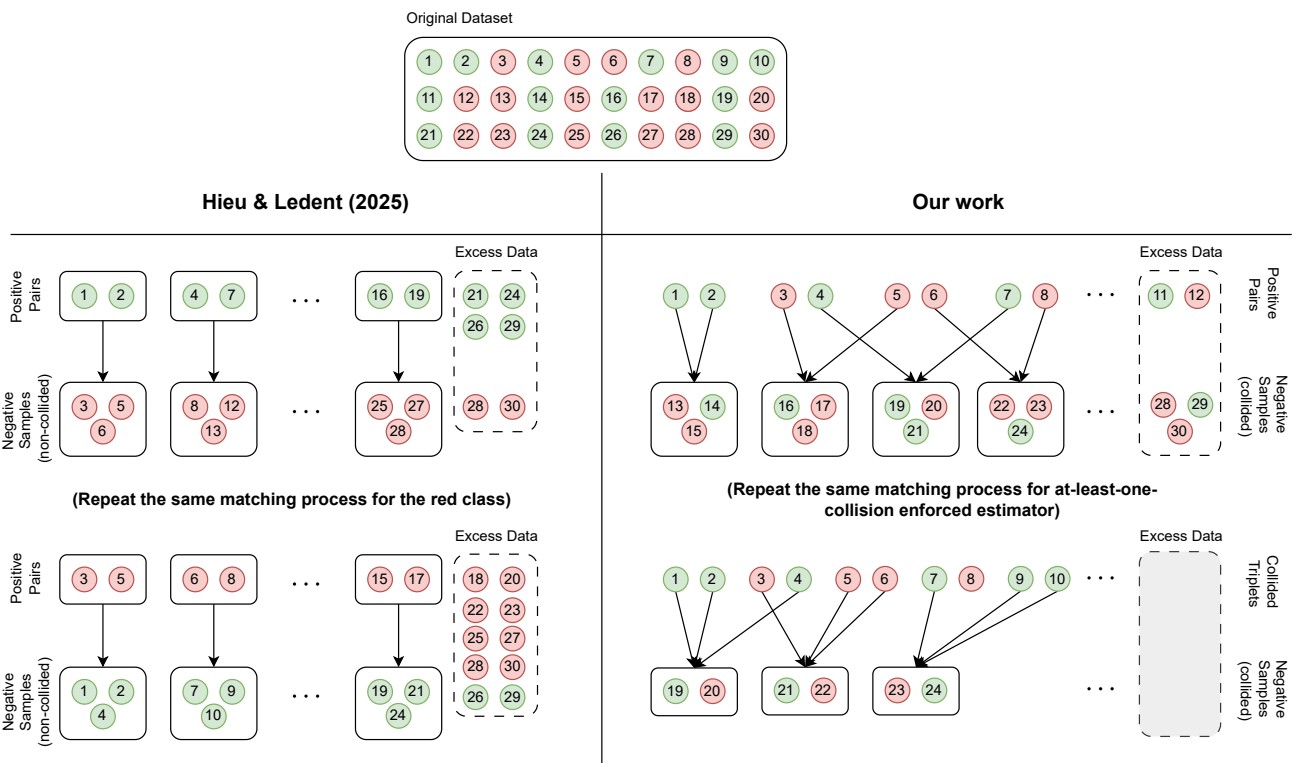

*Figure 3.* Comparison of the tuple selection process for each permutation of the labeled dataset $S$. For simplicity, only two classes are considered in this illustration (green and red). **Left**: Construction of the estimator $U_N^{\text{hl}}$ from Hieu & Ledent (2025): Tuples are selected class-wise for each component estimator defined in Hieu & Ledent (2025). $U_N^{\text{hl}}$ is expressed as an average of this construction over all permutations of the dataset, which is equivalent to a naive class-wise tuple selection from the empirical distribution (averaging all valid tuples for each class). **Right**: Construction of the auxiliary estimator $\bar{U}_N$ in our work. The tuples are selected to estimate both arbitrary-collision risk $\bar{U}_\Omega$ (top-right) and at-least-one-collision risk $\bar{U}_\Lambda$ (bottom-right). $\bar{U}_N$ is defined as a weighted combination of the averages of $\bar{U}_\Omega$ and $\bar{U}_\Lambda$ over all permutations of the dataset (cf. Eqns (24), (27)). In the final (natural) estimator $U_N$, the arbitrary and at-least-one-collision components are both constructed naturally by averaging all valid tuples. Unlike in the class-wise construction from Hieu & Ledent (2025), the natural and auxiliary estimators $\bar{U}_N$ and $U_N$ are not equal to each other (due to a mismatch between the empirical and population collision probabilities), but they can be shown to be asymptotically close (cf. Proposition D.6).

## C. Auxiliary Estimators

### C.1. Construction

In this section, we construct (asymptotically) unbiased estimators of $\mathrm{L}_\Omega(f)$ and $\mathrm{L}_\Lambda(f)$ for $f \in \mathcal{F}$ that are close to our proposed natural estimators $U_\Omega(f), U_\Lambda(f)$. For a bijective map $\pi : [N] \to [N]$, we define $S_\pi = \{X_{\pi(j)}\}_{j=1}^N$, i.e., the original dataset $S$ shuffled by permuting positions of data-points according to $\pi$. Let $n = 2\left\lfloor \frac{N}{k+2} \right\rfloor$ and $m = 3\left\lfloor \frac{N}{k+2} \right\rfloor$, we define $S_\pi^{(n)} = \{X_{\pi(j)}\}_{j=1}^n$ and $S_\pi^{(m)} = \{X_{\pi(j)}\}_{j=1}^m$, i.e., the first $n$ and $m$ data-points of the shuffled dataset $S_\pi$. Then, we define the following collections of tuples:

$$\Omega_r^\pi := \Big\{ \Big(X, X^+, \{X_i^-\}_{i=1}^k\Big) : X, X^+ \in S_r \cap S_\pi^{(n)}, \tag{22}$$

$$\{X_i^-\}_{i=1}^k \subseteq S \setminus S_\pi^{(n)}\Big\}.$$

$$\Lambda_r^\pi := \Big\{ \Big(X, X^+, \bar{X}, \{X_i^-\}_{i=1}^{k-1}\Big) : X, X^+, \bar{X} \in S_r \cap S_\pi^{(m)}, \tag{23}$$

$$\{X_i^-\}_{i=1}^{k-1} \subseteq S \setminus S_\pi^{(m)}\Big\}.$$

Essentially, Eqn. (22) denotes the collection of tuples whose anchor-positive pairs are selected from class-$r$ data-points from the first partition of $S_\pi$ while negative samples are selected from the other partition. Similarly, $\Lambda_r^\pi$ denotes the collection of

tuples where the anchor-positive-collided triplets are drawn from the first partition. Let $\Pi$ denote the set of all bijective maps $\pi : [N] \to [N]$, we define the auxiliary estimators $\bar{U}_\Omega(f)$ and $\bar{U}_\Lambda(f)$ as follows:

$$\bar{U}_\Omega(f) := \widehat{\mathbb{E}}_\Pi \left[ \sum_{r=1}^R \omega_r^\pi U_{\Omega_r^\pi}(f) \right], \qquad \bar{U}_\Lambda(f) := \widehat{\mathbb{E}}_\Pi \left[ \sum_{r=1}^R \lambda_r^\pi U_{\Lambda_r^\pi}(f) \right]. \tag{24}$$

Where for each $\pi \in \Pi$ and $r \in [R]$, we define:

$$U_{\Omega_r^\pi}(f) := \widehat{\mathbb{E}}_{\Omega_r^\pi} \left[ \ell_{\phi,f} \right], \quad \omega_r^\pi := \frac{\lfloor n_r^\pi / 2 \rfloor}{\sum_{q=1}^R \lfloor n_q^\pi / 2 \rfloor}. \tag{25}$$

$$U_{\Lambda_r^\pi}(f) := \widehat{\mathbb{E}}_{\Lambda_r^\pi} \left[ \ell_{\phi,f} \right], \quad \lambda_r^\pi := \frac{\lfloor m_r^\pi / 3 \rfloor}{\sum_{q=1}^R \lfloor m_q^\pi / 3 \rfloor}. \tag{26}$$

And we define $n_r^\pi := |S_r \cap S_\pi^{(n)}|$ and $m_r^\pi := |S_r \cap S_\pi^{(m)}|$, i.e., the number of class-$r$ data points in $S_\pi^{(n)}$ and $S_\pi^{(m)}$, respectively. Then, we can combine the auxiliary estimators as follows:

$$\bar{U}_N(f) := \frac{1}{1-\widehat{\tau}} \bar{U}_\Omega(f) - \frac{\widehat{\tau}}{1-\widehat{\tau}} \bar{U}_\Lambda(f). \tag{27}$$

In the following section, we will prove that the auxiliary estimators $\bar{U}_\Omega(f)$ and $\bar{U}_\Lambda(f)$ are indeed (asymptotically) unbiased. Then, we will provide formal proof for our claim that for large $N$, the difference between $\bar{U}_\Omega(f)$ and $U_\Omega(f)$ (as well as the difference between $\bar{U}_\Lambda(f)$ and $U_\Lambda(f)$) is negligible (i.e., Proposition D.6).

### C.2. Discussion of the Proofs

As illustrated in Figure 3, the construction of the new estimators $U_N$ and $\bar{U}_N$ differs enormously from that of the previous work. To the best of our knowledge, this is necessary to achieve our improved sample complexity. This radical change of direction inevitably leads to a variety of difficulties in the analysis:

(i) The first difficulty has already been mentioned in the main text: the selection of negative samples to compute the arbitrary-collision estimator $U_\Omega$ (as well as the at-least-one-collision estimator $U_\Lambda$) follows the **wrong distribution**. Specifically, if we are given a dataset $S$ drawn i.i.d. from $\bar{\mathcal{D}}^{\otimes N}$ (where $\bar{\mathcal{D}}$ denotes the full mixture of class-conditional distributions), then randomly hold out two data points from class $r \in [R]$ (to be used as positive and anchor positive), the remaining $N-2$ data points **no longer follow** $\bar{\mathcal{D}}$. Intuitively, this is because every time we hold out data points in class $r$, the probability of re-selecting another data point from class $r$ slightly decreases while probabilities of other classes slightly increase. In other words, the weights of all member distributions in the full mixture $\bar{\mathcal{D}}$ have been distorted. The more serious problem is that the subsequently selected negatives now depend on the pre-selected anchor-positive pair, leading to **within-tuple dependence** (between samples). Note that this is a different and more challenging dependence type from what we and the work before us are dealing with. Simpler Hoeffding-type arguments are aimed at **between-tuple** dependence (present in all estimators including $U_N$, $\bar{U}_N$ and $U_N^{\mathrm{hl}}$), which usually means that while the tuples are not independent of each other due to *repeating samples*, all member samples of every tuple are independent. In contrast, only $U_N$ exhibits *within tuple dependence*, which is the reason why we must construct the auxiliary estimator $\bar{U}_N$.

(ii) This issue directly motivates our decision to partition the dataset as illustrated in Figure 3 when constructing the auxiliary estimator $\bar{U}_N$. When we partition the dataset $S$ deterministically, i.e., shuffle $S$ by a permutation $\pi \in \Pi$ then separate $S$ into two partitions at a fixed index, we can treat each partition as a dataset drawn i.i.d. from $\bar{\mathcal{D}}$. However, one question might arise for many readers: **why do we have to construct $\bar{U}_\Omega$ (and $\bar{U}_\Lambda$) by averaging over all possible splits**? It is true that we could attempt to build $\bar{U}_\Omega$ as simply $\widetilde{U}_\Omega := \sum_{r=1}^R \omega_r^\pi U_{\Omega_r^\pi}(f)$ based on a particular split $\pi \in \Pi$. However, let us investigate how this would affect bounding the uniform bias $\|\widetilde{U}_\Omega - U_\Omega\|_{\mathcal{F}}$. Applying a similar argument as Lemma D.2, we have:

$$\left\| U_\Omega - \widetilde{U}_\Omega \right\|_{\mathcal{F}} \leqslant \sum_{r=1}^R \widehat{\rho}_r \sup_{f \in \mathcal{F}} \left| U_{\Omega_r}(f) - U_{\Omega_r^\pi}(f) \right| + \mathcal{B} \sum_{r=1}^R |\omega_r^\pi - \widehat{\rho}_r|.$$

This already reveals a significant problem: for each $r \in [R]$, the averages $U_{\Omega_r}$ and $U_{\Omega_r^\pi}$ are not supported on the same set of tuples. Furthermore, conditionally given the sample sizes $\{N_r\}_{r \in [R]}$, both $\Omega_r^\pi$ and $\omega_r^\pi$ are random due to the randomness of $\pi$, making it hard to bound either $|U_{\Omega_r}(f) - U_{\Omega_r^\pi}(f)|$ or $|\omega_r^\pi - \widehat{\rho}_r|$. On the other hand, if we average $U_{\Omega_r^\pi}$ over all permutations in $\Pi$, the task of controlling the uniform bias reduces to bounding the difference between two probability mass functions corresponding to the Hypergeometric$(N, N_r, k)$ and Hypergeometric$(N - 2, N_r - 2, k)$ distributions (see Lemma D.5).

(iii) The most technically cumbersome challenge arises in the concentration analysis of the auxiliary arbitrary-collision estimator $\bar{U}_\Omega$ (and the auxiliary at-least-one-collision estimator $\bar{U}_\Lambda$). Because we constructed $\bar{U}_\Omega$ as an average over all $\pi \in \Pi$, we cannot directly apply the classic decoupling techniques in Gine & Zinn (1984); Arcones & Gine (1993); Clémençon et al. (2008) as done in the previous work. In Hieu & Ledent (2025), each class-wise estimator is treated as a second-order U-statistic with a random sample size. While this randomness introduces some technical overhead, conditioning on the class counts immediately permits the use of standard decoupling arguments. After conditioning, the concentration of the empirical class probabilities can be handled separately and does not interact with the complexity of the function class. Concretely, upon conditioning on $\{N_r\}_{r \in [R]}$, each class-wise U-Statistic $U_{\Theta_r}$ concentrates to the correct class-wise risk $L_\phi^r(f)$, thereby enabling symmetrization and subsequent U-statistics decoupling. In contrast, in our setting, conditioning on the sample sizes alone does not guarantee that $\bar{U}_\Omega$ concentrates around its target risk $L_\phi$. As a result, our analysis requires a more delicate argument based on a sequence of auxiliary risks to bridge this gap.

(iv) Finally, since we combine $\bar{U}_\Omega$ and $\bar{U}_\Lambda$ via a de-biasing formula that involves the empirical collision probability $\widehat{\tau}$, we had to analyze the concentration (both absolute and multiplicative) of this quantity around the population counterpart $\tau$ (cf. Propositions F.1, F.2, F.4, F.5, F.7).

# D. Bridging Natural Estimators and Auxiliary Estimators

Recall from the main text that for a representation function $f \in \mathcal{F}$ we proposed the U-Statistic formulation $U_N(f)$ (Eqn. (13)), which is a combination of two estimators $U_\Omega(f)$ and $U_\Lambda(f)$ that estimate $\mathrm{L}_\Omega(f)$ and $\mathrm{L}_\Lambda(f)$, respectively.

Unfortunately, both $U_\Omega(f)$ and $U_\Lambda(f)$ are biased estimators. Therefore, we proposed $\bar{U}_\Omega(f)$ and $\bar{U}_\Lambda(f)$ as two auxiliary estimators and we claim that:

1. Both $\bar{U}_\Omega(f)$ and $\bar{U}_\Lambda(f)$ are asymptotically unbiased estimators of $\mathrm{L}_\Omega(f)$ and $\mathrm{L}_\Lambda(f)$.

2. They are different from $U_\Omega(f)$ and $U_\Lambda(f)$ by negligible factors.

Prior to presenting the proof for Proposition D.6, we first start by proving the asymptotic unbiasedness of $\bar{U}_\Omega(f)$ and $\bar{U}_\Lambda(f)$. Before that, for completeness of definitions, we define the class-wise risks $\mathrm{L}_\Omega^r$ and $\mathrm{L}_\Lambda^r$ as follows:

$$\mathrm{L}_\Omega^r(f) = \underset{\substack{X,X^+\sim\mathcal{D}_r^{\otimes 2} \\ \{X_i^-\}_{i=1}^k \sim \bar{\mathcal{D}}^{\otimes k}}}{\mathbb{E}} \left[ \ell_{\phi,f}\left( X, X^+, \{X_i^-\}_{i=1}^k \right) \right], \tag{28}$$

$$\mathrm{L}_\Lambda^r(f) = \underset{\substack{X,X^+,\bar{X}\sim\mathcal{D}_r^{\otimes 3} \\ \{X_i^-\}_{i=1}^{k-1} \sim \bar{\mathcal{D}}^{\otimes k-1}}}{\mathbb{E}} \left[ \ell_{\phi,f}\left( X, X^+, \bar{X}, \{X_i^-\}_{i=1}^{k-1} \right) \right]. \tag{29}$$

**Proposition D.1.** *For $f \in \mathcal{F}$, let $\bar{U}_\Omega(f)$ be defined as in Eqn. (24). Then, we have:*

$$\left( 1 - \frac{2}{n\rho_{\min}} \right) \mathrm{L}_\Omega(f) \leqslant \mathbb{E}[\bar{U}_\Omega(f)] \leqslant \left( 1 + \frac{R}{n-R} \right) \mathrm{L}_\Omega(f). \tag{30}$$

*Where $\rho_{\min} = \min_{r\in[R]} \rho_r$, i.e., the probability of the rarest class.*

**Proof.** Firstly, we decompose the risk $\mathrm{L}_\Omega(f)$ to a weighted sum of class-wise risk:

$$\mathrm{L}_\Omega(f) = \sum_{r=1}^R \rho_r \mathrm{L}_\Omega^r(f). \tag{31}$$

We observe that when we shuffle the dataset $S$ according to a permutation $\pi \in \Pi$ then select the first $n$ elements, this process is equivalent to selecting $n$ data points from $S$ without replacement. Therefore, for any $\pi_1, \pi_2 \in \Pi$ and $\pi_1 \neq \pi_2$, $S_{\pi_1}^{(n)} \overset{d}{=} S_{\pi_2}^{(n)}$ and $S \setminus S_\pi^{(n)} \overset{d}{=} S \setminus S_\pi^{(n)}$ [2]. Hence, for any distinct permutations $\pi, \bar{\pi} \in \Pi$, we have:

$$\mathbb{E}\left[ \sum_{r=1}^R \omega_r^\pi U_{\Omega_r^\pi}(f) \right] = \mathbb{E}\left[ \sum_{r=1}^R \omega_r^{\bar{\pi}} U_{\Omega_r^{\bar{\pi}}}(f) \right].$$

Now, fixing an arbitrary permutation $\theta \in \Pi$, we have:

$$\mathbb{E}[\bar{U}_\Omega(f)] = \mathbb{E}\left[ \frac{1}{N!} \sum_{\pi\in\Pi} \sum_{r=1}^R \omega_r^\pi U_{\Omega_r^\pi}(f) \right] = \frac{1}{N!} \sum_{\pi\in\Pi} \mathbb{E}\left[ \sum_{r=1}^R \omega_r^\pi U_{\Omega_r^\pi}(f) \right]$$

$$= \mathbb{E}\left[ \sum_{r=1}^R \omega_r^\theta U_{\Omega_r^\theta}(f) \right] = \sum_{r=1}^R \mathbb{E}\left[ \omega_r^\theta U_{\Omega_r^\theta}(f) \right].$$

For each $r \in [R]$, we have:

$$\mathbb{E}\left[ \omega_r^\theta U_{\Omega_r^\theta}(f) \right] = \underbrace{\mathbb{E}[\omega_r^\theta U_{\Omega_r^\theta}(f)|n_r^\theta \leqslant 1]}_{=0 \text{ since } \Omega_r^\theta = \emptyset} \mathbb{P}(n_r^\theta \leqslant 1) + \mathbb{E}[\omega_r^\theta U_{\Omega_r^\theta}(f)|n_r^\theta \geqslant 2]\mathbb{P}(n_r^\theta \geqslant 2)$$

$$= \mathbb{E}[\omega_r^\theta U_{\Omega_r^\theta}(f)|n_r^\theta \geqslant 2]\mathbb{P}(n_r^\theta \geqslant 2).$$

---

[2] Where $X \overset{d}{=} Y$ means "$X$ is identically distributed as $Y$".

Since $\omega_r^\theta$ and $U_{\Omega_r^\theta}(f)$ are conditionally independent given $n_r^\theta$, we can write:

$$\mathbb{E}\left[\omega_r^\theta U_{\Omega_r^\theta}(f)|n_r^\theta \geqslant 2\right] = \mathbb{E}\left[\omega_r^\theta|n_r^\theta \geqslant 2\right] \mathbb{E}\left[U_{\Omega_r^\theta}(f)|n_r^\theta \geqslant 2\right]$$
$$= \mathbb{E}\left[\omega_r^\theta|n_r^\theta \geqslant 2\right] \mathrm{L}_\Omega^r(f).$$

Furthermore, we have:

$$\mathbb{E}[\omega_r^\theta] = \mathbb{E}\left[\omega_r^\theta|n_r^\theta \geqslant 2\right]\mathbb{P}(n_r^\theta \geqslant 2) + \underbrace{\mathbb{E}\left[\omega_r^\theta|n_r^\theta \leqslant 1\right]}_{=0}\mathbb{P}(n_r^\theta \leqslant 1)$$
$$= \mathbb{E}\left[\omega_r^\theta|n_r^\theta \geqslant 2\right]\mathbb{P}(n_r^\theta \geqslant 2).$$

Therefore:

$$\mathbb{E}\left[\omega_r^\theta U_{\Omega_r^\theta}(f)\right] = \mathbb{E}[\omega_r^\theta U_{\Omega_r^\theta}(f)|n_r^\theta \geqslant 2]\mathbb{P}(n_r^\theta \geqslant 2)$$
$$= \mathbb{P}(n_r^\theta \geqslant 2)\mathbb{E}\left[\omega_r^\theta|n_r^\theta \geqslant 2\right]\mathrm{L}_\Omega^r(f)$$
$$= \mathbb{P}(n_r^\theta \geqslant 2) \cdot \frac{\mathbb{E}[\omega_r^\theta]}{\mathbb{P}(n_r^\theta \geqslant 2)}\mathrm{L}_\Omega^r(f)$$
$$= \mathbb{E}[\omega_r^\theta]\mathrm{L}_\Omega^r(f).$$

Using the following facts:

$$\frac{n_r^\theta - 1}{2} \leqslant \lfloor n_r^\theta/2 \rfloor \qquad \leqslant \frac{n_r^\theta}{2}$$
$$\frac{n - R}{2} = \sum_{q=1}^R \left[\frac{n_r}{2} - \frac{1}{2}\right] \leqslant \sum_{q=1}^R \lfloor n_q^\theta/2 \rfloor \leqslant \sum_{q=1}^R \frac{n_r^\theta}{2} = \frac{n}{2}.$$

We have:

$$\mathbb{E}[\omega_r^\theta] = \mathbb{E}\left[\frac{\lfloor n_r^\theta/2 \rfloor}{\sum_{q=1}^R \lfloor n_q^\theta/2 \rfloor}\right] \leqslant \mathbb{E}\left[\frac{n_r^\theta/2}{n/2 - R/2}\right] = \mathbb{E}\left[\frac{n_r^\theta}{n - R}\right]$$
$$= \mathbb{E}\left[\frac{\frac{n_r^\theta}{n}(n - R) + \frac{Rn_r^\theta}{n}}{n - R}\right]$$
$$= \mathbb{E}\left[\frac{n_r^\theta}{n}\left(1 + \frac{R}{n - R}\right)\right]$$
$$= \rho_r\left(1 + \frac{R}{n - R}\right) \qquad (n_r^\theta \sim \mathrm{Binom}(n, \rho_r)).$$

And:

$$\mathbb{E}[\omega_r^\theta] = \mathbb{E}\left[\frac{\lfloor n_r^\theta/2 \rfloor}{\sum_{q=1}^R \lfloor n_q^\theta/2 \rfloor}\right] \geqslant \mathbb{E}\left[\frac{(n_r^\theta - 2)/2}{n/2}\right] = \rho_r - \frac{2}{n}$$
$$\geqslant \rho_r\left(1 - \frac{2}{n\rho_{\min}}\right).$$

Combining the upper and lower bounds of $\mathbb{E}[\omega_r^\theta]$, we have:

$$\rho_r\left(1 - \frac{2}{n\rho_{\min}}\right)\mathrm{L}_\Omega^r(f) \leqslant \mathbb{E}\left[\omega_r^\theta U_{\Omega_r^\theta}(f)\right] \leqslant \rho_r\left(1 + \frac{R}{n - R}\right)\mathrm{L}_\Omega^r(f).$$

Finally, we have:

$$\left(1 - \frac{2}{n\rho_{\min}}\right)\sum_{r=1}^R \rho_r\mathrm{L}_\Omega^r(f) \leqslant \sum_{r=1}^R \mathbb{E}\left[\omega_r^\theta U_{\Omega_r^\theta}(f)\right] \leqslant \left(1 + \frac{R}{n - R}\right)\sum_{r=1}^R \rho_r\mathrm{L}_\Omega^r(f).$$

In other words, we have:

$$\left(1 - \frac{2}{n\rho_{\min}}\right) \mathrm{L}_\Omega(f) \leqslant \mathbb{E}[\bar{U}_\Omega(f)] \leqslant \left(1 + \frac{R}{n-R}\right) \mathrm{L}_\Omega(f),$$

as desired. As $N \to \infty$, we have $n \to \infty$. Hence, $\lim_{N\to\infty} \mathbb{E}[\bar{U}_\Omega(f)] = \lim_{n\to\infty} \mathbb{E}[\bar{U}_\Omega(f)]$ and:

$$\mathrm{L}_\Omega(f) = \lim_{n\to\infty}\left(1 - \frac{2}{n\rho_{\min}}\right)\mathrm{L}_\Omega(f) \leqslant \lim_{n\to\infty}\mathbb{E}[\bar{U}_\Omega(f)] \leqslant \lim_{n\to\infty}\left(1 + \frac{R}{n-R}\right)\mathrm{L}_\Omega(f) = \mathrm{L}_\Omega(f).$$

Therefore, $\lim_{n\to\infty}\mathbb{E}[\bar{U}_\Omega(f)] = \mathrm{L}_\Omega(f)$, making $\bar{U}_\Omega(f)$ asymptotically unbiased. $\qquad\square$

**Lemma D.2.** *For $f \in \mathcal{F}$, let $U_\Omega(f)$ be defined as in Eqn. (13) and $\bar{U}_\Omega(f)$ be defined as in Eqn. (24). Suppose that $|\ell_{\phi,f}| \leqslant \mathcal{B}$ for all $f \in \mathcal{F}$. Then, we have:*

$$\sup_{f\in\mathcal{F}} \left|U_\Omega(f) - \bar{U}_\Omega(f)\right| \leqslant \frac{2\mathcal{B}R}{n-R} + \sum_{r=1}^{R} \widehat{\rho}_r \sup_{f\in\mathcal{F}} \left|U_{\Omega_r}(f) - \widehat{\mathbb{E}}_\Pi\left[U_{\Omega_r^\pi}(f)\right]\right|, \tag{32}$$

*where for all $r \in [R]$, we define $\widehat{\rho}_r := \frac{N_r}{N}$ and $\widehat{\mathbb{E}}_\Pi[\cdot]$ denotes the average over permutations $\pi \in \Pi$.*

**Proof.** We have:

$$\sup_{f\in\mathcal{F}}\left|U_\Omega(f) - \bar{U}_\Omega(f)\right| = \sup_{f\in\mathcal{F}}\left|\sum_{r=1}^{R}\widehat{\rho}_r U_{\Omega_r}(f) - \widehat{\mathbb{E}}_\Pi\left[\sum_{r=1}^{R}\omega_r^\pi U_{\Omega_r^\pi}(f)\right]\right|$$

$$= \sup_{f\in\mathcal{F}}\left|\sum_{r=1}^{R}\widehat{\rho}_r U_{\Omega_r}(f) - \frac{1}{N!}\sum_{\pi\in\Pi}\sum_{r=1}^{R}\omega_r^\pi U_{\Omega_r^\pi}(f)\right|$$

$$= \sup_{f\in\mathcal{F}}\left|\sum_{r=1}^{R}\widehat{\rho}_r U_{\Omega_r}(f) - \sum_{r=1}^{R}\frac{1}{N!}\sum_{\pi\in\Pi}\omega_r^\pi U_{\Omega_r^\pi}(f)\right|.$$

Using triangle inequality, we have:

$$\sup_{f\in\mathcal{F}}\left|U_\Omega(f) - \bar{U}_\Omega(f)\right|$$

$$\leqslant \sum_{r=1}^{R}\sup_{f\in\mathcal{F}}\left|\widehat{\rho}_r U_{\Omega_r}(f) - \frac{1}{N!}\sum_{\pi\in\Pi}\omega_r^\pi U_{\Omega_r^\pi}(f)\right|$$

$$= \sum_{r=1}^{R}\sup_{f\in\mathcal{F}}\left|\widehat{\rho}_r U_{\Omega_r}(f) - \frac{1}{N!}\sum_{\pi\in\Pi}\widehat{\rho}_r U_{\Omega_r^\pi}(f) + \frac{1}{N!}\sum_{\pi\in\Pi}\left[\omega_r^\pi - \widehat{\rho}_r\right]U_{\Omega_r^\pi}(f)\right|$$

$$\leqslant \sum_{r=1}^{R}\widehat{\rho}_r \sup_{f\in\mathcal{F}}\left|U_{\Omega_r}(f) - \frac{1}{N!}\sum_{\pi\in\Pi}U_{\Omega_r^\pi}(f)\right| + \sum_{r=1}^{R}\sup_{f\in\mathcal{F}}\left|\frac{1}{N!}\sum_{\pi\in\Pi}\left[\omega_r^\pi - \widehat{\rho}_r\right]U_{\Omega_r^\pi}(f)\right|$$

$$= \sum_{r=1}^{R}\widehat{\rho}_r \sup_{f\in\mathcal{F}}\left|U_{\Omega_r}(f) - \widehat{\mathbb{E}}_\Pi\left[U_{\Omega_r^\pi}(f)\right]\right| + \sum_{r=1}^{R}\sup_{f\in\mathcal{F}}\left|\frac{1}{N!}\sum_{\pi\in\Pi}\left[\omega_r^\pi - \widehat{\rho}_r\right]U_{\Omega_r^\pi}(f)\right|.$$

Now, for each permutation $\pi \in \Pi$, we define $\widehat{\rho}_r^\pi := \frac{n_r^\pi}{n}$ where $n_r^\pi := |S_r \cap S_\pi^{(n)}|$, i.e., the number of data points of class $r$ from $S_\pi^{(n)}$. When we select a permutation $\pi \in \Pi$ uniformly, we can think of $n_r^\pi$ as a hypergeometric random variable ($n$ draws from a population of size $N$ with $N_r$ special items). Therefore, we can write:

$$\widehat{\rho}_r := \frac{N_r}{N} = \frac{1}{n}\mathbb{E}[n_r^\pi] = \frac{1}{n}\widehat{\mathbb{E}}_\Pi[n_r^\pi] = \frac{1}{N!}\sum_{\pi\in\Pi}\frac{n_r^\pi}{n}.$$

Hence, we have:

$$\sum_{r=1}^{R} \sup_{f \in \mathcal{F}} \left| \frac{1}{N!} \sum_{\pi \in \Pi} [\omega_r^\pi - \widehat{\rho}_r] U_{\Omega_r^\pi}(f) \right| \leqslant \mathcal{B} \sum_{r=1}^{R} \left| \frac{1}{N!} \sum_{\pi \in \Pi} [\omega_r^\pi - \widehat{\rho}_r] \right| = \mathcal{B} \sum_{r=1}^{R} \left| \widehat{\rho}_r - \frac{1}{N!} \sum_{\pi \in \Pi} \omega_r^\pi \right|$$

$$= \mathcal{B} \sum_{r=1}^{R} \left| \frac{1}{N!} \sum_{\pi \in \Pi} \widehat{\rho}_r^\pi - \frac{1}{N!} \sum_{\pi \in \Pi} \omega_r^\pi \right|$$

$$\leqslant \mathcal{B} \sum_{r=1}^{R} \frac{1}{N!} \sum_{\pi \in \Pi} |\widehat{\rho}_r^\pi - \omega_r^\pi|.$$

For each $r \in [R]$ and $\pi \in \Pi$, we have:

$$|\widehat{\rho}_r^\pi - \omega_r^\pi| = \left| \frac{n_r^\pi}{n} - \frac{\lfloor n_r^\pi/2 \rfloor}{\sum_{q=1}^{R} \lfloor n_q^\pi/2 \rfloor} \right|$$

$$\leqslant \left| \frac{n_r^\pi}{n} - \frac{2\lfloor n_r^\pi/2 \rfloor}{n} \right| + \left| \frac{2\lfloor n_r^\pi/2 \rfloor}{n} - \frac{\lfloor n_r^\pi/2 \rfloor}{\sum_{q=1}^{R} \lfloor n_q^\pi/2 \rfloor} \right|$$

$$= \frac{1}{n}(n_r^\pi - 2\lfloor n_r^\pi/2 \rfloor) + \lfloor n_r^\pi/2 \rfloor \left( \frac{1}{\sum_{q=1}^{R} \lfloor n_q/2 \rfloor} - \frac{2}{n} \right).$$

Now, use the fact that $n_r^\pi - 2\lfloor n_r^\pi/2 \rfloor \leqslant 1$ and $\sum_{q=1}^{R} \lfloor n_r^\pi/2 \rfloor \geqslant \frac{n-R}{2}$ (cf. Proposition D.1), we have:

$$|\widehat{\rho}_r^\pi - \omega_r^\pi| \leqslant \frac{1}{n} + 2\lfloor n_r^\pi/2 \rfloor \left( \frac{1}{n-R} - \frac{1}{n} \right) \leqslant \frac{1}{n} + \frac{2R\lfloor n_r^\pi/2 \rfloor}{n(n-R)}.$$

Therefore:

$$\sum_{r=1}^{R} |\widehat{\rho}_r^\pi - \omega_r^\pi| \leqslant \frac{R}{n} + \frac{2R}{n(n-R)} \underbrace{\sum_{r=1}^{R} \lfloor n_r^\pi/2 \rfloor}_{\leqslant n/2} \leqslant \frac{R}{n} + \frac{R}{n-R} \leqslant \frac{2R}{n-R}.$$

Finally, we have:

$$\sup_{f \in \mathcal{F}} \left| U_\Omega(f) - \bar{U}_\Omega(f) \right| \leqslant \mathcal{B} \frac{1}{N!} \sum_{\pi \in \Pi} \underbrace{\sum_{r=1}^{R} |\widehat{\rho}_r^\pi - \omega_r^\pi|}_{\leqslant 2R/(n-R)} + \sum_{r=1}^{R} \widehat{\rho}_r \sup_{f \in \mathcal{F}} \left| U_{\Omega_r}(f) - \widehat{\mathbb{E}}_\Pi \left[ U_{\Omega_r^\pi}(f) \right] \right|$$

$$\leqslant \frac{2\mathcal{B}R}{n-R} + \sum_{r=1}^{R} \widehat{\rho}_r \sup_{f \in \mathcal{F}} \left| U_{\Omega_r}(f) - \widehat{\mathbb{E}}_\Pi \left[ U_{\Omega_r^\pi}(f) \right] \right|,$$

as desired. $\qquad\square$

Before continuing on to the proof of Proposition D.6, we introduce the formal definition of class collision below.

**Definition D.3** (Class-collision)**.** Let $T_k$ be the map that returns the negative samples of a given tuple (i.e., the last $k$ elements in the tuple). Specifically:

$$T_k : \mathcal{X}^{k+2} \to \mathcal{X}^k, \tag{33}$$

$$\left( X, X^+, \{X_i^-\}_{i=1}^{k} \right) \mapsto (X_1^-, \ldots, X_k^-). \tag{34}$$

Let the map $\sharp_{\text{col}}^r : \mathcal{X}^{k+2} \to \{0, 1, \ldots, k\}$ be defined as follows:

$$\forall r \in [R] : \quad \sharp_{\text{col}}^r(t) := |S_r \cap T_k(t)|. \tag{35}$$

Then, for each $t = \left( X, X^+, \{X_i^-\}_{i=1}^{k} \right) \in \mathcal{X}^{k+2}$ such that $X, X^+ \sim \mathcal{D}_r^{\otimes 2}$, $\sharp_{\text{col}}^r(t)$ denotes the number of negative samples in $t$ that belong to class $r$, i.e., number of **class-collisions**.

**Lemma D.4.** *For a fixed $r \in [R]$, let $P_r, Q_r$ be probability mass functions over $\Omega_r$ such that:*

$$\forall t \in \Omega_r : \quad P_r(t) := \frac{1}{|\Omega_r|}, \quad Q_r(t) := \frac{1}{N!} \sum_{\pi \in \Pi} \mathbb{1}_{\Omega_r^\pi}(t) |\Omega_r^\pi|^{-1}. \tag{36}$$

*Then, for any $\overline{\kappa} \in \{0, 1, \ldots, k\}$, we have:*

$$\mathbb{E}_{t \sim P_r} \left[ \ell_{\phi,f}(t) \,\middle|\, \sharp_{\mathrm{col}}^r(t) = \overline{\kappa} \right] = \mathbb{E}_{t \sim Q_r} \left[ \ell_{\phi,f}(t) \,\middle|\, \sharp_{\mathrm{col}}^r(t) = \overline{\kappa} \right]. \tag{37}$$

**Proof.** For each $0 \leqslant \overline{\kappa} \leqslant k$, we define the set of tuples $\Omega_{r,\overline{\kappa}} \subset \Omega_r$ as follows:

$$\Omega_{r,\overline{\kappa}} := \left\{ t \in \Omega_r : \sharp_{\mathrm{col}}^r(t) = \overline{\kappa} \right\}, \tag{38}$$

i.e, the set of tuples with exactly $\overline{\kappa}$ class-collisions. Then, we have:

$$\begin{aligned}
\mathbb{E}_{t \sim P_r} \left[ \ell_{\phi,f}(t) \,\middle|\, \sharp_{\mathrm{col}}^r(t) = \overline{\kappa} \right] &= \frac{1}{P_r(\Omega_{r,\overline{\kappa}})} \sum_{t \in \Omega_{r,\overline{\kappa}}} P_r(t) \ell_{\phi,f}(t) \\
&= \frac{1}{|\Omega_{r,\overline{\kappa}}|/|\Omega_r|} \sum_{t \in \Omega_{r,\overline{\kappa}}} \frac{\ell_{\phi,f}(t)}{|\Omega_r|} \\
&= \frac{1}{|\Omega_{r,\overline{\kappa}}|} \sum_{t \in \Omega_{r,\overline{\kappa}}} \ell_{\phi,f}(t).
\end{aligned}$$

Now, we claim that for all $t_1, t_2 \in \Omega_{r,\overline{\kappa}}$ for $0 \leqslant \overline{\kappa} \leqslant k$, we have $Q_r(t_1) = Q_r(t_2)$. This can be proven using a simple symmetry argument. Suppose that we have two tuples:

$$\begin{aligned}
t_1 &= \left( X_{u_1}, X_{u_2}, \{X_{u_\ell}\}_{\ell=3}^{\overline{\kappa}+2}, \{X_{u_j}\}_{j=\overline{\kappa}+3}^{k+2} \right), \\
t_2 &= \left( X_{w_1}, X_{w_2}, \{X_{w_\ell}\}_{\ell=3}^{\overline{\kappa}+2}, \{X_{w_j}\}_{j=\overline{\kappa}+3}^{k+2} \right).
\end{aligned}$$

Where $\{u_j\}_{j=1}^{k+2}, \{w_j\}_{j=1}^{k+2}$ are sets of indices in $[N]$ such that $\sharp_{\mathrm{col}}^r(t_1) = \sharp_{\mathrm{col}}^r(t_2) = \overline{\kappa}$ and:

- $X_{u_1}, X_{u_2}, X_{w_1}, X_{w_2} \in S_r$.

- $X_{u_\ell}, X_{w_\ell} \in S_r$ for all $3 \leqslant \ell \leqslant \overline{\kappa} + 2$.

- $X_{u_j}, X_{w_j} \notin S_r$ for all $\overline{\kappa} + 3 \leqslant j \leqslant k + 2$.

Then, for any $\pi \in \Pi$, $t_1 \in \Omega_r^\pi$ if the following are satisfied:

$$\begin{aligned}
\pi(u_1) \leqslant n \quad &\text{and} \quad \pi(u_2) \leqslant n, \\
\pi(u_{j+2}) \geqslant n+1 \quad &\text{for all } 1 \leqslant j \leqslant k.
\end{aligned}$$

For all $\pi \in \Pi$, we define $\pi_{u,w}$ as the permutation where the indices of $\{u_j\}_{j=1}^{k+2}$ and $\{w_j\}_{j=1}^{k+2}$ swap places. Specifically:

$$\begin{cases}
\pi_{u,w}(u_j) &= \pi(w_j), \quad \forall 1 \leqslant j \leqslant k+2, \\
\pi_{u,w}(w_j) &= \pi(u_j), \quad \forall 1 \leqslant j \leqslant k+2, \\
\pi_{u,w}(i) &= \pi(i), \quad \forall i \in [N] \setminus \left[ \{u_j\}_{j=1}^{k+2} \cup \{w_j\}_{j=1}^{k+2} \right].
\end{cases} \quad {}^3$$

---

[3]To be completely clear: the swap is role-preserving. The anchor of $t_1$ is swapped with the anchor of $t_2$; the positive of $t_1$ is swapped with the positive of $t_2$; and each negative sample of $t_1$ is swapped with the corresponding negative sample of $t_2$. In particular, if a negative in $t_1$ collides with class $r$, then it is swapped with a negative in $t_2$ that also collides with class $r$. Thus, there is no situation where a collided sample is swapped with a true negative. This ensures that the collision pattern is preserved exactly under the mapping $\pi \mapsto \pi_{u,w}$.

Then, for any $\pi \in \Pi$ such that $t_1 \in \Omega_r^\pi$, we have $t_2 \in \Omega_r^{\pi_{u,w}}$[4]. Furthermore, we claim that since $|S_r \cap t_1| = |S_r \cap t_2| = \bar{\kappa} + 2$, we always have $|S_\pi^{(n)} \cap S_r| = |S_{\pi_{u,w}}^{(n)} \cap S_r|$. To prove this, we define:

$$\mathcal{I}_r := \left\{ j \in [N] : X_j \in S_r \right\}. \tag{39}$$

Then, we have:

$$|S_\pi^{(n)} \cap S_r| = \sum_{i \in \mathcal{I}_r} \mathbb{1}_{\{\pi(i) \leqslant n\}}, \quad |S_{\pi_{u,w}}^{(n)} \cap S_r| = \sum_{i \in \mathcal{I}_r} \mathbb{1}_{\{\pi_{u,w}(i) \leqslant n\}}.$$

Let $U = \{u_j\}_{j=1}^{k+2}$ and $W = \{w_j\}_{j=1}^{k+2}$. Furthermore, let $\bar{U} = U \setminus (U \cap W)$ and $\bar{W} = W \setminus (U \cap W)$. Noting that $\{u_j\}_{j=3}^{\bar{\kappa}+2}, \{w_j\}_{j=3}^{\bar{\kappa}+2} \subseteq \mathcal{I}_r$, we have:

$$
\begin{aligned}
&|S_{\pi_{u,w}}^{(n)} \cap S_r| \\
&= \sum_{i \in \mathcal{I}_r \setminus (U \cup W)} \mathbb{1}_{\{\pi_{u,w}(i) \leqslant n\}} + \sum_{i \in \mathcal{I}_r \cap (U \cap W)} \mathbb{1}_{\{\pi_{u,w}(i) \leqslant n\}} + \sum_{i \in \mathcal{I}_r \cap \bar{U}} \mathbb{1}_{\{\pi_{u,w}(i) \leqslant n\}} + \sum_{i \in \mathcal{I}_r \cap \bar{W}} \mathbb{1}_{\{\pi_{u,w}(i) \leqslant n\}} \\
&= \sum_{i \in \mathcal{I}_r \setminus (U \cup W)} \mathbb{1}_{\{\pi(i) \leqslant n\}} + \sum_{i \in \mathcal{I}_r \cap (U \cap W)} \mathbb{1}_{\{\pi(i) \leqslant n\}} + \sum_{i \in \mathcal{I}_r \cap \bar{U}} \mathbb{1}_{\{\pi_{u,w}(i) \leqslant n\}} + \sum_{i \in \mathcal{I}_r \cap \bar{W}} \mathbb{1}_{\{\pi_{u,w}(i) \leqslant n\}} \\
&= \sum_{i \in \mathcal{I}_r \setminus (U \Delta W)} \mathbb{1}_{\{\pi(i) \leqslant n\}} + \sum_{\substack{j : u_j \in \mathcal{I}_r \\ u_j \notin W}} \mathbb{1}_{\{\pi_{u,w}(u_j) \leqslant n\}} + \sum_{\substack{j : w_j \in \mathcal{I}_r \\ w_j \notin U}} \mathbb{1}_{\{\pi_{u,w}(w_j) \leqslant n\}} \\
&= \sum_{i \in \mathcal{I}_r \setminus (U \Delta W)} \mathbb{1}_{\{\pi(i) \leqslant n\}} + \sum_{\substack{3 \leqslant j \leqslant \bar{\kappa}+2 \\ u_j \notin W}} \mathbb{1}_{\{\pi(w_j) \leqslant n\}} + \sum_{\substack{3 \leqslant j \leqslant \bar{\kappa}+2 \\ w_j \notin U}} \mathbb{1}_{\{\pi(u_j) \leqslant n\}} \\
&= \sum_{i \in \mathcal{I}_r \setminus (U \Delta W)} \mathbb{1}_{\{\pi(i) \leqslant n\}} + \sum_{i \in W \setminus (W \cap U)} \mathbb{1}_{\{\pi(i) \leqslant n\}} + \sum_{i \in U \setminus (W \cap U)} \mathbb{1}_{\{\pi(i) \leqslant n\}} \\
&= \sum_{i \in \mathcal{I}_r \setminus (U \Delta W)} \mathbb{1}_{\{\pi(i) \leqslant n\}} + \sum_{i \in (U \Delta W)} \mathbb{1}_{\{\pi(i) \leqslant n\}} \\
&= \sum_{i \in \mathcal{I}_r} \mathbb{1}_{\{\pi(i) \leqslant n\}} = |S_\pi^{(n)} \cap S_r|.
\end{aligned}
$$

In the above, we used $\Delta$ to denote symmetric difference. Since $|S_\pi^{(n)} \cap S_r| = |S_{\pi_{u,w}}^{(n)} \cap S_r|$ for all $\pi \in \Pi$ satisfying $t_1 \in \Omega_r^\pi$, we have:

$$\forall \pi \in \Pi \text{ s.t. } t_1 \in \Omega_r^\pi : |\Omega_r^\pi| = |\Omega_r^{\pi_{u,w}}|.$$

As a result:

$$Q_r(t_2) = \sum_{\substack{\bar{\pi} \in \Pi \\ t_2 \in \Omega_r^{\bar{\pi}}}} |\Omega_r^{\bar{\pi}}|^{-1} = \sum_{\substack{\pi \in \Pi \\ t_1 \in \Omega_r^\pi}} |\Omega_r^{\pi_{u,w}}|^{-1} = \sum_{\substack{\pi \in \Pi \\ t_1 \in \Omega_r^\pi}} |\Omega_r^\pi|^{-1} = Q_r(t_1).$$

Therefore, we have:

$$
\begin{aligned}
\mathbb{E}_{t \sim Q_r}\left[ \ell_{\phi,f}(t) \Big| \sharp_{\mathrm{col}}^r(t) = \bar{\kappa} \right] &= \frac{1}{Q_r(\Omega_{r,\bar{\kappa}})} \sum_{t \in \Omega_{r,\bar{\kappa}}} Q_r(t) \ell_{\phi,f}(t) = \frac{1}{|\Omega_{r,\bar{\kappa}}|} \sum_{t \in \Omega_{r,\bar{\kappa}}} \ell_{\phi,f}(t) \\
&= \mathbb{E}_{t \sim P_r}\left[ \ell_{\phi,f}(t) \Big| \sharp_{\mathrm{col}}^r(t) = \bar{\kappa} \right].
\end{aligned}
$$

$\square$

---

[4]This means that we can generate all $\bar{\pi} \in \Pi$ where $t_2 \in \Omega_r^{\bar{\pi}}$ by swapping indices of the permutations $\pi \in \Pi$ that satisfy $t_1 \in \Omega_r^\pi$. However, note that this is true if and only if $\sharp_{\mathrm{col}}^r(t_1) = \sharp_{\mathrm{col}}^r(t_2)$.

**Proposition D.5.** *Let $\mathcal{F}$ denote a class of representation functions and $U_\Omega(f)$, $\bar{U}_\Omega(f)$ be U-Statistics formulations in Eqn. (13) and Eqn. (24), respectively. Suppose that $|\ell_{\phi,f}| \leqslant \mathcal{B}$ for all $f \in \mathcal{F}$ and $N_r \geqslant 2(k+1)$ for all $r \in [R]$. Then, we have:*

$$\sup_{f \in \mathcal{F}} \left| U_\Omega(f) - \bar{U}_\Omega(f) \right| \leqslant \frac{2\mathcal{B}R}{n-R} + \frac{6\mathcal{B}Rk}{N} + \frac{2\mathcal{B}k}{N-k-1}. \tag{40}$$

**Proof**. From Lemma D.2, our main task now is to bound the following weighted sum of uniform differences:

$$\sum_{r=1}^{R} \widehat{\rho}_r \sup_{f \in \mathcal{F}} \left| U_{\Omega_r}(f) - \widehat{\mathbb{E}}_\Pi \left[ U_{\Omega_r^\pi}(f) \right] \right|.$$

We can write $U_{\Omega_r}(f) = \mathbb{E}_{t \sim P_r}[\ell_{\phi,f}(t)]$ and $\widehat{\mathbb{E}}_\Pi[U_{\Omega_r^\pi}(f)] = \mathbb{E}_{t \sim Q_r}[\ell_{\phi,f}(t)]$ where $P_r, Q_r$ are probability mass functions defined in Eqn. (36). For each $0 \leqslant \overline{\kappa} \leqslant k$, by Lemma D.4, we can write:

$$\mathbb{E}_{t \sim P_r}\left[\ell_{\phi,f}(t)\middle|\sharp_{\text{col}}^r(t) = \overline{\kappa}\right] = \mathbb{E}_{t \sim Q_r}\left[\ell_{\phi,f}(t)\middle|\sharp_{\text{col}}^r(t) = \overline{\kappa}\right] := E_{r,\overline{\kappa}}(f). \tag{41}$$

Then, we have:

$$\sum_{r=1}^{R} \widehat{\rho}_r \sup_{f \in \mathcal{F}} \left| U_{\Omega_r}(f) - \widehat{\mathbb{E}}_\Pi \left[ U_{\Omega_r^\pi}(f) \right] \right| = \sum_{r=1}^{R} \widehat{\rho}_r \sup_{f \in \mathcal{F}} \left| \mathbb{E}_{t \sim P_r}[\ell_{\phi,f}(t)] - \mathbb{E}_{t \sim Q_r}[\ell_{\phi,f}(t)] \right|$$

$$= \sum_{r=1}^{R} \widehat{\rho}_r \sup_{f \in \mathcal{F}} \left| \sum_{\overline{\kappa}=0}^{k} \mathbb{P}_{t \sim P_r}(\sharp_{\text{col}}^r(t) = \overline{\kappa})E_{r,\overline{\kappa}}(f) - \sum_{\overline{\kappa}=0}^{k} \mathbb{P}_{t \sim Q_r}(\sharp_{\text{col}}^r(t) = \overline{\kappa})E_{r,\overline{\kappa}}(f) \right|$$

$$= \sum_{r=1}^{R} \widehat{\rho}_r \sup_{f \in \mathcal{F}} \left| \sum_{\overline{\kappa}=0}^{k} E_{r,\overline{\kappa}}(f)\left[ \mathbb{P}_{t \sim P_r}(\sharp_{\text{col}}^r(t) = \overline{\kappa}) - \mathbb{P}_{t \sim Q_r}(\sharp_{\text{col}}^r(t) = \overline{\kappa}) \right] \right|$$

$$\leqslant \sum_{r=1}^{R} \widehat{\rho}_r \sum_{\overline{\kappa}=0}^{k} \sup_{f \in \mathcal{F}} \left| E_{r,\overline{\kappa}}(f)\left[ \mathbb{P}_{t \sim P_r}(\sharp_{\text{col}}^r(t) = \overline{\kappa}) - \mathbb{P}_{t \sim Q_r}(\sharp_{\text{col}}^r(t) = \overline{\kappa}) \right] \right|$$

$$\leqslant \mathcal{B} \sum_{r=1}^{R} \widehat{\rho}_r \sum_{\overline{\kappa}=0}^{k} \left| \mathbb{P}_{t \sim P_r}(\sharp_{\text{col}}^r(t) = \overline{\kappa}) - \mathbb{P}_{t \sim Q_r}(\sharp_{\text{col}}^r(t) = \overline{\kappa}) \right|.$$

Now, the problem reduces to comparing class-collision probabilities under two probability mass functions $P_r, Q_r$ for each $r \in [R]$. Firstly, we claim that for each $r \in [R]$:

- For $t \sim P_r$, $\sharp_{\text{col}}^r(t) \sim \text{Hypergeometric}(N-2, N_r-2, k)$.

- For $t \sim Q_r$, $\sharp_{\text{col}}^r(t) \sim \text{Hypergeometric}(N, N_r, k)$.

Both of these claims will be elaborated further in Remark D.7. As a result, for each $r \in [R]$ and $0 \leqslant \overline{\kappa} \leqslant k$, we have:

$$\left| \mathbb{P}_{t \sim P_r}(\sharp_{\text{col}}^r(t) = \overline{\kappa}) - \mathbb{P}_{t \sim Q_r}(\sharp_{\text{col}}^r(t) = \overline{\kappa}) \right|$$

$$= \left| \frac{\binom{N_r-2}{\overline{\kappa}}\binom{N-N_r}{k-\overline{\kappa}}}{\binom{N-2}{k}} - \frac{\binom{N_r}{\overline{\kappa}}\binom{N-N_r}{k-\overline{\kappa}}}{\binom{N}{k}} \right|$$

$$= \mathbb{P}_{t \sim P_r}(\sharp_{\text{col}}^r(t) = \overline{\kappa}) \left| 1 - \frac{\binom{N_r}{\overline{\kappa}}}{\binom{N_r-2}{\overline{\kappa}}}\left[\frac{\binom{N}{k}}{\binom{N-2}{k}}\right]^{-1} \right|$$

$$= \mathbb{P}_{t \sim P_r}(\sharp_{\text{col}}^r(t) = \overline{\kappa}) \left| 1 - \frac{N_r(N_r-1)}{(N_r-\overline{\kappa})(N_r-\overline{\kappa}-1)}\left[\frac{N(N-1)}{(N-k)(N-k-1)}\right]^{-1} \right|.$$

Now, for notational brevity, we denote $\alpha, \beta$ as the following terms:

$$\alpha = \frac{N_r(N_r - 1)}{(N_r - \overline{\kappa})(N_r - \overline{\kappa} - 1)}, \quad \text{and} \quad \beta = \frac{N(N-1)}{(N-k)(N-k-1)}. \tag{42}$$

Note that:

$$\frac{N_r}{N_r - \overline{\kappa}} \leqslant \frac{N_r - 1}{N_r - \overline{\kappa} - 1} \leqslant \frac{N_r - 1}{N_r - k - 1}.$$

Therefore:

$$
\begin{aligned}
\alpha\beta^{-1} &= \beta^{-1}\left[\frac{N_r}{N_r - \overline{\kappa}} \cdot \frac{N_r - 1}{N_r - \overline{\kappa} - 1}\right] \\
&\leqslant \beta^{-1}\left[\frac{N_r - 1}{N_r - k - 1}\right]^2 = \beta^{-1}\left[1 + \frac{k}{N_r - k - 1}\right]^2 \\
&\overset{(*)}{\leqslant} \beta^{-1}\left[1 + \frac{3k}{N_r - k - 1}\right] \qquad ((1+\delta)^2 \leqslant 1 + 3\delta, \quad \forall \delta \in [0,1]) \\
&\overset{(a)}{\leqslant} 1 + \frac{3k}{N_r - k - 1} \qquad (\beta^{-1} \leqslant 1).
\end{aligned}
$$

Note that in step $(*)$, we assumed that $N_r \geqslant 2(k+1)$. Similarly, we have:

$$
\begin{aligned}
\alpha\beta^{-1} &\geqslant \beta^{-1} \geqslant \left[\frac{N-1}{N-k-1}\right]^{-2} = \left[\frac{1}{1 + \frac{k}{N-k-1}}\right]^2 \\
&\geqslant \left[1 - \frac{k}{N-k-1}\right]^2 \\
&\overset{(b)}{\geqslant} 1 - \frac{2k}{N-k-1} \qquad ((1-\delta)^2 \geqslant 1 - 2\delta, \quad \forall \delta \in [0,1]).
\end{aligned}
$$

By plugging inequalities $(a), (b)$ back to the collision probabilities difference bound, we have:

$$
\begin{aligned}
&\left|\mathbb{P}_{t \sim P_r}(\sharp_{\text{col}}^r(t) = \overline{\kappa}) - \mathbb{P}_{t \sim Q_r}(\sharp_{\text{col}}^r(t) = \overline{\kappa})\right| \leqslant \mathbb{P}_{t \sim P_r}(\sharp_{\text{col}}^r(t) = \overline{\kappa})\left|1 - \alpha\beta^{-1}\right| \\
&\leqslant \mathbb{P}_{t \sim P_r}(\sharp_{\text{col}}^r(t) = \overline{\kappa})\left(\frac{3k}{N_r - k - 1} + \frac{2k}{N - k - 1}\right).
\end{aligned}
$$

Therefore, we have:

$$
\begin{aligned}
&\sum_{r=1}^{R} \widehat{\rho}_r \sum_{\overline{\kappa}=0}^{k} \left|\mathbb{P}_{t \sim P_r}(\sharp_{\text{col}}^r(t) = \overline{\kappa}) - \mathbb{P}_{t \sim Q_r}(\sharp_{\text{col}}^r(t) = \overline{\kappa})\right| \\
&\leqslant \sum_{r=1}^{R} \widehat{\rho}_r \sum_{\overline{\kappa}=0}^{k} \mathbb{P}_{t \sim P_r}(\sharp_{\text{col}}^r(t) = \overline{\kappa})\left(\frac{3k}{N_r - k - 1} + \frac{2k}{N - k - 1}\right) \\
&= \sum_{r=1}^{R} \widehat{\rho}_r \left(\frac{3k}{N_r - k - 1} + \frac{2k}{N - k - 1}\right)\underbrace{\left[\sum_{\overline{\kappa}=0}^{k} \mathbb{P}_{t \sim P_r}(\sharp_{\text{col}}^r(t) = \overline{\kappa})\right]}_{=1} \\
&\leqslant \sum_{r=1}^{R} \widehat{\rho}_r \left(\frac{6k}{N_r} + \frac{2k}{N - k - 1}\right) \qquad \left(k + 1 \leqslant \frac{N_r}{2}\right) \\
&= \frac{6Rk}{N} + \frac{2k}{N - k - 1}.
\end{aligned}
$$

Finally, we have:

$$\sup_{f \in \mathcal{F}} \left| U_\Omega(f) - \bar{U}_\Omega(f) \right| \leqslant \frac{2\mathcal{B}R}{n-R} + \sum_{r=1}^{R} \widehat{\rho}_r \sup_{f \in \mathcal{F}} \left| U_{\Omega_r}(f) - \widehat{\mathbb{E}}_\Pi \left[ U_{\Omega_r^\pi}(f) \right] \right| \qquad \text{(Lemma D.2)}$$

$$\leqslant \frac{2\mathcal{B}R}{n-R} + \frac{6\mathcal{B}Rk}{N} + \frac{2\mathcal{B}k}{N-k-1},$$

as desired. $\qquad\qquad\qquad\qquad\qquad\qquad\qquad\qquad\qquad\qquad\qquad\qquad\qquad\qquad\qquad\qquad\qquad\qquad\qquad\qquad\qquad$ $\square$

**Proposition D.6.** *Let $\mathcal{F}$ denote a class of representation functions. Suppose that $|\ell_{\phi,f}| \leqslant \mathcal{B}$ for all $f \in \mathcal{F}$. Then:*

$$\| U_\Omega - \bar{U}_\Omega \|_\mathcal{F} \leqslant \mathcal{O}\left( \mathcal{B} \left[ \frac{R}{\lfloor \frac{N}{k} \rfloor - R} + \frac{Rk}{N-k} \right] \right), \tag{43}$$

$$\| U_\Lambda - \bar{U}_\Lambda \|_\mathcal{F} \leqslant \mathcal{O}\left( \mathcal{B} \left[ \frac{R}{\lfloor \frac{N}{k} \rfloor - R} + \frac{Rk}{N-k} \right] \right). \tag{44}$$

**Proof.** Now, the goal is to remove the assumption that $N_r \geqslant 2(k+1)$ for all $r \in [R]$. We first use the following abbreviations for each $r \in [R]$ and $0 \leqslant \overline{\kappa} \leqslant k$:

$$p_{r,\overline{\kappa}} = \mathbb{P}_{t \sim P_r}(\sharp_{\mathrm{col}}^r(t) = \overline{\kappa}), \qquad q_{r,\overline{\kappa}} = \mathbb{P}_{t \sim Q_r}(\sharp_{\mathrm{col}}^r(t) = \overline{\kappa}).$$

Then, we have:

$$\sum_{r=1}^{R} \widehat{\rho}_r \sum_{\overline{\kappa}=0}^{k} |p_{r,\overline{\kappa}} - q_{r,\overline{\kappa}}| = \sum_{r \in [R]: N_r \geqslant 3} \widehat{\rho}_r \sum_{\overline{\kappa}=0}^{k} |p_{r,\overline{\kappa}} - q_{r,\overline{\kappa}}| + \sum_{r \in [R]: N_r \leqslant 2} \widehat{\rho}_r \sum_{\overline{\kappa}=0}^{k} |p_{r,\overline{\kappa}} - q_{r,\overline{\kappa}}|$$

$$\leqslant \sum_{r \in [R]: N_r \geqslant 3} \widehat{\rho}_r \sum_{\overline{\kappa}=0}^{k} |p_{r,\overline{\kappa}} - q_{r,\overline{\kappa}}| + \frac{2}{N} \sum_{r \in [R]: N_r \leqslant 2} \sum_{\overline{\kappa}=0}^{k} |p_{r,\overline{\kappa}} - q_{r,\overline{\kappa}}|$$

$$= \sum_{r \in [R]: N_r \geqslant 3} \widehat{\rho}_r \sum_{\overline{\kappa}=0}^{k} |p_{r,\overline{\kappa}} - q_{r,\overline{\kappa}}| + \frac{2}{N} \sum_{r \in [R]: N_r \leqslant 2} |p_{r,0} - q_{r,0}|$$

$$\overset{(*)}{\leqslant} \sum_{r \in [R]: N_r \geqslant 3} \widehat{\rho}_r \sum_{\overline{\kappa}=0}^{k} |p_{r,\overline{\kappa}} - q_{r,\overline{\kappa}}| + \frac{2R}{N}.$$

Where the inequality in $(*)$ is due to the fact that $|p_{r,0} - q_{r,0}| \leqslant 1$ for all minority classes $r \in [R]$ such that $N_r \leqslant 2$. We can decompose the remaining sum as follows:

$$\sum_{r \in [R]: N_r \geqslant 3} \widehat{\rho}_r \sum_{\overline{\kappa}=0}^{k} |p_{r,\overline{\kappa}} - q_{r,\overline{\kappa}}| = \underbrace{\sum_{r \in [R]: N_r \geqslant 3} \widehat{\rho}_r \sum_{\substack{0 \leqslant \overline{\kappa} \leqslant k \\ N_r \geqslant 2(\overline{\kappa}+1)}} |p_{r,\overline{\kappa}} - q_{r,\overline{\kappa}}|}_{(A)} + \underbrace{\sum_{r \in [R]: N_r \geqslant 3} \widehat{\rho}_r \sum_{\substack{0 \leqslant \overline{\kappa} \leqslant k \\ N_r \leqslant 2\overline{\kappa}+1}} |p_{r,\overline{\kappa}} - q_{r,\overline{\kappa}}|}_{(B)}.$$

**Bounding (A):** By a more refined approach than the proof of Proposition D.5, for each $r \in [R]$ and $0 \leqslant \overline{\kappa} \leqslant k$ such that $N_r \geqslant 2(\overline{\kappa}+1)$, we have

$$\alpha\beta^{-1} = \beta^{-1} \left[ \frac{N_r}{N_r - \overline{\kappa}} \cdot \frac{N_r - 1}{N_r - \overline{\kappa} - 1} \right]$$

$$\leqslant \beta^{-1} \left[ \frac{N_r - 1}{N_r - \overline{\kappa} - 1} \right]^2 = \beta^{-1} \left[ 1 + \frac{\overline{\kappa}}{N_r - \overline{\kappa} - 1} \right]^2$$

$$\leqslant 1 + \frac{3\overline{\kappa}}{N_r - \overline{\kappa} - 1}.$$

As a result, for $r \in [R]$ and $0 \leqslant \overline{\kappa} \leqslant k$ such that $N_r \geqslant 2(\overline{\kappa} + 1)$:

$$|p_{r,\overline{\kappa}} - q_{r,\overline{\kappa}}| \leqslant p_{r,\overline{\kappa}} |1 - \alpha\beta^{-1}| \leqslant p_{r,\overline{\kappa}} \left( \frac{3\overline{\kappa}}{N_r - \overline{\kappa} - 1} + \frac{2k}{N - k - 1} \right).$$

Therefore, we have:

$$\sum_{r \in [R]: N_r \geqslant 3} \widehat{\rho}_r \sum_{\substack{0 \leqslant \overline{\kappa} \leqslant k \\ N_r \geqslant 2(\overline{\kappa}+1)}} |p_{r,\overline{\kappa}} - q_{r,\overline{\kappa}}|$$

$$\leqslant \sum_{r \in [R]: N_r \geqslant 3} \widehat{\rho}_r \sum_{\substack{0 \leqslant \overline{\kappa} \leqslant k \\ N_r \geqslant 2(\overline{\kappa}+1)}} p_{r,\overline{\kappa}} \left( \frac{3\overline{\kappa}}{N_r - \overline{\kappa} - 1} + \frac{2k}{N - k - 1} \right)$$

$$\leqslant \sum_{r \in [R]: N_r \geqslant 3} \widehat{\rho}_r \sum_{\substack{0 \leqslant \overline{\kappa} \leqslant k \\ N_r \geqslant 2(\overline{\kappa}+1)}} p_{r,\overline{\kappa}} \left( \frac{6\overline{\kappa}}{N_r} + \frac{2k}{N - k - 1} \right) \qquad \left( \text{Since } \overline{\kappa} + 1 \leqslant \frac{N_r}{2} \right)$$

$$\leqslant \sum_{r \in [R]: N_r \geqslant 3} \widehat{\rho}_r \sum_{\substack{0 \leqslant \overline{\kappa} \leqslant k \\ N_r \geqslant 2(\overline{\kappa}+1)}} p_{r,\overline{\kappa}} \left( \frac{6k}{N_r} + \frac{2k}{N - k - 1} \right)$$

$$= \sum_{r \in [R]: N_r \geqslant 3} \widehat{\rho}_r \left( \frac{6k}{N_r} + \frac{2k}{N - k - 1} \right) \underbrace{\sum_{\substack{0 \leqslant \overline{\kappa} \leqslant k \\ N_r \geqslant 2(\overline{\kappa}+1)}} p_{r,\overline{\kappa}}}_{\leqslant 1}$$

$$\leqslant \frac{2k}{N - k - 1} + \sum_{r \in [R]: N_r \geqslant 3} \widehat{\rho}_r \frac{6k}{N_r}$$

$$\leqslant \frac{2k}{N - k - 1} + \frac{6Rk}{N}.$$

**Bounding (B)**: Using the triangle inequality, we can write

$$\sum_{r \in [R]: N_r \geqslant 3} \widehat{\rho}_r \sum_{\substack{0 \leqslant \overline{\kappa} \leqslant k \\ N_r \leqslant 2\overline{\kappa}+1}} |p_{r,\overline{\kappa}} - q_{r,\overline{\kappa}}|$$

$$\leqslant \sum_{r \in [R]: N_r \geqslant 3} \widehat{\rho}_r \sum_{\substack{0 \leqslant \overline{\kappa} \leqslant k \\ N_r \leqslant 2\overline{\kappa}+1}} p_{r,\overline{\kappa}} + \sum_{r \in [R]: N_r \geqslant 3} \widehat{\rho}_r \sum_{\substack{0 \leqslant \overline{\kappa} \leqslant k \\ N_r \leqslant 2\overline{\kappa}+1}} q_{r,\overline{\kappa}}$$

$$= \sum_{r \in [R]: N_r \geqslant 3} \widehat{\rho}_r \mathbb{P}_{t \sim P_r} \left( 2\sharp_{\mathrm{col}}^r(t) + 1 \geqslant N_r \right) + \sum_{r \in [R]: N_r \geqslant 3} \widehat{\rho}_r \mathbb{P}_{t \sim Q_r} \left( 2\sharp_{\mathrm{col}}^r(t) + 1 \geqslant N_r \right).$$

Recall that for $t \sim Q_r$, $\sharp_{\mathrm{col}}^r(t) \sim \mathrm{Hypergeometric}(N, N_r, k)$. Therefore, $\underset{t \sim Q_r}{\mathbb{E}} [\sharp_{\mathrm{col}}^r(t)] = k\frac{N_r}{N} = k\widehat{\rho}_r$. Using the multiplicative Chernoff bound (Proposition J.7), we have:

$$\mathbb{P}_{t \sim Q_r} \left( \sharp_{\mathrm{col}}^r(t) \geqslant \frac{N_r - 1}{2} \right) = \mathbb{P}_{t \sim Q_r} \left( \sharp_{\mathrm{col}}^r(t) \geqslant k\frac{N_r}{N} \cdot \frac{N}{2k} \cdot \frac{N_r - 1}{N_r} \right)$$

$$\leqslant \mathbb{P}_{t \sim Q_r} \left( \sharp_{\mathrm{col}}^r(t) \geqslant k\frac{N_r}{N} \cdot \frac{N}{3k} \right) \qquad \left( \text{Since } \frac{N_r - 1}{N_r} \geqslant \frac{2}{3}, \forall N_r \geqslant 3 \right)$$

$$= \mathbb{P}_{t \sim Q_r} \left( \sharp_{\mathrm{col}}^r(t) \geqslant \left( 1 + \frac{N - 3k}{3k} \right) \underset{t \sim Q_r}{\mathbb{E}} [\sharp_{\mathrm{col}}^r(t)] \right).$$

By the multiplicative Chernoff bound (Proposition J.7), we have:

$$\mathbb{P}_{t \sim Q_r} \left( \sharp_{\text{col}}^r(t) \geqslant \left( 1 + \frac{N - 3k}{3k} \right) \underset{t \sim Q_r}{\mathbb{E}} \left[ \sharp_{\text{col}}^r(t) \right] \right)$$

$$\leqslant \exp \left( -\frac{1}{3} \left[ \frac{N - 3k}{3k} \right]^2 \underset{t \sim Q_r}{\mathbb{E}} \left[ \sharp_{\text{col}}^r(t) \right] \right) = \exp \left( -\frac{1}{3} \left[ \frac{N - 3k}{3k} \right]^2 k \widehat{\rho}_r \right).$$

Now, if $X \sim \text{Hypergeometric}(N, N_r, k)$ and $Y \sim \text{Hypergeometric}(N - 2, N_r - 2, k)$, we can easily show $X \succcurlyeq_{\text{st}} Y$ [5] by a simple coupling argument (Remark D.8). Therefore, we have:

$$\mathbb{P}_{t \sim P_r} \left( \sharp_{\text{col}}^r(t) \geqslant \frac{N_r - 1}{2} \right) \leqslant \mathbb{P}_{t \sim Q_r} \left( \sharp_{\text{col}}^r(t) \geqslant \frac{N_r - 1}{2} \right) \leqslant \exp \left( -\frac{1}{3} \left[ \frac{N - 3k}{3k} \right]^2 k \widehat{\rho}_r \right).$$

As a result:

$$\sum_{r \in [R] : N_r \geqslant 3} \widehat{\rho}_r \sum_{\substack{0 \leqslant \overline{\kappa} \leqslant k \\ N_r \leqslant 2\overline{\kappa} + 1}} |p_{r, \overline{\kappa}} - q_{r, \overline{\kappa}}|$$

$$\leqslant \sum_{r \in [R] : N_r \geqslant 3} \widehat{\rho}_r \mathbb{P}_{t \sim P_r} \left( 2 \sharp_{\text{col}}^r(t) + 1 \geqslant N_r \right) + \sum_{r \in [R] : N_r \geqslant 3} \widehat{\rho}_r \mathbb{P}_{t \sim Q_r} \left( 2 \sharp_{\text{col}}^r(t) + 1 \geqslant N_r \right)$$

$$\leqslant 2 \sum_{r \in [R] : N_r \geqslant 3} \widehat{\rho}_r \exp \left( -\frac{1}{3} \left[ \frac{N - 3k}{3k} \right]^2 k \widehat{\rho}_r \right)$$

$$\leqslant \frac{6}{e} \sum_{r \in [R] : N_r \geqslant 3} \widehat{\rho}_r \cdot \frac{9k^2}{k \widehat{\rho}_r (N - 3k)^2} \qquad \left( e^{-x} \leqslant \frac{1}{ex} \text{ for all } x \geqslant 0 \right)$$

$$= \frac{6}{e} \sum_{r \in [R] : N_r \geqslant 3} \frac{9k}{(N - 3k)^2}$$

$$\leqslant \frac{54 R k}{e (N - 3k)^2}.$$

Combining every inequalities, we have:

$$\sum_{r=1}^{R} \widehat{\rho}_r \sum_{\overline{\kappa}=0}^{k} |p_{r, \overline{\kappa}} - q_{r, \overline{\kappa}}| \leqslant \frac{2R}{N} + \frac{2k}{N - k - 1} + \frac{6Rk}{N} + \frac{54 R k}{e (N - 3k)^2}.$$

Hence:

$$\sup_{f \in \mathcal{F}} \left| U_\Omega(f) - \bar{U}_\Omega(f) \right| \leqslant \frac{2 \mathcal{B} R}{n - R} + \mathcal{B} \sum_{r=1}^{R} \widehat{\rho}_r \sum_{\overline{\kappa}=0}^{k} |p_{r, \overline{\kappa}} - q_{r, \overline{\kappa}}|$$

$$\leqslant \mathcal{B} \left[ \frac{2R}{n - R} + \frac{2R}{N} + \frac{2k}{N - k - 1} + \frac{6Rk}{N} + \frac{54 R k}{e (N - 3k)^2} \right]$$

$$= \mathcal{O} \left[ \mathcal{B} \left( \frac{R}{n - R} + R k \left[ \frac{1}{N - k} + \frac{1}{(N - k)^2} \right] \right) \right],$$

as desired. Of course, the proof is extensible to $\sup_{f \in \mathcal{F}} \left| U_\Lambda(f) - \bar{U}_\Lambda(f) \right|$. □

---

[5]Where $\succcurlyeq_{\text{st}}$ denotes stochastic dominance.

**Remark D.7.** For each $r \in [R]$, the probability mass functions $P_r, Q_r$ (Eqn. (36)) correspond to two selection procedures.

1. **Procedure** (1) **(Corresponding to $P_r$):** From the population of $N$ data points

    - We first choose two instances from $S_r$ as anchor-positive.
    - Then, we choose $k$ elements from the remaining $N - 2$ data points as negatives.

2. **Procedure** (2) **(Corresponding to $Q_r$):** From the population of $N$ data points

    - We first choose $n$ data points uniformly without replacement, denoted as the set $S^{(n)}$.
    - Then, we choose 2 elements from $S^{(n)} \cap S_r$ as anchor-positive.
    - Then, we choose $k$ elements from the $N - n$ data points, excluding everything in $S^{(n)}$, as negatives.

Obviously, the number of class-collisions in (1) follows the Hypergeometric$(N - 2, N_r - 2, k)$ distribution because after the anchor-positive is chosen, the negatives are selected from a population of $N - 2$ items with $N_r - 2$ "special items" (excluding the chosen anchor-positive).

On the other hand, procedure (2) is analogous to selecting $k$ elements straight from the original population of $N$ items with $N_r$ special items. That is, the "split-first" step does not affect the distribution of class-collisions.

**Remark D.8.** (Stochastic Dominance of Hypergeometric Variables) If we have two hypergeometric variables $X \sim$ Hypergeometric$(N, M, k)$ and $Y \sim$ Hypergeometric$(N - 2, M - 2, k)$ where $M \geqslant 3$. Then, we can show $X \succcurlyeq_{\mathrm{st}} Y$ via a simple coupling:

1. We start with a bag of $N$ balls with $M$ special balls and we mark two random special balls as "redundant". Next, we draw $k + 2$ balls without replacement from the $N$ balls.

2. To sample $X$, we take the first $k$ balls in the previously selected $k + 2$ balls.

3. To sample $Y$, we follow the following procedure:

    - If the first $k$ balls did not contain the "redundant" ones, it is also a valid draw of $Y$.
    - If one "redundant" showed up in the first $k$ draws, replace that with one of the last two balls (note that there must be one non-redundant balls among the last two).
    - If two "redundant showed up in the first $k$ draws, replace both with the last two balls.

Via this coupling, we will always have $X \geqslant Y$. Therefore, $X$ stochastically dominates $Y$.

# E. Concentration of the Auxiliary Estimators

In this section, we will prove that the proposed auxiliary estimators $\bar{U}_\Omega(f)$ and $\bar{U}_\Lambda(f)$ concentrates well to their respective targets. To do so, we introduce some definitions of auxiliary risks corresponding to $L_\Omega(f)$ and $L_\Lambda(f)$.

## E.1. Auxiliary Risks for $L_\Omega(f)$

Firstly, we introduce the following notations:

$$S^{(n)} = \{X_j\}_{j=1}^n, \text{ and } \forall r \in [R] : \begin{cases} n_r & := \left| S_r \cap S^{(n)} \right|, \\ \omega_r & := \lfloor n_r/2 \rfloor \left[ \sum_{q=1}^R \lfloor n_q/2 \rfloor \right]^{-1}, \\ \bar{\omega}_r & := \mathbb{E}[\omega_r]. \end{cases} \tag{45}$$

Furthermore, for every $\pi \in \Pi$, we denote:

$$S_\pi^{(n)} = \left\{ X_{\pi(j)} \right\}_{j=1}^n, \text{ and } \forall r \in [R] : \begin{cases} n_r^\pi & := |S_r \cap S_\pi^{(n)}|, \\ \omega_r^\pi & := \lfloor n_r^\pi/2 \rfloor \left[ \sum_{q=1}^R \lfloor n_q^\pi/2 \rfloor \right]^{-1}, \\ \omega_r^* & := \widehat{\mathbb{E}}_\Pi[\omega_r^\pi]. \end{cases} \tag{46}$$

Then, we define the following auxiliary risks:

$$L_\Omega^*(f) := \sum_{r=1}^R \omega_r^* L_\Omega^r(f), \quad \bar{L}_\Omega(f) := \sum_{r=1}^R \bar{\omega}_r L_\Omega^r(f). \tag{47}$$

$$\widehat{L}_\Omega(f|\pi) := \sum_{r=1}^R \omega_r^\pi L_\Omega^r(f), \quad \widehat{L}_\Omega(f) := \sum_{r=1}^R \omega_r L_\Omega^r(f). \tag{48}$$

From the definitions, we can observe that $L_\Omega^*(f) = \widehat{\mathbb{E}}_\Pi[L_\Omega^*(f|\pi)]$, $\bar{L}_\Omega(f) = \mathbb{E}[\widehat{L}_\Omega(f)] = \mathbb{E}[\widehat{L}_\Omega(f|\pi)]$ for any fixed permutation $\pi \in \Pi$ and $\widehat{L}_\Omega(f) = \widehat{L}_\Omega(f|\mathrm{id})$ where $\mathrm{id} \in \Pi$ is the identity permutation (i.e., $\mathrm{id}(j) = j, \forall j \in [N]$). Our proof strategy can be outlined in the below diagram:

$$\boxed{\begin{array}{cccc} \bar{U}_\Omega(f) & \xrightarrow{\text{c.t.}} & L_\Omega^*(f) & \xrightarrow{\text{c.t.}} & \bar{L}_\Omega(f) & \xrightarrow{\text{c.t.}} & L_\Omega(f) \\ \text{\footnotesize Eqn. (24)} & & \text{\footnotesize Eqn. (47)} & & \text{\footnotesize Eqn. (47)} & & \text{\footnotesize Eqn. (9)} \end{array}} \tag{49}$$

Where the arrows $\xrightarrow{\text{c.t.}}$ depicts the "concentrates to" relationship between risks/estimators.

## E.2. Auxiliary Risks for $L_\Lambda(f)$

Similar to the previous section, we introduce the following notations:

$$S^{(m)} = \{X_j\}_{j=1}^m, \text{ and } \forall r \in [R] : \begin{cases} m_r & := \left| S_r \cap S^{(m)} \right|, \\ \lambda_r & := \lfloor m_r/3 \rfloor \left[ \sum_{q=1}^R \lfloor m_q/3 \rfloor \right]^{-1}, \\ \bar{\lambda}_r & := \mathbb{E}[\lambda_r]. \end{cases} \tag{50}$$

Furthermore, for every $\pi \in \Pi$, we denote:

$$S_\pi^{(m)} = \left\{ X_{\pi(j)} \right\}_{j=1}^m, \text{ and } \forall r \in [R] : \begin{cases} m_r^\pi & := |S_r \cap S_\pi^{(m)}|, \\ \lambda_r^\pi & := \lfloor m_r^\pi/3 \rfloor \left[ \sum_{q=1}^R \lfloor m_q^\pi/3 \rfloor \right]^{-1}, \\ \lambda_r^* & := \widehat{\mathbb{E}}_\Pi[\lambda_r^\pi]. \end{cases} \tag{51}$$

Then, we define the following auxiliary risks:

$$\mathrm{L}^*_\Lambda(f) := \sum_{r=1}^R \lambda^*_r \mathrm{L}^r_\Lambda(f), \quad \overline{\mathrm{L}}_\Lambda(f) := \sum_{r=1}^R \overline{\lambda}_r \mathrm{L}^r_\Lambda(f). \tag{52}$$

$$\widehat{\mathrm{L}}_\Lambda(f|\pi) := \sum_{r=1}^R \lambda^\pi_r \mathrm{L}^r_\Lambda(f), \quad \widehat{\mathrm{L}}_\Lambda(f) := \sum_{r=1}^R \lambda_r \mathrm{L}^r_\Lambda(f). \tag{53}$$

Similarly, we have $\mathrm{L}^*_\Lambda(f) = \widehat{\mathbb{E}}_\Pi \left[ \mathrm{L}^*_\Lambda(f|\pi) \right], \overline{\mathrm{L}}_\Lambda(f) = \mathbb{E}[\widehat{\mathrm{L}}_\Lambda(f)] = \mathbb{E}[\widehat{\mathrm{L}}_\Lambda(f|\pi)]$ for any $\pi \in \Pi$ and $\widehat{\mathrm{L}}_\Lambda(f) = \widehat{\mathrm{L}}_\Lambda(f|\mathrm{id})$ where $\mathrm{id} \in \Pi$ is the identity permutation (i.e., $\mathrm{id}(j) = j, \forall j \in [N]$). We will prove that $\bar{U}_\Lambda(f)$ concentrates to $\mathrm{L}_\Lambda(f)$ by proving the following chain of concentration:

$$\boxed{\bar{U}_\Lambda(f) \xrightarrow[\text{Eqn. (24)}]{\text{c.t.}} \mathrm{L}^*_\Lambda(f) \xrightarrow[\text{Eqn. (52)}]{\text{c.t.}} \overline{\mathrm{L}}_\Lambda(f) \xrightarrow[\text{Eqn. (52)}]{\text{c.t.}} \mathrm{L}_\Lambda(f)}. \tag{54}$$

Where the arrows $\xrightarrow{\text{c.t.}}$ depicts the "concentrates to" relationship between risks/estimators.

### E.3. Concentration of $\mathrm{L}^*_\Omega(f)$ to $\mathrm{L}_\Omega(f)$

**Lemma E.1.** *Let $\mathcal{F}$ be a class of representation functions. For each $f \in \mathcal{F}$, let $\overline{\mathrm{L}}_\Omega(f)$ be defined in Eqn. (47) and $\mathrm{L}_\Omega(f)$ be defined in Eqn. (9). Then, we have:*

$$\sup_{f \in \mathcal{F}} \left| \overline{\mathrm{L}}_\Omega(f) - \mathrm{L}_\Omega(f) \right| \leqslant \frac{2\mathcal{B}R}{n-R}. \tag{55}$$

**Proof.** We have:

$$\sup_{f \in \mathcal{F}} \left| \overline{\mathrm{L}}_\Omega(f) - \mathrm{L}_\Omega(f) \right| \leqslant \sup_{f \in \mathcal{F}} \left| \sum_{r=1}^R \overline{\omega}_r \mathrm{L}^r_\Omega(f) - \sum_{r=1}^R \rho_r \mathrm{L}^r_\Omega(f) \right|$$

$$= \sup_{f \in \mathcal{F}} \left| \sum_{r=1}^R \mathrm{L}^r_\Omega(f)(\overline{\omega}_r - \rho_r) \right| \leqslant \sum_{r=1}^R \sup_{f \in \mathcal{F}} |\mathrm{L}^r_\Omega(f)(\overline{\omega}_r - \rho_r)|$$

$$\leqslant \sum_{r=1}^R |\overline{\omega}_r - \rho_r| \cdot \sup_{f \in \mathcal{F}} |\mathrm{L}^r_\Omega(f)|$$

$$\leqslant \mathcal{B} \sum_{r=1}^R |\overline{\omega}_r - \rho_r|.$$

Note that $\overline{\omega}_r = \mathbb{E}[\omega^\pi_r]$ for any $\pi \in \Pi$[6]. Therefore, we can write $\overline{\omega}_r = \mathbb{E}\left[ \widehat{\mathbb{E}}_\Pi[\omega^\pi_r] \right]$. As a result, for each $r \in [R]$, we have:

$$|\overline{\omega}_r - \rho_r| = \left| \mathbb{E}\left[ \widehat{\mathbb{E}}_\Pi[\omega^\pi_r] \right] - \mathbb{E}\left[ \widehat{\rho}_r \right] \right| \leqslant \mathbb{E}\left| \widehat{\mathbb{E}}_\Pi[\omega^\pi_r] - \widehat{\rho}_r \right|.$$

From Lemma D.2, we showed that $\sum_{r=1}^R |\omega^\pi_r - \widehat{\rho}^\pi_r| \leqslant \frac{2R}{n-R}$. Therefore:

$$\sum_{r=1}^R \left| \widehat{\mathbb{E}}_\Pi[\omega^\pi_r] - \widehat{\rho}_r \right| = \sum_{r=1}^R \left| \widehat{\mathbb{E}}_\Pi[\omega^\pi_r] - \widehat{\mathbb{E}}_\Pi[\widehat{\rho}^\pi_r] \right|$$

$$\leqslant \sum_{r=1}^R \widehat{\mathbb{E}}_\Pi \left[ |\omega^\pi_r - \widehat{\rho}^\pi_r| \right] = \widehat{\mathbb{E}}_\Pi \left[ \sum_{r=1}^R |\omega^\pi_r - \widehat{\rho}^\pi_r| \right]$$

$$\leqslant \frac{2R}{n-R}.$$

---

[6]Because for any $\pi \in \Pi$, $n^\pi_r \sim \mathrm{Bin}(n, \rho_r)$. In other words, $n^\pi_r$ has the same distribution no matter which permutation $\pi$ we choose.

Finally, we have:

$$\sup_{f \in \mathcal{F}} \left| \overline{L}_\Omega(f) - L_\Omega(f) \right| \leqslant \mathcal{B} \sum_{r=1}^R |\overline{\omega}_r - \rho_r| \leqslant \mathcal{B} \sum_{r=1}^R \mathbb{E} \left| \widehat{\mathbb{E}}_\Pi[\omega_r^\pi] - \widehat{\rho}_r \right|$$

$$= \mathcal{B} \mathbb{E} \left[ \sum_{r=1}^R \left| \widehat{\mathbb{E}}_\Pi[\omega_r^\pi] - \widehat{\rho}_r \right| \right]$$

$$\leqslant \frac{2\mathcal{B}R}{n-R},$$

as desired. $\qquad\square$

**Lemma E.2.** *Let $\mathcal{F}$ be a class of representation functions. For each $f \in \mathcal{F}$, let $\overline{L}_\Omega(f), L_\Omega^*(f)$ be defined in Eqn. (47). Then, for any $\Delta \in (0,1)$ we have:*

$$\sup_{f \in \mathcal{F}} \left| \overline{L}_\Omega(f) - L_\Omega^*(f) \right| \leqslant \mathcal{B} \left[ \frac{4R}{n-R} + \frac{8R}{3N} \ln 2R/\Delta + \sqrt{\frac{8(R-1)\ln 2R/\Delta}{3N}} \right], \tag{56}$$

*with probability of at least $1 - \Delta$.*

**Proof.** Similar to the initial analysis of Lemma E.1, we have:

$$\sup_{f \in \mathcal{F}} \left| \overline{L}_\Omega(f) - L_\Omega^*(f) \right| \leqslant \sup_{f \in \mathcal{F}} \left| \sum_{r=1}^R \overline{\omega}_r L_\Omega^r(f) - \sum_{r=1}^R \omega_r^* L_\Omega^r(f) \right| = \sup_{f \in \mathcal{F}} \left| \sum_{r=1}^R L_\Omega^r(f)(\overline{\omega}_r - \omega_r^*) \right|$$

$$\leqslant \sum_{r=1}^R \sup_{f \in \mathcal{F}} |L_\Omega^r(f)(\overline{\omega}_r - \omega_r^*)|$$

$$\leqslant \sum_{r=1}^R |\overline{\omega}_r - \omega_r^*| \cdot \sup_{f \in \mathcal{F}} |L_\Omega^r(f)|$$

$$\leqslant \mathcal{B} \sum_{r=1}^R |\overline{\omega}_r - \omega_r^*|.$$

Then, for each $r \in [R]$, we have:

$$|\overline{\omega}_r - \omega_r^*| \leqslant |\omega_r^* - \widehat{\rho}_r| + |\widehat{\rho}_r - \rho_r| + |\overline{\omega}_r - \rho_r|$$

$$= \left| \widehat{\mathbb{E}}_\Pi[\omega_r^\pi] - \widehat{\rho}_r \right| + \left| \mathbb{E} \left[ \widehat{\mathbb{E}}_\Pi[\omega_r^\pi] \right] - \mathbb{E}[\widehat{\rho}_r] \right| + |\widehat{\rho}_r - \rho_r|$$

$$\leqslant \left| \widehat{\mathbb{E}}_\Pi[\omega_r^\pi] - \widehat{\rho}_r \right| + \mathbb{E} \left| \widehat{\mathbb{E}}_\Pi[\omega_r^\pi] - \widehat{\rho}_r \right| + |\widehat{\rho}_r - \rho_r|.$$

As a result, we have:

$$\sum_{r=1}^R |\overline{\omega}_r - \omega_r^*| \leqslant \sum_{r=1}^R \left| \widehat{\mathbb{E}}_\Pi[\omega_r^\pi] - \widehat{\rho}_r \right| + \sum_{r=1}^R \mathbb{E} \left| \widehat{\mathbb{E}}_\Pi[\omega_r^\pi] - \widehat{\rho}_r \right| + \sum_{r=1}^R |\widehat{\rho}_r - \rho_r|$$

$$\leqslant \frac{2R}{n-R} + \frac{2R}{n-R} + \sum_{r=1}^R |\widehat{\rho}_r - \rho_r|$$

$$= \frac{4R}{n-R} + \sum_{r=1}^R |\widehat{\rho}_r - \rho_r|.$$

The semi-last inequality comes from $\sum_{r=1}^R \left| \widehat{\mathbb{E}}_\Pi[\omega_r^\pi] - \widehat{\rho}_r \right| \leqslant \frac{2R}{n-R}$, which we proved in Lemma D.2 and Lemma E.1. For each $r \in [R]$, we can write:

$$\widehat{\rho}_r := \frac{N_r}{N} = \frac{1}{N} \sum_{j=1}^N B_{r,j}, \quad \text{where } B_{r,j} \sim \text{Bern}(\rho_r), \forall j \in [N].$$

Then, by the high-probability Bernstein bound, for all $\delta \in (0,1)$ we have the following inequality with probability of at least $1 - \delta$:

$$|\widehat{\rho}_r - \rho_r| = \left| \frac{1}{N} \sum_{j=1}^{N} [B_{r,j} - \rho_r] \right| \leqslant \frac{8}{3N} \ln 2/\delta + \sigma_r \sqrt{\frac{8 \ln 2/\delta}{3N}},$$

where $\sigma_r^2$ is the following mean of second moments:

$$\sigma_r^2 = \frac{1}{N} \sum_{j=1}^{N} \mathbb{E} \left[ (B_{r,j} - \rho_r)^2 \right] = \frac{1}{N} \sum_{j=1}^{N} \mathrm{Var}(B_{r,j}) = \rho_r (1 - \rho_r).$$

Then, by the union bound, with probability of at least $1 - R\delta$, we have:

$$\sum_{r=1}^{R} |\widehat{\rho}_r - \rho_r| \leqslant \frac{8R}{3N} \ln 2/\delta + \sqrt{\frac{8 \ln 2/\delta}{3N}} \sum_{r=1}^{R} \sqrt{\rho_r (1 - \rho_r)}$$

$$\leqslant \frac{8R}{3N} \ln 2/\delta + \sqrt{\frac{8 \ln 2/\delta}{3N}} \sqrt{\left( \sum_{r=1}^{R} \rho_r \right) \left( \sum_{r=1}^{R} (1 - \rho_r) \right)} \quad \text{(Cauchy-Schwarz)}$$

$$\leqslant \frac{8R}{3N} \ln 2/\delta + \sqrt{\frac{8(R-1) \ln 2/\delta}{3N}}.$$

For a fixed $\Delta \in (0,1)$, by setting $\delta = \Delta/R$, we have:

$$\sum_{r=1}^{R} |\widehat{\rho}_r - \rho_r| \leqslant \frac{8R}{3N} \ln 2R/\Delta + \sqrt{\frac{8(R-1) \ln 2R/\Delta}{3N}},$$

with probability of at least $1 - \Delta$. As a result:

$$\sup_{f \in \mathcal{F}} \left| \overline{\mathrm{L}}_\Omega(f) - \mathrm{L}_\Omega^*(f) \right| \leqslant \mathcal{B} \sum_{r=1}^{R} |\overline{\omega}_r - \omega_r^*| \leqslant \mathcal{B} \left[ \frac{4R}{n - R} + \sum_{r=1}^{R} |\widehat{\rho}_r - \rho_r| \right]$$

$$\leqslant \mathcal{B} \left[ \frac{4R}{n - R} + \frac{8R}{3N} \ln 2R/\Delta + \sqrt{\frac{8(R-1) \ln 2R/\Delta}{3N}} \right],$$

with probability of at least $1 - \Delta$, as desired. $\qquad \square$

**Proposition E.3.** *Let $\mathcal{F}$ be a class of representation functions. For each $f \in \mathcal{F}$, let $\mathrm{L}_\Omega(f)$ be defined in Eqn.* (9) *and $\mathrm{L}_\Omega^*(f)$ be defined in Eqn.* (47)*. Then, for any $\Delta \in (0,1)$, we have:*

$$\sup_{f \in \mathcal{F}} |\mathrm{L}_\Omega^*(f) - \mathrm{L}_\Omega(f)| \leqslant \mathcal{B} \left[ \frac{6R}{n - R} + \frac{8R}{3N} \ln 2R/\Delta + \sqrt{\frac{8(R-1) \ln 2R/\Delta}{3N}} \right], \tag{57}$$

*with probability of at least $1 - \Delta$.*

**Proof.** This is a simple consequence of Lemma E.1 and Lemma E.2. For any $\Delta \in (0,1)$, we have:

$$\sup_{f \in \mathcal{F}} |\mathrm{L}_\Omega^*(f) - \mathrm{L}_\Omega(f)| \leqslant \underbrace{\sup_{f \in \mathcal{F}} \left| \mathrm{L}_\Omega^*(f) - \overline{\mathrm{L}}_\Omega(f) \right|}_{\text{Lemma E.2}} + \underbrace{\sup_{f \in \mathcal{F}} \left| \overline{\mathrm{L}}_\Omega(f) - \mathrm{L}_\Omega(f) \right|}_{\text{Lemma E.1}}$$

$$\leqslant \mathcal{B} \left[ \frac{4R}{n - R} + \frac{8R}{3N} \ln 2R/\Delta + \sqrt{\frac{8(R-1) \ln 2R/\Delta}{3N}} \right] + \frac{2\mathcal{B}R}{n - R}$$

$$= \mathcal{B} \left[ \frac{6R}{n - R} + \frac{8R}{3N} \ln 2R/\Delta + \sqrt{\frac{8(R-1) \ln 2R/\Delta}{3N}} \right],$$

with probability of at least $1 - \Delta$, as desired. $\qquad \square$

## E.4. Concentration of $\bar{U}_\Omega(f)$ to $\mathrm{L}_\Omega^*(f)$

**Lemma E.4.** *Let $\mathcal{F}$ be a class of representation functions. For each $f \in \mathcal{F}$, let $\mathrm{L}_\Omega^*(f)$ be defined in Eqn. (47) and $\bar{U}_\Omega(f)$ be defined in Eqn. (24). Then, for any convex $\varphi : \mathbb{R} \to \mathbb{R}$, we have:*

$$\mathbb{E}\varphi\left(\sup_{f \in \mathcal{F}} \left|\bar{U}_\Omega(f) - \mathrm{L}_\Omega^*(f)\right|\right) \leqslant \mathbb{E}\varphi\left(\sup_{f \in \mathcal{F}} \left|\sum_{r=1}^R \omega_r U_{\Omega_r^{\mathrm{id}}}(f) - \widehat{\mathrm{L}}_\Omega(f)\right|\right). \tag{58}$$

**Proof.** For notational brevity, for every $\pi \in \Pi$, we denote $\widehat{U}_\Omega(f|\pi)$ as:

$$\widehat{U}_\Omega(f|\pi) = \sum_{r=1}^R \omega_r^\pi U_{\Omega_r^\pi}(f).$$

Essentially, $\bar{U}_\Omega(f) := \widehat{\mathbb{E}}_\Pi\left[\widehat{U}_\Omega(f|\pi)\right]$. Then, we have:

$$\mathbb{E}\varphi\left(\sup_{f \in \mathcal{F}} \left|\bar{U}_\Omega(f) - \mathrm{L}_\Omega^*(f)\right|\right) = \mathbb{E}\varphi\left(\sup_{f \in \mathcal{F}} \left|\widehat{\mathbb{E}}_\Pi\left[\widehat{U}_\Omega(f|\pi)\right] - \widehat{\mathbb{E}}_\Pi\left[\widehat{\mathrm{L}}_\Omega(f|\pi)\right]\right|\right)$$

$$= \mathbb{E}\varphi\left(\sup_{f \in \mathcal{F}} \left|\widehat{\mathbb{E}}_\Pi\left[\widehat{U}_\Omega(f|\pi) - \widehat{\mathrm{L}}_\Omega(f|\pi)\right]\right|\right)$$

$$\leqslant \mathbb{E}\varphi\left(\widehat{\mathbb{E}}_\Pi\left[\sup_{f \in \mathcal{F}} \left|\widehat{U}_\Omega(f|\pi) - \widehat{\mathrm{L}}_\Omega(f|\pi)\right|\right]\right) \quad \left(\sup\left|\widehat{\mathbb{E}}_\Pi[\cdot]\right| \leqslant \widehat{\mathbb{E}}_\Pi\left[\sup|\cdot|\right]\right)$$

$$\leqslant \widehat{\mathbb{E}}_\Pi\left[\mathbb{E}\varphi\left(\sup_{f \in \mathcal{F}} \left|\widehat{U}_\Omega(f|\pi) - \widehat{\mathrm{L}}_\Omega(f|\pi)\right|\right)\right] \quad \text{(Jensen's Inequality)}.$$

For any two distinct permutations $\pi_1, \pi_2 \in \Pi$, we always have:

$$S_{\pi_1}^{(n)} \overset{d}{=} S_{\pi_2}^{(n)}, \quad \text{and} \quad S \setminus S_{\pi_1}^{(n)} \overset{d}{=} S \setminus S_{\pi_2}^{(n)}.$$

Where $X \overset{d}{=} Y$ means "$X$ is identically distributed as $Y$". Therefore:

$$\forall \pi_1, \pi_2 \in \Pi : \widehat{U}_\Omega(f|\pi_1) \overset{d}{=} \widehat{U}_\Omega(f|\pi_2) \text{ and } \widehat{\mathrm{L}}_\Omega(f|\pi_1) \overset{d}{=} \widehat{\mathrm{L}}_\Omega(f|\pi_2).$$

In other words, for any $\pi_1, \pi_2 \in \Pi$:

$$\mathbb{E}\varphi\left(\sup_{f \in \mathcal{F}} \left|\widehat{U}_\Omega(f|\pi_1) - \widehat{\mathrm{L}}_\Omega(f|\pi_1)\right|\right) = \mathbb{E}\varphi\left(\sup_{f \in \mathcal{F}} \left|\widehat{U}_\Omega(f|\pi_2) - \widehat{\mathrm{L}}_\Omega(f|\pi_2)\right|\right).$$

Therefore, we have:

$$\mathbb{E}\varphi\left(\sup_{f \in \mathcal{F}} \left|\bar{U}_\Omega(f) - \mathrm{L}_\Omega^*(f)\right|\right) \leqslant \widehat{\mathbb{E}}_\Pi\left[\mathbb{E}\varphi\left(\sup_{f \in \mathcal{F}} \left|\widehat{U}_\Omega(f|\pi) - \widehat{\mathrm{L}}_\Omega(f|\pi)\right|\right)\right]$$

$$= \mathbb{E}\varphi\left(\sup_{f \in \mathcal{F}} \left|\widehat{U}_\Omega(f|\mathrm{id}) - \widehat{\mathrm{L}}_\Omega(f|\mathrm{id})\right|\right)$$

$$= \mathbb{E}\varphi\left(\sup_{f \in \mathcal{F}} \left|\sum_{r=1}^R \omega_r U_{\Omega_r^{\mathrm{id}}}(f) - \widehat{\mathrm{L}}_\Omega(f)\right|\right),$$

as desired. $\qquad\square$

From the above lemma, we can reduce the concentration analysis problem of $\bar{U}_\Omega(f)$ around $\mathrm{L}_\Omega^*(f)$ to analyzing the concentration of $\sum_{r=1}^R \omega_r U_{\Omega_r^{\mathrm{id}}}(f)$ around $\widehat{\mathrm{L}}_\Omega(f)$. From here on, we use the notation $S^{(n)} = \{X_j\}_{j=1}^n$ and $\bar{S}^{(n)} = \{Z_j\}_{j=1}^{N-n}$ where $Z_j = X_{n+j}$ for all $1 \leqslant j \leqslant N - n$.

For notational brevity, we first denote:

$$\widehat{U}_\Omega(f) := \sum_{r=1}^R \omega_r U_{\Omega_r^{\mathrm{id}}}(f). \tag{59}$$

Then, we note that $\widehat{U}_\Omega(f)$ can also be re-written as follows:

$$\widehat{U}_\Omega(f) = \frac{1}{n!(N-n)!} \sum_{\substack{\xi \in \Pi[n] \\ \zeta \in \Pi[N-n]}} \frac{1}{\bar{n}} \sum_{r=1}^R \sum_{j=1}^{\bar{n}_r} \ell_{\phi,f}\left(X_{2j-1}^{r,\xi}, X_{2j}^{r,\xi}, \left\{Z_{k(j+n_r^*)+i}^\zeta\right\}_{i=1}^k\right), \tag{60}$$

where $\bar{n}_r = \lfloor n_r/2 \rfloor$, $\bar{n} = \sum_{r=1}^R \bar{n}_r$, $n_r^* = \sum_{q=1}^{r-1} \bar{n}_r$ and $\Pi[n], \Pi[N-n]$ denotes the sets of all bijections $[n] \to [n]$ and $[N-n] \to [N-n]$, respectively. We also use the following notations for brevity:

1. $X_u^r$ denotes the $u^{\mathrm{th}}$ data-point that belongs to class $r \in [R]$ in $\{X_j\}_{j=1}^n$.

2. $X_u^{r,\xi}$ denotes the $u^{\mathrm{th}}$ data-point that belongs to class $r \in [R]$ in $\{X_{\xi(j)}\}_{j=1}^n$.

3. $Z_u^\zeta = Z_{\xi(u)}$, i.e., $\left\{Z_{ku+i}^\zeta\right\}_{i=1}^k$ is the $u^{\mathrm{th}}$ block of $k$-elements extracted in order from the shuffled dataset $\left\{Z_{\zeta(j)}^\zeta\right\}_{j=1}^{N-n}$.

Furthermore, for the proofs of subsequent results, we also define the following dataset of tuples:

$$S_{\mathrm{tup}} := \bigcup_{r=1}^R \left\{\left(X_{2j-1}^r, X_{2j}^r, \left\{Z_{k(j+n_r^*)+i}\right\}_{i=1}^k\right)\right\}_{j=1}^{\bar{n}_r}. \tag{61}$$

**Lemma E.5.** *Let $\mathcal{F}$ be a class of representation functions. For each $f \in \mathcal{F}$, let $\widehat{U}_\Omega(f)$ be the estimator defined in Eqn. (59) (or Eqn. (60)) and $\widehat{\mathrm{L}}_\Omega(f)$ be defined in Eqn. (48). Then, for any convex $\varphi: \mathbb{R} \to \mathbb{R}$, we have:*

$$\mathbb{E}_S \varphi\left(\sup_{f \in \mathcal{F}} \left|\widehat{U}_\Omega(f) - \widehat{\mathrm{L}}_\Omega(f)\right|\right) \leqslant \mathbb{E}_{S_{\mathrm{tup}}, \boldsymbol{\Sigma}_{\bar{n}}} \varphi\left(2\mathcal{R}_\mathcal{H}^{S_{\mathrm{tup}}, \boldsymbol{\Sigma}_{\bar{n}}}\right), \tag{62}$$

*where $\boldsymbol{\Sigma}_{\bar{n}} = \{\sigma_j\}_{j=1}^{\bar{n}}$ is a sequence of independent Rademacher variables, $\mathcal{H}$ is defined in Eqn. (7) and the random variable $\mathcal{R}_\mathcal{H}^{S_{\mathrm{tup}}, \boldsymbol{\Sigma}_{\bar{n}}}$ is defined as follows:*

$$\mathcal{R}_\mathcal{H}^{S_{\mathrm{tup}}, \boldsymbol{\Sigma}_{\bar{n}}} := \sup_{f \in \mathcal{F}} \left|\frac{1}{\bar{n}} \sum_{r=1}^R \sum_{j=1}^{\bar{n}_r} \sigma_{j+n_r^*} \ell_{\phi,f}\left(X_{2j-1}^r, X_{2j}^r, \left\{Z_{k(j+n_r^*)+i}\right\}_{i=1}^k\right)\right|. \tag{63}$$

**Proof.** For brevity, we denote $C = n!(N-n)!$ and $\Pi_{n,N} = \Pi[n] \times \Pi[N-n]$. Then, we have:

$$\mathbb{E}_S \varphi\left(\sup_{f \in \mathcal{F}} \left|\widehat{U}_\Omega(f) - \widehat{\mathrm{L}}_\Omega(f)\right|\right)$$

$$= \mathbb{E}_S \varphi\left(\sup_{f \in \mathcal{F}} \left|\frac{1}{C} \sum_{\xi,\zeta \in \Pi_{n,N}} \frac{1}{\bar{n}} \sum_{r=1}^R \sum_{j=1}^{\bar{n}_r} \ell_{\phi,f}\left(X_{2j-1}^{r,\xi}, X_{2j}^{r,\xi}, \left\{Z_{k(j+n_r^*)+i}^\zeta\right\}_{i=1}^k\right) - \widehat{\mathrm{L}}_\Omega(f)\right|\right)$$

$$\leqslant \mathbb{E}_S \varphi\left(\frac{1}{C} \sum_{\xi,\zeta \in \Pi_{n,N}} \sup_{f \in \mathcal{F}} \left|\frac{1}{\bar{n}} \sum_{r=1}^R \sum_{j=1}^{\bar{n}_r} \ell_{\phi,f}\left(X_{2j-1}^{r,\xi}, X_{2j}^{r,\xi}, \left\{Z_{k(j+n_r^*)+i}^\zeta\right\}_{i=1}^k\right) - \widehat{\mathrm{L}}_\Omega(f)\right|\right)$$

$$\leqslant \frac{1}{C} \sum_{\xi,\zeta \in \Pi_{n,N}} \mathbb{E}_S \varphi\left(\sup_{f \in \mathcal{F}} \left|\frac{1}{\bar{n}} \sum_{r=1}^R \sum_{j=1}^{\bar{n}_r} \ell_{\phi,f}\left(X_{2j-1}^{r,\xi}, X_{2j}^{r,\xi}, \left\{Z_{k(j+n_r^*)+i}^\zeta\right\}_{i=1}^k\right) - \widehat{\mathrm{L}}_\Omega(f)\right|\right) \quad \text{(Jensen's).}$$

For each $\xi \in \Pi[n]$ and $\zeta \in \Pi[N-n]$, we define $S_{\text{tup}}^{\xi,\zeta}$ as the set of tuples:

$$S_{\text{tup}}^{\xi,\zeta} := \bigcup_{r=1}^{R} \left\{ \left( X_{2j-1}^{r,\xi}, X_{2j}^{r,\xi}, \left\{ Z_{k(j+n_r^*)+i}^{\zeta} \right\}_{i=1}^{k} \right) \right\}_{j=1}^{\bar{n}_r}, \tag{64}$$

$$\text{and } \Psi\left( S_{\text{tup}}^{\xi,\zeta} \right) = \frac{1}{\bar{n}} \sum_{r=1}^{R} \sum_{j=1}^{\bar{n}_r} \ell_{\phi,f} \left( X_{2j-1}^{r,\xi}, X_{2j}^{r,\xi}, \left\{ Z_{k(j+n_r^*)+i}^{\zeta} \right\}_{i=1}^{k} \right). \tag{65}$$

Then, for any $\xi_1, \xi_2 \in \Pi[n]$ and $\zeta_1, \zeta_2 \in \Pi[N-n]$, we will always have $S_{\text{tup}}^{\xi_1,\zeta_1} \stackrel{d}{=} S_{\text{tup}}^{\xi_2,\zeta_2}$. Therefore, we have:

$$\mathbb{E}_S \left[ \Psi\left( S_{\text{tup}}^{\xi_1,\zeta_1} \right) \right] = \mathbb{E}_S \left[ \Psi\left( S_{\text{tup}}^{\xi_2,\zeta_2} \right) \right].$$

Therefore:

$$\mathbb{E}_S \varphi \left( \sup_{f \in \mathcal{F}} \left| \widehat{U}_\Omega(f) - \widehat{L}_\Omega(f) \right| \right) \leqslant \frac{1}{C} \sum_{\xi,\zeta \in \Pi_{n,N}} \mathbb{E}_S \varphi \left( \sup_{f \in \mathcal{F}} \left| \Psi\left( S_{\text{tup}}^{\xi,\zeta} \right) - \widehat{L}_\Omega(f) \right| \right)$$

$$= \mathbb{E}_{S_{\text{tup}}} \varphi \left( \sup_{f \in \mathcal{F}} \left| \Psi\left( S_{\text{tup}} \right) - \widehat{L}_\Omega(f) \right| \right).$$

Now, for each $r \in [R]$, recall that $n_r = |S_r \cap S^{(n)}|$ and $n_r \sim \text{Bin}(n, \rho_r)$. Let $\mathbf{n} = \{n_r\}_{1 \leqslant r \leqslant R}$ be the vector of class-wise sample sizes in the first partition $S^{(n)}$. We observe that:

$$\mathbb{E}_{S_{\text{tup}}|\mathbf{n}} \left[ \Psi\left( S_{\text{tup}} \right) \right] = \mathbb{E}_{S_{\text{tup}}|\mathbf{n}} \left[ \frac{1}{\bar{n}} \sum_{r=1}^{R} \sum_{j=1}^{\bar{n}_r} \ell_{\phi,f} \left( X_{2j-1}^{r,\xi}, X_{2j}^{r,\xi}, \left\{ Z_{k(j+n_r^*)+i}^{\zeta} \right\}_{i=1}^{k} \right) \right]$$

$$= \frac{1}{\bar{n}} \sum_{r=1}^{R} \sum_{j=1}^{\bar{n}_r} \mathbb{E}_{S_{\text{tup}}|\mathbf{n}} \left[ \ell_{\phi,f} \left( X_{2j-1}^{r,\xi}, X_{2j}^{r,\xi}, \left\{ Z_{k(j+n_r^*)+i}^{\zeta} \right\}_{i=1}^{k} \right) \right]$$

$$= \sum_{r=1}^{R} \frac{\bar{n}_r}{\bar{n}} L_\Omega^r(f) = \sum_{r=1}^{R} \omega_r L_\Omega^r(f)$$

$$= \widehat{L}_\Omega(f).$$

In other words, given the sample sizes $\mathbf{n}$, $\Psi(S_{\text{tup}})$ has expectation $\widehat{L}_\Omega(f)$. Therefore, we can apply the symmetrization trick (given $\mathbf{n}$) here. Specifically:

$$\mathbb{E}_S \varphi \left( \sup_{f \in \mathcal{F}} \left| \widehat{U}_\Omega(f) - \widehat{L}_\Omega(f) \right| \right) = \mathbb{E}_{\mathbf{n}} \mathbb{E}_{S|\mathbf{n}} \varphi \left( \sup_{f \in \mathcal{F}} \left| \widehat{U}_\Omega(f) - \widehat{L}_\Omega(f) \right| \right)$$

$$\leqslant \mathbb{E}_{\mathbf{n}} \mathbb{E}_{S_{\text{tup}}|\mathbf{n}} \varphi \left( \sup_{f \in \mathcal{F}} \left| \Psi\left( S_{\text{tup}} \right) - \widehat{L}_\Omega(f) \right| \right)$$

$$\leqslant \mathbb{E}_{\mathbf{n}} \mathbb{E}_{S_{\text{tup}}, \mathbf{\Sigma}_{\bar{n}}|\mathbf{n}} \varphi \left( \sup_{f \in \mathcal{F}} \left| \frac{1}{\bar{n}} \sum_{r=1}^{R} \sum_{j=1}^{\bar{n}_r} \sigma_{j+n_r^*} \ell_{\phi,f} \left( X_{2j-1}^{r}, X_{2j}^{r}, \left\{ Z_{k(j+n_r^*)+i}^{r} \right\}_{i=1}^{k} \right) \right| \right) \quad \text{(Lemma L.7)}$$

$$= \mathbb{E}_{S_{\text{tup}}, \mathbf{\Sigma}_{\bar{n}}} \varphi \left( \sup_{f \in \mathcal{F}} \left| \frac{1}{\bar{n}} \sum_{r=1}^{R} \sum_{j=1}^{\bar{n}_r} \sigma_{j+n_r^*} \ell_{\phi,f} \left( X_{2j-1}^{r}, X_{2j}^{r}, \left\{ Z_{k(j+n_r^*)+i}^{r} \right\}_{i=1}^{k} \right) \right| \right)$$

$$\leqslant \mathbb{E}_{S_{\text{tup}}, \mathbf{\Sigma}_{\bar{n}}} \varphi \left( 2\mathcal{R}_{\mathcal{H}}^{S_{\text{tup}}, \mathbf{\Sigma}_{\bar{n}}} \right),$$

as desired. $\qquad \square$

**Proposition E.6.** *Let $\mathcal{F}$ be a class of representation functions. For each $f \in \mathcal{F}$, let $\bar{U}_\Omega(f)$ be defined in Eqn. (24) and $\mathrm{L}_\Omega^*(f)$ be defined in Eqn. (47). Then, for any $\Delta \in (0,1)$, we have:*

$$\sup_{f \in \mathcal{F}} \left| \bar{U}_\Omega(f) - \mathrm{L}_\Omega^*(f) \right| \leqslant 2\mathfrak{R}_{\mu_*}(\mathcal{H}) + 8\mathcal{B}\sqrt{\frac{\ln 1/\Delta}{n - R}}, \tag{66}$$

*with probability of at least $1 - \Delta$ where $\mu_*$ is a distribution of tuples set defined as:*

$$\mu_* := \arg\max_{\mu \in \mathcal{U}} \mathfrak{R}_\mu(\mathcal{H}), \tag{67}$$

$$\mathcal{U} = \left\{ \bigotimes_{r=1}^{R} \left[ \mathcal{D}_r^{\otimes 2} \otimes \bar{\mathcal{D}}^{\otimes k} \right]^{\otimes \lfloor q_r/2 \rfloor} : q_r \in \mathbb{Z}_{\geqslant 0}, \forall r \in [R] \text{ and } \sum_{r=1}^{R} q_r = n \right\}. \tag{68}$$

**Proof.** Let $\mathcal{R}_\mathcal{H}^{S_{\mathrm{tup}}, \Sigma_{\bar{n}}}$ be defined in Lemma E.5 and let $\mathbf{n} = \{n_r\}_{1 \leqslant r \leqslant R}$ be the random vector of class-wise sample sizes in $S^{(n)}$. Applying Lemma E.4 and Lemma E.5 with $\varphi(x) = e^{\lambda x}$ (for $\lambda > 0$), we have:

$$
\begin{aligned}
\mathbb{E}_S \exp\left( \lambda \sup_{f \in \mathcal{F}} \left| \bar{U}_\Omega(f) - \mathrm{L}_\Omega^*(f) \right| \right) &\leqslant \mathbb{E}_S \exp\left( \lambda \sup_{f \in \mathcal{F}} \left| \widehat{U}_\Omega(f) - \widehat{\mathrm{L}}_\Omega(f) \right| \right) && \text{(Lemma E.4)} \\
&\leqslant \mathbb{E}_{S_{\mathrm{tup}}, \Sigma_{\bar{n}}} \exp\left( 2\lambda \mathcal{R}_\mathcal{H}^{S_{\mathrm{tup}}, \Sigma_{\bar{n}}} \right) && \text{(Lemma E.5)} \\
&= \mathbb{E}_\mathbf{n} \mathbb{E}_{S_{\mathrm{tup}}, \Sigma_{\bar{n}} | \mathbf{n}} \exp\left( 2\lambda \mathcal{R}_\mathcal{H}^{S_{\mathrm{tup}}, \Sigma_{\bar{n}}} \right) \\
&\leqslant \mathbb{E}_\mathbf{n} \exp\left( 2\lambda \mathbb{E}_{S_{\mathrm{tup}}, \Sigma_{\bar{n}} | \mathbf{n}} \left[ \mathcal{R}_\mathcal{H}^{S_{\mathrm{tup}}, \Sigma_{\bar{n}}} \right] + \frac{32\lambda^2 \mathcal{B}^2}{\bar{n}} \right) && \text{(Lemma L.10)} \\
&= \mathbb{E}_\mathbf{n} \exp\left( 2\lambda \mathfrak{R}_{\bar{\mu}_\mathbf{n}}(\mathcal{H}) + \frac{32\lambda^2 \mathcal{B}^2}{\bar{n}} \right).
\end{aligned}
$$

where we define the tuples distribution $\bar{\mu}_\mathbf{n}$ as:

$$\bar{\mu}_\mathbf{n} := \bigotimes_{r=1}^{R} \left[ \mathcal{D}_r^{\otimes 2} \otimes \bar{\mathcal{D}}^{\otimes k} \right]^{\otimes \lfloor n_r/2 \rfloor}.$$

Which is the distribution of $S_{\mathrm{tup}}$ given the observations of the sample sizes $\mathbf{n}$. Obviously, $\bar{\mu}_\mathbf{n} \in \mathcal{U}$ for every observation of $\mathbf{n}$. Hence, we have:

$$
\begin{aligned}
\mathbb{E}_S \exp\left( \lambda \sup_{f \in \mathcal{F}} \left| \bar{U}_\Omega(f) - \mathrm{L}_\Omega^*(f) \right| \right) &\leqslant \mathbb{E}_\mathbf{n} \exp\left( 2\lambda \mathfrak{R}_{\bar{\mu}_\mathbf{n}}(\mathcal{H}) + \frac{32\lambda^2 \mathcal{B}^2}{\bar{n}} \right) \\
&\leqslant \sup_{\mu \in \mathcal{U}} \exp\left( 2\lambda \mathfrak{R}_\mu(\mathcal{H}) + \frac{32\lambda^2 \mathcal{B}^2}{\bar{n}} \right) \\
&= \exp\left( 2\lambda \mathfrak{R}_{\mu_*}(\mathcal{H}) + \frac{32\lambda^2 \mathcal{B}^2}{\bar{n}} \right).
\end{aligned}
$$

Using the Chernoff bound, for all $\varepsilon, \lambda > 0$, we have:

$$
\begin{aligned}
\mathbb{P}\left( \sup_{f \in \mathcal{F}} \left| \bar{U}_\Omega(f) - \mathrm{L}_\Omega^*(f) \right| \geqslant \varepsilon \right) &\leqslant e^{-\lambda \varepsilon} \mathbb{E}_S \exp\left( \lambda \sup_{f \in \mathcal{F}} \left| \bar{U}_\Omega(f) - \mathrm{L}_\Omega^*(f) \right| \right) \\
&\leqslant \exp\left( 2\lambda \mathfrak{R}_{\mu_*}(\mathcal{H}) + \frac{32\lambda^2 \mathcal{B}^2}{\bar{n}} - \lambda \varepsilon \right).
\end{aligned}
$$

Setting $\Delta = \exp\left( 2\lambda \mathfrak{R}_{\mu_*}(\mathcal{H}) + \frac{32\lambda^2 \mathcal{B}^2}{\bar{n}} - \lambda \varepsilon \right)$ and solve for $\varepsilon$, we have:

$$\varepsilon = 2\lambda \mathfrak{R}_{\mu_*}(\mathcal{H}) + \frac{32\lambda \mathcal{B}^2}{\bar{n}} + \lambda^{-1} \ln 1/\Delta.$$

Solving for the optimal value $\lambda$, we have $\lambda^{-1} = 4\mathcal{B}\sqrt{\frac{2}{\bar{n}\ln 1/\Delta}}$. Plugging this value back to $\varepsilon$ yields:

$$\sup_{f\in\mathcal{F}}\left|\bar{U}_\Omega(f) - \mathrm{L}_\Omega^*(f)\right| \leqslant 2\mathfrak{R}_{\mu_*}(\mathcal{H}) + 8\mathcal{B}\sqrt{\frac{\ln 1/\Delta}{2\bar{n}}}$$

$$\leqslant 2\mathfrak{R}_{\mu_*}(\mathcal{H}) + 8\mathcal{B}\sqrt{\frac{\ln 1/\Delta}{n - R}},$$

with probability of at least $1 - \Delta$, as desired. $\qquad\square$

**Theorem E.7.** *Let $\mathcal{F}$ be a class of representation functions. For each $f \in \mathcal{F}$, let $\bar{U}_\Omega(f)$ be defined in Eqn. (24) and $\mathrm{L}_\Omega(f)$ be defined in Eqn. (9). Then, for any $\Delta \in (0, 1)$, we have:*

$$\sup_{f\in\mathcal{F}}\left|\bar{U}_\Omega(f) - \mathrm{L}_\Omega(f)\right| \tag{69}$$

$$\leqslant 2\mathfrak{R}_{\mu_*}(\mathcal{H}) + \mathcal{B}\left[\frac{6R}{n-R} + \frac{8R}{3N}\ln 4R/\Delta + \sqrt{\frac{8(R-1)\ln 4R/\Delta}{3N}} + 8\sqrt{\frac{\ln 2/\Delta}{n-R}}\right],$$

*with probability of at least $1 - \Delta$.*

**Proof.** We have:

$$\sup_{f\in\mathcal{F}}\left|\bar{U}_\Omega(f) - \mathrm{L}_\Omega(f)\right| \leqslant \sup_{f\in\mathcal{F}}\left|\bar{U}_\Omega(f) - \mathrm{L}_\Omega^*(f)\right| + \sup_{f\in\mathcal{F}}\left|\mathrm{L}_\Omega^*(f) - \overline{\mathrm{L}}_\Omega(f)\right| + \sup_{f\in\mathcal{F}}\left|\overline{\mathrm{L}}_\Omega(f) - \mathrm{L}_\Omega(f)\right|.$$

For a given $\delta \in (0, 1)$, by Lemma E.1, Proposition E.3 and Proposition E.6, we have:

$$\sup_{f\in\mathcal{F}}\left|\overline{\mathrm{L}}_\Omega(f) - \mathrm{L}_\Omega(f)\right| \leqslant \frac{2\mathcal{B}R}{n-R},$$

$$\sup_{f\in\mathcal{F}}\left|\mathrm{L}_\Omega^*(f) - \overline{\mathrm{L}}_\Omega(f)\right| \leqslant \mathcal{B}\left[\frac{6R}{n-R} + \frac{8R}{3N}\ln 2R/\delta + \sqrt{\frac{8(R-1)\ln 2R/\delta}{3N}}\right] \text{ with prob. } 1 - \delta,$$

$$\sup_{f\in\mathcal{F}}\left|\bar{U}_\Omega(f) - \mathrm{L}_\Omega^*(f)\right| \leqslant 2\mathfrak{R}_{\mu_*}(\mathcal{H}) + 8\mathcal{B}\sqrt{\frac{\ln 1/\delta}{n-R}} \text{ with prob. } 1 - \delta.$$

Combining everything with the union bound, we have:

$$\sup_{f\in\mathcal{F}}\left|\bar{U}_\Omega(f) - \mathrm{L}_\Omega(f)\right| \leqslant 2\mathfrak{R}_{\mu_*}(\mathcal{H}) + \mathcal{B}\left[\frac{6R}{n-R} + \frac{8R}{3N}\ln 2R/\delta + \sqrt{\frac{8(R-1)\ln 2R/\delta}{3N}} + 8\sqrt{\frac{\ln 1/\delta}{n-R}}\right],$$

with probability of at least $1 - 2\delta$. Setting $\delta = \Delta/2$ yields the desired bound. $\qquad\square$

## E.5. Concentration of $\mathrm{L}_\Lambda^*(f)$ to $\mathrm{L}_\Lambda(f)$

The concentration analysis of $\bar{U}_\Lambda(f)$ is almost verbatim to the previous sub-sections. Therefore, for readers who have grasped the overall strategy, we recommend skipping this sub-section and the next sub-section.

**Lemma E.8.** *Let $\mathcal{F}$ be a class of representation functions. For each $f \in \mathcal{F}$, let $\overline{\mathrm{L}}_\Lambda(f)$ be defined in Eqn. (52) and $\mathrm{L}_\Lambda(f)$ be defined in Eqn. (10). Then, we have:*

$$\sup_{f\in\mathcal{F}}\left|\overline{\mathrm{L}}_\Lambda(f) - \mathrm{L}_\Lambda(f)\right| \leqslant \frac{4\mathcal{B}R}{m - 2R}. \tag{70}$$

**Proof.** Similar to Lemma E.1, we can easily show that:

$$\sup_{f\in\mathcal{F}}\left|\overline{\mathrm{L}}_\Lambda(f) - \mathrm{L}_\Lambda(f)\right| \leqslant \mathcal{B}\sum_{r=1}^{R}\left|\bar{\lambda}_r - \rho_r\right| \leqslant \mathcal{B}\sum_{r=1}^{R}\mathbb{E}\left|\widehat{\mathbb{E}}_\Pi[\lambda_r^\pi] - \widehat{\rho}_r\right|.$$

Then, using the fact that we can write $\widehat{\rho}_r = \widehat{\mathbb{E}}_\Pi[m_r^\pi/m]$, we have:

$$\left|\widehat{\mathbb{E}}_\Pi[\lambda_r^\pi] - \widehat{\rho}_r\right| = \left|\widehat{\mathbb{E}}_\Pi[\lambda_r^\pi] - \widehat{\mathbb{E}}_\Pi[m_r^\pi/m]\right| \leqslant \widehat{\mathbb{E}}_\Pi\left|\lambda_r^\pi - m_r^\pi/m\right|.$$

Then, for each $r \in [R]$ and $\pi \in \Pi$, we have:

$$\begin{aligned}
\left|m_r^\pi/m - \lambda_r^\pi\right| &= \left|\frac{m_r^\pi}{m} - \frac{\lfloor m_r^\pi/3\rfloor}{\sum_{q=1}^R \lfloor m_q^\pi/3\rfloor}\right| \\
&\leqslant \left|\frac{m_r^\pi}{m} - \frac{3\lfloor m_r^\pi/3\rfloor}{m}\right| + \left|\frac{3\lfloor m_r^\pi/3\rfloor}{m} - \frac{\lfloor m_r^\pi/3\rfloor}{\sum_{q=1}^R \lfloor m_q^\pi/3\rfloor}\right| \\
&= \frac{1}{m}(m_r^\pi - 3\lfloor m_r^\pi/3\rfloor) + \lfloor m_r^\pi/3\rfloor\left(\frac{1}{\sum_{q=1}^R \lfloor m_q^\pi/3\rfloor} - \frac{3}{m}\right).
\end{aligned}$$

Using the fact that $m_r^\pi - 3\lfloor m_r^\pi/3\rfloor \leqslant 2$ and $\sum_{q=1}^R \lfloor m_r^\pi/3\rfloor \geqslant \frac{m-2R}{3}$, we have:

$$\left|m_r^\pi/m - \lambda_r^\pi\right| \leqslant \frac{2}{m} + 3\lfloor m_r^\pi/3\rfloor\left(\frac{1}{m-2R} - \frac{1}{m}\right) \leqslant \frac{2}{m} + \frac{6R\lfloor m_r^\pi/3\rfloor}{m(m-2R)}.$$

Hence, we have:

$$\begin{aligned}
\sup_{f \in \mathcal{F}} \left|\overline{\mathrm{L}}_\Lambda(f) - \mathrm{L}_\Lambda(f)\right| &\leqslant \mathcal{B}\sum_{r=1}^R \mathbb{E}\left|\widehat{\mathbb{E}}_\Pi[\lambda_r^\pi] - \widehat{\rho}_r\right| \leqslant \mathcal{B}\sum_{r=1}^R \mathbb{E}\left[\widehat{\mathbb{E}}_\Pi\left|\lambda_r^\pi - m_r^\pi/m\right|\right] \\
&\leqslant \frac{2\mathcal{B}R}{m} + \frac{6\mathcal{B}R}{m(m-2R)}\mathbb{E}\left[\widehat{\mathbb{E}}_\Pi\left[\underbrace{\sum_{r=1}^R \lfloor m_r^\pi/3\rfloor}_{\leqslant m/3}\right]\right] \\
&\leqslant \frac{2\mathcal{B}R}{m} + \frac{2\mathcal{B}R}{m-2R} \leqslant \frac{4\mathcal{B}R}{m-2R},
\end{aligned}$$

as desired. $\qquad\square$

**Lemma E.9.** *Let $\mathcal{F}$ be a class of representation functions. For each $f \in \mathcal{F}$, let $\overline{\mathrm{L}}_\Lambda(f), \mathrm{L}_\Lambda^*(f)$ be defined in Eqn. (52). Then, for any $\Delta \in (0,1)$ we have:*

$$\sup_{f \in \mathcal{F}} \left|\overline{\mathrm{L}}_\Lambda(f) - \mathrm{L}_\Lambda^*(f)\right| \leqslant \mathcal{B}\left[\frac{8R}{m-2R} + \frac{8R}{3N}\ln 2R/\Delta + \sqrt{\frac{8(R-1)\ln 2R/\Delta}{3N}}\right], \tag{71}$$

*with probability of at least $1 - \Delta$.*

**Proof.** Similar to Lemma E.2, we can easily obtain:

$$\begin{aligned}
\sup_{f \in \mathcal{F}} \left|\overline{\mathrm{L}}_\Lambda(f) - \mathrm{L}_\Lambda^*(f)\right| &\leqslant \mathcal{B}\sum_{r=1}^R |\overline{\lambda}_r - \lambda_r^*| \\
&\leqslant \mathcal{B}\left[\sum_{r=1}^R |\lambda_r^* - \widehat{\rho}_r| + \sum_{r=1}^R |\overline{\lambda}_r - \rho_r| + \sum_{r=1}^R |\widehat{\rho}_r - \rho_r|\right] \\
&\leqslant \mathcal{B}\left[\sum_{r=1}^R \left|\widehat{\mathbb{E}}_\Pi[\lambda_r^\pi] - \widehat{\rho}_r\right| + \sum_{r=1}^R \mathbb{E}\left|\widehat{\mathbb{E}}_\Pi[\lambda_r^\pi] - \widehat{\rho}_r\right| + \sum_{r=1}^R |\widehat{\rho}_r - \rho_r|\right] \\
&\leqslant \frac{4\mathcal{B}R}{m-2R} + \frac{4\mathcal{B}R}{m-2R} + \mathcal{B}\sum_{r=1}^R |\widehat{\rho}_r - \rho_r| \quad \text{(Lemma E.8)} \\
&= \frac{8\mathcal{B}R}{m-2R} + \mathcal{B}\sum_{r=1}^R |\widehat{\rho}_r - \rho_r|.
\end{aligned}$$

Using Bernstein's bound to analyze the concentration of $\widehat{\rho}_r$ around $\rho_r$ as usual, for any $\Delta \in (0, 1)$, we have:

$$\sum_{r=1}^{R} |\widehat{\rho}_r - \rho_r| \leqslant \frac{8R}{3N} \ln 2R/\Delta + \sqrt{\frac{8(R-1) \ln 2R/\Delta}{3N}},$$

with probability of at least $1 - \Delta$. As a result, we have:

$$\sup_{f \in \mathcal{F}} \left| \overline{\mathrm{L}}_\Lambda(f) - \mathrm{L}_\Lambda^*(f) \right| \leqslant \frac{8\mathcal{B}R}{m - 2R} + \mathcal{B} \sum_{r=1}^{R} |\widehat{\rho}_r - \rho_r|$$

$$\leqslant \mathcal{B} \left[ \frac{8R}{m - 2R} + \frac{8R}{3N} \ln 2R/\Delta + \sqrt{\frac{8(R-1) \ln 2R/\Delta}{3N}} \right],$$

with probability of at least $1 - \Delta$, as desired. $\qquad \square$

### E.6. Concentration of $\bar{U}_\Lambda(f)$ to $\mathrm{L}_\Lambda^*(f)$

Applying the same strategy as Lemma E.4, Lemma E.5 and Proposition E.6, we have the following analogous result.

**Proposition E.10.** *Let $\mathcal{F}$ be a class of representation functions. For each $f \in \mathcal{F}$, let $\bar{U}_\Lambda(f)$ be defined in Eqn. (24) and $\mathrm{L}_\Lambda^*(f)$ be defined in Eqn. (52). Then, for any $\Delta \in (0, 1)$, we have:*

$$\sup_{f \in \mathcal{F}} \left| \bar{U}_\Lambda(f) - \mathrm{L}_\Lambda^*(f) \right| \leqslant 2\mathfrak{R}_{\nu_*}(\mathcal{H}) + 8\mathcal{B} \sqrt{\frac{\ln 1/\Delta}{m - 2R}}, \tag{72}$$

*with probability of at least $1 - \Delta$ where $\nu_*$ is a distribution of tuples set defined as:*

$$\nu_* := \arg \max_{\nu \in \mathcal{V}} \mathfrak{R}_\nu(\mathcal{H}), \tag{73}$$

$$\mathcal{V} = \left\{ \bigotimes_{r=1}^{R} \left[ \mathcal{D}_r^{\otimes 3} \otimes \bar{\mathcal{D}}^{\otimes k-1} \right]^{\otimes \lfloor q_r/3 \rfloor} : q_r \in \mathbb{Z}_{\geqslant 0}, \forall r \in [R] \text{ and } \sum_{r=1}^{R} q_r = m \right\}. \tag{74}$$

Then, we also have the following result.

**Theorem E.11.** *Let $\mathcal{F}$ be a class of representation functions. For each $f \in \mathcal{F}$, let $\bar{U}_\Lambda(f)$ be defined in Eqn. (24) and $\mathrm{L}_\Lambda(f)$ be defined in Eqn. (10). Then, for any $\Delta \in (0, 1)$, we have:*

$$\sup_{f \in \mathcal{F}} \left| \bar{U}_\Lambda(f) - \mathrm{L}_\Lambda(f) \right| \tag{75}$$

$$\leqslant 2\mathfrak{R}_{\nu_*}(\mathcal{H}) + \mathcal{B} \left[ \frac{12R}{m - 2R} + \frac{8R}{3N} \ln 4R/\Delta + \sqrt{\frac{8(R-1) \ln 4R/\Delta}{3N}} + 8\sqrt{\frac{\ln 2/\Delta}{m - 2R}} \right],$$

*with probability of at least $1 - \Delta$.*

## F. Concentration of the Collision Probability Estimator

Our final ingredient for the proposed U-Statistic formulation is an estimator for the collision probability $\tau$ (Eqn. (12)). In this work, we propose the following plug-in estimator:

$$\widehat{\tau} := 1 - \sum_{r=1}^{R} \widehat{\rho}_r \left[1 - \widehat{\rho}_r\right]^k, \quad \forall r \in [R] : \widehat{\rho}_r = \frac{N_r}{N}. \tag{76}$$

**Proposition F.1.** *Let $\tau$ be the collision probability defined in Eqn. (12) and $\widehat{\tau}$ be the plug-in estimator in Eqn. (76). For any $\Delta \in (0, 1)$, we have:*

$$|\tau - \widehat{\tau}| \leqslant \frac{|R - (k+1)|}{\sqrt{R}} \left(1 - \frac{1}{R}\right)^{k-1} \left[ \frac{2 \ln(R+1)/\Delta}{3N} + \sqrt{\frac{2(1 - \|\rho\|_2^2) \ln(R+1)/\Delta}{N}} \right], \tag{77}$$

*with probability of at least $1 - \Delta$ where $\|\rho\|_2^2 = \sum_{r=1}^{R} \rho_r^2$.*

**Proof.** Let $\Phi : \mathbb{R}^R \to [0, 1]$ be defined as follows:

$$\Phi(q_1, \ldots, q_R) = 1 - \sum_{r=1}^{R} q_r \left[1 - q_r\right]^k.$$

Then, we have:

$$
\begin{aligned}
|\tau - \widehat{\tau}| &= |\Phi(\rho_1, \ldots, \rho_R) - \Phi(\widehat{\rho}_1, \ldots, \widehat{\rho}_R)| \\
&\leqslant \max_{\{q_r\}_{r=1}^{R} \in \Delta_R} \|\nabla\Phi(q_1, \ldots, q_R)\|_2 \cdot \|\mathbf{u} - \mathbb{E}[\mathbf{u}]\|_2,
\end{aligned}
$$

where $\mathbf{u} = \{\widehat{\rho}_1, \ldots, \widehat{\rho}_R\} \in \Delta_R$ is the vector of probabilities and $\Delta_R$ is defined as the probability simplex with $R$ probabilities:

$$\Delta_R := \left\{ \{q_r\}_{r=1}^{R} : q_r \geqslant 0, \forall r \in [R] \text{ and } \sum_{r=1}^{R} q_r = 1 \right\}. \tag{78}$$

First, we bound the $\ell^2$-norm of the Jacobian $\|\nabla\Phi(q_1, \ldots, q_R)\|_2$. For each $r \in [R]$, we have:

$$\frac{\partial\Phi}{\partial q_r} = kq_r(1 - q_r)^{k-1} - (1 - q_r)^k.$$

Now, let us solve the following constrained maximization problem:

$$\max_{\{q_r\}_{r=1}^{R} \in \Delta_R} \|\nabla\Phi(q_1, \ldots, q_R)\|_2^2 = \max_{\{q_r\}_{r=1}^{R} \in \Delta_R} \sum_{r=1}^{R} (1 - q_r)^{2(k-1)} \left[q_r(k+1) - 1\right]^2.$$

Then, for a Lagrange multiplier $\alpha > 0$, we have the following Lagrangian:

$$\mathcal{L}(q_1, \ldots, q_R; \alpha) = kq_r(1 - q_r)^{k-1} - (1 - q_r)^k + \alpha \left[\sum_{r=1}^{R} q_r - 1\right].$$

Therefore, for each $r \in [R]$, we have:

$$
\begin{aligned}
\frac{\partial\mathcal{L}(q_1, \ldots, q_R; \alpha)}{\partial q_r} &= \frac{\partial}{\partial q_r} \left[kq_r(1 - q_r)^{k-1} - (1 - q_r)^k\right] + \alpha \\
&= 2k(1 - q_r)^{k-1} - k(k-1)q_r(1 - q_r)^{k-2} + \alpha.
\end{aligned}
$$

Setting $\frac{\partial\mathcal{L}(q_1, \ldots, q_R; \alpha)}{\partial q_r} = 0$, we have $\alpha = k(k-1)q_r(1 - q_r)^{k-2} - 2k(1 - q_r)^{k-1}$ for all $r \in [R]$. Therefore, we have the optimal value of $\|\nabla\Phi(q_1, \ldots, q_R)\|_2^2$ when $q_1 = q_2 = \cdots = q_R = \frac{1}{R}$. As a result, for all set of probabilities $\{q_r\}_{r=1}^{R} \in \Delta_R$, we have:

$$
\begin{aligned}
\|\nabla\Phi(q_1, \ldots, q_R)\|_2^2 &\leqslant \sqrt{R\left(1 - \frac{1}{R}\right)^{2(k-1)}\left(1 - \frac{k+1}{R}\right)^2} \\
&= \frac{|R - (k+1)|}{\sqrt{R}}\left(1 - \frac{1}{R}\right)^{k-1}.
\end{aligned}
$$

Now, we analyze the concentration of $\mathbf{u} = \{\widehat{\rho}_1, \ldots, \widehat{\rho}_R\}$ around its mean. Let $\mathcal{B}_R = \{e_1, \ldots, e_R\}$ be the canonical basis of $\mathbb{R}^R$. Then, we can think of $\rho$ as a distribution over $\mathcal{B}_R$ such that for a random vector $X \sim \rho$, we have:

$$\forall r \in [R] : \mathbb{P}(X = e_r) = \rho_r.$$

Then, we can write $\mathbf{u} - \mathbb{E}[\mathbf{u}]$ as:

$$\mathbf{u} - \mathbb{E}[\mathbf{u}] = \frac{1}{N}\sum_{j=1}^{N}(X_j - \mathbb{E}[\mathbf{u}]) \quad \text{where} \quad X_j \overset{\text{i.i.d.}}{\sim} \rho, \forall j \in [R].$$

Let $Z_j = X_j - \mathbb{E}[\mathbf{u}]$ for all $j \in [N]$. Using the matrix Bernstein inequality (Proposition J.6), for all $\lambda > 0$, we have:

$$\mathbb{P}\left(\|\mathbf{u} - \mathbb{E}[\mathbf{u}]\|_2 \geqslant \lambda\right) = \mathbb{P}\left(\left\|\frac{1}{N}\sum_{j=1}^{N} Z_j\right\|_2 \geqslant \lambda\right) \leqslant (R+1)\exp\left(-\frac{N^2\lambda^2/2}{\sum_{j=1}^{N}\sigma_j^2 + N\lambda/3}\right),$$

where $\sigma_j^2 = \max\left(\|\mathbb{E}[Z_j Z_j^\top]\|_\sigma, \mathbb{E}[Z_j^\top Z_j]\right)$ for all $j \in [N]$. From here, we use the notation $\rho = \mathbb{E}[\mathbf{u}]$ for brevity. Now, for all $j \in [N]$ we have:

$$\mathbb{E}[Z_j^\top Z_j] = \mathbb{E}[\|Z_j\|_2^2] = \sum_{r=1}^{R}\rho_r\|e_r - \rho\|_2^2$$

$$= \sum_{r=1}^{R}\rho_r\left[(1-\rho_r)^2 + \sum_{q\neq r}^{R}\rho_q^2\right] = \sum_{r=1}^{R}\rho_r\left[(1-\rho_r)^2 + \|\rho\|_2^2 - \rho_r^2\right]$$

$$= \sum_{r=1}^{R}\rho_r\left[\|\rho\|_2^2 + 1 - 2\rho_r\right] = \|\rho\|_2^2 + 1 - 2\|\rho\|_2^2$$

$$= 1 - \|\rho\|_2^2.$$

Since the matrix $\mathbb{E}[Z_j Z_j^\top]$ is positive semi-definite, for all $j \in [N]$, we have:

$$\|\mathbb{E}[Z_j Z_j^\top]\|_\sigma \leqslant \text{trace}(\mathbb{E}[Z_j Z_j^\top]) = \text{trace}\left(\sum_{r=1}^{R}\rho_r(e_r - \rho)(e_r - \rho)^\top\right)$$

$$= \text{trace}\left(\text{diag}(\rho) - \rho\rho^\top\right) = 1 - \|\rho\|_2^2.$$

From the above, we have $\sigma_j^2 = 1 - \|\rho\|_2^2$ for all $j \in [N]$. As a result, we have:

$$\mathbb{P}\left(\left\|\frac{1}{N}\sum_{j=1}^{N} Z_j\right\|_2 \geqslant \lambda\right) \leqslant (R+1)\exp\left(-\frac{N^2\lambda^2/2}{N - N\|\rho\|_2^2 + N\lambda/3}\right)$$

$$= (R+1)\exp\left(-\frac{N\lambda^2/2}{1 - \|\rho\|_2^2 + \lambda/3}\right).$$

Setting the right-hand-side of the above inequality to $\Delta \in (0,1)$, we have:

$$\frac{N\lambda^2}{2(1 - \|\rho\|_2^2 + \lambda/3)} = \ln\frac{R+1}{\Delta} \implies N\lambda^2 - \frac{2\ln(R+1)/\Delta}{3}\lambda - 2(1 - \|\rho\|_2^2)\ln\frac{R+1}{\Delta} = 0.$$

Solving the above quadratic equation yields:

$$\lambda = \frac{\frac{2\ln(R+1)/\Delta}{3} + \sqrt{\frac{4\ln^2(R+1)/\Delta}{9} + 8N(1 - \|\rho\|_2^2)\ln\frac{R+1}{\Delta}}}{2N}$$

$$= \frac{\ln(R+1)/\Delta}{3N} + \frac{1}{N}\sqrt{\frac{\ln^2(R+1)/\Delta}{9} + 2N(1 - \|\rho\|_2^2)\ln\frac{R+1}{\Delta}}.$$

In other words, we have:

$$\|\mathbf{u} - \rho\|_2 \leqslant \frac{\ln(R+1)/\Delta}{3N} + \frac{1}{N}\sqrt{\frac{\ln^2(R+1)/\Delta}{9} + 2N(1 - \|\rho\|_2^2)\ln\frac{R+1}{\Delta}}$$

$$\leqslant \frac{\ln(R+1)/\Delta}{3N} + \frac{1}{N}\left[\frac{\ln(R+1)/\Delta}{3} + \sqrt{2N(1 - \|\rho\|_2^2)\ln\frac{R+1}{\Delta}}\right]$$

$$= \frac{2\ln(R+1)/\Delta}{3N} + \sqrt{\frac{2(1 - \|\rho\|_2^2)\ln(R+1)/\Delta}{N}},$$

with probability of $1 - \Delta$. Combine this with the bound on $\|\nabla \Phi(q_1, \ldots, q_R)\|_2$, we have:

$$
\begin{aligned}
|\tau - \widehat{\tau}| &\leqslant \frac{|R - (k+1)|}{\sqrt{R}} \left(1 - \frac{1}{R}\right)^{k-1} \cdot \|\mathbf{u} - \rho\|_2 \\
&\leqslant \frac{|R - (k+1)|}{\sqrt{R}} \left(1 - \frac{1}{R}\right)^{k-1} \left[\frac{2\ln(R+1)/\Delta}{3N} + \sqrt{\frac{2(1 - \|\rho\|_2^2)\ln(R+1)/\Delta}{N}}\right],
\end{aligned}
$$

with probability of at least $1 - \Delta$, as desired. $\qquad\square$

**Proposition F.2** (Multiplicative Concentration of $1 - \widehat{\tau}$). *Let $\tau$ be the collision probability defined in Eqn.* (12) *and $\widehat{\tau}$ be the plug-in estimator in Eqn.* (76). *For any $\Delta \in (0, 1)$, with probability of at least $1 - \Delta$:*

$$
1 - \widehat{\tau} \geqslant \frac{C}{2}\left[1 + \frac{1 - \gamma_{4k}}{\gamma_{4k}}\bar{C}\right]^{-1}(1 - \tau) \qquad \text{as long as } N \geqslant 8\gamma_{4k}^{-1}\ln\left(\frac{2}{\Delta}\right) + 12k\ln\left(\frac{2R}{\Delta}\right), \qquad (79)
$$

$$
1 - \tau \geqslant \frac{C}{2}\left[1 + \frac{1 - \widehat{\gamma}_{4k}}{\widehat{\gamma}_{4k}}\bar{C}\right]^{-1}(1 - \widehat{\tau}) \qquad \text{as long as } N \geqslant 3\gamma_{2k}^{-1}\ln\left(\frac{2}{\Delta}\right) + 16k\ln\left(\frac{2R}{\Delta}\right). \qquad (80)
$$

*with probability of at least $1 - \Delta$ where $\gamma_\alpha, \widehat{\gamma}_\alpha, C, \bar{C}$ (where $\alpha \geqslant 1$) are defined as follows:*

$$
\gamma_\alpha := \sum_{r \in [R]: \rho_r \leqslant \frac{1}{\alpha}} \rho_r, \qquad \widehat{\gamma}_\alpha := \sum_{r \in [R]: \widehat{\rho}_r \leqslant \frac{1}{\alpha}} \widehat{\rho}_r, \qquad C := \inf_{k \geqslant 1}\left(1 - \frac{1}{2k}\right)^k, \qquad \bar{C} := \sup_{k \geqslant 1}\left(1 - \frac{1}{4k}\right)^k.
$$

**Proof.** (Prove $1 - \tau \lesssim 1 - \widehat{\tau}$ whp.) We define the indices set $\Gamma$ and $\widehat{\Gamma}$ as follows:

$$
\Gamma := \left\{r \in [R] : \rho_r \leqslant \frac{1}{4k}\right\}, \qquad \widehat{\Gamma} := \left\{r \in [R] : \widehat{\rho}_r \leqslant \frac{1}{2k}\right\}.
$$

Then, by definition, we have $\gamma_{4k} := \sum_{r \in \Gamma} \rho_r$.

**Claim (i)**: For $\delta \in (0, 1)$, we have $\mathbb{P}(\Gamma \subseteq \widehat{\Gamma}) \geqslant 1 - \delta$ as long as $N \geqslant 12k\ln\left(\frac{R}{\delta}\right)$.

Fix one class $r \in \Gamma$. We can write $\widehat{\rho}_r = \frac{1}{N}\sum_{j=1}^N X_j$ where $X_j \sim \text{Bern}(\rho_r)$. Let $\{U_j\}_{j=1}^N$ be a sequence of uniform variables ($U_j \sim \text{Uniform}(0, 1), \forall j \in [N]$). Then, for each $j \in [N]$, we have $X_j \overset{d}{=} \mathbb{1}_{\{U_j \leqslant \rho_r\}}$ by coupling. Therefore, we can write $\widehat{\rho}_r$ as follows:

$$
\widehat{\rho}_r = \frac{1}{N}\sum_{j=1}^N X_j \overset{d}{=} \frac{1}{N}\sum_{j=1}^N \mathbb{1}_{\{U_j \leqslant \rho_r\}}.
$$

Then, we have:

$$
\begin{aligned}
\mathbb{P}\left(\widehat{\rho}_r \geqslant \frac{1}{2k}\right) &= \mathbb{P}\left(\frac{1}{N}\sum_{j=1}^N X_j \geqslant \frac{1}{2k}\right) = \mathbb{P}\left(\frac{1}{N}\sum_{j=1}^N \mathbb{1}_{\{U_j \leqslant \rho_r\}} \geqslant \frac{1}{2k}\right) \\
&= \mathbb{P}\left(\frac{\sharp\{j \in [N] : U_j \leqslant \rho_r\}}{N} \geqslant \frac{1}{2k}\right) \\
&\leqslant \mathbb{P}\left(\frac{\sharp\{j \in [N] : U_j \leqslant \frac{1}{4k}\}}{N} \geqslant \frac{1}{2k}\right) \qquad \left(\text{Since } \rho_r \leqslant \frac{1}{4k}\right) \\
&= \mathbb{P}\left(\frac{1}{N}\sum_{j=1}^N \mathbb{1}_{\{U_j \leqslant \frac{1}{4k}\}} \geqslant \frac{1}{2k}\right).
\end{aligned}
$$

For each $j \in [N]$, we have $\mathbb{1}_{\{U_j \leqslant \frac{1}{4k}\}} \sim \text{Bern}(\frac{1}{4k})$ (by coupling). Therefore, we can use the multiplicative Chernoff (Proposition J.7) bound as follows:

$$
\mathbb{P}\left(\widehat{\rho}_r \geqslant \frac{1}{2k}\right) \leqslant \mathbb{P}\left(\frac{1}{N}\sum_{j=1}^N \mathbb{1}_{\{U_j \leqslant \frac{1}{4k}\}} \geqslant \frac{1}{2k}\right) \leqslant \exp\left(-\frac{N}{12k}\right).
$$

Then, using the union bound, we have:

$$\mathbb{P}\left(\exists r \in \Gamma : \widehat{\rho}_r \geqslant \frac{1}{2k}\right) \leqslant |\Gamma| \exp\left(-\frac{N}{12k}\right) \leqslant R \exp\left(-\frac{N}{12k}\right).$$

Hence, if we let $Re^{-N/12k} \leqslant \delta$, then as long as $N \geqslant 12k \ln\left(\frac{R}{\delta}\right)$, we have $\widehat{\rho}_r \geqslant \frac{1}{2k}$ for all $r \in \Gamma$ wp. of at least $1 - \delta$. In other words, as long as $N \geqslant 12k \ln\left(\frac{R}{\delta}\right)$, $\Gamma \subseteq \widehat{\Gamma}$ wp. of at least $1 - \delta$.

**Claim (ii):** Conditionally given that $\Gamma \subseteq \widehat{\Gamma}$ and for any constant $\delta \in (0, 1)$, we have $1 - \widehat{\tau} \geqslant \frac{C}{2}\gamma_{4k}$ and $1 - \tau \leqslant \alpha\left[1 + \frac{1-\gamma_{4k}}{\gamma_{4k}}\bar{C}\right]$ with probability of at least $1 - \delta$ as long as $N \geqslant 8\gamma_{4k}^{-1} \ln\left(\frac{1}{\delta}\right)$.

Firstly, we note that given $\Gamma \subseteq \widehat{\Gamma}$, then for all $r \in \Gamma$, $\widehat{\rho}_r \leqslant \frac{1}{2k}$. Therefore:

$$\begin{aligned}
1 - \widehat{\tau} = \sum_{r=1}^{R} \widehat{\rho}_r (1 - \widehat{\rho}_r)^k &\geqslant \sum_{r \in \Gamma} \widehat{\rho}_r (1 - \widehat{\rho}_r)^k \\
&\geqslant \left(1 - \frac{1}{2k}\right)^k \sum_{r \in \Gamma} \widehat{\rho}_r \geqslant \inf_{k \geqslant 1}\left(1 - \frac{1}{2k}\right)^k \sum_{r \in \Gamma} \widehat{\rho}_r \quad \text{(Since } \Gamma \subseteq \widehat{\Gamma}) \\
&= C \sum_{r \in \Gamma} \widehat{\rho}_r.
\end{aligned}$$

Furthermore, we have:

$$\mathbb{P}\left(\sum_{r \in \Gamma} \widehat{\rho}_r \leqslant \frac{\gamma_{4k}}{2}\right) = \mathbb{P}\left(\sum_{r \in \Gamma} \widehat{\rho}_r \leqslant \frac{1}{2}\mathbb{E}\left[\sum_{r \in \Gamma} \widehat{\rho}_r\right]\right) \leqslant \exp\left(-\frac{N\gamma_{4k}}{8}\right) \quad \text{(Multiplicative Chernoff)}.$$

Then, if we let $e^{-N\gamma_{4k}/8} \leqslant \delta$, then we have $N \geqslant 8\gamma_{4k}^{-1} \ln\left(\frac{1}{\delta}\right)$. Therefore, as long as $N \geqslant 8\gamma_{4k}^{-1} \ln\left(\frac{1}{\delta}\right)$, we have $\sum_{r \in \Gamma} \widehat{\rho}_r \geqslant \frac{\gamma_{4k}}{2}$, making $1 - \widehat{\tau} \geqslant \frac{C}{2}\gamma_{4k}$, wp. of at least $1 - \delta$. Furthermore:

$$\begin{aligned}
1 - \tau = \sum_{r \in \Gamma} \rho_r (1 - \rho_r)^k + \sum_{\bar{r} \notin \Gamma} \rho_{\bar{r}} (1 - \rho_{\bar{r}})^k &\leqslant \gamma_{4k} + \sum_{\bar{r}:\rho_{\bar{r}} > \frac{1}{4k}} \rho_{\bar{r}} (1 - \rho_{\bar{r}})^k \quad ((1 - \rho_r)^k \leqslant 1, \forall r \in [R]) \\
&\leqslant \gamma_{4k} + \left(1 - \frac{1}{4k}\right)^k \sum_{\bar{r}:\rho_{\bar{r}} > \frac{1}{4k}} \rho_{\bar{r}} = \gamma_{4k} + (1 - \gamma_{4k})\left(1 - \frac{1}{4k}\right)^k \\
&\leqslant \gamma_{4k} + (1 - \gamma_{4k}) \sup_{k \geqslant 1}\left(1 - \frac{1}{4k}\right)^k = \gamma_{4k} + (1 - \gamma_{4k})\bar{C} \\
&= \gamma_{4k}\left[1 + \frac{1 - \gamma_{4k}}{\gamma_{4k}}\bar{C}\right].
\end{aligned}$$

**Combining claims (i) and (ii):** As long as we have $N \geqslant 8\gamma_{4k}^{-1} \ln\left(\frac{1}{\delta}\right) + 12k \ln\left(\frac{R}{\delta}\right)$, then $1 - \widehat{\tau} \geqslant \frac{C}{2}\gamma_{4k}$ and $\gamma_{4k} \geqslant (1 - \tau)\left[1 + \frac{1-\gamma_{4k}}{\gamma_{4k}}\bar{C}\right]^{-1}$ hold simultaneously with probability of at least $1 - 2\delta$. Therefore:

$$1 - \widehat{\tau} \geqslant \frac{C\gamma_{4k}}{2} \geqslant \frac{C}{2}\left[1 + \frac{1 - \gamma_{4k}}{\gamma_{4k}}\bar{C}\right]^{-1}(1 - \tau),$$

with probability of at least $1 - 2\delta$. Setting $\delta = \Delta/2$ yields the desired sample complexity. $\qquad\square$

**Proof.** (Prove $1 - \widehat{\tau} \lesssim 1 - \tau$ whp.) Now, we define the following indices sets:

$$\Gamma_c := \left\{r \in [R] : \rho_r \geqslant \frac{1}{2k}\right\}, \qquad \widehat{\Gamma}_c := \left\{r \in [R] : \widehat{\rho}_r \geqslant \frac{1}{4k}\right\}.$$

Then, by definition, $\widehat{\gamma}_{4k} := \sum_{r \notin \widehat{\Gamma}_c} \widehat{\rho}_r$ and $\gamma_{2k} = \sum_{r \notin \Gamma_c} \rho_r$.

**Claim (i)**: For $\delta \in (0,1)$, $\mathbb{P}(\Gamma_c \subseteq \widehat{\Gamma}_c) \geqslant 1 - \delta$ as long as $N \geqslant 16k \ln(\frac{R}{\delta})$. Fix $r \in \Gamma_c$, then we have $\rho_r \geqslant \frac{1}{2k}$. Then, we can write $\widehat{\rho}_r = \frac{1}{N} \sum_{j=1}^{N} X_j$ where $X_j \sim \text{Bern}(\rho_r)$ for all $j \in [N]$. Using the same coupling strategy as Proposition F.2, i.e., writing $\widehat{\rho}_r \overset{d}{=} \frac{1}{N} \sum_{j=1}^{N} \mathbb{1}_{\{U_j \leqslant \rho_r\}}$, we have:

$$\mathbb{P}\left(\widehat{\rho}_r \leqslant \frac{1}{4k}\right) = \mathbb{P}\left(\frac{1}{N} \sum_{j=1}^{N} X_j \leqslant \frac{1}{4k}\right) = \mathbb{P}\left(\frac{1}{N} \sum_{j=1}^{N} \mathbb{1}_{\{U_j \leqslant \rho_r\}} \leqslant \frac{1}{4k}\right)$$

$$\leqslant \mathbb{P}\left(\frac{1}{N} \sum_{j=1}^{N} \mathbb{1}_{\{U_j \leqslant \frac{1}{2k}\}} \leqslant \frac{1}{4k}\right) \qquad \text{(By coupling)}$$

$$\leqslant \exp\left(-\frac{N}{16k}\right) \qquad \text{(Multiplicative Chernoff)}.$$

Therefore, by the union bound:

$$\mathbb{P}\left(\exists r \in \Gamma_c : \widehat{\rho}_r \leqslant \frac{1}{4k}\right) \leqslant |\Gamma_c| \exp\left(-\frac{N}{16k}\right) \leqslant R \exp\left(-\frac{N}{16k}\right).$$

Therefore, if $Re^{-N/16k} \leqslant \delta$ then as long as $N \geqslant 16k \ln\left(\frac{R}{\delta}\right)$, we have $\Gamma_c \subseteq \widehat{\Gamma}_c$ wp. of at least $1 - \delta$. **Claim (ii)**: Conditionally given $\Gamma_c \subseteq \widehat{\Gamma}_c$, for any $\delta \in (0,1)$, we have $1 - \widehat{\tau} \leqslant \widehat{\gamma}_{4k}\left[1 + \frac{1 - \widehat{\gamma}_{4k}}{\widehat{\gamma}_{4k}} \bar{C}\right]$ and $1 - \tau \geqslant \frac{C}{2} \widehat{\gamma}_{4k}$ wp. of at least $1 - \delta$ as long as $N \geqslant 3\gamma_{2k}^{-1} \ln\left(\frac{1}{\delta}\right)$. We note that:

$$1 - \tau = \sum_{r=1}^{R} \rho_r (1 - \rho_r)^k \geqslant \sum_{r \notin \Gamma_c} \rho_r (1 - \rho_r)^k$$

$$\geqslant \left(1 - \frac{1}{2k}\right)^k \sum_{r \notin \Gamma_c} \rho_r \geqslant \inf_{k \geqslant 1} \left(1 - \frac{1}{2k}\right)^k \sum_{r \notin \Gamma_c} \rho_r$$

$$= C \sum_{r \notin \Gamma_c} \rho_r = C\gamma_{2k}.$$

By the multiplicative Chernoff bound (Proposition J.7), we have:

$$\mathbb{P}\left(\gamma_{2k} \leqslant \frac{1}{2} \sum_{r \notin \Gamma_c} \widehat{\rho}_r\right) = \mathbb{P}\left(\sum_{r \notin \Gamma_c} \widehat{\rho}_r \geqslant 2\mathbb{E}\left[\sum_{r \notin \Gamma_c} \widehat{\rho}_r\right]\right) \leqslant \exp\left(-\frac{N\gamma_{2k}}{3}\right).$$

Therefore, as long as $N \geqslant 3\gamma_{2k}^{-1} \ln\left(\frac{1}{\delta}\right)$, we have $\gamma_{2k} \geqslant \frac{1}{2} \sum_{r \notin \Gamma_c} \widehat{\rho}_r$ and:

$$1 - \tau \geqslant \frac{C}{2} \sum_{r \notin \Gamma_c} \widehat{\rho}_r \geqslant \frac{C}{2} \sum_{r \notin \widehat{\Gamma}_c} \widehat{\rho}_r = \frac{C}{2} \widehat{\gamma}_{4k} \qquad ([R] \setminus \Gamma_c \supseteq [R] \setminus \widehat{\Gamma}_c).$$

Furthermore, we have:

$$1 - \widehat{\tau} = \sum_{r \notin \widehat{\Gamma}_c} \widehat{\rho}_r (1 - \widehat{\rho}_r)^k + \sum_{\bar{r} \in \widehat{\Gamma}_c} \widehat{\rho}_{\bar{r}} (1 - \widehat{\rho}_{\bar{r}})^k$$

$$\leqslant \sum_{r \notin \widehat{\Gamma}_c} \widehat{\rho}_r + \left(1 - \frac{1}{4k}\right)^k \sum_{\bar{r} \in \widehat{\Gamma}_c} \widehat{\rho}_{\bar{r}}$$

$$= \widehat{\gamma}_{4k} + \left(1 - \frac{1}{4k}\right)^k (1 - \widehat{\gamma}_{4k}) \qquad \left(\widehat{\gamma}_{4k} := \sum_{r \notin \widehat{\Gamma}_c} \widehat{\rho}_r\right)$$

$$\leqslant \widehat{\gamma}_{4k} + (1 - \widehat{\gamma}_{4k}) \sup_{k \geqslant 1} \left(1 - \frac{1}{4k}\right)^k \qquad \left(\bar{C} := \sup_{k \geqslant 1} \left(1 - \frac{1}{4k}\right)^k\right)$$

$$= \widehat{\gamma}_{4k} \left[1 + \frac{1 - \widehat{\gamma}_{4k}}{\widehat{\gamma}_{4k}} \bar{C}\right].$$

Combining claims **(i)** and **(ii)** via the union bound: As long as we have $N \geqslant 3\gamma_{2k}^{-1} \ln\left(\frac{1}{\delta}\right) + 16k \ln\left(\frac{R}{\delta}\right)$, then $1 - \tau \geqslant \frac{C}{2}\widehat{\gamma}_{4k}$ and $1 - \widehat{\tau} \leqslant \widehat{\gamma}_{4k}\left[1 + \frac{1-\widehat{\gamma}_{4k}}{\widehat{\gamma}_{4k}}\bar{C}\right]$ holds simultaneously with probability of at least $1 - 2\delta$. Therefore, we have:

$$1 - \tau \geqslant \frac{C}{2}\widehat{\gamma}_{4k} \geqslant \frac{C}{2}\left[1 + \frac{1-\widehat{\gamma}_{4k}}{\widehat{\gamma}_{4k}}\bar{C}\right]^{-1}(1 - \widehat{\tau}),$$

with probability of at least $1 - 2\delta$. Setting $\delta = \Delta/2$ yields the desired sample complexity. $\qquad \square$

**Remark F.3.** From the above result, if we have $\gamma_{4k} \in \mathcal{O}(1)$, then as long as $N \geqslant \mathcal{O}\left(k \ln\left(\frac{R}{\delta}\right)\right)$, we have a "well-behaved" multiplicative control for $1 - \widehat{\tau}$. This means that if the probability $\gamma_{4k}$ is not vanishingly small, meaning the majority of the classes have small probabilities, the above result is sufficient. In practical extreme multi-class settings, this is a reasonable assumption especially when $k \ll R$. However, if we want to get rid of the prior assumption that the majority of the classes are small, we can rely on the result that follows.

**Proposition F.4** (Multiplicative Concentration of $1 - \widehat{\tau}$). *Let $\tau$ be the collision probability defined in Eqn.* (12) *and $\widehat{\tau}$ be the plug-in estimator in Eqn.* (76). *Suppose $\sum_{r:\rho_r \leqslant \frac{1}{4k^2}} \rho_r(1 - \rho_r)^k \leqslant \frac{1-\tau}{2}$. Then, for any $\Delta \in (0,1)$, with probability of at least $1 - \Delta$, we have:*

$$\text{When } \rho_{\max} \leqslant \frac{1}{2} : 1 - \widehat{\tau} \geqslant \frac{1-\tau}{4} \quad \text{as long as } N \geqslant 132k^2 \ln\left(\frac{2R}{\Delta}\right). \tag{81}$$

$$\text{When } \rho_{\max} > \frac{1}{2} : 1 - \widehat{\tau} \geqslant \frac{1-\tau}{8} \quad \text{as long as } N \geqslant 164k^2 \ln\left(\frac{4R}{\Delta}\right) + \frac{25\ln\left(\frac{2}{\Delta}\right)}{1 - \rho_{\max}}. \tag{82}$$

**Proof.** (Prove $1 - \tau \lesssim 1 - \widehat{\tau}$) We define the set $\bar{\Gamma}_*, \Gamma \subseteq [R]$ as follows:

$$\bar{\Gamma}_* := \left\{r \in [R] : \frac{1}{4k^2} < \rho_r \leqslant \frac{1}{2}\right\}, \qquad \Gamma := \left\{r \in [R] : \rho_r \leqslant \frac{1}{4k^2}\right\}.$$

We first make the observation that $\Gamma \cup \bar{\Gamma}_* = [R]$ if $\rho_{\max} = \max_{r \in [R]} \rho_r \leqslant \frac{1}{2}$. Otherwise, we have $\Gamma \cup \bar{\Gamma}_* = [R] \setminus \{r_*\}$ where $r_* = \arg\max_{r \in [R]} \rho_r$.

**Claim (i):** For any $\delta \in (0,1)$, as long as we have $N \geqslant 32k^2 \ln\left(\frac{R}{\delta}\right)$, then $(1 - \widehat{\rho}_r)^k \geqslant \frac{1}{\sqrt{2}}(1 - \rho_r)^k$ with probability of at least $1 - \delta$ for all $r \in \bar{\Gamma}_*$. For all $r \in [R]$ such that $\rho_r \leqslant \frac{1}{2}$, we have:

$$\mathbb{P}\left([1 - \widehat{\rho}_r]^k \leqslant \frac{[1 - \rho_r]^k}{\sqrt{2}}\right) = \mathbb{P}\left([1 - \widehat{\rho}_r]^k \leqslant \left[2^{-\frac{1}{2k}}(1 - \rho_r)\right]^k\right) = \mathbb{P}\left(1 - \widehat{\rho}_r \leqslant 2^{-\frac{1}{2k}}(1 - \rho_r)\right)$$

$$\leqslant \exp\left(-\frac{1}{2}\left(1 - \frac{1}{2^{\frac{1}{2k}}}\right)^2 N(1 - \rho_r)\right) \qquad \text{(Multiplicative Chernoff)}$$

$$\leqslant \exp\left(-\frac{N}{4}\left(1 - \frac{1}{2^{\frac{1}{2k}}}\right)^2\right) \qquad \left(\text{Since } \rho_r \leqslant \frac{1}{2}\right)$$

$$\leqslant \exp\left(-\frac{N}{4}\left[\frac{1}{2}\left(\frac{1}{2k}\right)\right]^2\right) = \exp\left(-\frac{N}{32k^2}\right).$$

Therefore, by the union bound:

$$\mathbb{P}\left(\exists r \in \bar{\Gamma}_* : [1 - \widehat{\rho}_r]^k \leqslant \frac{[1 - \rho_r]^k}{\sqrt{2}}\right) \leqslant |\bar{\Gamma}_*| \exp\left(-\frac{N}{32k^2}\right) \leqslant R \exp\left(-\frac{N}{32k^2}\right).$$

Hence, if $\delta \geqslant Re^{-N/32k^2}$ then as long as $N \geqslant 32k^2 \ln\left(\frac{R}{\delta}\right)$, we have $[1 - \widehat{\rho}_r]^k \geqslant \frac{[1 - \rho_r]^k}{\sqrt{2}}$ for all $r \in \bar{\Gamma}_*$.

**Claim (ii):** For any $\delta \in (0,1)$, as long as $N \geqslant 100k^2 \ln\left(\frac{R}{\delta}\right)$, then $\widehat{\rho}_r \geqslant \frac{\rho_r}{\sqrt{2}}$ with probability of at least $1 - \delta$ for all $r \in \bar{\Gamma}_*$.

By multiplicative Chernoff bound, we have:

$$\mathbb{P}\left(\widehat{\rho}_r \leqslant \frac{1}{\sqrt{2}}\rho_r\right) \leqslant \exp\left(-\frac{1}{2}\left(1-\frac{1}{\sqrt{2}}\right)^2 N\rho_r\right)$$

$$\leqslant \exp\left(-\frac{1}{8}\left(1-\frac{1}{\sqrt{2}}\right)^2 \frac{N}{k^2}\right) \qquad \left(\text{Since } \rho_r > \frac{1}{4k^2}\right)$$

$$\leqslant \exp\left(-\frac{N}{100k^2}\right).$$

Then, by the union bound:

$$\mathbb{P}\left(\exists r \in \bar{\Gamma}_* : \widehat{\rho}_r \leqslant \frac{\rho_r}{\sqrt{2}}\right) \leqslant |\bar{\Gamma}_*| \exp\left(-\frac{N}{100k^2}\right) \leqslant R \exp\left(-\frac{N}{100k^2}\right).$$

As a result, as long as $N \geqslant 100k^2 \ln\left(\frac{R}{\delta}\right)$, then $\widehat{\rho}_r \geqslant \frac{\rho_r}{\sqrt{2}}$ for all $r \in \bar{\Gamma}_*$.

**Case 1** ($\rho_{\max} \leqslant \frac{1}{2}$): In this case, we have $1 - \tau = \sum_{r \in \bar{\Gamma}_*} \rho_r(1-\rho_r)^k + \sum_{r \in \Gamma} \rho_r(1-\rho_r)^k$. Combining claims **(i)** and **(ii)**: as long as we have $N \geqslant 132k^2 \ln\left(\frac{R}{\delta}\right)$, with probability of at least $1 - 2\delta$, the following event holds:

$$1 - \widehat{\tau} \geqslant \sum_{r \in \bar{\Gamma}_*} \widehat{\rho}_r(1-\widehat{\rho}_r)^k \geqslant \frac{1}{2}\sum_{r \in \bar{\Gamma}_*} \rho_r(1-\rho_r)^k \qquad \text{(By claims (i), (ii))}$$

$$\geqslant \frac{1}{2}\left[(1-\tau) - \sum_{r \in \Gamma} \rho_r(1-\rho_r)^k\right]$$

$$\geqslant \frac{1}{4}(1-\tau) \qquad \left(\sum_{r \in \Gamma} \rho_r(1-\rho_r)^k \leqslant \frac{1-\tau}{2}\right).$$

Finally, setting $\delta = \Delta/2$ yields the desired sample complexity for $\rho_{\max} \leqslant \frac{1}{2}$ case.

**Case 2** ($\rho_{\max} > \frac{1}{2}$): Let us denote $r_* = \arg\max_{r \in [R]} \rho_r$. Using the strategy in claim **(ii)**, for all $r \in [R]$ such that $r \neq r_*$, we have:

$$\mathbb{P}\left([1-\widehat{\rho}_r]^k \leqslant \frac{\rho_{\max}^k}{\sqrt{2}}\right) \leqslant \mathbb{P}\left([1-\widehat{\rho}_r]^k \leqslant \frac{[1-\rho_r]^k}{\sqrt{2}}\right) \qquad (1 - \rho_r \geqslant \rho_{\max}, \forall r \neq r_*)$$

$$\leqslant \exp\left(-\frac{N}{32k^2}\right) \qquad \text{(Multiplicative Chernoff)}.$$

Therefore, by the union bound, we have:

$$\mathbb{P}\left(\exists r \neq r_* : [1-\widehat{\rho}_r]^k \leqslant \frac{\rho_{\max}^k}{\sqrt{2}}\right) \leqslant (R-1)\exp\left(-\frac{N}{32k^2}\right).$$

Hence, as long as $N \geqslant 32k^2 \ln\left(\frac{R-1}{\delta}\right)$, with probability of at least $1 - \delta$, we have:

$$\sum_{r \neq r_*} \widehat{\rho}_r(1-\widehat{\rho}_r)^k \geqslant \frac{1}{\sqrt{2}}\rho_{\max}^k \sum_{r \neq r_*} \widehat{\rho}_r = \frac{1}{\sqrt{2}}\rho_{\max}^k(1-\widehat{\rho}_{r_*}).$$

Using multiplicative Chernoff's bound again on $1 - \widehat{\rho}_{r_*}$, we have:

$$\mathbb{P}\left(1 - \widehat{\rho}_{r_*} \leqslant \frac{1-\rho_{\max}}{\sqrt{2}}\right) \leqslant \exp\left(-\frac{N}{2}\left(1-\frac{1}{\sqrt{2}}\right)^2(1-\rho_{\max})\right)$$

$$\leqslant \exp\left(-\frac{N(1-\rho_{\max})}{25}\right).$$

Therefore, as long as $N \geqslant 25(1 - \rho_{\max})^{-1} \ln\left(\frac{1}{\delta}\right)$, with probability of at least $1 - \delta$, we have $1 - \widehat{\rho}_{r_*} \geqslant \frac{1-\rho_{\max}}{\sqrt{2}}$. Combine this with the above, as long as $N \geqslant 32k^2 \ln\left(\frac{R-1}{\delta}\right) + 25(1 - \rho_{\max})^{-1} \ln\left(\frac{1}{\delta}\right)$, with probability of at least $1 - 2\delta$, we have:

$$\sum_{r \neq r_*} \widehat{\rho}_r (1 - \widehat{\rho}_r)^k \geqslant \frac{1}{2}\rho_{\max}^k(1 - \rho_{\max}) \geqslant \frac{1}{2}\rho_{\max}(1 - \rho_{\max})^k \qquad (1 - \rho_{\max} \leqslant \rho_{\max}).$$

Combine with claims **(i)**, **(ii)**, as long as we have $N \geqslant 132k^2 \ln\left(\frac{R}{\delta}\right) + 32k^2 \ln\left(\frac{R-1}{\delta}\right) + \frac{25 \ln\left(\frac{1}{\delta}\right)}{1-\rho_{\max}}$, with probability of at least $1 - 4\delta$, we have:

$$
\begin{aligned}
1 - \widehat{\tau} &\geqslant \frac{1}{2}\left[\frac{1}{2}\rho_{\max}(1 - \rho_{\max})^k + \frac{1}{2}\sum_{r \in \overline{\Gamma}_*} \rho_r(1 - \rho_r)^k\right] \\
&= \frac{1}{4}\left[\frac{1}{2}\rho_{\max}(1 - \rho_{\max})^k + \frac{1}{2}\sum_{r \in \overline{\Gamma}_*} \rho_r(1 - \rho_r)^k\right] \\
&\geqslant \frac{1}{8}(1 - \tau).
\end{aligned}
$$

Finally, setting $\delta = \Delta/4$ yields the desired sample complexity for case $\rho_{\max} > \frac{1}{2}$. $\qquad\square$

**Proposition F.5.** *Let $\tau$ be the collision probability defined in Eqn. (12) and $\widehat{\tau}$ be the plug-in estimator in Eqn. (76). Suppose $\sum_{r:\rho_r \leqslant \frac{1}{4k^2}} \rho_r(1 - \rho_r)^k \geqslant \frac{1-\tau}{2}$. Then, for any $\Delta \in (0,1)$, with probability of at least $1 - \Delta$:*

$$1 - \widehat{\tau} \geqslant \frac{1 - \tau}{8} \quad \text{as long as } N \geqslant 12k^2 \ln\left(\frac{2R}{\Delta}\right) + \frac{16 \ln\left(\frac{2}{\Delta}\right)}{1 - \tau}. \tag{83}$$

**Proof.** Let us define the following indices sets:

$$\Gamma := \left\{r \in [R] : \rho_r \leqslant \frac{1}{4k^2}\right\}, \qquad \widehat{\Gamma} := \left\{r \in [R] : \widehat{\rho}_r \leqslant \frac{1}{2k^2}\right\}.$$

Then, by the initial assumption, we have $\sum_{r \in \Gamma} \rho_r(1 - \rho_r)^k \geqslant \frac{1-\tau}{2}$.

**Claim**: For any $\delta \in (0,1)$, $\mathbb{P}(\Gamma \subseteq \widehat{\Gamma}) \geqslant 1 - \delta$ as long as $N \geqslant 12k^2 \ln\left(\frac{R}{\delta}\right)$.

Let us fix any $r \in \Gamma$. Then, we can write $\widehat{\rho}_r = \frac{1}{N}\sum_{j=1}^N X_j$ where $X_j \sim \text{Bern}(\rho_r)$. Then, by coupling, we have $X_j \stackrel{d}{=} \mathbb{1}_{\{U_j \leqslant \rho_r\}}$ where $U_j \sim \text{Uniform}(0,1)$ for all $j \in [N]$. Therefore:

$$
\begin{aligned}
\mathbb{P}\left(\widehat{\rho}_r \geqslant \frac{1}{2k^2}\right) = \mathbb{P}\left(\frac{1}{N}\sum_{j=1}^N X_j \geqslant \frac{1}{2k^2}\right) &= \mathbb{P}\left(\frac{1}{N}\sum_{j=1}^N \mathbb{1}_{\{U_j \leqslant \rho_r\}} \geqslant \frac{1}{2k^2}\right) \\
&\leqslant \mathbb{P}\left(\frac{1}{N}\sum_{j=1}^N \mathbb{1}_{\{U_j \leqslant \frac{1}{4k^2}\}} \geqslant \frac{1}{2k^2}\right) \quad \text{(By coupling)} \\
&\leqslant \exp\left(-\frac{N}{12k^2}\right) \quad \text{(Multiplicative Chernoff)}.
\end{aligned}
$$

By the union bound:

$$\mathbb{P}\left(\exists r \in \Gamma : \widehat{\rho}_r \geqslant \frac{1}{2k^2}\right) \leqslant |\Gamma| \exp\left(-\frac{N}{12k^2}\right) \leqslant R\exp\left(-\frac{N}{12k^2}\right).$$

As a result, if we let $\delta \geqslant Re^{-N/12k^2}$ then as long as $N \geqslant 12k^2 \ln\left(\frac{R}{\delta}\right)$, we have $\widehat{\rho}_r \leqslant \frac{1}{2k^2}, \forall r \in \Gamma$ with probability of at least $1 - \delta$. In other words, $\Gamma \subseteq \widehat{\Gamma}$ with probability of at least $1 - \delta$.

From the above claim, under the high probability event that $\Gamma \subseteq \widehat{\Gamma}$, we have:

$$1 - \widehat{\tau} \geqslant \sum_{r \in \widehat{\Gamma}} \widehat{\rho}_r (1 - \widehat{\rho}_r)^k \geqslant \left(1 - \frac{1}{2k^2}\right)^k \sum_{r \in \widehat{\Gamma}} \widehat{\rho}_r$$

$$\geqslant \left(1 - \frac{1}{2k^2}\right)^k \sum_{r \in \Gamma} \widehat{\rho}_r.$$

Now, by the multiplicative Chernoff bound:

$$\mathbb{P}\left(\sum_{r \in \Gamma} \widehat{\rho}_r \leqslant \frac{1}{2} \sum_{r \in \Gamma} \rho_r\right) \leqslant \exp\left(-\frac{N}{8} \sum_{r \in \Gamma} \rho_r\right) \leqslant \exp\left(-\frac{N}{8} \sum_{r \in \Gamma} \rho_r (1 - \rho_r)^k\right)$$

$$\leqslant \exp\left(-\frac{N(1 - \tau)}{16}\right).$$

Hence, if we let $\delta \geqslant e^{-N(1-\tau)/16}$ then as long as $N \geqslant 16(1-\tau)^{-1} \ln\left(\frac{1}{\delta}\right)$ we have $\sum_{r \in \Gamma} \widehat{\rho}_r \geqslant \frac{1}{2} \sum_{r \in \Gamma} \rho_r$ with probability of at least $1 - \delta$. Combining the above with **claim (i)**, as long as we have $N \geqslant 12k^2 \ln\left(\frac{R}{\delta}\right) + 16(1-\tau)^{-1} \ln\left(\frac{1}{\delta}\right)$, we have:

$$1 - \widehat{\tau} \geqslant \left(1 - \frac{1}{2k^2}\right)^k \sum_{r \in \Gamma} \widehat{\rho}_r \geqslant \frac{1}{2}\left(1 - \frac{1}{2k^2}\right)^k \sum_{r \in \Gamma} \rho_r \qquad \left(\sum_{r \in \Gamma} \widehat{\rho}_r \geqslant \frac{1}{2} \sum_{r \in \Gamma} \rho_r \text{ wp. } \geqslant 1 - \delta\right)$$

$$\geqslant \frac{1}{2}\left(1 - \frac{1}{2k^2}\right)^k \sum_{r \in \Gamma} \rho_r (1 - \rho_r)^k$$

$$\geqslant \frac{1 - \tau}{4}\left(1 - \frac{1}{2k^2}\right)^k$$

$$\geqslant \frac{1 - \tau}{8},$$

with probability of at least $1 - 2\delta$. Setting $\delta = \Delta/2$ yields the desired sample complexity. $\qquad \square$

**Remark F.6.** In Table 7, we summarize the multiplicative concentration results for the plug-in estimator $\widehat{\tau}$. We note that under the assumptions of Proposition F.4, we can easily show that $1 - \tau \lesssim 1 - \rho_{\max}$. Specifically, suppose $k \geqslant 2$, we have:

$$\frac{1 - \tau}{2} \leqslant \sum_{r:\rho_r > \frac{1}{4k^2}} \rho_r (1 - \rho_r)^k = \rho_{\max}(1 - \rho_{\max})^k + \sum_{r:\rho_r > \frac{1}{4k^2}, \rho_r \neq \rho_{\max}} \rho_r (1 - \rho_r)^k$$

$$\leqslant \frac{1}{4}(1 - \rho_{\max})^{k-1} + \sum_{r:\rho_r > \frac{1}{4k^2}, \rho_r \neq \rho_{\max}} \rho_r (1 - \rho_r)^k \qquad \left(x(1 - x) \leqslant \frac{1}{4}, \forall x \in [0, 1]\right)$$

$$\leqslant \frac{1}{4} \cdot \frac{1 - \rho_{\max}}{2^{k-2}} + \left(1 - \frac{1}{4k^2}\right)^k \sum_{r:\rho_r > \frac{1}{4k^2}, \rho_r \neq \rho_{\max}} \rho_r \qquad \left(\text{Since } \rho_{\max} \geqslant \frac{1}{2}\right)$$

$$\leqslant \frac{1}{2^k}(1 - \rho_{\max}) + \left(1 - \frac{1}{4k^2}\right)^k (1 - \rho_{\max})$$

$$= (1 - \rho_{\max})\left[\frac{1}{2^k} + \left(1 - \frac{1}{4k^2}\right)^k\right].$$

When $k = 1$ then we have $\rho_{\max}(1 - \rho_{\max})^k = \rho_{\max}(1 - \rho_{\max}) \leqslant 1 - \rho_{\max}$. Therefore, for any value of $k \geqslant 1$, we have the following general upper bound:

$$\frac{1 - \tau}{2} \leqslant (1 - \rho_{\max})\left[\frac{1}{2^{k-1}} + \left(1 - \frac{1}{4k^2}\right)^k\right] \leqslant 2(1 - \rho_{\max}).$$

*Table 7.* Multiplicative concentration results of the form $\mathbb{P}\left(1 - \widehat{\tau} \geqslant c(1 - \tau)\right) \geqslant 1 - \Delta$.

| Prop. | Constant ($c$) | Assumption | Sample Complexity |
|---|---|---|---|
| F.2 | $\frac{C}{2}\left[1 + \frac{1-\gamma_{4k}}{\gamma_{4k}}\bar{C}\right]^{-1}$ | N/A | $\mathcal{O}\left(\gamma_{4k}^{-1}\ln\left(\frac{1}{\Delta}\right) + k\ln\left(\frac{R}{\Delta}\right)\right)$ |
| F.7 | $\frac{1}{8}$ | N/A | $\mathcal{O}\left(k^2\ln\left(\frac{R}{\Delta}\right) + \frac{\ln\left(\frac{1}{\Delta}\right)}{1-\tau}\right)$ |
| F.4 | $\frac{1}{8}$ | $\sum_{r:\rho_r \leqslant \frac{1}{4k^2}}\rho_r(1-\rho_r)^k \leqslant \frac{1-\tau}{2}$ | $\mathcal{O}\left(k^2\ln\left(\frac{R}{\Delta}\right) + \frac{\ln\left(\frac{1}{\Delta}\right)}{1-\rho_{\max}}\right)$ |
| F.5 | $\frac{1}{8}$ | $\sum_{r:\rho_r \leqslant \frac{1}{4k^2}}\rho_r(1-\rho_r)^k \geqslant \frac{1-\tau}{2}$ | $\mathcal{O}\left(k^2\ln\left(\frac{R}{\Delta}\right) + \frac{\ln\left(\frac{1}{\Delta}\right)}{1-\tau}\right)$ |

Hence, **in all cases**, $1 - \tau \leqslant 4(1 - \rho_{\max})$ or $\frac{1}{1-\tau} \gtrsim \frac{1}{1-\rho_{\max}}$. Therefore, in both Proposition F.4 and Proposition F.5, the sample complexity can admit the following worst-case order:

$$N \geqslant \mathcal{O}\left(k^2\ln\left(\frac{R}{\Delta}\right) + \frac{\ln\left(\frac{1}{\Delta}\right)}{1-\tau}\right).$$

Therefore, we have the following final general result for multiplicative concentration of $1 - \widehat{\tau}$.

**Proposition F.7.** *Let $\tau$ be the collision probability defined in Eqn. (12) and $\widehat{\tau}$ be the plug-in estimator in Eqn. (76). Then, for any $\Delta \in (0, 1)$, with probability of at least $1 - \Delta$:*

$$1 - \widehat{\tau} \geqslant \frac{1-\tau}{8} \quad \text{as long as } N \geqslant 164k^2\ln\left(\frac{4R}{\Delta}\right) + \frac{100\ln\left(\frac{2}{\Delta}\right)}{1-\tau}. \tag{84}$$

**Proof.** Combining Proposition F.4 and Proposition F.5, we have $1 - \widehat{\tau} \geqslant \frac{1-\tau}{8}$ with probability of at least $1 - \Delta$ as long as $N$ is large enough to cover both cases when $\sum_{r:\rho_r \leqslant \frac{1}{4k^2}}\rho_r(1-\rho_r)^k \geqslant \frac{1-\tau}{2}$ and $\sum_{r:\rho_r \leqslant \frac{1}{4k^2}}\rho_r(1-\rho_r)^k \leqslant \frac{1-\tau}{2}$. Taking the max of minimum required sample complexity for both cases, we have:

$$\max\left\{164k^2\ln\left(\frac{4R}{\Delta}\right) + \frac{25\ln\left(\frac{2}{\Delta}\right)}{1-\rho_{\max}}, 12k^2\ln\left(\frac{2R}{\Delta}\right) + \frac{16\ln\left(\frac{2}{\Delta}\right)}{1-\tau}\right\}$$

$$\leqslant 164k^2\ln\left(\frac{4R}{\Delta}\right) + \max\left\{\frac{25\ln\left(\frac{2}{\Delta}\right)}{1-\rho_{\max}}, \frac{16\ln\left(\frac{2}{\Delta}\right)}{1-\tau}\right\}$$

$$\leqslant 164k^2\ln\left(\frac{4R}{\Delta}\right) + 25\ln\left(\frac{2}{\Delta}\right)\max\left\{\frac{1}{1-\rho_{\max}}, \frac{1}{1-\tau}\right\}$$

$$\leqslant 164k^2\ln\left(\frac{4R}{\Delta}\right) + \frac{100\ln\left(\frac{2}{\Delta}\right)}{1-\tau} \qquad (1 - \tau \leqslant 4(1 - \rho_{\max})).$$

Therefore, as long as $N \geqslant 164k^2\ln\left(\frac{4R}{\Delta}\right) + \frac{100\ln\left(\frac{2}{\Delta}\right)}{1-\tau}$, we have $1 - \widehat{\tau} \geqslant \frac{1-\tau}{8}$ with probability of at least $1 - \Delta$. $\qquad\square$

# G. Proof of the Main Results

**Remark G.1** (On the Boundedness of worst-case Rademacher complexities). Before presenting the proofs, we would like to make some remarks about the boundedness of $\mathfrak{R}_{\mu_*}(\mathcal{H})$ and $\mathfrak{R}_{\nu_*}(\mathcal{H})$, which are both important objects to the main results. Recall that $\mathfrak{R}_{\mu_*}(\mathcal{H})$ denotes the worst-case expected Rademacher complexity over the distribution set $\mathcal{U}$ defined as follows:

$$\mathcal{U} = \left\{\bigotimes_{r=1}^{R}\left[\mathcal{D}_r^{\otimes 2} \otimes \bar{\mathcal{D}}^{\otimes k}\right]^{\otimes \lfloor q_r/2\rfloor} : q_r \in \mathbb{Z}_{\geqslant 0}, \forall r \in [R] \text{ and } \sum_{r=1}^{R}q_r = n\right\}.$$

Essentially, $\mathcal{U}$ denotes the space of distributions over $(k+2)$-tuples collections with at least $n - R$ and at most $n$ tuples. For any $\mu \in \mathcal{U}$ and $S_{\text{tup}} \sim \mu$. Suppose that $|S_{\text{tup}}| = \widetilde{n}$, we have $n - R \leqslant \widetilde{n} \leqslant n$. By Dudley entropy integral (Theorem L.5):

$$
\begin{aligned}
\widehat{\mathfrak{R}}_{S_{\text{tup}}}(\mathcal{H}) &\leqslant \frac{4}{n} + \frac{12}{\sqrt{\widetilde{n}}} \int_{\frac{1}{n}}^{\mathcal{B}} \sqrt{\ln 2\mathcal{N}(\mathcal{H}, \epsilon, \mathrm{L}_2(S_{\text{tup}}))} d\epsilon \leqslant \frac{4}{n} + \frac{12}{\sqrt{n - R}} \int_{\frac{1}{n}}^{\mathcal{B}} \sqrt{\ln 2\mathcal{N}(\mathcal{H}, \epsilon, \mathrm{L}_2(S_{\text{tup}}))} d\epsilon \\
&\leqslant \frac{4}{n} + \frac{12}{\sqrt{n - R}} \sup_{S_{\text{tup}}^* \in (\mathcal{X}^{k+2})^{\widetilde{n}}} \int_{\frac{1}{n}}^{\mathcal{B}} \sqrt{\ln 2\mathcal{N}(\mathcal{H}, \epsilon, \mathrm{L}_2(S_{\text{tup}}^*))} d\epsilon \\
&\leqslant \frac{4}{n} + \frac{12}{\sqrt{n - R}} \sup_{S_{\text{tup}}^* \in (\mathcal{X}^{k+2})^n} \int_{\frac{1}{n}}^{\mathcal{B}} \sqrt{\ln 2\mathcal{N}(\mathcal{H}, \epsilon, \mathrm{L}_2(S_{\text{tup}}^*))} d\epsilon \\
&= \frac{4}{n} + \frac{12\mathfrak{C}_n(\mathcal{H})}{\sqrt{n - R}} \leqslant \frac{4}{n} + \frac{12\mathfrak{C}_N(\mathcal{H})}{\sqrt{n - R}}.
\end{aligned}
$$

Therefore, we have the data-independent bound as follows:

$$
\forall \mu \in \mathcal{U} : \mathfrak{R}_\mu(\mathcal{H}) = \mathbb{E}_{S_{\text{tup}} \sim \mu} \left[ \widehat{\mathfrak{R}}_{S_{\text{tup}}}(\mathcal{H}) \right] \leqslant \frac{4}{n} + \frac{12\mathfrak{C}_N(\mathcal{H})}{\sqrt{n - R}}, \tag{85}
$$

$$
\forall \nu \in \mathcal{V} : \mathfrak{R}_\nu(\mathcal{H}) = \mathbb{E}_{S_{\text{tup}} \sim \nu} \left[ \widehat{\mathfrak{R}}_{S_{\text{tup}}}(\mathcal{H}) \right] \leqslant \frac{4}{m} + \frac{12\mathfrak{C}_N(\mathcal{H})}{\sqrt{m - 2R}} \qquad \text{(Similar arguments)}. \tag{86}
$$

We can use the tighter complexity terms $\mathfrak{C}_n(\mathcal{H}), \mathfrak{C}_m(\mathcal{H})$ to bound $\mathfrak{R}_{\mu_*}(\mathcal{H})$ and $\mathfrak{R}_{\nu_*}(\mathcal{H})$. However, the improvement is at most logarithmic in $k$. Hence, we use $\mathfrak{C}_N(\mathcal{H})$ for both terms so that the subsequent results simplify elegantly.

**Remark G.2.** We note that all main results of this work can be made tighter via **worst-case Rademacher complexity**. Specifically, given a class $\mathcal{G}$ of functions $g : \mathcal{Z} \to \mathbb{R}$, the worst-case Rademacher complexity $\mathfrak{R}_m^{\text{wc}}(\mathcal{G})$ is defined as:

$$
\mathfrak{R}_m^{\text{wc}}(\mathcal{G}) := \sup_{S \in \mathcal{Z}^m} \widehat{\mathfrak{R}}_S(\mathcal{G}). \tag{87}
$$

Then, for any $\mu \in \mathcal{U}$ and $S_{\text{tup}} \sim \mu$, $|S_{\text{tup}}| = \widetilde{n}$, we have:

$$
\forall \mu \in \mathcal{U} : \mathfrak{R}_\mu(\mathcal{H}) = \mathbb{E}_{S_{\text{tup}} \sim \mu} \left[ \widehat{\mathfrak{R}}_{S_{\text{tup}}}(\mathcal{H}) \right] \leqslant \mathfrak{R}_{\widetilde{n}}^{\text{wc}}(\mathcal{H}) \leqslant \mathfrak{R}_{n-R}^{\text{wc}}(\mathcal{H}) \qquad \text{(By monotonicity)}. \tag{88}
$$

Then, the worst-case complexity $\mathfrak{R}_{n-R}^{\text{wc}}(\mathcal{H})$ can be bounded by metric entropy (as we do in this work) or using the contraction principle (Maurer, 2016; Foster & Rakhlin, 2019; Lei et al., 2023; 2026; Lei & Xie, 2026).

**Theorem G.3.** *Let $\mathcal{F}$ be a class of representation functions and let $\bar{U}_N(f)$ be defined for each $f \in \mathcal{F}$ in Eqn. (16). Then, for any $\Delta \in (0, 1)$, we have:*

$$
\begin{aligned}
\sup_{f \in \mathcal{F}} \left| \bar{U}_N(f) - \mathrm{L}_\phi(f) \right| &\leqslant \frac{2\mathfrak{R}_{\mu_*}(\mathcal{H}) + 2\widehat{\tau}\mathfrak{R}_{\nu_*}(\mathcal{H})}{1 - \widehat{\tau}} + \frac{8\mathcal{B}}{1 - \widehat{\tau}} \left[ \sqrt{\frac{\ln 12/\Delta}{n - R}} + \sqrt{\frac{\ln 12/\Delta}{m - 2R}} \right] \\
&+ \frac{2\mathcal{B}}{1 - \widehat{\tau}} \left[ \frac{3R}{n - R} + \frac{6R}{m - 2R} + \frac{8R}{3N} \ln\left(\frac{24R}{\Delta}\right) + \sqrt{\frac{8(R - 1)\ln\left(\frac{24R}{\Delta}\right)}{3N}} \right] \\
&+ \frac{4\mathcal{B}\Psi(R, k)}{C(1 - \widehat{\tau})^2} \left[ 1 + \frac{1 - \widehat{\gamma}_{4k}}{\widehat{\gamma}_{4k}} \bar{C} \right] \left[ \frac{2\ln\left(\frac{6(R+1)}{\Delta}\right)}{3N} + \sqrt{\frac{2(1 - \|\rho\|_2^2)\ln\left(\frac{6(R+1)}{\Delta}\right)}{N}} \right],
\end{aligned} \tag{89}
$$

*with probability of at least $1 - \Delta$ as long as $N \geqslant 3\gamma_{2k}^{-1} \ln\left(\frac{12}{\Delta}\right) + 16k \ln\left(\frac{12R}{\Delta}\right)$ where the constants $C, \bar{C}, \gamma_{2k}, \widehat{\gamma}_{4k}$ are defined in Proposition F.2 and $\Psi(R, k) = \frac{|R - (k+1)|}{\sqrt{R}} \left(1 - \frac{1}{R}\right)^{k-1}$. Furthermore, when $R \geqslant k$, as long as we have $N \geqslant k \times \max\left\{ \frac{3\gamma_{2k}^{-1} \ln(12/\Delta)}{k} + 16\ln(12R/\Delta), R \right\}$, we have:*

$$
\sup_{f \in \mathcal{F}} \left| \bar{U}_N(f) - \mathrm{L}_\phi(f) \right| \leqslant \mathcal{O}\left[ \frac{\mathfrak{C}_N(\mathcal{H})}{1 - \widehat{\tau}} \sqrt{\frac{k}{N}} + \frac{\mathcal{B}\sqrt{R}}{(1 - \widehat{\tau})^2} \left[ 1 + \frac{1 - \widehat{\gamma}_{4k}}{\widehat{\gamma}_{4k}} \right] \sqrt{\frac{(1 - \|\rho\|_2^2)\ln R/\Delta}{N}} \right], \tag{90}
$$

*with probability of at least $1 - \Delta$.*

**Proof.** We have:

$$
\sup_{f \in \mathcal{F}} \left| \bar{U}_N(f) - \mathcal{L}_\phi(f) \right| = \sup_{f \in \mathcal{F}} \left| \frac{1}{1 - \widehat{\tau}} \bar{U}_\Omega(f) - \frac{\widehat{\tau}}{1 - \widehat{\tau}} \bar{U}_\Lambda(f) - \frac{1}{1 - \tau} \mathcal{L}_\Omega(f) + \frac{\tau}{1 - \tau} \mathcal{L}_\Lambda(f) \right|
$$

$$
\leqslant \underbrace{\sup_{f \in \mathcal{F}} \left| \frac{1}{1 - \widehat{\tau}} \bar{U}_\Omega(f) - \frac{1}{1 - \tau} \mathcal{L}_\Omega(f) \right|}_{\alpha} + \underbrace{\sup_{f \in \mathcal{F}} \left| \frac{\widehat{\tau}}{1 - \widehat{\tau}} \bar{U}_\Lambda(f) - \frac{\tau}{1 - \tau} \mathcal{L}_\Lambda(f) \right|}_{\beta}.
$$

Now, we proceed to bound the $\alpha$ and $\beta$ terms. Firstly, we have:

$$
\alpha \leqslant \frac{1}{1 - \widehat{\tau}} \sup_{f \in \mathcal{F}} \left| \bar{U}_\Omega(f) - \mathcal{L}_\Omega(f) \right| + \sup_{f \in \mathcal{F}} \left| \mathcal{L}_\Omega(f) \right| \cdot \left| \frac{1}{1 - \widehat{\tau}} - \frac{1}{1 - \tau} \right|
$$

$$
\leqslant \frac{1}{1 - \widehat{\tau}} \sup_{f \in \mathcal{F}} \left| \bar{U}_\Omega(f) - \mathcal{L}_\Omega(f) \right| + \frac{\mathcal{B} |\widehat{\tau} - \tau|}{(1 - \tau)(1 - \widehat{\tau})}.
$$

By Proposition F.1 (absolute error of $\widehat{\tau}$) and Theorem E.7 (absolute error of $\bar{U}_\Omega(f)$), we have:

$$
\alpha \leqslant \frac{1}{1 - \widehat{\tau}} \left\{ 2 \mathfrak{R}_{\mu_*}(\mathcal{H}) + \mathcal{B} \left[ \frac{6R}{n - R} + \frac{8R}{3N} \ln 4R/\delta + \sqrt{\frac{8(R - 1) \ln 4R/\delta}{3N}} + 8 \sqrt{\frac{\ln 2/\delta}{n - R}} \right] \right\}
$$

$$
+ \frac{\mathcal{B} \Psi(R, k)}{(1 - \tau)(1 - \widehat{\tau})} \left[ \frac{2 \ln(R + 1)/\delta}{3N} + \sqrt{\frac{2(1 - \|\rho\|_2^2) \ln(R + 1)/\delta}{N}} \right],
$$

with probability of at least $1 - 2\delta$ (union bound) and $\Psi(R, k) = \frac{|R - (k+1)|}{\sqrt{R}} \left( 1 - \frac{1}{R} \right)^{k-1}$. Furthermore, by Proposition F.2 (relative error of $\widehat{\tau}$), we have $1 - \tau \geqslant \frac{C}{2} \left[ 1 + \frac{1 - \widehat{\gamma}_{4k}}{\widehat{\gamma}_{4k}} \bar{C} \right]^{-1} (1 - \widehat{\tau})$, with probability of at least $1 - \delta$ as long as $N \geqslant 3 \gamma_{2k}^{-1} \ln \left( \frac{2}{\delta} \right) + 16k \ln \left( \frac{2R}{\Delta} \right)$. As a result:

$$
\alpha \leqslant \frac{1}{1 - \widehat{\tau}} \left\{ 2 \mathfrak{R}_{\mu_*}(\mathcal{H}) + \mathcal{B} \left[ \frac{6R}{n - R} + \frac{8R}{3N} \ln 4R/\delta + \sqrt{\frac{8(R - 1) \ln 4R/\delta}{3N}} + 8 \sqrt{\frac{\ln 2/\delta}{n - R}} \right] \right\}
$$

$$
+ \frac{2 \mathcal{B} \Psi(R, k)}{C(1 - \widehat{\tau})^2} \left[ 1 + \frac{1 - \widehat{\gamma}_{4k}}{\widehat{\gamma}_{4k}} \bar{C} \right] \left[ \frac{2 \ln(R + 1)/\delta}{3N} + \sqrt{\frac{2(1 - \|\rho\|_2^2) \ln(R + 1)/\delta}{N}} \right],
$$

with probability of at least $1 - 3\delta$ (by union bound) as long as $N \geqslant 3 \gamma_{2k}^{-1} \ln \left( \frac{2}{\delta} \right) + 16k \ln \left( \frac{2R}{\delta} \right)$. Similarly, we can bound $\beta$ using the concentration result of $\bar{U}_\Lambda(f)$ in Theorem E.11 and Proposition F.1, Proposition F.2. Specifically, we have:

$$
\beta \leqslant \frac{\widehat{\tau}}{1 - \widehat{\tau}} \sup_{f \in \mathcal{F}} \left| \bar{U}_\Lambda(f) - \mathcal{L}_\Lambda(f) \right| + \frac{\mathcal{B} |\widehat{\tau} - \tau|}{(1 - \tau)(1 - \widehat{\tau})}
$$

$$
\leqslant \frac{\widehat{\tau}}{1 - \widehat{\tau}} \left\{ 2 \mathfrak{R}_{\nu_*}(\mathcal{H}) + \mathcal{B} \left[ \frac{12R}{m - 2R} + \frac{8R}{3N} \ln 4R/\delta + \sqrt{\frac{8(R - 1) \ln 4R/\delta}{3N}} + 8 \sqrt{\frac{\ln 2/\delta}{m - 2R}} \right] \right\}
$$

$$
+ \frac{2 \mathcal{B} \Psi(R, k)}{C(1 - \widehat{\tau})^2} \left[ 1 + \frac{1 - \widehat{\gamma}_{4k}}{\widehat{\gamma}_{4k}} \bar{C} \right] \left[ \frac{2 \ln(R + 1)/\delta}{3N} + \sqrt{\frac{2(1 - \|\rho\|_2^2) \ln(R + 1)/\delta}{N}} \right]
$$

$$
\leqslant \frac{2 \widehat{\tau} \mathfrak{R}_{\nu_*}(\mathcal{H})}{1 - \widehat{\tau}} + \frac{\mathcal{B}}{1 - \widehat{\tau}} \left[ \frac{12R}{m - 2R} + \frac{8R}{3N} \ln 4R/\delta + \sqrt{\frac{8(R - 1) \ln 4R/\delta}{3N}} + 8 \sqrt{\frac{\ln 2/\delta}{m - 2R}} \right] \quad (\widehat{\tau} \leqslant 1)
$$

$$
+ \frac{2 \mathcal{B} \Psi(R, k)}{C(1 - \widehat{\tau})^2} \left[ 1 + \frac{1 - \widehat{\gamma}_{4k}}{\widehat{\gamma}_{4k}} \bar{C} \right] \left[ \frac{2 \ln(R + 1)/\delta}{3N} + \sqrt{\frac{2(1 - \|\rho\|_2^2) \ln(R + 1)/\delta}{N}} \right],
$$

with probability of at least $1 - 3\delta$ as long as $N \geqslant 3\gamma_{2k}^{-1} \ln\left(\frac{2}{\delta}\right) + 16k \ln\left(\frac{2R}{\delta}\right)$. Combining the bounds of $\alpha$ and $\beta$, we have:

$$\sup_{f \in \mathcal{F}} \left|\bar{U}_N(f) - \mathrm{L}_\phi(f)\right| \leqslant \frac{2\mathfrak{R}_{\mu_*}(\mathcal{H}) + 2\widehat{\tau}\mathfrak{R}_{\nu_*}(\mathcal{H})}{1 - \widehat{\tau}} + \frac{8\mathcal{B}}{1 - \widehat{\tau}} \left[\sqrt{\frac{\ln 2/\delta}{n - R}} + \sqrt{\frac{\ln 2/\delta}{m - 2R}}\right]$$

$$+ \frac{2\mathcal{B}}{1 - \widehat{\tau}} \left[\frac{3R}{n - R} + \frac{6R}{m - 2R} + \frac{8R}{3N} \ln 4R/\delta + \sqrt{\frac{8(R - 1)\ln 4R/\delta}{3N}}\right]$$

$$+ \frac{4\mathcal{B}\Psi(R, k)}{C(1 - \widehat{\tau})^2} \left[1 + \frac{1 - \widehat{\gamma}_{4k}}{\widehat{\gamma}_{4k}}\bar{C}\right] \left[\frac{2\ln(R + 1)/\delta}{3N} + \sqrt{\frac{2(1 - \|\rho\|_2^2)\ln(R + 1)/\delta}{N}}\right]$$

$$\leqslant \mathcal{O}\left[\frac{\mathfrak{R}_{\mu_*}(\mathcal{H}) + \widehat{\tau}\mathfrak{R}_{\nu_*}(\mathcal{H})}{1 - \widehat{\tau}} + \frac{\mathcal{B}}{1 - \widehat{\tau}}\left[\sqrt{\frac{\ln 1/\delta}{n - R}} + \sqrt{\frac{\ln 1/\delta}{m - R}}\right]\right.$$

$$\left.+ \frac{\mathcal{B}\Psi(R, k)}{(1 - \widehat{\tau})^2}\left[1 + \frac{1 - \widehat{\gamma}_{4k}}{\widehat{\gamma}_{4k}}\right]\sqrt{\frac{(1 - \|\rho\|_2^2)\ln R/\delta}{N}}\right],$$

with probability of at least $1 - 6\delta$ as long as $N \geqslant 3\gamma_{2k}^{-1} \ln\left(\frac{2}{\delta}\right) + 16k \ln\left(\frac{2R}{\delta}\right)$. Setting $\delta = \Delta/6$ yields the desired complete bound. Furthermore, as long as $N \geqslant 2Rk$, we have:

$$\sqrt{\frac{1}{n - R}} = \sqrt{\frac{1}{2\lfloor\frac{N}{k}\rfloor - R}} \leqslant \sqrt{\frac{1}{N/k - R}} \leqslant \sqrt{\frac{k}{N - Rk}} \in \mathcal{O}\left(\sqrt{\frac{k}{N}}\right).$$

Similarly, we also have $\frac{1}{\sqrt{m - 2R}} \leqslant \mathcal{O}\left(\sqrt{\frac{k}{N}}\right)$. Therefore, as long as $N \geqslant k \times \max\left\{\frac{3\gamma_{2k}^{-1}\ln(12/\Delta)}{k} + 16\ln(12R/\Delta), R\right\}$ (so that both minimum requirements $N \geqslant 2Rk$ and $N \geqslant 3\gamma_{2k}^{-1}\ln\left(\frac{12}{\Delta}\right) + 16k\ln\left(\frac{12R}{\Delta}\right)$ are satisfied), we have:

$$\mathfrak{R}_{\mu_*}(\mathcal{H}) \leqslant \mathcal{O}\left[\mathfrak{C}_N(\mathcal{H})\sqrt{\frac{k}{N}}\right], \qquad \mathfrak{R}_{\nu_*}(\mathcal{H}) \leqslant \mathcal{O}\left[\mathfrak{C}_N(\mathcal{H})\sqrt{\frac{k}{N}}\right].$$

As a result, as long as $N \geqslant k \times \max\left\{\frac{3\gamma_{2k}^{-1}\ln(12/\Delta)}{k} + 16\ln(12R/\Delta), R\right\}$ and given that $R \geqslant k$, we have:

$$\sup_{f \in \mathcal{F}} \left|\bar{U}_N(f) - \mathrm{L}_\phi(f)\right| \leqslant \mathcal{O}\left[\frac{1 + \widehat{\tau}}{1 - \widehat{\tau}}\mathfrak{C}_N(\mathcal{H})\sqrt{\frac{k}{N}} + \frac{\mathcal{B}\sqrt{k}}{1 - \widehat{\tau}}\left[\sqrt{\frac{\ln 1/\Delta}{N}} + \sqrt{\frac{\ln 1/\Delta}{N}}\right]\right.$$

$$\left.+ \frac{\mathcal{B}\Psi(R, k)}{(1 - \widehat{\tau})^2}\left[1 + \frac{1 - \widehat{\gamma}_{4k}}{\widehat{\gamma}_{4k}}\right]\sqrt{\frac{(1 - \|\rho\|_2^2)\ln R/\Delta}{N}}\right]$$

$$\leqslant \mathcal{O}\left[\frac{\mathfrak{C}_N(\mathcal{H})}{1 - \widehat{\tau}}\sqrt{\frac{k}{N}} + \frac{\mathcal{B}\sqrt{R}}{(1 - \widehat{\tau})^2}\left[1 + \frac{1 - \widehat{\gamma}_{4k}}{\widehat{\gamma}_{4k}}\right]\sqrt{\frac{(1 - \|\rho\|_2^2)\ln R/\Delta}{N}}\right],$$

with probability of at least $1 - \Delta$, as desired. In the last inequality, we have $\Psi(R, k) \in \mathcal{O}(\sqrt{R})$ given that $R \geqslant k$. $\qquad \square$

**Theorem G.4** (cf. Table 1). *Let $\mathcal{F}$ be a class of representation functions and let $\bar{U}_N(f)$ be defined for each $f \in \mathcal{F}$ in Eqn. (16). Then, for any $\Delta \in (0, 1)$, we have:*

$$\sup_{f \in \mathcal{F}} \left|\bar{U}_N(f) - \mathrm{L}_\phi(f)\right| \leqslant \frac{2\mathfrak{R}_{\mu_*}(\mathcal{H}) + 2\tau\mathfrak{R}_{\nu_*}(\mathcal{H})}{1 - \tau} + \frac{8\mathcal{B}}{1 - \tau}\left[\sqrt{\frac{\ln 12/\Delta}{n - R}} + \sqrt{\frac{\ln 12/\Delta}{m - 2R}}\right]$$

$$+ \frac{2\mathcal{B}}{1 - \tau}\left[\frac{3R}{n - R} + \frac{6R}{m - 2R} + \frac{8R}{3N}\ln\left(\frac{24R}{\Delta}\right) + \sqrt{\frac{8(R - 1)\ln\left(\frac{24R}{\Delta}\right)}{3N}}\right] \qquad (91)$$

$$+ \frac{4\mathcal{B}\Psi(R, k)}{C(1 - \tau)^2}\left[1 + \frac{1 - \gamma_{4k}}{\gamma_{4k}}\bar{C}\right]\left[\frac{2\ln\left(\frac{6(R+1)}{\Delta}\right)}{3N} + \sqrt{\frac{2(1 - \|\rho\|_2^2)\ln\left(\frac{6(R+1)}{\Delta}\right)}{N}}\right],$$

*with probability of at least* $1 - \Delta$ *as long as* $N \geqslant 8\gamma_{4k}^{-1} \ln\left(\frac{12}{\Delta}\right) + 12k \ln\left(\frac{12R}{\Delta}\right)$ *where the constants* $C, \bar{C}, \gamma_{2k}, \gamma_{4k}$ *are defined in Proposition F.2 and* $\Psi(R, k) = \frac{|R - (k+1)|}{\sqrt{R}}\left(1 - \frac{1}{R}\right)^{k-1}$. *Furthermore, when* $R \geqslant k$, *as long as we have* $N \geqslant k \times \max\left\{\frac{8\gamma_{4k}^{-1} \ln(12/\Delta)}{k} + 12\ln(12R/\Delta), R\right\}$, *we have:*

$$\sup_{f \in \mathcal{F}} |\bar{U}_N(f) - \mathrm{L}_\phi(f)| \leqslant \mathcal{O}\left[\frac{\mathfrak{C}_N(\mathcal{H})}{1 - \tau}\sqrt{\frac{k}{N}} + \frac{\mathcal{B}\sqrt{R}}{(1 - \tau)^2}\left[1 + \frac{1 - \gamma_{4k}}{\gamma_{4k}}\right]\sqrt{\frac{(1 - \|\rho\|_2^2)\ln R/\Delta}{N}}\right], \tag{92}$$

*with probability of at least* $1 - \Delta$.

**Proof.** Similar to Theorem G.3, we have:

$$\sup_{f \in \mathcal{F}} \left|\bar{U}_N(f) - \mathrm{L}_\phi(f)\right| \leqslant \underbrace{\sup_{f \in \mathcal{F}}\left|\frac{1}{1 - \widehat{\tau}}\bar{U}_\Omega(f) - \frac{1}{1 - \tau}\mathrm{L}_\Omega(f)\right|}_{\alpha} + \underbrace{\sup_{f \in \mathcal{F}}\left|\frac{\widehat{\tau}}{1 - \widehat{\tau}}\bar{U}_\Lambda(f) - \frac{\tau}{1 - \tau}\mathrm{L}_\Lambda(f)\right|}_{\beta}.$$

Bounding the $\alpha$ term, we have:

$$\alpha \leqslant \frac{1}{1 - \tau}\sup_{f \in \mathcal{F}}\left|\bar{U}_\Omega(f) - \mathrm{L}_\Omega(f)\right| + \sup_{f \in \mathcal{F}}|\bar{U}_\Omega(f)| \cdot \left|\frac{1}{1 - \widehat{\tau}} - \frac{1}{1 - \tau}\right|$$

$$\leqslant \frac{1}{1 - \tau}\sup_{f \in \mathcal{F}}\left|\bar{U}_\Omega(f) - \mathrm{L}_\Omega(f)\right| + \frac{\mathcal{B}|\widehat{\tau} - \tau|}{(1 - \tau)(1 - \widehat{\tau})}.$$

Then, By Proposition F.1 and Theorem E.7, we have:

$$\alpha \leqslant \frac{1}{1 - \tau}\left\{2\mathfrak{R}_{\mu_*}(\mathcal{H}) + \mathcal{B}\left[\frac{6R}{n - R} + \frac{8R}{3N}\ln 4R/\delta + \sqrt{\frac{8(R - 1)\ln 4R/\delta}{3N}} + 8\sqrt{\frac{\ln 2/\delta}{n - R}}\right]\right\}$$

$$+ \frac{\mathcal{B}\Psi(R, k)}{(1 - \tau)(1 - \widehat{\tau})}\left[\frac{2\ln(R + 1)/\delta}{3N} + \sqrt{\frac{2(1 - \|\rho\|_2^2)\ln(R + 1)/\delta}{N}}\right],$$

with probability of at least $1 - 2\delta$ (union bound) and $\Psi(R, k) = \frac{|R - (k+1)|}{\sqrt{R}}\left(1 - \frac{1}{R}\right)^{k-1}$. Furthermore, by Proposition F.2 (relative error of $\widehat{\tau}$), we have $1 - \widehat{\tau} \geqslant \frac{C}{2}\left[1 + \frac{1 - \gamma_{4k}}{\gamma_{4k}}\bar{C}\right]^{-1}(1 - \tau)$ with probability of at least $1 - \delta$ as long as $N \geqslant 8\gamma_{4k}^{-1} \ln\left(\frac{2}{\delta}\right) + 12k\ln\left(\frac{2R}{\delta}\right)$. Therefore:

$$\alpha \leqslant \frac{1}{1 - \tau}\left\{2\mathfrak{R}_{\mu_*}(\mathcal{H}) + \mathcal{B}\left[\frac{6R}{n - R} + \frac{8R}{3N}\ln 4R/\delta + \sqrt{\frac{8(R - 1)\ln 4R/\delta}{3N}} + 8\sqrt{\frac{\ln 2/\delta}{n - R}}\right]\right\}$$

$$+ \frac{2\mathcal{B}\Psi(R, k)}{C(1 - \tau)^2}\left[1 + \frac{1 - \gamma_{4k}}{\gamma_{4k}}\bar{C}\right]\left[\frac{2\ln(R + 1)/\delta}{3N} + \sqrt{\frac{2(1 - \|\rho\|_2^2)\ln(R + 1)/\delta}{N}}\right],$$

with probability of at least $1 - 3\delta$ (by union bound) as long as $N \geqslant 8\gamma_{4k}^{-1}\ln\left(\frac{2}{\delta}\right) + 12k\ln\left(\frac{2R}{\delta}\right)$. Similarly, we have:

$$\beta \leqslant \frac{\tau}{1 - \tau}\left\{2\mathfrak{R}_{\nu_*}(\mathcal{H}) + \mathcal{B}\left[\frac{12R}{m - 2R} + \frac{8R}{3N}\ln 4R/\delta + \sqrt{\frac{8(R - 1)\ln 4R/\delta}{3N}} + 8\sqrt{\frac{\ln 2/\delta}{m - 2R}}\right]\right\}$$

$$+ \frac{2\mathcal{B}\Psi(R, k)}{C(1 - \tau)^2}\left[1 + \frac{1 - \gamma_{4k}}{\gamma_{4k}}\bar{C}\right]\left[\frac{2\ln(R + 1)/\delta}{3N} + \sqrt{\frac{2(1 - \|\rho\|_2^2)\ln(R + 1)/\delta}{N}}\right]$$

$$\leqslant \frac{2\tau\mathfrak{R}_{\nu_*}(\mathcal{H})}{1 - \tau} + \frac{\mathcal{B}}{1 - \tau}\left[\frac{12R}{m - 2R} + \frac{8R}{3N}\ln 4R/\delta + \sqrt{\frac{8(R - 1)\ln 4R/\delta}{3N}} + 8\sqrt{\frac{\ln 2/\delta}{m - 2R}}\right] \quad (\tau \leqslant 1)$$

$$+ \frac{2\mathcal{B}\Psi(R, k)}{C(1 - \tau)^2}\left[1 + \frac{1 - \gamma_{4k}}{\gamma_{4k}}\bar{C}\right]\left[\frac{2\ln(R + 1)/\delta}{3N} + \sqrt{\frac{2(1 - \|\rho\|_2^2)\ln(R + 1)/\delta}{N}}\right],$$

with probability of at least $1 - 3\delta$ as long as $N \geqslant 8\gamma_{4k}^{-1} \ln\left(\frac{2}{\delta}\right) + 12k \ln\left(\frac{2R}{\delta}\right)$. Combining the bounds on $\alpha, \beta$ via the union bound, we have:

$$\sup_{f \in \mathcal{F}} \left| \bar{U}_N(f) - \mathrm{L}_\phi(f) \right| \leqslant \frac{2\mathfrak{R}_{\mu_*}(\mathcal{H}) + 2\tau\mathfrak{R}_{\nu_*}(\mathcal{H})}{1 - \tau} + \frac{8\mathcal{B}}{1 - \tau} \left[ \sqrt{\frac{\ln 2/\delta}{n - R}} + \sqrt{\frac{\ln 2/\delta}{m - 2R}} \right]$$

$$+ \frac{2\mathcal{B}}{1 - \tau} \left[ \frac{3R}{n - R} + \frac{6R}{m - 2R} + \frac{8R}{3N} \ln 4R/\delta + \sqrt{\frac{8(R - 1)\ln 4R/\delta}{3N}} \right]$$

$$+ \frac{4\mathcal{B}\Psi(R, k)}{C(1 - \tau)^2} \left[ 1 + \frac{1 - \gamma_{4k}}{\gamma_{4k}}\bar{C} \right] \left[ \frac{2\ln(R + 1)/\delta}{3N} + \sqrt{\frac{2(1 - \|\rho\|_2^2)\ln(R + 1)/\delta}{N}} \right]$$

$$\leqslant \mathcal{O}\left[ \frac{\mathfrak{R}_{\mu_*}(\mathcal{H}) + \tau\mathfrak{R}_{\nu_*}(\mathcal{H})}{1 - \tau} + \frac{\mathcal{B}}{1 - \tau} \left[ \sqrt{\frac{\ln 1/\delta}{n - R}} + \sqrt{\frac{\ln 1/\delta}{m - R}} \right] \right.$$

$$\left. + \frac{\mathcal{B}\Psi(R, k)}{(1 - \tau)^2} \left[ 1 + \frac{1 - \gamma_{4k}}{\gamma_{4k}} \right] \sqrt{\frac{(1 - \|\rho\|_2^2)\ln R/\delta}{N}} \right],$$

with probability of at least $1 - 6\delta$ as long as $N \geqslant 8\gamma_{4k}^{-1} \ln\left(\frac{2}{\delta}\right) + 12k \ln\left(\frac{2R}{\delta}\right)$. Setting $\delta = \Delta/6$ yields the desired complete bound. Applying the same arguments to obtain $\mathcal{O}$-notation bound in Theorem G.3: When $R \geqslant k$, as long as $N \geqslant k \times \max\left\{ \frac{8\gamma_{4k}^{-1}\ln(12/\Delta)}{k} + 12\ln(12R/\Delta), R \right\}$, we have:

$$\sup_{f \in \mathcal{F}} |\bar{U}_N(f) - \mathrm{L}_\phi(f)| \leqslant \mathcal{O}\left[ \frac{\mathfrak{C}_N(\mathcal{H})}{1 - \tau}\sqrt{\frac{k}{N}} + \frac{\mathcal{B}\sqrt{R}}{(1 - \tau)^2} \left[ 1 + \frac{1 - \gamma_{4k}}{\gamma_{4k}} \right] \sqrt{\frac{(1 - \|\rho\|_2^2)\ln R/\Delta}{N}} \right],$$

with probability of at least $1 - \Delta$. $\qquad\square$

**Theorem G.5** (cf. Theorem 4.5, Table 1). *Let $\mathcal{F}$ be a class of representation functions and let $\bar{U}_N(f)$ be defined for each $f \in \mathcal{F}$ in Eqn. (16). Then, for any $\Delta \in (0, 1)$, we have:*

$$\sup_{f \in \mathcal{F}} \left| \bar{U}_N(f) - \mathrm{L}_\phi(f) \right| \leqslant \frac{2\mathfrak{R}_{\mu_*}(\mathcal{H}) + 2\tau\mathfrak{R}_{\nu_*}(\mathcal{H})}{1 - \tau} + \frac{8\mathcal{B}}{1 - \tau} \left[ \sqrt{\frac{\ln 12/\Delta}{n - R}} + \sqrt{\frac{\ln 12/\Delta}{m - 2R}} \right]$$

$$+ \frac{2\mathcal{B}}{1 - \tau} \left[ \frac{3R}{n - R} + \frac{6R}{m - 2R} + \frac{8R}{3N} \ln\left(\frac{24R}{\Delta}\right) + \sqrt{\frac{8(R - 1)\ln\left(\frac{24R}{\Delta}\right)}{3N}} \right] \qquad (93)$$

$$+ \frac{16\mathcal{B}\Psi(R, k)}{(1 - \tau)^2} \left[ \frac{2\ln\left(\frac{6(R+1)}{\Delta}\right)}{3N} + \sqrt{\frac{2(1 - \|\rho\|_2^2)\ln\left(\frac{6(R+1)}{\Delta}\right)}{N}} \right],$$

*with probability of at least $1 - \Delta$ as long as $N \geqslant 164k^2 \ln\left(\frac{24R}{\Delta}\right) + \frac{100\ln\left(\frac{12}{\Delta}\right)}{1 - \tau}$ where $\Psi(R, k) = \frac{|R - (k+1)|}{\sqrt{R}}\left(1 - \frac{1}{R}\right)^{k-1}$. Furthermore, when $R \geqslant k$, as long as $N \geqslant k \times \max\left\{ 164k \ln\left(\frac{24R}{\Delta}\right) + \frac{100\ln\left(\frac{12}{\Delta}\right)}{k(1 - \tau)}, R \right\}$, we have:*

$$\sup_{f \in \mathcal{F}} |\bar{U}_N(f) - \mathrm{L}_\phi(f)| \leqslant \mathcal{O}\left[ \frac{\mathfrak{C}_N(\mathcal{H})}{1 - \tau}\sqrt{\frac{k}{N}} + \frac{\mathcal{B}\sqrt{R}}{(1 - \tau)^2}\sqrt{\frac{(1 - \|\rho\|_2^2)\ln R/\Delta}{N}} \right],$$

*with probability of at least $1 - \Delta$.*

**Proof.** Let $\delta \in (0, 1)$. Suppose that the minimum sample complexity $N \geqslant 164k^2 \ln\left(\frac{4R}{\delta}\right) + \frac{100\ln\left(\frac{2}{\delta}\right)}{1 - \tau}$ is satisfied. Let us reuse the quantities $\alpha, \beta$ as defined in Theorem G.4:

$$\alpha = \sup_{f \in \mathcal{F}} \left| \frac{1}{1 - \hat{\tau}}\bar{U}_\Omega(f) - \frac{1}{1 - \tau}\mathrm{L}_\Omega(f) \right|, \qquad \beta = \sup_{f \in \mathcal{F}} \left| \frac{\hat{\tau}}{1 - \hat{\tau}}\bar{U}_\Lambda(f) - \frac{\tau}{1 - \tau}\mathrm{L}_\Lambda(f) \right|.$$

Combine Proposition F.1 (absolute error of $\widehat{\tau}$), Theorem E.7 (absolute error of $\bar{U}_\Omega(f)$) and Proposition F.7 (generic multiplicative error of $\widehat{\tau}$) via the union bound, we have:

$$
\begin{aligned}
\alpha &\leqslant \frac{1}{1-\tau}\left\{2\mathfrak{R}_{\mu_*}(\mathcal{H}) + \mathcal{B}\left[\frac{6R}{n-R} + \frac{8R}{3N}\ln 4R/\delta + \sqrt{\frac{8(R-1)\ln 4R/\delta}{3N}} + 8\sqrt{\frac{\ln 2/\delta}{n-R}}\right]\right\} \\
&\quad + \frac{\mathcal{B}\Psi(R,k)}{(1-\tau)(1-\widehat{\tau})}\left[\frac{2\ln(R+1)/\delta}{3N} + \sqrt{\frac{2(1-\|\rho\|_2^2)\ln(R+1)/\delta}{N}}\right] \\
&\leqslant \frac{1}{1-\tau}\left\{2\mathfrak{R}_{\mu_*}(\mathcal{H}) + \mathcal{B}\left[\frac{6R}{n-R} + \frac{8R}{3N}\ln 4R/\delta + \sqrt{\frac{8(R-1)\ln 4R/\delta}{3N}} + 8\sqrt{\frac{\ln 2/\delta}{n-R}}\right]\right\} \\
&\quad + \frac{8\mathcal{B}\Psi(R,k)}{(1-\tau)^2}\left[\frac{2\ln(R+1)/\delta}{3N} + \sqrt{\frac{2(1-\|\rho\|_2^2)\ln(R+1)/\delta}{N}}\right],
\end{aligned}
$$

with probability of at least $1-3\delta$. Similarly, combining Proposition F.1 (absolute error of $\widehat{\tau}$), Proposition F.7 (generic multiplicative error of $\widehat{\tau}$) and Theorem E.11 (absolute error of $\bar{U}_\Lambda(f)$) via union bound, we have:

$$
\begin{aligned}
\beta &\leqslant \frac{\tau}{1-\tau}\left\{2\mathfrak{R}_{\nu_*}(\mathcal{H}) + \mathcal{B}\left[\frac{12R}{m-2R} + \frac{8R}{3N}\ln 4R/\delta + \sqrt{\frac{8(R-1)\ln 4R/\delta}{3N}} + 8\sqrt{\frac{\ln 2/\delta}{m-2R}}\right]\right\} \\
&\quad + \frac{8\mathcal{B}\Psi(R,k)}{(1-\tau)^2}\left[\frac{2\ln(R+1)/\delta}{3N} + \sqrt{\frac{2(1-\|\rho\|_2^2)\ln(R+1)/\delta}{N}}\right] \\
&\leqslant \frac{2\tau\mathfrak{R}_{\nu_*}(\mathcal{H})}{1-\tau} + \frac{\mathcal{B}}{1-\tau}\left[\frac{12R}{m-2R} + \frac{8R}{3N}\ln 4R/\delta + \sqrt{\frac{8(R-1)\ln 4R/\delta}{3N}} + 8\sqrt{\frac{\ln 2/\delta}{m-2R}}\right] \\
&\quad + \frac{8\mathcal{B}\Psi(R,k)}{(1-\tau)^2}\left[\frac{2\ln(R+1)/\delta}{3N} + \sqrt{\frac{2(1-\|\rho\|_2^2)\ln(R+1)/\delta}{N}}\right],
\end{aligned}
$$

with probability of at least $1-3\delta$. Finally, combining the bounds of $\alpha, \beta$ via union bound, we have:

$$
\begin{aligned}
\sup_{f\in\mathcal{F}}\left|\bar{U}_N(f) - \mathrm{L}_\phi(f)\right| &\leqslant \frac{2\mathfrak{R}_{\mu_*}(\mathcal{H}) + 2\tau\mathfrak{R}_{\nu_*}(\mathcal{H})}{1-\tau} + \frac{8\mathcal{B}}{1-\tau}\left[\sqrt{\frac{\ln 2/\delta}{n-R}} + \sqrt{\frac{\ln 2/\delta}{m-2R}}\right] \\
&\quad + \frac{2\mathcal{B}}{1-\tau}\left[\frac{3R}{n-R} + \frac{6R}{m-2R} + \frac{8R}{3N}\ln 4R/\delta + \sqrt{\frac{8(R-1)\ln 4R/\delta}{3N}}\right] \\
&\quad + \frac{16\mathcal{B}\Psi(R,k)}{(1-\tau)^2}\left[\frac{2\ln(R+1)/\delta}{3N} + \sqrt{\frac{2(1-\|\rho\|_2^2)\ln(R+1)/\delta}{N}}\right], \\
&\leqslant \mathcal{O}\left[\frac{\mathfrak{R}_{\mu_*}(\mathcal{H}) + \tau\mathfrak{R}_{\nu_*}(\mathcal{H})}{1-\tau} + \frac{\mathcal{B}}{1-\tau}\left[\sqrt{\frac{\ln 1/\delta}{n-R}} + \sqrt{\frac{\ln 1/\delta}{m-R}}\right]\right. \\
&\quad \left. + \frac{\mathcal{B}\Psi(R,k)}{(1-\tau)^2}\sqrt{\frac{(1-\|\rho\|_2^2)\ln R/\delta}{N}}\right],
\end{aligned}
$$

with probability of at least $1-6\delta$ as long as $N \geqslant 164k^2\ln\left(\frac{4R}{\delta}\right) + \frac{100\ln\left(\frac{2}{\delta}\right)}{1-\tau}$. Setting $\delta = \Delta/6$ yields the desired complete bound. Applying the same arguments to obtain $\mathcal{O}$-notation bound in Theorem G.3: When $R \geqslant k$, as long as $N \geqslant k \times \max\left\{164k\ln\left(\frac{24R}{\Delta}\right) + \frac{100\ln\left(\frac{12}{\Delta}\right)}{k(1-\tau)}, R\right\}$, we have:

$$
\sup_{f\in\mathcal{F}}\left|\bar{U}_N(f) - \mathrm{L}_\phi(f)\right| \leqslant \mathcal{O}\left[\frac{\mathfrak{C}_N(\mathcal{H})}{1-\tau}\sqrt{\frac{k}{N}} + \frac{\mathcal{B}\sqrt{R}}{(1-\tau)^2}\sqrt{\frac{(1-\|\rho\|_2^2)\ln R/\Delta}{N}}\right],
$$

with probability of at least $1-\Delta$. $\qquad\square$

# H. Refined Analysis for the Estimator $U_N^{\mathrm{hl}}(f)$ from Hieu & Ledent (2025)

To re-iterate from the main text, in Hieu & Ledent (2025), the estimator of the population contrastive risk is constructed for each $f \in \mathcal{F}$ as follows:

$$U_N^{\mathrm{hl}}(f) := \sum_{r=1}^{R} \widehat{\rho}_r U_\Theta^r(f) \quad \forall r \in [R] : U_\Theta^r(f) = \widehat{\mathbb{E}}_{\Theta_r}\left[\ell_{\phi,f}\right], \tag{94}$$

where $\Theta_r$ is the set of collision-free tuples corresponding to class $r \in [R]$:

$$\Theta_r := \left\{ \left(X, X^+, \{X_i^-\}_{i=1}^k\right) : X, X^+ \in S_r \ \& \ \{X_i^-\}_{i=1}^k \subseteq S \setminus S_r \right\}. \tag{95}$$

Basically, for every class $r \in [R]$, the class-wise estimator $U_\Theta^r(f)$ is the (asymptotically) unbiased estimator for the class-wise collision-free risk defined as follows:

$$\mathrm{L}_\phi^r(f) := \mathop{\mathbb{E}}_{\substack{X, X^+ \sim \mathcal{D}_r^{\otimes 2} \\ \{X_i^-\}_{i=1}^k \sim \bar{\mathcal{D}}_r^{\otimes k}}} \left[\ell_{\phi,f}\left(X, X^+, \{X_i^-\}_{i=1}^k\right)\right]. \tag{96}$$

In this section, we will refine the analysis from Hieu & Ledent (2025) and mitigate the $\mathcal{O}\left(\frac{1}{\sqrt{\rho_{\min}}}\right)$ dependency of the generalization gap. Firstly, we re-state their following result:

**Proposition H.1.** *(Hieu & Ledent (2025, Proposition D.5)) Let $U_\Theta^r(f)$ be the class-wise risk estimate defined in Eqn. (95). Then, for any $\Delta \in (0, 1)$, we have:*

$$\mathbb{P}\left(\sup_{f \in \mathcal{F}} \left|U_\Theta^r(f) - \mathrm{L}_\phi^r(f)\right| \geqslant 2\widehat{\mathfrak{R}}_{T_r^{\mathrm{iid}}}(\mathcal{H}) + 10\mathcal{B}\sqrt{\frac{\ln 4/\Delta}{2(1 \vee \bar{N}_r)}} \middle| N_r \right) \leqslant \Delta, \tag{97}$$

*where $\bar{N}_r = \left\lfloor \min\left(\frac{N_r}{2}, \frac{N - N_r}{k}\right) \right\rfloor$ and $T_r^{\mathrm{iid}}$ is the set of $\bar{N}_r$ independent tuples selected "greedily" from the labeled dataset $S$. Specifically:*

$$T_r^{\mathrm{iid}} := \left\{ \left(X_{2j-1}^r, X_{2j}^r, \{\bar{X}_{jk-k+i}^r\}_{i=1}^k\right) \right\}_{j=1}^{\bar{N}_r}, \tag{98}$$

*where:*

1. *$X_u^r$ denotes the $u^{\mathrm{th}}$ data-point in $S_r$.*

2. *$\bar{X}_u^r$ denotes the $u^{\mathrm{th}}$ data-point in $\bar{S}_r = S \setminus S_r$.*

**Remark H.2.** For the sake of definition completeness, we define the empirical Rademacher complexity $\widehat{\mathfrak{R}}_{T_r^{\mathrm{iid}}}(\mathcal{H}) = 0$ when $T_r^{\mathrm{iid}} = \emptyset$. In other words, the complexity is 0 when evaluated on an empty dataset.

**Theorem H.3** (cf. Table 1). *Let $\mathcal{F}$ be a class of representation functions and $U_N^{\mathrm{hl}}(f)$ be defined for each $f \in \mathcal{F}$ in Eqn. (94). Suppose that we have $\rho_r \leqslant \frac{1}{k+2}$ for all $r \in [R]$. Then, for any $\Delta \in (0, 1)$, as long as $N \geqslant 3(k+2)^2 \ln \frac{3R}{\Delta}$, we have:*

$$\sup_{f \in \mathcal{F}} \left|U_N^{\mathrm{hl}}(f) - \mathrm{L}_\phi(f)\right| \tag{99}$$

$$\leqslant 24\mathfrak{C}_N(\mathcal{H})\sqrt{\frac{R}{N}} + \frac{\mathcal{B}R}{N} + \frac{8}{N} + \frac{8\mathcal{B}R}{3N}\ln\left(\frac{6R^2}{\Delta}\right) + \mathcal{B}\sqrt{\frac{8(R-1)\ln 6R^2/\Delta}{3N}} + 10\mathcal{B}\sqrt{\frac{2R\ln 12R/\Delta}{N}}$$

$$\leqslant \mathcal{O}\left[\mathfrak{C}_N(\mathcal{H})\sqrt{\frac{R}{N}} + \mathcal{B}\sqrt{\frac{R\ln(R/\Delta)}{N}}\right],$$

*with probability of at least $1 - \Delta$ where $\mathfrak{C}_N(\mathcal{H})$ is a complexity measure of $\mathcal{H}$ defined as:*

$$\mathfrak{C}_N(\mathcal{H}) := \sup_{S_{\mathrm{tup}} \in (\mathcal{X}^{k+2})^N} \int_{\frac{1}{N}}^{\mathcal{B}} \sqrt{\ln 2\mathcal{N}(\mathcal{H}, \epsilon, \mathrm{L}_2(S_{\mathrm{tup}}))} d\epsilon, \tag{100}$$

*i.e., the worst-case Dudley integral over all possible $(k+2)$-tuple datasets of size $N$.*

**Proof.** Firstly, we have:

$$\sup_{f \in \mathcal{F}} \left| U_N^{\text{hl}}(f) - L_\phi(f) \right| = \sup_{f \in \mathcal{F}} \left| \sum_{r=1}^R \widehat{\rho}_r U_\Theta^r(f) - \sum_{r=1}^R \rho_r L_\phi^r(f) \right|$$

$$= \sup_{f \in \mathcal{F}} \left| \sum_{r=1}^R \widehat{\rho}_r \left[ U_\Theta^r(f) - L_\phi^r(f) \right] + \sum_{r=1}^R L_\phi^r(f) \left[ \widehat{\rho}_r - \rho_r \right] \right|$$

$$\leqslant \sum_{r=1}^R \widehat{\rho}_r \sup_{f \in \mathcal{F}} \left| U_\Theta^r(f) - L_\phi^r(f) \right| + \sum_{r=1}^R \sup_{f \in \mathcal{F}} |L_\phi^r(f)| \cdot |\widehat{\rho}_r - \rho_r|$$

$$\leqslant \sum_{r=1}^R \widehat{\rho}_r \sup_{f \in \mathcal{F}} \left| U_\Theta^r(f) - L_\phi^r(f) \right| + \mathcal{B} \sum_{r=1}^R |\widehat{\rho}_r - \rho_r|.$$

From the proof of Lemma E.2, we already know that for all $\delta \in (0,1)$, we have:

$$\sum_{r=1}^R |\widehat{\rho}_r - \rho_r| \leqslant \frac{8R}{3N} \ln 2R/\delta + \sqrt{\frac{8(R-1)\ln 2R/\delta}{3N}},$$

with probability of at least $1 - \delta$. Now, we proceed to analyze the weighted sum of absolute deviation. From Proposition H.1, we have:

$$\sum_{r=1}^R \widehat{\rho}_r \sup_{f \in \mathcal{F}} |U_\Theta^r(f) - L_\phi^r(f)| = \sum_{\bar{r}:N_{\bar{r}}=1} \frac{\sup_{f \in \mathcal{F}} |L_\phi^{\bar{r}}(f)|}{N} + \sum_{r:N_r \geqslant 2} \widehat{\rho}_r \sup_{f \in \mathcal{F}} |U_\Theta^r(f) - L_\phi^r(f)|$$

$$\leqslant \frac{\mathcal{B}R}{N} + \sum_{r:N_r \geqslant 2} \widehat{\rho}_r \sup_{f \in \mathcal{F}} |U_\Theta^r(f) - L_\phi^r(f)|.$$

**Claim**: As long as $N \geqslant 3(k+2)^2 \ln \frac{1}{\delta}$ then $\widehat{\rho}_r \leqslant \frac{2}{k+2}$ for all $r \in [R]$ with probability of at least $1 - R\delta$. Firstly, we define $\{U_j\}_{j=1}^N$ as the sequence of i.i.d. random variables where $U_j \sim \text{Uniform}(0,1)$ for all $j \in [N]$. For all $r \in [R]$, we have:

$$\mathbb{P}\left(\widehat{\rho}_r \geqslant \frac{2}{k+2}\right) = \mathbb{P}\left(\frac{N_r}{N} \geqslant \frac{2}{k+2}\right) = \mathbb{P}\left(\frac{1}{N} \sum_{j=1}^N \mathbb{1}_{\{U_j \leqslant \rho_r\}} \geqslant \frac{2}{k+2}\right)$$

$$\leqslant \mathbb{P}\left(\frac{1}{N} \sum_{j=1}^N \mathbb{1}_{\{U_j \leqslant \frac{1}{k+2}\}} \geqslant \frac{2}{k+2}\right) \qquad \text{(By coupling)}$$

$$\leqslant \exp\left(-\frac{N}{3(k+2)^2}\right) \qquad \text{(Multiplicative Chernoff)}.$$

Therefore, by the union bound:

$$\mathbb{P}\left(\exists r \in [R] : \widehat{\rho}_r \geqslant \frac{2}{k+2}\right) \leqslant R \exp\left(-\frac{N}{3(k+2)^2}\right).$$

Therefore, if we let $\delta \geqslant \exp\left(-\frac{N}{3(k+2)^2}\right)$, then we have $N \geqslant 3(k+2)^2 \ln \frac{1}{\delta}$, as desired. Combine the above claim with Proposition H.1, the following events:

$$\begin{cases} \widehat{\rho}_r \leqslant \frac{2}{k+2}, & \forall r \in [R], \\ \sup_{f \in \mathcal{F}} \left| U_\Theta^r(f) - L_\phi^r(f) \right| \leqslant 2\widehat{\mathfrak{R}}_{T_r^{\text{iid}}}(\mathcal{H}) + 10\mathcal{B}\sqrt{\frac{\ln 4/\delta}{2\bar{N}_r}}, & \forall r \in [R]. \end{cases}$$

hold simultaneously with probability of at least $1 - 2R\delta$. We also note that $\widehat{\rho}_r \leqslant \frac{2}{k+2}$ is equivalent to $\frac{N_r}{2} \leqslant \frac{N-N_r}{k}$. Therefore, given that $\widehat{\rho}_r \leqslant \frac{2}{k+2}$ for all $r \in [R]$, we have $|T_r^{\text{iid}}| = \bar{N}_r = \lfloor N_r/2 \rfloor$. Therefore, with probability of at least

$1 - 2R\delta$, we have:

$$\sum_{r=1}^{R} \widehat{\rho}_r \sup_{f \in \mathcal{F}} |U_\Theta^r(f) - \mathrm{L}_\phi^r(f)|$$

$$\leqslant \frac{\mathcal{B}R}{N} + \sum_{r:N_r \geqslant 2} \widehat{\rho}_r \sup_{f \in \mathcal{F}} |U_\Theta^r(f) - \mathrm{L}_\phi^r(f)|$$

$$\leqslant \frac{\mathcal{B}R}{N} + \sum_{r:N_r \geqslant 2} \widehat{\rho}_r \left[ 2\widehat{\mathfrak{R}}_{T_r^{\mathrm{iid}}}(\mathcal{H}) + 10\mathcal{B}\sqrt{\frac{\ln 4/\delta}{2\lfloor N_r/2 \rfloor}} \right]$$

$$\overset{(*)}{\leqslant} \frac{\mathcal{B}R}{N} + \sum_{r:N_r \geqslant 2} \widehat{\rho}_r \left[ \frac{8}{N} + \frac{12}{\lfloor N_r/2 \rfloor^{\frac{1}{2}}} \int_{\frac{1}{N}}^{\mathcal{B}} \sqrt{\ln 2\mathcal{N}(\mathcal{H}, \epsilon, \mathrm{L}_2(T_r^{\mathrm{iid}}))} d\epsilon + 10\mathcal{B}\sqrt{\frac{\ln 4/\delta}{2\lfloor N_r/2 \rfloor}} \right]$$

$$\leqslant \frac{\mathcal{B}R}{N} + \frac{8}{N} + \sum_{r:N_r \geqslant 2} \widehat{\rho}_r \left[ \frac{12}{\sqrt{N_r/4}} \int_{\frac{1}{N}}^{\mathcal{B}} \sqrt{\ln 2\mathcal{N}(\mathcal{H}, \epsilon, \mathrm{L}_2(T_r^{\mathrm{iid}}))} d\epsilon + 10\mathcal{B}\sqrt{\frac{2\ln 4/\delta}{N_r}} \right]$$

$$= \frac{\mathcal{B}R}{N} + \frac{8}{N} + \sum_{r:N_r \geqslant 2} \widehat{\rho}_r \left[ \frac{24}{\sqrt{N_r}} \int_{\frac{1}{N}}^{\mathcal{B}} \sqrt{\ln 2\mathcal{N}(\mathcal{H}, \epsilon, \mathrm{L}_2(T_r^{\mathrm{iid}}))} d\epsilon + 10\mathcal{B}\sqrt{\frac{2\ln 4/\delta}{N_r}} \right]$$

$$\leqslant \frac{\mathcal{B}R}{N} + \frac{8}{N} + \sum_{r:N_r \geqslant 2} \widehat{\rho}_r \left[ \frac{24\mathfrak{C}_N(\mathcal{H})}{\sqrt{N_r}} + 10\mathcal{B}\sqrt{\frac{2\ln 4/\delta}{N_r}} \right].$$

In $(*)$, we used the Dudley entropy bound (Theorem L.5) with $\alpha = \frac{1}{N}$. Combine the above bound with the high probability bound on $\sum_{r=1}^{R} |\widehat{\rho}_r - \rho_r|$, as long as we have $N \geqslant 3(k+2)^2 \ln \frac{1}{\delta}$, we have:

$$\sup_{f \in \mathcal{F}} \left| U_N^{\mathrm{hl}}(f) - \mathrm{L}_\phi^r(f) \right| \leqslant \left[ 24\mathfrak{C}_N(\mathcal{H}) + 10\mathcal{B}\sqrt{2\ln \frac{4}{\delta}} \right] \sum_{r:N_r \geqslant 2} \frac{\widehat{\rho}_r}{\sqrt{N_r}} + \frac{\mathcal{B}R}{N} + \frac{8}{N}$$

$$+ \frac{8\mathcal{B}R}{3N} \ln \frac{2R}{\delta} + \mathcal{B}\sqrt{\frac{8(R-1)\ln 2R/\delta}{3N}},$$

with probability of at least $1 - (2R+1)\delta$. Furthermore, we have:

$$\sum_{r:N_r \geqslant 2} \frac{\widehat{\rho}_r}{\sqrt{N_r}} = \sum_{r:N_r \geqslant 2} \sqrt{\frac{\widehat{\rho}_r}{N}} \leqslant \left( \sum_{r:N_r \geqslant 2} \widehat{\rho}_r \right)^{\frac{1}{2}} \left( \sum_{r:N_r \geqslant 2} \frac{1}{N} \right)^{\frac{1}{2}} \leqslant \sqrt{\frac{R}{N}}.$$

Therefore, as long as $N \geqslant 3(k+2)^2 \ln \frac{1}{\delta}$, we have:

$$\sup_{f \in \mathcal{F}} \left| U_N^{\mathrm{hl}}(f) - \mathrm{L}_\phi(f) \right|$$

$$\leqslant \sum_{r=1}^{R} \widehat{\rho}_r \sup_{f \in \mathcal{F}} \left| U_\Theta^r(f) - \mathrm{L}_\phi^r(f) \right| + \mathcal{B}\sum_{r=1}^{R} |\widehat{\rho}_r - \rho_r|$$

$$\leqslant \left[ 24\mathfrak{C}_N(\mathcal{H}) + 10\mathcal{B}\sqrt{2\ln \frac{4}{\delta}} \right] \sqrt{\frac{R}{N}} + \frac{\mathcal{B}R}{N} + \frac{8}{N} + \frac{8\mathcal{B}R}{3N} \ln \frac{2R}{\delta} + \mathcal{B}\sqrt{\frac{8(R-1)\ln 2R/\delta}{3N}}$$

$$= 24\mathfrak{C}_N(\mathcal{H})\sqrt{\frac{R}{N}} + \frac{\mathcal{B}R}{N} + \frac{8}{N} + \frac{8\mathcal{B}R}{3N} \ln \frac{2R}{\delta} + \mathcal{B}\sqrt{\frac{8(R-1)\ln 2R/\delta}{3N}} + 10\mathcal{B}\sqrt{\frac{2R\ln 4/\delta}{N}},$$

with probability of at least $1 - (2R+1)\delta$. Crudely setting $\delta = \Delta/3R$ yields the desired bound. $\qquad \square$

**Remark H.4.** The complexity measure $\mathfrak{C}_N(\mathcal{H})$ is easily bounded for linear classes or neural networks with bounded spectral norms and bounded activation Lipschitz constants. For example, if $\mathcal{F}$ is a linear class defined as follows:

$$\mathcal{F} = \left\{ \mathbf{x} \mapsto A\mathbf{x} : A \in \mathbb{R}^{d \times m}, \|A^\top\|_{2,1} \leqslant a, \|A\|_\sigma \leqslant s, \|\mathbf{x}\|_2 \leqslant b \right\}. \tag{101}$$

Then, from Hieu & Ledent (2025), we have:

$$\mathfrak{C}_N(\mathcal{H}) \lesssim \eta sab^2 \ln^{\frac{1}{2}} \left( \eta sab^2 N(k+2)d \right) \ln(N\mathcal{B}) \leqslant \widetilde{\mathcal{O}} \left( \eta sab^2 \right), \tag{102}$$

where $\eta > 0$ is the $\ell_\infty$-Lipschitz constant of the contrastive loss $\phi$. On the other hand, if $\mathcal{F}$ is denoted as a class of deep neural networks of weights with bounded spectral norms:

$$\mathcal{F} = \left\{ \mathbf{x} \mapsto A_L \varphi_{L-1} \left( A_{L-1} \varphi_{L-2}(\dots \sigma_1(A_1 \mathbf{x}) \dots) \right) : A_\ell \in \mathbb{R}^{d_\ell \times d_{\ell-1}}, \|A_\ell\|_\sigma \leqslant s_\ell, \forall \ell \in [L] \right\}, \tag{103}$$

Then, we have the following metric entropy bound (Long & Sedghi, 2020; Graf et al., 2022; Hieu et al., 2024):

$$\mathfrak{C}_N(\mathcal{H}) \lesssim W^{\frac{1}{2}} \ln^{\frac{1}{2}} \left( 1 + \eta L N b \prod_{\ell=1}^{L} s_\ell^2 \right) \leqslant \widetilde{\mathcal{O}}([LW]^{\frac{1}{2}}), \tag{104}$$

where $W := \sum_{\ell=1}^{L} d_\ell \times d_{\ell-1}$, i.e., total number of specified parameters.

**Theorem H.5** (Theorem 4.1, Table 1). *Let $\mathcal{F}$ be a class of representation functions and let $U_N^{\mathrm{hl}}(f)$ be defined for each $f \in \mathcal{F}$ as in Eqn. (94). Suppose that $\rho_r \leqslant \frac{1}{2}$ for all $r \in [R]$. Then, for any $\Delta \in (0,1)$, as long as $N \geqslant 6(k+1)^2 \ln \frac{3R}{\Delta}$:*

$$\sup_{f \in \mathcal{F}} \left| U_N^{\mathrm{hl}}(f) - \mathrm{L}_\phi(f) \right| \leqslant \left[ 24\mathfrak{C}_N(\mathcal{H}) + 10\mathcal{B}\sqrt{2\ln\left(\frac{12R}{\Delta}\right)} \right] \left[ \widehat{\theta}_{k+2}^{\frac{1}{2}} \sqrt{\frac{2R}{N}} + (1 - \widehat{\theta}_{k+2})\sqrt{\frac{2(k+1)}{N}} \right]$$

$$+ \frac{\mathcal{B}R}{N} + \frac{8}{N} + \frac{8\mathcal{B}R}{3N} \ln\left(\frac{6R^2}{\Delta}\right) + \mathcal{B}\sqrt{\frac{8(R-1)\ln 6R^2/\Delta}{3N}} \tag{105}$$

$$\leqslant \mathcal{O}\left[ \mathfrak{C}_N(\mathcal{H}) \left[ \widehat{\theta}_{k+2}^{\frac{1}{2}} \sqrt{\frac{R}{N}} + (1 - \widehat{\theta}_{k+2})\sqrt{\frac{k}{N}} \right] + \mathcal{B}\sqrt{\frac{R\ln(R/\Delta)}{N}} \right]. \tag{106}$$

*with probability of at least $1 - \Delta$, where $\widehat{\theta}_{k+2} := \sum_{r:\widehat{\rho}_r \leqslant \frac{2}{k+2}} \widehat{\rho}_r$.*

**Proof.** Let $\delta \in (0,1)$. From the proof of Theorem H.3, we have:

$$\sum_{r=1}^{R} |\widehat{\rho}_r - \rho_r| \leqslant \frac{8R}{3N} \ln 2R/\delta + \sqrt{\frac{8(R-1)\ln 2R/\delta}{3N}}, \qquad\qquad \text{wp. } \geqslant 1 - \delta,$$

$$\sup_{f \in \mathcal{F}} \left| U_\Theta^r(f) - \mathrm{L}_\phi^r(f) \right| \leqslant 2\widehat{\mathfrak{R}}_{T_r^{\mathrm{iid}}}(\mathcal{H}) + 10\mathcal{B}\sqrt{\frac{\ln 4/\delta}{2\bar{N}_r}}, \quad \forall r \in [R] \qquad \text{wp. } \geqslant 1 - R\delta.$$

Therefore, by the union bound, with probability of at least $1 - (R+1)\delta$, we have:

$$\sup_{f \in \mathcal{F}} \left| U_N^{\mathrm{hl}}(f) - \mathrm{L}_\phi(f) \right|$$

$$\leqslant \sum_{r=1}^{R} \widehat{\rho}_r \sup_{f \in \mathcal{F}} \left| U_\Theta^r(f) - \mathrm{L}_\phi^r(f) \right| + \mathcal{B}\sum_{r=1}^{R} |\widehat{\rho}_r - \rho_r|$$

$$\leqslant \frac{\mathcal{B}R}{N} + \sum_{r:N_r \geqslant 2} \widehat{\rho}_r \left[ 2\widehat{\mathfrak{R}}_{T_r^{\mathrm{iid}}}(\mathcal{H}) + 10\mathcal{B}\sqrt{\frac{\ln 4/\delta}{2\bar{N}_r}} \right] + \frac{8\mathcal{B}R}{3N} \ln 2R/\delta + \mathcal{B}\sqrt{\frac{8(R-1)\ln 2R/\delta}{3N}}$$

$$\leqslant \frac{\mathcal{B}R}{N} + \frac{8}{N} + \left[ 24\mathfrak{C}_N(\mathcal{H}) + 10\mathcal{B}\sqrt{2\ln\frac{4}{\delta}} \right] \sum_{r:N_r \geqslant 2} \frac{\widehat{\rho}_r}{\sqrt{\bar{N}_r}} + \frac{8\mathcal{B}R}{3N} \ln 2R/\delta + \mathcal{B}\sqrt{\frac{8(R-1)\ln 2R/\delta}{3N}}.$$

Then, we have:

$$\sum_{r:N_r \geqslant 2} \frac{\widehat{\rho}_r}{\sqrt{\bar{N}_r}} = \sum_{r:\frac{2}{N} \leqslant \widehat{\rho}_r \leqslant \frac{2}{k+2}} \frac{\widehat{\rho}_r}{\sqrt{N_r/2}} + \sum_{\bar{r}:\frac{2}{k+2} < \widehat{\rho}_{\bar{r}} \leqslant 1} \frac{\widehat{\rho}_{\bar{r}}}{\sqrt{(N - N_{\bar{r}})/k}}.$$

Now, let $\alpha > 1$ be a real number and $\{U_j\}_{j=1}^N$ be a sequence of independent random variables such that $U_j \sim \text{Uniform}(0,1)$ for all $j \in [N]$. Then, by coupling:

$$\widehat{\rho}_r = \frac{N_r}{N} \stackrel{d}{=} \frac{1}{N} \sum_{j=1}^N \mathbb{1}_{\{U_j \leqslant \rho_r\}}.$$

Hence, for all $r \in [R]$, we have:

$$\mathbb{P}\left(\widehat{\rho}_r \geqslant \frac{1}{2}\left(1 + \frac{1}{\alpha}\right)\right) = \mathbb{P}\left(\frac{N_r}{N} \geqslant \frac{1}{2}\left(1 + \frac{1}{\alpha}\right)\right)$$

$$= \mathbb{P}\left(\frac{1}{N}\sum_{j=1}^N \mathbb{1}_{\{U_j \leqslant \rho_r\}} \geqslant \frac{1}{2}\left(1 + \frac{1}{\alpha}\right)\right)$$

$$\leqslant \mathbb{P}\left(\frac{1}{N}\sum_{j=1}^N \mathbb{1}_{\{U_j \leqslant \frac{1}{2}\}} \geqslant \frac{1}{2}\left(1 + \frac{1}{\alpha}\right)\right) \qquad \text{(By Coupling)}$$

$$\leqslant \exp\left(-\frac{N}{6\alpha^2}\right) \qquad \text{(Multiplicative Chernoff)}.$$

Therefore, by the union bound:

$$\mathbb{P}\left(\exists r \in [R] : \widehat{\rho}_r \geqslant \frac{1}{2}\left(1 + \frac{1}{\alpha}\right)\right) \leqslant R \exp\left(-\frac{N}{6\alpha^2}\right).$$

Hence, as long as $N \geqslant 6\alpha^2 \ln\frac{1}{\delta}$, we have $\widehat{\rho}_r \leqslant \frac{1}{2}\left(1 + \frac{1}{\alpha}\right)$ for all $r \in [R]$ with probability of at least $1 - R\delta$. Therefore, as long as $N \geqslant 6\alpha^2 \ln\frac{1}{\delta}$, we have:

$$\sum_{r:N_r \geqslant 2} \frac{\widehat{\rho}_r}{\sqrt{\bar{N}_r}} = \sum_{r:\frac{2}{N} \leqslant \widehat{\rho}_r \leqslant \frac{2}{k+2}} \frac{\widehat{\rho}_r}{\sqrt{N_r/2}} + \sum_{\bar{r}:\frac{2}{k+2} < \widehat{\rho}_{\bar{r}} \leqslant 1} \frac{\widehat{\rho}_{\bar{r}}}{\sqrt{(N-N_{\bar{r}})/k}}$$

$$\leqslant \sum_{r:\frac{2}{N} \leqslant \widehat{\rho}_r \leqslant \frac{2}{k+2}} \sqrt{\frac{2\widehat{\rho}_r}{N}} + \sum_{\bar{r}:\frac{2}{k+2} < \widehat{\rho}_{\bar{r}} \leqslant 1} \widehat{\rho}_{\bar{r}}\sqrt{\frac{k}{N - \left(1 + \frac{1}{\alpha}\right)\frac{N}{2}}}$$

$$= \sum_{r:\frac{2}{N} \leqslant \widehat{\rho}_r \leqslant \frac{2}{k+2}} \sqrt{\frac{2\widehat{\rho}_r}{N}} + \sum_{\bar{r}:\frac{2}{k+2} < \widehat{\rho}_{\bar{r}} \leqslant 1} \widehat{\rho}_{\bar{r}}\sqrt{\frac{k}{\frac{N}{2}\left(1 - \frac{1}{\alpha}\right)}}$$

$$= \sum_{r:\frac{2}{N} \leqslant \widehat{\rho}_r \leqslant \frac{2}{k+2}} \sqrt{\frac{2\widehat{\rho}_r}{N}} + \sum_{\bar{r}:\frac{2}{k+2} < \widehat{\rho}_{\bar{r}} \leqslant 1} \widehat{\rho}_{\bar{r}}\sqrt{\frac{k}{N} \cdot \frac{2\alpha}{\alpha - 1}}$$

$$= \sum_{r:\frac{2}{N} \leqslant \widehat{\rho}_r \leqslant \frac{2}{k+2}} \sqrt{\frac{2\widehat{\rho}_r}{N}} + (1 - \widehat{\theta}_{k+2})\sqrt{\frac{2\alpha k}{N(\alpha - 1)}}$$

$$\leqslant \left(\sum_{r:\frac{2}{N} \leqslant \widehat{\rho}_r \leqslant \frac{2}{k+2}} \widehat{\rho}_r\right)^{\frac{1}{2}} \left(\sum_{r:\frac{2}{N} \leqslant \widehat{\rho}_r \leqslant \frac{2}{k+2}} \frac{2}{N}\right)^{\frac{1}{2}} + (1 - \widehat{\theta}_{k+2})\sqrt{\frac{2\alpha k}{N(\alpha - 1)}} \quad \text{(Cauchy-Schwarz)}$$

$$\leqslant \sqrt{\widehat{\theta}_{k+2}} \cdot \sqrt{\frac{2R}{N}} + (1 - \widehat{\theta}_{k+2})\sqrt{\frac{2\alpha k}{N(\alpha - 1)}},$$

with probability of at least $1 - R\delta$. Choosing $\alpha = k + 1$, then as long as $N \geqslant 6(k+1)^2 \ln\frac{1}{\delta}$, with probability of at least $1 - R\delta$, we have:

$$\sum_{r:N_r \geqslant 2} \frac{\widehat{\rho}_r}{\sqrt{\bar{N}_r}} \leqslant \widehat{\theta}_{k+2}^{\frac{1}{2}}\sqrt{\frac{2R}{N}} + (1 - \widehat{\theta}_{k+2})\sqrt{\frac{2(k+1)}{N}}.$$

Combine with the bound on $\sup_{f\in\mathcal{F}}|U_N^{\mathrm{hl}}(f) - \mathrm{L}_\phi(f)|$ via union bound, as long as $N \geqslant 6(k+1)^2 \ln\frac{1}{\delta}$, with probability of at least $1 - (2R+1)\delta$, we have:

$$
\begin{aligned}
&\sup_{f\in\mathcal{F}}\left|U_N^{\mathrm{hl}}(f) - \mathrm{L}_\phi(f)\right| \\
&\leqslant \sum_{r=1}^{R} \widehat{\rho}_r \sup_{f\in\mathcal{F}}\left|U_\Theta^r(f) - \mathrm{L}_\phi^r(f)\right| + \mathcal{B}\sum_{r=1}^{R}|\widehat{\rho}_r - \rho_r| \\
&\leqslant \frac{\mathcal{B}R}{N} + \frac{8}{N} + \left[24\mathfrak{C}_N(\mathcal{H}) + 10\mathcal{B}\sqrt{2\ln\frac{4}{\delta}}\right]\sum_{r:N_r\geqslant 2}\frac{\widehat{\rho}_r}{\sqrt{\bar{N}_r}} + \frac{8\mathcal{B}R}{3N}\ln 2R/\delta + \mathcal{B}\sqrt{\frac{8(R-1)\ln 2R/\delta}{3N}} \\
&\leqslant \frac{\mathcal{B}R}{N} + \frac{8}{N} + \left[24\mathfrak{C}_N(\mathcal{H}) + 10\mathcal{B}\sqrt{2\ln\frac{4}{\delta}}\right]\left[\widehat{\theta}_{k+2}^{\frac{1}{2}}\sqrt{\frac{2R}{N}} + (1 - \widehat{\theta}_{k+2})\sqrt{\frac{2(k+1)}{N}}\right] \\
&\quad + \frac{8\mathcal{B}R}{3N}\ln 2R/\delta + \mathcal{B}\sqrt{\frac{8(R-1)\ln 2R/\delta}{3N}}.
\end{aligned}
$$

Crudely setting $\delta = \Delta/3R$ yields the desired bound. It is straightforward to get rid of the assumption that $\rho_r \leqslant \frac{1}{2}$. We have the following result. $\qquad\square$

**Theorem H.6** (Theorem H.5 without $\rho_r \leqslant \frac{1}{2}$ assumption)**.** *Let $\mathcal{F}$ be a class of representation functions and let $U_N^{\mathrm{hl}}(f)$ be defined for each $f \in \mathcal{F}$ as in Eqn. (94). Then, for any $\Delta \in (0,1)$, as long as $N \geqslant \frac{2(k+1)^2}{1-\rho_{\max}}\ln\frac{3R}{\Delta}$, we have:*

$$
\begin{aligned}
\sup_{f\in\mathcal{F}}\left|U_N^{\mathrm{hl}}(f) - \mathrm{L}_\phi(f)\right| &\leqslant \left[24\mathfrak{C}_N(\mathcal{H}) + 10\mathcal{B}\sqrt{2\ln\left(\frac{12R}{\Delta}\right)}\right]\left[\widehat{\theta}_{k+2}^{\frac{1}{2}}\sqrt{\frac{2R}{N}} + (1 - \widehat{\theta}_{k+2})\sqrt{\frac{k+1}{N(1-\rho_{\max})}}\right] \\
&\quad + \frac{\mathcal{B}R}{N} + \frac{8}{N} + \frac{8\mathcal{B}R}{3N}\ln\left(\frac{6R^2}{\Delta}\right) + \mathcal{B}\sqrt{\frac{8(R-1)\ln 6R^2/\Delta}{3N}} \qquad (107) \\
&\leqslant \mathcal{O}\left[\mathfrak{C}_N(\mathcal{H})\left[\widehat{\theta}_{k+2}^{\frac{1}{2}}\sqrt{\frac{R}{N}} + (1 - \widehat{\theta}_{k+2})\sqrt{\frac{k}{N(1-\rho_{\max})}}\right] + \mathcal{B}\sqrt{\frac{R\ln(R/\Delta)}{N}}\right]. \qquad (108)
\end{aligned}
$$

*with probability of at least $1 - \Delta$, where $\widehat{\theta}_{k+2} := \sum_{r:\widehat{\rho}_r\leqslant\frac{2}{k+2}}\widehat{\rho}_r$.*

**Proof**. Using similar arguments in Theorem H.5, for all $\delta \in (0,1)$, with probability of at least $1 - (R+1)\delta$, we have:

$$
\begin{aligned}
&\sup_{f\in\mathcal{F}}\left|U_N^{\mathrm{hl}}(f) - \mathrm{L}_\phi(f)\right| \\
&\leqslant \sum_{r=1}^{R}\widehat{\rho}_r\sup_{f\in\mathcal{F}}\left|U_\Theta^r(f) - \mathrm{L}_\phi^r(f)\right| + \mathcal{B}\sum_{r=1}^{R}|\widehat{\rho}_r - \rho_r| \\
&\leqslant \frac{\mathcal{B}R}{N} + \frac{8}{N} + \left[24\mathfrak{C}_N(\mathcal{H}) + 10\mathcal{B}\sqrt{2\ln\frac{4}{\delta}}\right]\sum_{r:N_r\geqslant 2}\frac{\widehat{\rho}_r}{\sqrt{\bar{N}_r}} + \frac{8\mathcal{B}R}{3N}\ln 2R/\delta + \mathcal{B}\sqrt{\frac{8(R-1)\ln 2R/\delta}{3N}}.
\end{aligned}
$$

Then, let $\rho_{\max} = \max_{r\in[R]}\rho_r$, we split the sum $\sum_{r:N_r\geqslant 2}\frac{\widehat{\rho}_r}{\sqrt{\bar{N}_r}}$ as follows:

$$
\sum_{r:N_r\geqslant 2}\frac{\widehat{\rho}_r}{\sqrt{\bar{N}_r}} = \sum_{r:\frac{2}{N}\leqslant\widehat{\rho}_r\leqslant\frac{2}{k+2}}\frac{\widehat{\rho}_r}{\sqrt{N_r/2}} + \sum_{\bar{r}:\frac{2}{k+2}<\widehat{\rho}_{\bar{r}}\leqslant 1}\frac{\widehat{\rho}_{\bar{r}}}{\sqrt{(N-N_{\bar{r}})/k}}
$$

Then, using the same coupling trick. Let $\{U_j\}_{j=1}^{N}$ be a sequence of independent uniform random variables where

$U_j \sim \text{Uniform}(0, 1)$ for all $j \in [N]$. Then, for $\alpha > 1$, we have:

$$\mathbb{P}\left(1 - \widehat{\rho}_r \leqslant (1 - \rho_{\max})\left(1 - \frac{1}{\alpha}\right)\right) \leqslant \mathbb{P}\left(1 - \widehat{\rho}_r \leqslant (1 - \rho_r)\left(1 - \frac{1}{\alpha}\right)\right) \qquad (1 - \rho_r \geqslant 1 - \rho_{\max})$$

$$\leqslant \mathbb{P}\left(\frac{1}{N}\sum_{j=1}^{N}\mathbb{1}_{\{U_j \leqslant 1 - \rho_r\}} \leqslant (1 - \rho_r)\left(1 - \frac{1}{\alpha}\right)\right) \quad \text{(By Coupling)}$$

$$\leqslant \exp\left(-\frac{N(1 - \rho_r)}{2\alpha^2}\right) \leqslant \exp\left(-\frac{N(1 - \rho_{\max})}{2\alpha^2}\right)$$

$$= \exp\left(-\frac{N}{2\alpha^2(1 - \rho_{\max})^{-1}}\right).$$

Therefore, by the union bound:

$$\mathbb{P}\left(\exists r \in [R] : \widehat{1} - \rho_r \leqslant (1 - \rho_{\max})\left(1 - \frac{1}{\alpha}\right)\right) \leqslant R\exp\left(-\frac{N}{2\alpha^2(1 - \rho_{\max})^{-1}}\right).$$

Therefore, as long as $N \geqslant \frac{2\alpha^2}{1 - \rho_{\max}}\ln\frac{1}{\delta}$, we have $\widehat{\rho}_r \geqslant (1 - \rho_{\max})\left(1 - \frac{1}{\alpha}\right)$ for all $r \in [R]$ with probability of at least $1 - R\delta$. As a result, we have:

$$\sum_{r:N_r \geqslant 2}\frac{\widehat{\rho}_r}{\sqrt{\bar{N}_R}} = \sum_{r:\frac{2}{N} \leqslant \widehat{\rho}_r \leqslant \frac{2}{k+2}}\frac{\widehat{\rho}_r}{\sqrt{N_r/2}} + \sum_{\bar{r}:\frac{2}{k+2} < \widehat{\rho}_{\bar{r}} \leqslant 1}\frac{\widehat{\rho}_{\bar{r}}}{\sqrt{(N - N_{\bar{r}})/k}}$$

$$\leqslant \sum_{r:\frac{2}{N} \leqslant \widehat{\rho}_r \leqslant \frac{2}{k+2}}\sqrt{\frac{2\widehat{\rho}_r}{N}} + \sum_{\bar{r}:\frac{2}{k+2} < \widehat{\rho}_{\bar{r}} \leqslant 1}\widehat{\rho}_{\bar{r}}\sqrt{\frac{k}{N(1 - \widehat{\rho}_{\bar{r}})}}$$

$$\leqslant \widehat{\theta}_{k+2}^{\frac{1}{2}}\sqrt{\frac{2R}{N}} + \sum_{\bar{r}:\frac{2}{k+2} < \widehat{\rho}_{\bar{r}} \leqslant 1}\widehat{\rho}_{\bar{r}}\sqrt{\frac{k}{N(1 - \widehat{\rho}_{\bar{r}})}} \qquad \text{(Cauchy-Schwarz)}$$

$$\leqslant \widehat{\theta}_{k+2}^{\frac{1}{2}}\sqrt{\frac{2R}{N}} + \sum_{\bar{r}:\frac{2}{k+2} < \widehat{\rho}_{\bar{r}} \leqslant 1}\widehat{\rho}_{\bar{r}}\sqrt{\frac{k}{N(1 - \rho_{\max})\left(1 - \frac{1}{\alpha}\right)}}$$

$$= \widehat{\theta}_{k+2}^{\frac{1}{2}}\sqrt{\frac{2R}{N}} + (1 - \widehat{\theta}_{k+2})\sqrt{\frac{\alpha k}{N(1 - \rho_{\max})(\alpha - 1)}},$$

with probability of at least $1 - R\delta$ as long as $N \geqslant \frac{2\alpha^2}{1 - \rho_{\max}}\ln\frac{1}{\delta}$. Choosing $\alpha = k + 1$ yields:

$$\sum_{r:N_r \geqslant 2}\frac{\widehat{\rho}_r}{\sqrt{\bar{N}_R}} \leqslant \widehat{\theta}_{k+2}^{\frac{1}{2}}\sqrt{\frac{2R}{N}} + (1 - \widehat{\theta}_{k+2})\sqrt{\frac{k+1}{N(1 - \rho_{\max})}},$$

with probability of at least $1 - R\delta$ as long as $N \geqslant \frac{2(k+1)^2}{1 - \rho_{\max}}\ln\frac{1}{\delta}$. Combining this with the very first high-probability event, we have:

$$\sup_{f \in \mathcal{F}}\left|U_N^{\text{hl}}(f) - L_\phi(f)\right|$$

$$\leqslant \frac{\mathcal{B}R}{N} + \frac{8}{N} + \left[24\mathfrak{C}_N(\mathcal{H}) + 10\mathcal{B}\sqrt{2\ln\frac{4}{\delta}}\right]\sum_{r:N_r \geqslant 2}\frac{\widehat{\rho}_r}{\sqrt{\bar{N}_r}} + \frac{8\mathcal{B}R}{3N}\ln 2R/\delta + \mathcal{B}\sqrt{\frac{8(R - 1)\ln 2R/\delta}{3N}}$$

$$\leqslant \frac{\mathcal{B}R}{N} + \frac{8}{N} + \left[24\mathfrak{C}_N(\mathcal{H}) + 10\mathcal{B}\sqrt{2\ln\frac{4}{\delta}}\right]\left[\widehat{\theta}_{k+2}^{\frac{1}{2}}\sqrt{\frac{2R}{N}} + (1 - \widehat{\theta}_{k+2})\sqrt{\frac{k+1}{N(1 - \rho_{\max})}}\right]$$

$$+ \frac{8\mathcal{B}R}{3N}\ln 2R/\delta + \mathcal{B}\sqrt{\frac{8(R - 1)\ln 2R/\delta}{3N}}.$$

Setting $\delta = \Delta/3R$ yields the desired bound. $\qquad\qquad\square$

# I. Experiment Details

## I.1. Sub-sampled Estimation of U-Statistics

In the main text, we have established that the proposed U-Statistic of this work ($U_N$) and that of prior work ($U_N^{\text{hl}}$) require massive numbers of evaluations, making it infeasible to calculate either of them once let alone repeating such computations over many training iterations. Fortunately, we can construct sub-sampled estimators that converge to the full U-Statistic with probability one.

Let us recall the general construction. Given a finite population $\{\mathbf{z}_j\}_{j=1}^K \subset \mathcal{Z}$, a function $h : \mathcal{Z} \to \mathbb{R}$, a weighting function $w : \mathcal{Z} \to \mathbb{R}$ and $A_K^w(h)$ be defined as follows:

$$A_K^w(h) := \sum_{j=1}^K w(\mathbf{z}_j)h(\mathbf{z}_j), \qquad \sum_{j=1}^K w(\mathbf{z}_j) = 1. \tag{109}$$

Then, given an i.i.d. draw $\{\mathbf{z}_\ell^*\}_{\ell=1}^M$ from a distribution $q$ over $\{\mathbf{z}_j\}_{j=1}^K$, the sub-sampled average:

$$\widehat{A}_M^q(h) = \frac{1}{M} \sum_{\ell=1}^M \frac{w(\mathbf{z}_\ell^*)}{q(\mathbf{z}_\ell^*)} h(\mathbf{z}_\ell^*), \tag{110}$$

is an unbiased estimator of $A_K^w(h)$, which means $\widehat{A}_M^q(h) \to A_K^w(h)$ with probability one.

**Sub-sampled Estimator of $U_N^{\text{hl}}$:** Let $\Theta := \bigcup_{r=1}^R \Theta_r$, i.e., the total collection of collision-free tuples and a representation function $f \in \mathcal{F}$. The U-Statistic $U_N^{\text{hl}}(f)$ can be written in the form of Eqn. (109) as follows

$$U_N^{\text{hl}}(f) := \sum_{t \in \Theta} w(t)\ell_{\phi,f}(t), \qquad \forall t \in \Theta_r : w(t) = \frac{\widehat{\rho}_r}{|\Theta_r|}. \tag{111}$$

Since $w$ itself is a distribution over $\Theta$, we can take $q = w$ and this is also the sampling distribution proposed in Hieu & Ledent (2025). Furthermore, because we choose $q = w$, the weightage of all tuples (in the empirical risk) drawn from the distribution $q$ are identical and all equal to $1$. The sub-sampling procedure of tuples using the distribution $q$ is given by Algorithm 1, which is also described in Hieu & Ledent (2025, Section 3.3).

---

**Algorithm 1** Sub-sampled Estimation for $U_N^{\text{hl}}(f)$

---

1: Given a representation function $f \in \mathcal{F}$
2: **for** $j$ in $1 \ldots M$ **do**
3:   Select $r \in [R]$ with probability $\widehat{\rho}_r$
4:   Select $t_j \in \Theta_r$ with equal probabilities
5: **end for**
6: $\text{ERM}(f) \leftarrow \frac{1}{M} \sum_{j=1}^M \ell_{\phi,f}(t_j)$.
7: **return** $\text{ERM}(f)$

---

**Sub-sampled Estimator of $U_N$:** Let $\Omega := \bigcup_{r=1}^R \Omega_r$, i.e., the total collection of collision-allowed tuples. We can write $U_N(f)$ as follows

$$U_N(f) := \sum_{t \in \Omega} w(t)\ell_{\phi,f}(t), \qquad \forall t \in \Omega_r : w(t) = \frac{\widehat{\rho}_r}{|\Omega_r|}\left(\frac{1}{1-\widehat{\tau}} - \mathbb{1}_{\Lambda_r}(t)\frac{|\Omega_r|}{|\Lambda_r|}\frac{\widehat{\tau}}{1-\widehat{\tau}}\right). \tag{112}$$

Let us calculate the ratio $|\Omega_r|/|\Lambda_r|$ explicitly:

$$\begin{aligned}
\frac{|\Omega_r|}{|\Lambda_r|} &= \frac{\binom{N_r}{2} \times \binom{N-2}{k}}{\binom{N_r}{3} \times \binom{N-3}{k-1}} = \frac{\frac{1}{2}N_r(N_r-1)}{\frac{1}{6}N_r(N_r-1)(N_r-2)} \times \frac{(N-2)!}{k!(N-k-2)!} \times \frac{(k-1)!(N-k-2)!}{(N-3)!} \\
&= \frac{3}{N_r-2} \times \frac{(N-2)!(k-1)!}{(N-3)!k!} \\
&= \frac{3(N-2)}{k(N_r-2)} \approx \frac{3}{k\widehat{\rho}_r}.
\end{aligned}$$

Therefore, for a tuple $t \in \Omega_r$, we have:

$$w(t) = \frac{\widehat{\rho}_r}{|\Omega_r|} \left( \frac{1}{1 - \widehat{\tau}} - \mathbb{1}_{\Lambda_r}(t) \frac{3\widehat{\tau}(N - 2)}{k(1 - \widehat{\tau})(N_r - 2)} \right) \approx \frac{\widehat{\rho}_r}{|\Omega_r|} \left( \frac{1}{1 - \widehat{\tau}} - \mathbb{1}_{\Lambda_r}(t) \frac{3\widehat{\tau}}{k\widehat{\rho}_r(1 - \widehat{\tau})} \right).$$

Now, we note that it is possible that $w(t) < 0$ if we pick a collided tuple $t$ from a small class $r \in [R]$. Specifically, given that $t \in \Omega_r$ is a collided tuple (i.e., $t \in \Lambda_r$) where $N_r \leqslant \frac{3\widehat{\tau}(N-2)}{k} + 2$. Then:

$$N_r - 2 \leqslant \frac{3\widehat{\tau}(N - 2)}{k} \implies \frac{3\widehat{\tau}(N - 2)}{k(N_r - 2)} \geqslant 1 \implies w(t) = \frac{\widehat{\rho}_r \left( 1 - \frac{3\widehat{\tau}(N-2)}{k(N_r-2)} \right)}{(1 - \widehat{\tau}) \times |\Omega_r|} \leqslant 0.$$

The negative weight can potentially cause optimization instability during training. One work-around is to design the sampling distribution $q$ in a way that avoids collided tuples from small class entirely. In this work, we propose $q$ as follows:

$$\forall t \in \Omega_r : q(t) = \widehat{\rho}_r \times \begin{cases} \mathbb{1}_{\Theta_r}(t)|\Theta_r|^{-1} & \text{if } N_r \leqslant \frac{3\widehat{\tau}(N-2)}{k} + 2, \\ \frac{1}{|\Omega_r|} & \text{otherwise.} \end{cases} \tag{113}$$

With the above chosen sampling distribution, for a given tuple $t$ whose anchor-positive pair belongs to a class $r \in [R]$ such that $N_r > \frac{3\widehat{\tau}(N-2)}{k} + 2$ (meaning that $r$ is one of the major classes), the weightage of $t$ in the sub-sampled empirical risk is:

$$\frac{w(t)}{q(t)} = \frac{1}{1 - \widehat{\tau}} - \frac{|\Omega_r|}{|\Lambda_r|} \frac{\widehat{\tau}}{1 - \widehat{\tau}} \mathbb{1}_{\Lambda_r}(t) = \frac{1}{1 - \widehat{\tau}} - \frac{3\widehat{\tau}(N - 2)}{k(1 - \widehat{\tau})(N_r - 2)} \mathbb{1}_{\Lambda_r}(t).$$

On the other hand, if $N_r \leqslant \frac{3\widehat{\tau}(N-2)}{k} + 2$, we have:

$$\begin{aligned} \frac{w(t)}{q(t)} &= \frac{|\Theta_r|}{|\Omega_r|} (1 - \widehat{\tau})^{-1} = \frac{\binom{N_r}{2}\binom{N - N_r}{k}}{\binom{N_r}{2}\binom{N - 2}{k}} (1 - \widehat{\tau})^{-1} \\ &= \frac{\binom{N - N_r}{k}}{\binom{N - 2}{k}} (1 - \widehat{\tau})^{-1} = \frac{(N - N_r)!}{k!(N - N_r - k)!} \times \frac{k!(N - k - 2)!}{(N - 2)!} (1 - \widehat{\tau})^{-1} \\ &= \frac{(N - N_r)!(N - k - 2)!}{(N - 2)!(N - N_r - k)!} (1 - \widehat{\tau})^{-1} \\ &= \frac{1}{1 - \widehat{\tau}} \prod_{\ell=0}^{k-1} \frac{N - \ell - N_r}{N - \ell - 2}. \end{aligned}$$

With all of the above computations handled, the calculation of sub-sampled estimation for $U_N(f)$ can be described in Algorithm 2.

---

**Algorithm 2** Sub-sampled Estimation for $U_N(f)$

---

1: Given a representation function $f \in \mathcal{F}$
2: Calculate $\widehat{\tau} := \sum_{r=1}^{R} \widehat{\rho}_r (1 - \widehat{\rho}_r)^k$
3: **for** $j$ in $1 \ldots M$ **do**
4:    Select $r \in [R]$ with probability $\widehat{\rho}_r$
5:    **if** $N_r \leqslant \frac{3\widehat{\tau}(N-2)}{k} + 2$ **then**
6:       Select $t_j$ from $\Theta_r$ with equal probabilities          {Avoid collision for small classes}
7:       $\omega_j \leftarrow \frac{1}{1-\widehat{\tau}} \prod_{\ell=0}^{k-1} \frac{N-\ell-N_r}{N-\ell-2}$
8:    **else**
9:       Select $t_j$ from $\Omega_r$ with equal probabilities          {Allow collision for large classes}
10:       $\omega_j \leftarrow \frac{1}{1-\widehat{\tau}} - \frac{3\widehat{\tau}(N-2)}{k(1-\widehat{\tau})(N_r-2)} \mathbb{1}_{\Lambda_r}(t)$
11:    **end if**
12: **end for**
13: $\text{ERM}(f) \leftarrow \frac{1}{M} \sum_{j=1}^{M} \omega_j \ell_{\phi,f}(t_j)$
14: **return** $\text{ERM}(f)$

---

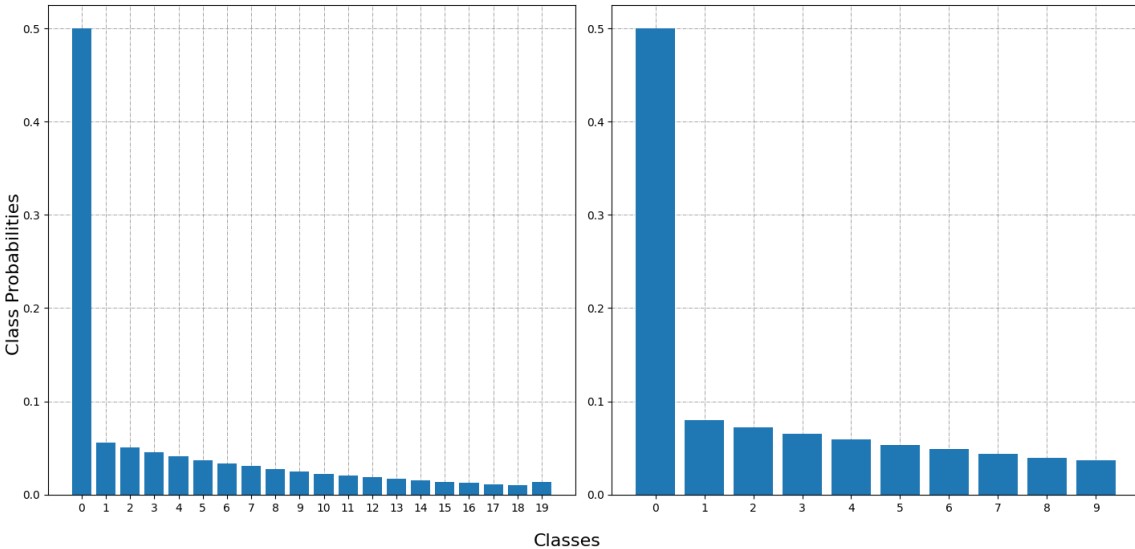

*Figure 4.* Class distributions used for synthetic (left) and real (right) data experiments. For the real data experiments, we used the same class distribution structure as depicted in the figure above to create subsets from all real datasets (MNIST, FashionMNIST and CIFAR10).

**Remark I.1.** From Algorithm 2, there is an interesting effect of the chosen sub-sampling distribution $q$ in Eqn. (113) on how the sub-sampled estimation of $U_N$ behave on tail classes. Specifically:

(i) For tuples whose anchor-positive pairs come from major classes, by default, they will have the largest weights (of $\frac{1}{1-\widehat{\tau}}$) if no class collision is found in their negative samples.

(ii) However, since these major classes are so plentiful in the labeled dataset, they also have the highest probabilities of class collision. Therefore, even though these major-class-anchor-positive-pair tuples are frequently sub-sampled for ERM, a lot of them are penalized for class collision (by a subtraction term of $\frac{3\widehat{\tau}(N-2)}{k(1-\widehat{\tau})(N_r-2)}$).

(iii) On the other hand, tuples with rare classes as their anchor-positive pairs receive lower weights by default (of $\frac{1}{1-\widehat{\tau}} \prod_{\ell=0}^{k-1} \frac{N-\ell-N_r}{N-\ell-2}$). However, since we avoid class-collision for these classes altogether, they are not at all penalized and often possess higher weights than tuples with major-class anchor-positive pairs.

These effects interestingly make the sub-sampled estimator of $U_N$ possess a class-rarity-aware balancing feature that makes it favor tail classes more than $U_N^{\mathrm{hl}}$. However, we affirm our position that we **do not claim novelty** for this behavior since there has been a strong line of work in importance sampling and weighted empirical risk minimization prior to this work. We are primarily interested in providing sharper excess risk bounds.

### I.2. Experiment Settings

In this work, we experimented with both synthetic and real datasets to compare the generalization capabilities of representation functions on tail classes when trained using empirical risks computed in Algorithm 1 and Algorithm 2. For the synthetic dataset, we randomly draw data points from a mixture of Gaussian distributions $\bar{\mathcal{D}} = \sum_{r=1}^{R} \rho_r \mathcal{N}(\cdot|\mu_r, \sigma_r^2)$ where $\rho_{\max} = 0.5, \sigma_r^2 = 1$ for all $r \in [R]$ and other minor class probabilities decay exponentially. For real data experiments, we take small subsets of MNIST, FashionMNIST and CIFAR10 datasets such that, just like the synthetic dataset, tailed-ness is enforced. The specific class distributions of both synthetic and real datasets are visualized in Figure 4. For the full experiment configurations, please see Table 8.

**Hardware**: For all experiments, we use an Ubuntu desktop (Intel(R) Xeon(R) W-2133 CPU) with an **NVIDIA GeForce RTX 2080** GPU, **32GB RAM** and **12GB of GPU Memory**.

*Table 8.* Experiment settings for real and synthetic data experiments presented in the main text.

|  | Description | Synthetic | Real |
|---|---|---|---|
| $R$ | Number of classes | 20 | 10 |
| $N$ | Number of labeled data points | 5000 | 5000 |
| $M$ | Number of sub-sampled tuples | 3000 | 3000 |
| $k$ | Number of negative samples | 5 | 3, 5, 7 |

## I.3. Additional Experiments

In order to further verify that the proposed estimator $U_N$ outperforms the class-wise estimator $U_N^{\mathrm{hl}}$ in typical extreme multi-class scenarios, we conducted further experiments on MNIST, FashionMNIST, CIFAR10 and CIFAR100 with convolutional neural networks (CNNs). For MNIST, FashionMNIST and CIFAR10, we simulate the class-imbalanced distribution illustrated in Figure 4 as originally done for deep neural networks. For CIFAR100, we use the entire balanced dataset without modifying the class distribution. The CNN architecture used in this experiment comprises of three (Conv $\rightarrow$ BN $\rightarrow$ ReLU $\rightarrow$ MaxPool) blocks with $32 \rightarrow 64 \rightarrow 128$ channels, respectively. Aside from the comparison to $U_N^{\mathrm{hl}}$, we also ran this experiment on SupCon (Khosla et al., 2020) to provide a more comprehensive comparative study. Overall, the debiased estimator $U_N$ proposed in this work outperforms the class-wise estimator proposed in Hieu & Ledent (2025), as expected, with only a few exceptions on CIFAR10. Notably, in typical extreme multi-class scenario (CIFAR100), the performance of $U_N$ is better than that of both $U_N^{\mathrm{hl}}$ and SupCon.

*Table 9.* Average (macro) precisions, recalls and F1-scores of a simple *convolutional* neural network trained using sub-sampled estimations of $U_N$, $U_N^{\mathrm{hl}}$ and SupCon as empirical risks. The one with higher score between the two estimators $U_N$ and $U_N^{\mathrm{hl}}$ is highlighted in bold font. The metrics of SupCon is underlined if it outperforms both $U_N$, $U_N^{\mathrm{hl}}$.

| Metr. | ♯Neg. | MNIST (imbl) $U_N$ | $U_N^{\mathrm{hl}}$ | SupCon | FashionMNIST (imbl) $U_N$ | $U_N^{\mathrm{hl}}$ | SupCon | CIFAR10 (imbl) $U_N$ | $U_N^{\mathrm{hl}}$ | SupCon | CIFAR100 $U_N$ | $U_N^{\mathrm{hl}}$ | SupCon |
|---|---|---|---|---|---|---|---|---|---|---|---|---|---|
| | $k=3$ | **0.9835** | 0.9238 | | **0.4146** | 0.3487 | | **0.8128** | 0.7794 | | **0.3490** | 0.2836 | |
| Prec. | $k=5$ | **0.9898** | 0.5031 | 0.9874 | **0.9212** | 0.3560 | 0.9382 | **0.8078** | 0.8045 | 0.8057 | **0.3239** | 0.3196 | 0.2772 |
| | $k=7$ | **0.9903** | 0.2845 | | **0.9253** | 0.3765 | | 0.7977 | **0.8033** | | **0.3173** | 0.2778 | |
| | $k=3$ | **0.9809** | 0.7599 | | **0.4998** | 0.4076 | | **0.5649** | 0.5404 | | **0.3298** | 0.2675 | |
| Rec. | $k=5$ | **0.9869** | 0.4315 | 0.9801 | **0.8629** | 0.4071 | 0.8435 | **0.5507** | 0.5324 | 0.6045 | **0.3422** | 0.2710 | 0.3095 |
| | $k=7$ | **0.9791** | 0.3711 | | **0.8451** | 0.3966 | | **0.5623** | 0.5515 | | **0.2895** | 0.2544 | |
| | $k=3$ | **0.9821** | 0.7771 | | **0.4195** | 0.3222 | | **0.6592** | 0.6328 | | **0.3335** | 0.2723 | |
| F1 | $k=5$ | **0.9883** | 0.4209 | 0.9837 | **0.8862** | 0.3212 | 0.8745 | **0.6406** | 0.6366 | 0.6813 | **0.3306** | 0.2929 | 0.2919 |
| | $k=7$ | **0.9846** | 0.3141 | | **0.8744** | 0.3111 | | 0.6481 | **0.6505** | | **0.2997** | 0.2640 | |

# J. Concentration of Sums of Independent Variables

In this section, we recall some concentration inequalities required for our analysis. The results in this section as well as Sections K and L are classic and no claim of originality is made for these last three sections, which are included for completeness and self-containedness.

## J.1. Hoeffding's Inequality

Concentration inequalities are the central tools for most of the modern results in learning theory. One of the most popular results was dated back to the work of Hoeffding (1948), which focused concentration inequalities for sum of bounded independent random variables. We restate the key result below.

**Theorem J.1.** *(Hoeffding's Inequality - Hoeffding (1948, Theorem 2)) Let $X_1, \ldots, X_n$ be independent random variables with expected values $\mu_1, \ldots, \mu_n$ and $a_i \leqslant X_i \leqslant b_i$ with probability one for all $1 \leqslant i \leqslant n$. Let $\overline{X} = \frac{1}{n} \sum_{i=1}^{n} X_i$ and $\mu = \mathbb{E}[\overline{X}]$. Then, for any constant $\epsilon > 0$:*

$$\mathbb{P}(\overline{X} - \mu \geqslant \epsilon) \leqslant \exp\left(-\frac{2n^2\epsilon^2}{\sum_{i=1}^{n}(b_i - a_i)^2}\right). \tag{114}$$

The central idea for proving Theorem J.1 is to use the Chernoff bound to obtain the initial upper bound, which is a product of moment generating functions. Specifically, let $S_n = X_1 + \cdots + X_n = n\overline{X}$, we have:

$$\begin{aligned}
\mathbb{P}(\overline{X} - \mu \geqslant \epsilon) &= \mathbb{P}(S_n - n\mu \geqslant n\epsilon) \\
&\leqslant e^{-n\epsilon t}\mathbb{E}[e^{t(S_n - n\mu)}] \quad \text{(Chernoff Bound)} \\
&= e^{-n\epsilon t} \prod_{i=1}^{n} \mathbb{E}e^{t(X_i - \mu_i)} \quad \text{(By independence of } X_i\text{'s).}
\end{aligned}$$

Then, we can proceed to bound each moment generating function $\mathbb{E}e^{t(X_i - \mu_i)}$ separately using the Hoeffding's lemma. Another counterpart of Hoeffding's inequality is the Bernstein bound, which tend to be tighter, especially when the variances of the random variables considered are small. We state and provide the necessary technical lemma to prove the Bernstein bound below.

## J.2. Bernstein's Bound

**Lemma J.2.** *Let $X$ be a random variable with $\mathbb{E}[X] = 0$ and $\mathbb{E}[X^2] = \kappa^2$. Suppose that there is some $c > 0$ such that $|X| \leqslant c$. Then, for any $t > 0$, we have:*

$$\mathbb{E}e^{tX} \leqslant \exp\left(t^2\kappa^2\left(\frac{e^{tc} - 1 - tc}{t^2c^2}\right)\right). \tag{115}$$

**Proof.** Let $G = \sum_{k=2}^{\infty} \frac{t^{k-2}\mathbb{E}[X^k]}{k!\kappa^2}$. Using Taylor's expansion, we have:

$$\begin{aligned}
\mathbb{E}e^{tX} &= \mathbb{E}\left[1 + tX + \sum_{k=2}^{\infty} \frac{t^k X^k}{k!}\right] = 1 + \mathbb{E}\left[t^2 X^2 \sum_{k=2}^{\infty} \frac{t^{k-2} X^k}{k! X^2}\right] \\
&= 1 + t^2\kappa^2 \sum_{k=1}^{\infty} \frac{t^{k-2}\mathbb{E}[X^k]}{k!\kappa^2} = 1 + t^2\kappa^2 G.
\end{aligned}$$

For $k \geqslant 2$, we have $\mathbb{E}[X^k] = \mathbb{E}[X^2 X^{k-2}] \leqslant \kappa^2 c^{k-2}$. Therefore,

$$\begin{aligned}
G &\leqslant \sum_{k=2}^{\infty} \frac{t^{k-2} c^{k-2} \kappa^2}{k!\kappa^2} = \sum_{k=2}^{\infty} \frac{t^{k-2} c^{k-2}}{k!} \\
&= \frac{1}{t^2c^2}\left[\sum_{k=0}^{\infty} \frac{t^k c^k}{k!} - 1 - tc\right] = \frac{e^{tc} - 1 - tc}{t^2c^2}.
\end{aligned}$$

As a result, using the inequality $1 + x \leqslant e^x$, we have:

$$\mathbb{E}e^{tX} = 1 + t^2\kappa^2 G \leqslant e^{t^2\kappa^2 G} \leqslant \exp\left(t^2\kappa^2\left(\frac{e^{tc} - 1 - tc}{t^2 c^2}\right)\right),$$

as desired. □

**Theorem J.3** (Bernstein Bound (Boucheron et al. (2013) or Vershynin (2018) - Theorem 2.8.4)). *Let $X_1, \dots, X_n$ be independent random variables with 0 means. For all $i \in [n]$, let $\mathbb{E}[X_i^2] = \sigma_i^2$ and assume that $|X_i| \leqslant M$ for some $M > 0$. Then, for all $\epsilon > 0$, we have:*

$$\mathbb{P}(|\overline{X}| \geqslant \epsilon) \leqslant 2\exp\left(-\frac{n\epsilon^2/2}{\sigma^2 + M\epsilon/3}\right), \tag{116}$$

*where $\sigma^2 = \frac{1}{n}\sum_{i=1}^n \sigma_i^2$ and $\overline{X} = \frac{1}{n}\sum_{i=1}^n X_i$.*

**Proof.** First, we prove that $\mathbb{P}(\overline{X} \geqslant \epsilon) \leqslant \exp\left(-\frac{n\epsilon^2/2}{\sigma^2 + M\epsilon/3}\right)$. Using the Chernoff bound as usual:

$$\begin{aligned}
\mathbb{P}(\overline{X} \geqslant \epsilon) = \mathbb{P}\left(\sum_{i=1}^n X_i \geqslant n\epsilon\right) &\leqslant e^{-n\epsilon t}\prod_{i=1}^n \mathbb{E}^{tX_i} \qquad (t > 0) \\
&\leqslant e^{-n\epsilon t}\prod_{i=1}^n \exp\left(t^2\sigma_i^2\left(\frac{e^{tM} - 1 - tM}{t^2 M^2}\right)\right) \qquad \text{(Lemma J.2)} \\
&= e^{-n\epsilon t}\exp\left(nt^2\sigma^2\left(\frac{e^{tM} - 1 - tM}{t^2 M^2}\right)\right).
\end{aligned}$$

Solving for the optimal value of $t$, we have $t = M^{-1}\ln(1 + M\epsilon/\sigma^2)$. Let $f(x) = (1 + x)\ln(1 + x) - x$. Plugging the optimal value of $t$ back to the above right-hand-side, we have:

$$\mathbb{P}(\overline{X} \geqslant \epsilon) \leqslant \exp\left(-\frac{n\sigma^2}{M^2}f\left(\frac{M\epsilon}{\sigma^2}\right)\right) \leqslant \exp\left(-\frac{n\epsilon^2/2}{\sigma^2 + M\epsilon/3}\right),$$

where the last inequality holds because $f(x) \geqslant \frac{x^2/2}{1 + x/3}$ for all $x \geqslant 0$. We repeat the same arguments to prove that $\mathbb{P}(-\overline{X} \geqslant \epsilon) \leqslant \exp\left(-\frac{n\epsilon^2/2}{\sigma^2 + M\epsilon/3}\right)$ and combine with the above result to obtain the desired two-sided inequality. □

**Proposition J.4.** *(High-probability Bernstein Bound - See Ledent & Alves (2024, Prop. F.21)) Let $X_1, \dots, X_n$ be independent random variables with means 0. For $i \in [n]$, let $\mathbb{E}[X_i^2] = \sigma_i^2$ and assume that $|X_i| \leqslant M$ for some $M > 0$. Then, for any $\delta \in (0, 1)$, with probability of at least $1 - \delta$:*

$$|\overline{X}| \leqslant \frac{8M}{3n}\ln 2/\delta + \sigma\sqrt{\frac{8\ln 2/\delta}{3n}}, \tag{117}$$

*where $\sigma^2 = \frac{1}{n}\sum_{i=1}^n \sigma_i^2$ and $\overline{X} = \frac{1}{n}\sum_{i=1}^n X_i$.*

**Proof.** Fix $\delta \in (0, 1)$ and let $\epsilon > 0$. We divide the arguments into cases when $M\epsilon \geqslant \sigma^2$ and when $M\epsilon \leqslant \sigma^2$. For the case $M\epsilon \geqslant \sigma^2$, we have $\mathbb{P}(|\overline{X}| \geqslant \epsilon) \leqslant 2\exp\left(-\frac{n\epsilon^2/2}{4M\epsilon/3}\right)$. Setting the right-hand-side of the inequality to $\delta$ and solve for $\epsilon$, we have:

$$\frac{n\epsilon^2/2}{4M\epsilon/3} = \ln 2/\delta \implies \epsilon = \frac{8M}{3n}\ln 2/\delta.$$

In other words, with probability of at least $1 - \delta$, $|\overline{X}| \leqslant \frac{8M}{3n}\ln 2/\delta$. Similarly, for the case $\sigma^2 \geqslant M\epsilon$, we have $\mathbb{P}(|\overline{X}| \geqslant \epsilon) \leqslant 2\exp\left(-\frac{n\epsilon^2/2}{4\sigma^2/3}\right)$. Setting the right-hand-side to $\delta$ yields:

$$\frac{n\epsilon^2/2}{4\sigma^2/3} = \ln 2/\delta \implies \epsilon = \sigma\sqrt{\frac{8\ln 2/\delta}{3n}}.$$

Or, $|\overline{X}| \leqslant \sigma\sqrt{\frac{8\ln 2/\delta}{3n}}$ with probability of at least $1 - \delta$. Hence, for all cases, we have:

$$|\overline{X}| \leqslant \frac{8M}{3n}\ln 2/\delta + \sigma\sqrt{\frac{8\ln 2/\delta}{3n}}$$

with probability of at least $1 - \delta$. $\qquad\square$

**Proposition J.5.** *(Bernstein's Condition) Let $X$ be a random variable with variance $\mathrm{Var}(X) = \sigma^2$ and $\mathbb{E}[X] = \mu$. Suppose that $|X - \mu| \leqslant M$ with probability one. Then, we have:*

$$\mathbb{E}e^{\lambda(X-\mu)} \leqslant \exp\left(\frac{\lambda^2\sigma^2/2}{1 - M|\lambda|/3}\right), \quad \forall|\lambda| < \frac{3}{M}.$$

**Proof.** Using Taylor expansion on the moment-generating function $\mathbb{E}e^{\lambda(X-\mu)}$, we have:

$$\mathbb{E}e^{\lambda(X-\mu)} = 1 + \sum_{k=1}^{\infty}\frac{\lambda^k\mathbb{E}[(X-\mu)^k]}{k!} = 1 + \sum_{k=2}^{\infty}\frac{\lambda^k\mathbb{E}[(X-\mu)^k]}{k!}$$

$$= 1 + \sum_{k=2}^{\infty}\frac{\lambda^k\mathbb{E}[(X-\mu)^2(X-\mu)^{k-2}]}{k!}$$

$$\leqslant 1 + \sum_{k=2}^{\infty}\frac{\lambda^k\mathbb{E}[(X-\mu)^2|X-\mu|^{k-2}]}{k!}$$

$$\leqslant 1 + \sum_{k=2}^{\infty}\frac{\lambda^k\mathbb{E}[(X-\mu)^2 M^{k-2}]}{k!}$$

$$= 1 + \sum_{k=2}^{\infty}\frac{\lambda^k\sigma^2 M^{k-2}}{k!} \leqslant 1 + \lambda^2\sigma^2\sum_{k=1}^{\infty}\frac{(M|\lambda|)^{k-2}}{k!}.$$

Using the inequality $k! \geqslant 2 \cdot 3^{k-2}$ for any $k \geqslant 2$, for any $|\lambda| < 3/M$ we have:

$$\mathbb{E}e^{\lambda(X-\mu)} \leqslant 1 + \lambda^2\sigma^2\sum_{k=2}^{\infty}\frac{(M|\lambda|)^{k-2}}{k!}$$

$$\leqslant 1 + \frac{\lambda^2\sigma^2}{2}\sum_{k=2}^{\infty}\left(\frac{M|\lambda|}{3}\right)^{k-2} = 1 + \frac{\lambda^2\sigma^2/2}{1 - M|\lambda|/3}$$

$$\leqslant \exp\left(\frac{\lambda^2\sigma^2/2}{1 - M|\lambda|/3}\right) \qquad (1 + x \leqslant e^x).$$

Using the same approach, we can also show that:

$$\mathbb{E}e^{\lambda(\mu-X)} \leqslant \exp\left(\frac{\lambda^2\sigma^2/2}{1 - M|\lambda|/3}\right), \quad \forall|\lambda| < \frac{3}{M}.$$

Hence, for all $|\lambda| < \frac{3}{M}$, we have:

$$\mathbb{E}e^{\lambda|X-\mu|} \leqslant \mathbb{E}e^{\lambda(X-\mu)} + \mathbb{E}e^{\lambda(\mu-X)} \leqslant 2\exp\left(\frac{\lambda^2\sigma^2/2}{1 - M|\lambda|/3}\right).$$

$\qquad\square$

The above inequality can be extended to the following matrix Bernstein inequality.

**Proposition J.6** (Non-commutative Matrix Bernstein Inequality (cf. Ledent et al. (2021a), Proposition F.3 or Recht (2011), Theorem 4))**.** *Let* $X_1, \ldots, X_n$ *be independent zero-mean matrices of dimension* $m \times n$. *For all* $1 \leqslant k \leqslant n$, *assume that* $\|X_k\|_\sigma \leqslant M$ [7] *almost surely. Denote* $\rho_k^2 = \max\left(\|\mathbb{E}[X_k X_k^\top]\|_\sigma, \|\mathbb{E}[X_k^\top X_k]\|_\sigma\right)$. *Then, for any* $\lambda > 0$:

$$\mathbb{P}\left(\left\|\sum_{k=1}^n X_k\right\|_\sigma \geqslant \lambda\right) \leqslant (m+n)\exp\left(-\frac{\lambda^2/2}{\sum_{k=1}^n \rho_k^2 + M\lambda/3}\right). \tag{118}$$

### J.3. Chernoff's Bound

**Proposition J.7** (Multiplicative Chernoff Bound (cf. Lemma F.15, Ledent & Alves (2024)))**.** *Let* $X_1, \ldots, X_N$ *be independent random variables taking* ***values in*** $\{0,1\}$. *Let* $S_N = \sum_{j=1}^N X_j$ *and* $\mu = \mathbb{E}[S_N]$. *Then, for any* $\delta > 0$:

$$\mathbb{P}(S_N \geqslant (1+\delta)\mu) \leqslant \left(\frac{e^\delta}{(1+\delta)^{1+\delta}}\right)^\mu. \tag{119}$$

*Similarly, we can show that for* $\delta \in (0,1)$:

$$\mathbb{P}(S_N \leqslant (1-\delta)\mu) \leqslant \left(\frac{e^{-\delta}}{(1-\delta)^{1-\delta}}\right)^\mu. \tag{120}$$

**Proof.** For $t > 0$, we have:

$$\begin{aligned}
\mathbb{P}(S_N \geqslant (1+\delta)\mu) &= \mathbb{P}\left(e^{tS_N} \geqslant e^{t\mu(1+\delta)}\right) \\
&\leqslant e^{-t\mu(1+\delta)} M_{S_N}(t) \qquad \text{(Markov's Inequality)} \\
&\leqslant e^{-t\mu(1+\delta)} \prod_{j=1}^N M_{X_j}(t) \qquad \text{(Independence).}
\end{aligned}$$

Then, suppose that for each $j \in [N]$, $\mathbb{E}[X_j] = p_j$. Then, we have:

$$M_{X_j}(t) = 1 + p_j(e^t - 1).$$

Plugging this back to the above inequality, we have:

$$\begin{aligned}
\mathbb{P}(S_N \geqslant (1+\delta)\mu) &\leqslant e^{-t\mu(1+\delta)} \prod_{j=1}^N [1 + p_j(e^t - 1)] \leqslant e^{-t\mu(1+\delta)} \prod_{j=1}^N e^{p_j(e^t - 1)} \qquad (1 + x \leqslant e^x) \\
&= \frac{\exp\left[(e^t - 1)\sum_{j=1}^N p_j\right]}{\exp[t\mu(1+\delta)]} \\
&= \left(\frac{\exp[e^t - 1]}{\exp[t(1+\delta)]}\right)^\mu \qquad (\sum_{j=1}^N p_j = \mu).
\end{aligned}$$

Solving for the optimal value of $t$, we have $t = \ln(1+\delta)$. Plugging this to the inequality:

$$\mathbb{P}(S_N \geqslant (1+\delta)\mu) \leqslant \left(\frac{e^\delta}{(1+\delta)^{1+\delta}}\right)^\mu, \tag{121}$$

as desired. We can use the same proof strategy to prove the other inequality for $\delta \in (0,1)$. $\square$

**Corollary J.8** (Multiplicative Chernoff Bound)**.** *Let* $S_N$ *be defined in Theorem J.7, we have:*

$$\mathbb{P}(S_N \geqslant (1+\delta)\mu) \leqslant e^{-\delta^2\mu/(2+\delta)}, \qquad\qquad 0 \leqslant \delta. \tag{122}$$

$$\mathbb{P}(S_N \leqslant (1-\delta)\mu) \leqslant e^{-\delta^2\mu/2}, \qquad\qquad 0 \leqslant \delta \leqslant 1. \tag{123}$$

$$\mathbb{P}(|S_N - \mu| \geqslant \delta\mu) \leqslant 2e^{-\delta^2\mu/3}, \qquad\qquad 0 \leqslant \delta \leqslant 1. \tag{124}$$

---

[7] Where $\|\cdot\|_\sigma$ denotes the spectral norm.

**Proof.** Using the logarithm inequality $\ln(1 + \delta) \geqslant \frac{2\delta}{2+\delta}$, from Eqn.119, we have:

$$\mathbb{P}(S_N \geqslant (1+\delta)\mu) \leqslant \left(\frac{e^\delta}{(1+\delta)^{1+\delta}}\right)^\mu = e^{\mu\delta}(1+\delta)^{-\mu(1+\delta)}$$

$$\leqslant \exp\left(\mu\delta - \frac{2\mu\delta(1+\delta)}{2+\delta}\right) = e^{-\delta^2\mu/(2+\delta)}.$$

Similarly, using the inequality $\ln(1 - \delta) \geqslant \frac{2\delta}{2-\delta}$ for $\delta \in (0, 1)$, from Eqn. (120), we have:

$$\mathbb{P}(S_N \leqslant (1-\delta)\mu) \leqslant \left(\frac{e^{-\delta}}{(1-\delta)^{1-\delta}}\right)^\mu = e^{-\mu\delta}(1-\delta)^{-\mu(1-\delta)}$$

$$\leqslant \exp\left(-\mu\delta + \frac{2\mu\delta(1-\delta)}{2-\delta}\right) = e^{-\delta^2\mu/(2-\delta)}$$

$$\leqslant e^{-\delta^2\mu/2}.$$

Finally, for $\delta \in (0, 1)$, we have:

$$\mathbb{P}(|S_N - \mu| \geqslant \delta\mu) \leqslant \mathbb{P}(S_N - \mu \geqslant \delta\mu) + \mathbb{P}(S_N - \mu \leqslant -\delta\mu)$$

$$\leqslant e^{-\delta^2\mu/(2+\delta)} + e^{-\delta^2\mu/2}$$

$$\leqslant 2e^{-\delta^2\mu/3}.$$

$\square$

## K. Concentration of U-Statistics & V-Statistics

In this section, we recall some classic results on U-statistics and V-statistics, which form key building blocks of our approach. One limitation of inequalities in Hoeffding (1948) or Vershynin (2018, Theorem 2.8.4) is that the initial Chernoff bound can be decomposed elegantly to a product of moment generating functions thanks to the independence assumption. However, this approach is inapplicable to more complex sum structures involving dependence. To leverage the arguments of Theorem J.1, we focus on the following specific class of sums of dependent random variables:

$$S = \sum_{i=1}^{N} \rho_i S_i, \text{ where } \rho_i \geqslant 0, \forall i \in [N] \text{ and } \sum_{i=1}^{N} \rho_i = 1, \tag{125}$$

where each $S_i$ is a **sum of (bounded) independent random variables** and these sums are **not mutually independent**. Suppose that for all $i \in [N]$, $S_i$ is a sum of $n_i$ independent random variables. Furthermore, for simplicity, suppose that all random variables in the sum $S_i$ have the same mean $\mu_i$ and $\mathbb{E}S_i = n_i\mu_i$, which means $\mathbb{E}S = \sum_{i=1}^{N} \rho_i n_i \mu_i$. By the Chernoff bound, for $t > 0$:

$$\mathbb{P}(S - \mathbb{E}S \geqslant \epsilon) \leqslant e^{-t\epsilon}\mathbb{E}\exp\left(t(S - \mathbb{E}S)\right) = e^{-t\epsilon}\mathbb{E}\exp\left(\sum_{i=1}^{N} \rho_i t(S_i - n_i\mu_i)\right)$$

$$\leqslant e^{-t\epsilon}\mathbb{E}\left[\sum_{i=1}^{N} \rho_i \exp\left(t(S_i - n_i\mu_i)\right)\right] \quad \text{(Jensen's Inequality)}$$

$$= e^{-t\epsilon}\sum_{i=1}^{N} \rho_i \mathbb{E}\exp\left(t(S_i - n_i\mu_i)\right).$$

Since for all $i \in [N]$, $S_i$ is a sum of independent random variables, the inequality in Theorem J.1 apply. Specifically, suppose that for $i \in [N]$, $S_i = \sum_{j=1}^{n_i} X_{ij}$, we have:

$$\mathbb{E}\exp\left(t(S_i - n_i\mu_i)\right) = \mathbb{E}\exp\left(\sum_{j=1}^{n_i} t(X_{ij} - \mu_i)\right) = \prod_{j=1}^{n_i} \mathbb{E}e^{t(X_{ij}-\mu_i)}.$$

Therefore, for each $i \in [N]$, we can proceed to use the same arguments in Thm. J.1 to bound the expectation $\mathbb{E}e^{t(X_{ij}-\mu_i)}$. In the following section, we will consider some classes of statistics that takes similar form in Eqn. (125).

### K.1. U-Statistics

**One-sample U-Statistics**: Let $X_1, \ldots, X_n$ be identically and independently distributed random variables on $\mathcal{X}$ and $g : \mathcal{X}^r \to \mathbb{R}$ be a function symmetric in its arguments, i.e., $g(x_1, \ldots, x_r) = g(x_{i_1}, \ldots, x_{i_r})$ for all $1 \leqslant i_1, \ldots, i_r \leqslant r$ and $i_j \neq i_k$ for all $j, k \in [r]$. The one-sample U-Statistic (of order $r$) used to estimate $\mathbb{E}g(X_1, \ldots, X_r)$ is defined as:

$$U_n(g) = \frac{1}{\binom{n}{r}} \sum_{i_1, \ldots, i_r \in C_{n,r}} g(X_{i_1}, \ldots, X_{i_r}), \tag{126}$$

where $C_{n,r}$ denotes the set of all $r$-element tuples selected without replacement from the indices set $[n]$. Let $q = \lfloor n/r \rfloor$ and define $V(X_1, \ldots, X_n)$ as follows:

$$\widetilde{U}(X_1, \ldots, X_n) = \frac{1}{q} \sum_{j=1}^{q} g(X_{jr-r+1}, \ldots, X_{jr}). \tag{127}$$

Then, clearly $\widetilde{U}(X_1, \ldots, X_n)$ is an average over the values of $g$ applied on independent $r$-element blocks of the sequence of random variables $X_1, \ldots, X_n$. Then, we can re-define $U_n(g)$ as follows:

$$U_n(g) = \frac{1}{n!} \sum_{\pi \in \Pi[n]} \widetilde{U}(X_{\pi,1}, \ldots, X_{\pi,n}), \quad \Pi[n] = \left\{ \pi : [n] \to [n] \,\middle|\, \pi \text{ bijective} \right\}, \tag{128}$$

which is consistent with the form in Eqn. (125). This decoupling technique was later rigorously utilized by later works (Arcones & Gine, 1993; de la Peña & Giné, 1998) to derive concentration inequalities for U-Statistics. If $g(X_1, \ldots, X_r) \in [a, b]$ with probability one for some $a, b \in \mathbb{R}$, applying Theorem J.1, for all $\epsilon > 0$, we have:

$$\mathbb{P}(U_n(g) - \mathbb{E}U_n(g) \geqslant \epsilon) \leqslant e^{-2q\epsilon^2/(b-a)^2}. \tag{129}$$

**Two-sample (or Multi-sample) U-Statistics**: Let $X_1, \ldots, X_n$ be a sample drawn i.i.d. from a distribution over $\mathcal{X}$ and $Y_1, \ldots, Y_m$ be another sample drawn i.i.d. from another distribution over $\mathcal{X}$. Let $g : \mathcal{X}^{r+s} \to \mathbb{R}$ be a kernel that is symmetric in the first $r$ arguments and the subsequent $s$ arguments ($r \leqslant n$ and $s \leqslant m$). Then, we define the two-sample U-Statistic of orders $r$ and $s$ as follows:

$$U_{n,m}(g) = \frac{1}{\binom{n}{r} \times \binom{m}{s}} \sum_{\substack{i_1, \ldots, i_r \in C_{n,r} \\ j_1, \ldots, j_s \in C_{m,s}}} g(X_{i_1}, \ldots, X_{i_r}, Y_{j_1}, \ldots, Y_{j_s}). \tag{130}$$

Let $q = \min(\lfloor n/r \rfloor, \lfloor m/s \rfloor)$. Using the same decoupling trick, we have:

$$\mathbb{P}(U_{n,m}(g) - \mathbb{E}U_{n,m}(g) \geqslant \epsilon) \leqslant e^{-2q\epsilon^2/(b-a)^2}, \tag{131}$$

if we know that $g(X_1, \ldots, X_r, Y_1, \ldots, Y_s) \in [a, b]$ with probability one for some $a, b \in \mathbb{R}$.

### K.2. V-Statistics

Let $X_1, \ldots, X_n$ be identically and independently distributed random variables on $\mathcal{X}$ and $g : \mathcal{X}^r \to \mathbb{R}$ be a function symmetric in its arguments. The one-sample V-Statistics of order $r$ is defined as the average over all $r$-tuples sampled *with replacement* from $X_1, \ldots, X_n$:

$$V_{n,r}(g) = \frac{1}{n^r} \sum_{i_1=1}^{n} \cdots \sum_{i_r=1}^{n} g(X_{i_1}, \ldots, X_{i_r}) \tag{132}$$

$$= \frac{1}{n^r} \sum_{t \in [n]^r} g(\mathbf{X}_t), \tag{133}$$

where $[n]^r$ is the set of $r$-tuples sampled with replacement and we use $\mathbf{X}_t = (X_{t_1}, \ldots, X_{t_r})$ for any tuple of indices $t = (t_1, \ldots, t_n)$. The V-Statistics can be re-formulated as a U-Statistic, i.e., an average over the the $r$-tuples sampled *without*

*replacement* from $X_1, \ldots, X_n$. Let $C_{n,r}$ denote the set of $r$-tuples selected with no replacement from the indices set $[n]$, we can re-write the V-Statistics as follows:

$$V_{n,r}(g) = \frac{1}{\binom{n}{r}} \sum_{t \in C_{n,r}} \binom{n}{r} n^{-r} \sum_{b \in S(t)} \omega_b g(\mathbf{X}_b), \tag{134}$$

where $S(t)$ is the set of $r$-tuples selected with replacement from $t \in C_{n,r}$ and $\omega_b = \binom{n-u(b)}{r-u(b)}^{-1}$ where $1 \leqslant u(b) \leqslant r$ is the number of unique elements from $b$. The rescaling factor $\omega_b$ accounts for the re-appearance of $b$ in other tuples $t' \in C_{n,r}$. We can further break down $V_{n,r}(g)$ as follows:

$$V_{n,r}(g) = \frac{1}{\binom{n}{r}} \sum_{t \in C_{n,r}} \binom{n}{r} n^{-r} \sum_{u=1}^{r} \sum_{b \in S_u(t)} \frac{1}{\binom{n-u}{r-u}} g(\mathbf{X}_b), \tag{135}$$

where $S_u(t) \subset S(t)$ is the subset of tuples with $u$ unique elements. For $1 \leqslant u \leqslant r$, we have:

$$|S_u(t)| = u! S(r,u) \binom{r}{u},$$

where $S(r,u)$ is the Stirling number of the second kind. To justify the formula for the cardinality of $S_u(t)$, we can select a $r$-tuple with $u$ unique elements from an original set of $r$ distinct elements as follows:

- Select $u$ unique elements for the subset ($\binom{r}{u}$ ways).

- Among the selected unique elements, partition the $r$ positions of the tuple to $u$ bins, which correspond to the unique elements ($S(r,u)$ ways).

- Since the unique elements can be in any arbitrary order, there is an additional multiplicative factor of $u!$.

We can further verify that $\sum_{u=1}^{r} |S_u(t)| = r^r$, which is the total number of ways to sample from any $t \in C_{n,r}$ with replacement. Now, we define $g^* : \mathcal{X}^r \to \mathbb{R}$ as

$$\forall t \in C_{n,r} : g^*(\mathbf{X}_t) = \binom{n}{r} n^{-r} \sum_{u=1}^{r} \sum_{b \in S_u(t)} \frac{1}{\binom{n-u}{r-u}} g(\mathbf{X}_b), \tag{136}$$

we can say that $V_{n,r}(g)$ is a U-Statistic with kernel $g^*$. Now, we would like to show that $g^*(\mathbf{X}_t)$ is a weighted average of $g(\mathbf{X}_b)$ for $b \in S(t)$. In other words, we would like to show that $\sum_{u=1}^{r} \sum_{b \in S_u(t)} \frac{\binom{n}{r}}{n^r \binom{n-u}{r-u}} = 1$. We have:

$$\begin{aligned}
\sum_{u=1}^{r} \sum_{b \in S_u(t)} \frac{1}{\binom{n-u}{r-u}} &= \sum_{u=1}^{r} |S_u(t)| \cdot \frac{1}{\binom{n-u}{r-u}} = \sum_{u=1}^{r} u! S(r,u) \binom{r}{u} \cdot \frac{1}{\binom{n-u}{r-u}} \\
&= \sum_{u=1}^{r} u! S(r,u) \frac{r!}{u!(r-u)!} \cdot \frac{(r-u)!(n-r)!}{(n-u)!} \\
&= \sum_{u=1}^{r} S(r,u) \frac{r!(n-r)!}{(n-u)!} = \binom{n}{r}^{-1} \sum_{u=1}^{r} S(r,u) \frac{n!}{(n-u)!}.
\end{aligned} \tag{137}$$

It is a well-known property of Stirling number that $\sum_{u=1}^{r} S(r,u) \frac{n!}{(n-u)!} = n^r$ (For example, see Adell (2024))[8]. Hence, we have $\sum_{u=1}^{r} \sum_{b \in S_u(t)} \frac{1}{\binom{n-u}{r-u}} = \frac{n^r}{\binom{n}{r}}$, as desired.

---

[8]Similarly, $\sum_{u=1}^{r} |S_u(t)| = \sum_{u=1}^{r} u! S(r,u) \binom{r}{u} = \sum_{u=1}^{r} S(r,u) \frac{r!}{(r-u)!} = r^r$ for all $t \in C_{n,r}$, which is the number of ways to sample $r$ elements with replacement from a set with $t$ elements.

# L. Classic Learning Theory Results

## L.1. Rademacher Complexity

**Definition L.1** (Rademacher Complexity). Let $\mathcal{Z}$ be a vector space and $\mathcal{G}$ be a class of functions $g : \mathcal{Z} \to [a, b]$ where $a, b \in \mathbb{R}$ and $a < b$. Let $S = \{\mathbf{z}_1, \ldots, \mathbf{z}_n\}$ be independent random variables such that $\mathbf{z}_i \sim P_i, \forall i \in [n]$. Then, the *empirical Rademacher complexity* of $\mathcal{G}$ is defined as:

$$\widehat{\mathfrak{R}}_S(\mathcal{G}) = \mathbb{E}_{\mathbf{\Sigma}_n} \left[ \sup_{g \in \mathcal{G}} \frac{1}{n} \left| \sum_{i=1}^n \sigma_i g(\mathbf{z}_i) \right| \right], \tag{138}$$

where $\mathbf{\Sigma}_n = (\sigma_1, \ldots, \sigma_n)$ is a vector of $n$ independent Rademacher variables. Additionally, the *expected Rademacher complexity* of $\mathcal{G}$ is defined as follows:

$$\mathfrak{R}_{\mathcal{Q}}(\mathcal{G}) = \mathbb{E}_S \left[ \widehat{\mathfrak{R}}_S(\mathcal{G}) \right], \tag{139}$$

where $\mathcal{Q} = \bigotimes_{i=1}^n P_i$ is the distribution of $S$. Intuitively, the Rademacher complexity is a measure of a function class' richness. If a class $\mathcal{G}$ is sufficiently diverse, there is a higher chance that given a random sequence of signs (represented by the sequence of Rademacher variables), we will be able to find a function $g \in \mathcal{G}$ that matches the signs. Hence, the Rademacher complexity will be large.

**Lemma L.2.** *Let $\mathcal{Z}$ be a vector space and $\mathcal{G}$ be a class of functions $g : \mathcal{Z} \to [a, b]$ where $a, b \in \mathbb{R}$ and $a < b$. Let $S = \{\mathbf{z}_1, \ldots, \mathbf{z}_n\}$ be a dataset of independent random variables such that each $\mathbf{z}_i \sim P_i, \forall i \in [n]$. Then, for any $\delta \in (0, 1)$, with probability of at least $1 - \delta$, we have:*

$$\mathfrak{R}_{\mathcal{Q}}(\mathcal{G}) \leqslant \widehat{\mathfrak{R}}_S(\mathcal{G}) + (b - a) \sqrt{\frac{\ln 1/\delta}{2n}}, \tag{140}$$

*where $\mathcal{Q} = \bigotimes_{i=1}^n P_i$ is the distribution of the whole dataset $S$.*

**Proof.** Let $\phi : \mathcal{Z}^n \to \mathbb{R}_+$ be defined as $\phi(x_1, \ldots, x_n) = \mathbb{E}_{\mathbf{\Sigma}_n} \left[ \sup_{g \in \mathcal{G}} \frac{1}{n} \left| \sum_{j=1}^n \sigma_j g(x_j) \right| \right]$ where $x_i \in \mathcal{Z}$ for all $1 \leqslant i \leqslant n$. Then, for all $1 \leqslant 1 \leqslant n$, we have:

$$\sup_{x_i, x_i' \in \mathcal{Z}} |\phi(x_1, \ldots, x_i, \ldots, x_n) - \phi(x_1, \ldots, x_i', \ldots, x_n)| \leqslant \frac{b - a}{n}.$$

Hence, by McDiarmid's inequality (McDiarmid, 1989), for any $\epsilon > 0$, we have:

$$\mathbb{P} \Big( \underbrace{\mathbb{E}\phi(\mathbf{z}_1, \ldots, \mathbf{z}_n)}_{\mathfrak{R}_{\mathcal{Q}}(\mathcal{G})} - \underbrace{\phi(\mathbf{z}_1, \ldots, \mathbf{z}_n)}_{\widehat{\mathfrak{R}}_S(\mathcal{G})} \geqslant \epsilon \Big) \leqslant \exp \left( -\frac{2n\epsilon^2}{(b - a)^2} \right).$$

Setting the right-hand-side to $\delta$, we have $\epsilon = (b - a) \sqrt{\frac{\ln 1/\delta}{2n}}$ and we obtained the desired bound. $\qquad\square$

**Proposition L.3** (Rademacher Complexity Bound). *Let $\mathcal{Z}$ be a vector space and $\mathcal{G}$ be a class of functions $g : \mathcal{Z} \to [a, b]$ where $a, b \in \mathbb{R}$ and $a < b$. Let $S = \{\mathbf{z}_1, \ldots, \mathbf{z}_n\}$ be a dataset of independent variables such that $\mathbf{z}_i \sim P_i, \forall i \in [n]$. Then, for any $\delta \in (0, 1)$, with probability of at least $1 - \delta$:*

$$\sup_{g \in \mathcal{G}} \left| \Theta(g) - \frac{1}{n} \sum_{i=1}^n g(\mathbf{z}_i) \right| \leqslant 2\widehat{\mathfrak{R}}_S(\mathcal{G}) + 3(b - a) \sqrt{\frac{\ln 4/\delta}{2n}}. \tag{141}$$

*where $\Theta(g) = \frac{1}{n} \sum_{i=1}^m \mathbb{E}_{\mathbf{z} \sim P_i}[g(\mathbf{z})]$.*

**Proof.** Let $\phi : \mathcal{Z}^n \to \mathbb{R}$ be defined as $\phi(x_1, \ldots, x_n) = \sup_{g \in \mathcal{G}} \left[ \Theta(g) - \frac{1}{n} \sum_{i=1}^n g(x_i) \right]$ and $\mathcal{Q} = \bigotimes_{i=1}^n P_i$, i.e., the distribution of $S$. Using McDiarmid's inequality, for all $\Delta \in (0, 1)$, the following inequality holds with probability of at

least $1 - \Delta/2$:

$$\phi(\mathbf{z}_1, \ldots, \mathbf{z}_n) \leqslant \mathbb{E}_S \phi(\mathbf{z}_1, \ldots, \mathbf{z}_n) + (b - a) \sqrt{\frac{\ln 2/\Delta}{2n}}$$

$$\leqslant 2 \mathfrak{R}_{\mathcal{Q}}(\mathcal{G}) + (b - a) \sqrt{\frac{\ln 2/\Delta}{2n}}. \qquad \text{(Symmetrization - Lemma L.7)}$$

Furthermore, we have $\mathfrak{R}_{\mathcal{Q}}(\mathcal{G}) \leqslant \widehat{\mathfrak{R}}_S(\mathcal{G}) + (b - a) \sqrt{\frac{\ln 2/\Delta}{2n}}$ with probability of at least $1 - \Delta/2$ (Lemma L.2). Hence, by the union bound, with probability of at least $1 - \Delta$, we have:

$$\sup_{g \in \mathcal{G}} \left[ \Theta(g) - \frac{1}{n} \sum_{i=1}^{n} g(x_i) \right] \leqslant 2 \widehat{\mathfrak{R}}_S(\mathcal{G}) + 3(b - a) \sqrt{\frac{\ln 2/\Delta}{2n}}.$$

Let $\phi(x_1, \ldots, x_n) = \sup_{g \in \mathcal{G}} \left[ \frac{1}{n} \sum_{i=1}^{n} g(x_i) - \Theta(g) \right]$ and repeat the above argument, we have the inequality in the other direction with probability of at least $1 - \Delta$. Hence, by the union bound, with probability of at least $1 - 2\Delta$, we have the following two-sided inequality:

$$\sup_{g \in \mathcal{G}} \left| \Theta(g) - \frac{1}{n} \sum_{i=1}^{n} g(\mathbf{z}_i) \right| \leqslant 2 \widehat{\mathfrak{R}}_S(\mathcal{G}) + 3(b - a) \sqrt{\frac{\ln 2/\Delta}{2n}}.$$

Setting $\Delta = \delta/2$ completes the proof. $\qquad \square$

### L.2. Massart Lemma & Dudley's Entropy Integral

**Lemma L.4** (Massart's Finite Lemma). *Given $S = \{\mathbf{z}_1, \ldots, \mathbf{z}_n\}$ be a sample of independent random variables. Let $\mathcal{G}$ be a finite function class. Then, we have:*

$$\mathbb{E}_{\boldsymbol{\Sigma}_n} \left[ \sup_{g \in \mathcal{G}} \frac{1}{n} \left| \sum_{i=1}^{n} \sigma_i g(\mathbf{z}_i) \right| \right] \leqslant B \sqrt{\frac{2 \ln 2 |\mathcal{G}|}{n}}, \qquad (142)$$

*where $\boldsymbol{\Sigma}_n = (\sigma_j)_{j \in [n]}$ are independent Rademacher variables, $B = \sup_{g \in \mathcal{G}} \left( \frac{1}{n} \sum_{i=1}^{n} |g(\mathbf{z}_i)|^2 \right)^{1/2}$.*

**Proof.** For each $h \in \mathcal{G}$, we denote $\theta_h = \frac{1}{n} \sum_{i=1}^{n} \sigma_i g(\mathbf{z}_i)$. Then, for $\lambda > 0$, we have:

$$\lambda \mathbb{E}_{\boldsymbol{\Sigma}_n} \left[ \sup_{h \in \mathcal{G}} \frac{1}{n} \left| \sum_{i=1}^{n} \sigma_i g(\mathbf{z}_i) \right| \right] = \lambda \mathbb{E}_{\boldsymbol{\Sigma}_n} \left[ \max_{h \in \mathcal{G}} |\theta_h| \right] \quad (\sup \to \max \text{ due to finite } \mathcal{G})$$

$$= \ln \exp \lambda \mathbb{E}_{\boldsymbol{\Sigma}_n} \left[ \max_{h \in \mathcal{G}} |\theta_h| \right]$$

$$= \ln \exp \lambda \mathbb{E}_{\boldsymbol{\Sigma}_n} \left[ \max_{h \in \mathcal{G}} \max(\theta_h, -\theta_h) \right]$$

$$\leqslant \ln \mathbb{E}_{\boldsymbol{\Sigma}_n} \exp \left( \lambda \max_{h \in \mathcal{G}} \max(\theta_h, -\theta_h) \right) \quad \text{(Jensen's Inequality)}$$

$$= \ln \mathbb{E}_{\boldsymbol{\Sigma}_n} \max_{h \in \mathcal{G}} \exp \left( \lambda \max(\theta_h, -\theta_h) \right) \quad (\exp(\lambda x) \text{ is increasing})$$

$$\leqslant \ln \sum_{h \in \mathcal{G}} \mathbb{E}_{\boldsymbol{\Sigma}_n} \left[ \exp(\lambda \theta_h) + \exp(-\lambda \theta_h) \right] \quad (\exp \max \leqslant \exp \text{ sum})$$

$$= \ln 2 \sum_{h \in \mathcal{G}} \mathbb{E}_{\boldsymbol{\Sigma}_n} \left[ \exp(\lambda \theta_h) \right] \quad \text{(By symmetry of } \sigma)$$

$$= \ln 2 \sum_{h \in \mathcal{G}} \prod_{i=1}^{n} \mathbb{E}_{\boldsymbol{\Sigma}_n} \left[ \exp \left( \frac{\lambda}{n} \sigma_i g(\mathbf{z}_i) \right) \right] \quad (\sigma_i \text{ are independent})$$

$$= \ln 2 \sum_{h \in \mathcal{G}} \prod_{i=1}^{n} \frac{\exp \left( -\frac{\lambda}{n} g(\mathbf{z}_i) \right) + \exp \left( \frac{\lambda}{n} g(\mathbf{z}_i) \right)}{2}.$$

Using the inequality $e^x + e^{-x} \leqslant 2e^{\frac{x^2}{2}}$, we have:

$$\lambda \mathbb{E}_{\mathbf{\Sigma}_n} \left[ \sup_{h \in \mathcal{G}} \frac{1}{n} \left| \sum_{i=1}^n \sigma_i g(\mathbf{z}_i) \right| \right] \leqslant \ln 2 \sum_{h \in \mathcal{G}} \prod_{i=1}^n \exp\left( \frac{\lambda^2 g(\mathbf{z}_i)^2}{2n^2} \right)$$

$$= \ln 2 \sum_{h \in \mathcal{G}} \exp\left( \frac{\lambda^2 \sum_{i=1}^n g(\mathbf{z}_i)^2}{2n^2} \right)$$

$$\leqslant \ln 2 |\mathcal{G}| \cdot \exp\left( \frac{\lambda^2 B^2}{2n} \right)$$

$$= \ln 2 |\mathcal{G}| + \frac{\lambda^2 B^2}{2n}.$$

From the above, let $\lambda = B^{-1} \sqrt{2n \ln 2|\mathcal{G}|}$ and we obtain the desired bound. $\qquad\square$

**Theorem L.5** (Dudley's Entropy Integral). *Let $\mathcal{G}$ be a real-valued function class and the dataset $S = \{\mathbf{z}_1, \ldots, \mathbf{z}_n\}$ consists of independent random variables. Then, we have:*

$$\widehat{\mathfrak{R}}_S(\mathcal{G}) \leqslant \inf_{\alpha > 0} \left( 4\alpha + 12 \int_\alpha^B \sqrt{\frac{\ln 2\mathcal{N}(\mathcal{G}, \epsilon, \mathrm{L}_2(S))}{n}} d\epsilon \right), \tag{143}$$

*where $B = \sup_{g \in \mathcal{G}} \left( \frac{1}{n} \sum_{i=1}^n g(\mathbf{z}_i)^2 \right)^{1/2}$ and $\widehat{\mathfrak{R}}_S(\mathcal{G})$ is the empirical Rademacher complexity of the function class $\mathcal{G}$ (given the dataset $S$).*

**Proof.** The above result is derived by a standard chaining argument (See, for example, Ledent et al. (2021b, Proposition 22) or Bartlett et al. (2017, Lemma A.5)). Then, apply Massart's finite lemma. The difference is that instead of using the standard Massart's lemma for the regular notion of Rademacher Complexity (without absolute value), we apply Lemma L.4. $\qquad\square$

**Lemma L.6.** *(Hieu et al. (2024)) Let $\mathcal{F}$ be a class of representation functions $f : \mathcal{X} \to \mathbb{R}^d$ and $\ell : \mathbb{R}^k \to \mathbb{R}_+$ be a contrastive loss function that is $\ell^\infty$-Lipschitz with constant $\eta > 0$. Let $S = \left\{ (\mathbf{x}_j, \mathbf{x}_j^+, \mathbf{x}_{j1:k}^-) \right\}_{j=1}^m$ be a set of tuples. Then, we have:*

$$\ln \mathcal{N}\left( \mathcal{H}, \epsilon, \mathrm{L}_2(S) \right) \leqslant \ln \mathcal{N}\left( \mathcal{F}, \frac{\epsilon}{4\eta\Gamma}, \mathrm{L}_{\infty,2}(\widetilde{S}) \right), \tag{144}$$

$$\Gamma = \sup_{f \in \mathcal{F}} \sup_{\mathbf{x} \in \widetilde{S}} \|f(\mathbf{x})\|_2. \tag{145}$$

*where $\widetilde{S}$ is the set of all vectors, including anchors, positive and negative samples in $S$.*

## L.3. Symmetrization Inequality

**Lemma L.7** (Symmetrization). *Let $S = \{\mathbf{z}_1, \ldots, \mathbf{z}_n\}$ be a sample of independent random variables such that $\mathbf{z}_i \sim P_i, \forall i \in [n]$. Let $\mathcal{G}$ denote a function class. Then, for any real-valued non-decreasing function $\varphi$, we have:*

$$\mathbb{E}_S \varphi \left[ \sup_{g \in \mathcal{G}} \left| \frac{1}{n} \sum_{i=1}^n g(\mathbf{z}_i) - \Theta(g) \right| \right] \leqslant \mathbb{E}_{S, \mathbf{\Sigma}_n} \varphi \left[ 2\mathcal{R}_{\mathcal{G}}^{S, \mathbf{\Sigma}_n} \right], \tag{146}$$

*where $\mathbf{\Sigma}_n = (\sigma_j)_{j \in [n]}$ are independent Rademacher variables and $\Theta(g) = \frac{1}{n} \sum_{i=1}^m \mathbb{E}_{\mathbf{z} \sim P_i}[g(\mathbf{z})]$.*

**Proof.** Denote $\mathbb{E}_S$ as the expectation taken over $S$. We introduce another sample $S' = \{\mathbf{z}_1', \ldots, \mathbf{z}_n'\}$, called the "ghost" sample, such that $\mathbf{z}_i'$ is identically distributed as (and independent of) $\mathbf{z}_i$ for all $i \in [n]$. Then, we can write $\Theta(g)$ as

$\Theta(g) = \mathbb{E}_{S'}\left[\frac{1}{n}\sum_{i=1}^{n} g(\mathbf{z}_i')\right]$. Then, we have:

$$\mathbb{E}_S\varphi\left[\sup_{g\in\mathcal{G}}\left|\frac{1}{n}\sum_{i=1}^{n} g(\mathbf{z}_i) - \Theta(g)\right|\right] = \mathbb{E}_S\varphi\left[\sup_{g\in\mathcal{G}}\left|\frac{1}{n}\sum_{i=1}^{n} g(\mathbf{z}_i) - \mathbb{E}_{S'}\left[\frac{1}{n}\sum_{i=1}^{n} g(\mathbf{z}_i')\right]\right|\right]$$

$$= \mathbb{E}_S\varphi\left[\sup_{g\in\mathcal{G}}\left|\mathbb{E}_{S'}\left[\frac{1}{n}\sum_{i=1}^{n} g(\mathbf{z}_i) - \frac{1}{n}\sum_{i=1}^{n} g(\mathbf{z}_i')\right]\right|\right]$$

$$\leqslant \mathbb{E}_{S,S'}\varphi\left[\sup_{g\in\mathcal{G}}\left|\frac{1}{n}\sum_{i=1}^{n}\left(g(\mathbf{z}_i) - g(\mathbf{z}_i')\right)\right|\right] \quad \text{(Jensen's Ineq.)}$$

Now, since for all $i \in [n]$, $\mathbf{z}_i$ and $\mathbf{z}_i'$ are identically distributed. Then, let $\sigma_i$ be a Rademacher random variable, $g(\mathbf{z}_i) - g(\mathbf{z}_i')$, $g(\mathbf{z}_i') - g(\mathbf{z}_i)$ and $\sigma_i\left(g(\mathbf{z}_i) - g(\mathbf{z}_i')\right)$ are all identically distributed random variables. Hence, let $\boldsymbol{\Sigma}_n = (\sigma_j)_{j\in[n]}$ be independent Rademacher variables, we have:

$$\mathbb{E}_S\varphi\left[\sup_{g\in\mathcal{G}}\left|\frac{1}{n}\sum_{i=1}^{n} g(\mathbf{z}_i) - \Theta(g)\right|\right] \leqslant \mathbb{E}_{S,S'}\varphi\left[\sup_{g\in\mathcal{G}}\left|\frac{1}{n}\sum_{i=1}^{n}\left(g(\mathbf{z}_i) - g(\mathbf{z}_i')\right)\right|\right]$$

$$= \mathbb{E}_{S,S',\boldsymbol{\Sigma}_n}\varphi\left[\sup_{g\in\mathcal{G}}\left|\frac{1}{n}\sum_{i=1}^{n}\sigma_i\left(g(\mathbf{z}_i) - g(\mathbf{z}_i')\right)\right|\right]$$

$$= \mathbb{E}_{S,S',\boldsymbol{\Sigma}_n}\varphi\left[\sup_{g\in\mathcal{G}}\left|\frac{1}{n}\sum_{i=1}^{n}\sigma_i g(\mathbf{z}_i) + \frac{1}{n}\sum_{i=1}^{n}(-\sigma_i)g(\mathbf{z}_i')\right|\right].$$

Since the Rademacher variables $\sigma_i$ are symmetric, $\sigma_i$ and $-\sigma_i$ are identically distributed for all $i \in [n]$. Hence, we have:

$$\mathbb{E}_S\varphi\left[\sup_{g\in\mathcal{G}}\left|\frac{1}{n}\sum_{i=1}^{n} g(\mathbf{z}_i) - \Theta(g)\right|\right]$$

$$\leqslant \mathbb{E}_{S,S',\boldsymbol{\Sigma}_n}\varphi\left[\sup_{g\in\mathcal{G}}\left|\frac{1}{n}\sum_{i=1}^{n}\sigma_i g(\mathbf{z}_i) + \frac{1}{n}\sum_{i=1}^{n}\sigma_i g(\mathbf{z}_i')\right|\right] \quad (\sigma_i \text{ are symmetric})$$

$$\leqslant \mathbb{E}_{S,S',\boldsymbol{\Sigma}_n}\varphi\left[\sup_{g\in\mathcal{G}}\left|\frac{1}{n}\sum_{i=1}^{n}\sigma_i g(\mathbf{z}_i)\right| + \sup_{g\in\mathcal{G}}\left|\frac{1}{n}\sum_{i=1}^{n}\sigma_i g(\mathbf{z}_i')\right|\right]$$

$$\leqslant \frac{1}{2}\mathbb{E}_{S,\boldsymbol{\Sigma}_n}\varphi\left[\sup_{g\in\mathcal{G}}\left|\frac{2}{n}\sum_{i=1}^{n}\sigma_i g(\mathbf{z}_i)\right|\right] + \frac{1}{2}\mathbb{E}_{S',\boldsymbol{\Sigma}_n}\varphi\left[\sup_{g\in\mathcal{G}}\left|\frac{2}{n}\sum_{i=1}^{n}\sigma_i g(\mathbf{z}_i')\right|\right] \quad \text{(Jensen's Ineq.)}$$

$$= \mathbb{E}_{S,\boldsymbol{\Sigma}_n}\varphi\left[2\mathcal{R}_{\mathcal{G}}^{S,\boldsymbol{\Sigma}_n}\right],$$

as desired. $\qquad\square$

## L.4. Sub-Gaussianity of Rademacher Complexity

**Definition L.8** (Sub-Gaussian Random Variable)**.** Let $X$ be a random variable with mean $\mathbb{E}[X] = \mu$. $X$ is then called sub-Gaussian if there exists $\xi > 0$ such that:

$$M_X(t) \leqslant \exp\left(t\mu + \frac{t^2\xi^2}{2}\right), \quad \forall t > 0, \tag{147}$$

where $M_X$ denotes the moment generating function of $X$. We call $X$ a sub-Gaussian random variable with variance proxy $\xi^2$, denoted as $X \in \mathcal{SG}(\xi^2)$.

**Lemma L.9** ((Boucheron et al., 2013), Theorem 2.1)**.** *Let $X$ be a random variable with mean $\mathbb{E}[X] = \mu$. If there exists $\xi > 0$ such that the following holds:*

$$\mathbb{P}(|X - \mu| \geqslant t) \leqslant 2\exp\left(-\frac{t^2}{2\xi^2}\right), \tag{148}$$

*then the random variable $X$ is sub-Gaussian. Specifically, $X \in \mathcal{SG}(16\xi^2)$.*

**Proof**. Let $Z = X - \mu$ be the centered random variable derived by translating $X$ by its mean. Firstly, we prove that Eqn. (148) implies that $\mathbb{E}|Z|^{2q} \leqslant q!(4\xi^2)^q$ for all integers $q \geqslant 1$. Using the identity $\mathbb{E}|Z|^q = \int_0^\infty qt^{q-1}\mathbb{P}(|Z| \geqslant t)dt$, we have:

$$\mathbb{E}|Z|^{2q} = 2q \int_0^\infty t^{2q-1}\mathbb{P}(|Z| \geqslant t)dt$$
$$\leqslant 4q \int_0^\infty t^{2q-1} \exp\left(-\frac{t^2}{2\xi^2}\right) dt.$$

Letting $u = \frac{t^2}{2\xi^2}$, hence $t^2 = 2u\xi^2$ and $dt = \frac{\xi^2 du}{t}$, the above integral becomes:

$$\mathbb{E}|Z|^{2q} \leqslant 4q\xi^2 \int_0^\infty t^{2q-2}e^{-u}du$$
$$= 4q\xi^2 \int_0^\infty (2u\xi^2)^{q-1}e^{-u}du$$
$$= 2q \cdot (2\xi^2)^q \underbrace{\int_0^\infty u^{q-1}e^{-u}du}_{\Gamma(q)}$$
$$= 2q!(2\xi^2)^q \leqslant q!(4\xi^2)^q.$$

Let $\tilde{Z}$ be the i.i.d. copy of $Z$. Hence, $Z - \tilde{Z}$ is symmetric about 0, which means that $\mathbb{E}[(Z - \tilde{Z})^p] = 0$ for odd-order $p$-moments. Therefore, For all $\lambda > 0$, we have:

$$M_Z(\lambda)M_{-\tilde{Z}}(\lambda) = M_{Z-\tilde{Z}}(\lambda) \quad \text{(Due to independence)}$$
$$= \mathbb{E} \exp\left(\lambda(Z - \tilde{Z})\right)$$
$$= 1 + \sum_{q=1}^\infty \frac{\lambda^{2q}\mathbb{E}\left[(Z - \tilde{Z})^{2q}\right]}{(2q)!}.$$

By the convexity of $f(z) = z^{2q}$, for all $t \in (0, 1)$, we have:

$$\left[tZ + (1 - t)(-\tilde{Z})\right]^{2q} \leqslant tZ^{2q} + (1 - t)\tilde{Z}^{2q}.$$

Setting $t = \frac{1}{2}$, we have:

$$\left[\frac{Z - \tilde{Z}}{2}\right]^{2q} \leqslant \frac{Z^{2q} + \tilde{Z}^{2q}}{2} \implies (Z - \tilde{Z})^{2q} \leqslant 2^{2q-1}(Z^{2q} + \tilde{Z}^{2q}).$$

As a result, we have $\mathbb{E}[(Z - \tilde{Z})^{2q}] \leqslant 2^{2q-1}(\mathbb{E}[Z^{2q}] + \mathbb{E}[\tilde{Z}^{2q}]) = 2^{2q}\mathbb{E}[Z^{2q}]$. Plugging this back to the formula of $M_{Z-\tilde{Z}}(\lambda)$, we have:

$$\mathbb{E}[e^{\lambda Z}]\mathbb{E}[e^{-\lambda\tilde{Z}}] \leqslant 1 + \sum_{q=1}^\infty \frac{\lambda^{2q}2^{2q}\mathbb{E}[Z^{2q}]}{(2q)!}$$
$$\leqslant 1 + \sum_{q=1}^\infty \frac{\lambda^{2q}2^{2q}(4\xi^2)^q q!}{(2q)!}.$$

Since $\mathbb{E}[e^{-\lambda\tilde{Z}}] \geqslant 1$ for all $\lambda > 0$ and

$$\frac{(2q)!}{q!} = \prod_{j=1}^q (q + j) \geqslant \prod_{j=1}^q (2j) = 2^q q!,$$

We have:

$$\mathbb{E}[e^{\lambda Z}] \leqslant 1 + \sum_{q=1}^{\infty} \frac{(2\lambda^2 \cdot 4\xi^2)^q}{q!} = 1 + \sum_{q=1}^{\infty} \frac{(8\lambda^2 \xi^2)^q}{q!} = e^{8\lambda^2 \xi^2}.$$

As a result, we have:

$$M_X(t) = e^{\lambda\mu}\mathbb{E}[e^{\lambda Z}] \leqslant e^{\lambda\mu + 8\lambda^2\xi^2}.$$

Hence, by definition, we have $X \in \mathcal{SG}(16\xi^2)$ as desired. $\qquad\square$

**Lemma L.10.** *Let $\mathcal{F}$ be a class of bounded functions $f : \mathcal{X} \to [0, \mathcal{B}]$ and let $S = \{\mathbf{x}_1, \ldots, \mathbf{x}_n\}$ be sampled i.i.d. from a given distribution $\mathcal{P}$. Let $\mathbf{\Sigma}_n = \{\sigma_j\}_{j=1}^n$ be a sequence of independent Rademacher variables and define the following random variable:*

$$\mathcal{R}_{\mathcal{F}}^{S,\mathbf{\Sigma}_n} = \sup_{f \in \mathcal{F}} \left| \frac{1}{n} \sum_{j=1}^n \sigma_j f(\mathbf{x}_j) \right|, \tag{149}$$

*which is a function of both $S$ and $\mathbf{\Sigma}_n$. Then, we have:*

$$\mathbb{E}_{S,\mathbf{\Sigma}_n} \exp\left(t\mathcal{R}_{\mathcal{F}}^{S,\mathbf{\Sigma}_n}\right) \leqslant \exp\left(t\mathfrak{R}_n(\mathcal{F}) + \frac{16t^2\mathcal{B}^2}{2n}\right), \quad \forall t > 0,$$

*where $\mathfrak{R}_n(\mathcal{F})$ is the expected Rademacher complexity, defined as $\mathfrak{R}_n(\mathcal{F}) = \mathbb{E}_{S,\mathbf{\Sigma}_n}\left[\mathcal{R}_{\mathcal{F}}^{S,\mathbf{\Sigma}_n}\right]$.*

**Proof**. Let $S_i$ be the copy of $S$ with the $i^{th}$ element replaced with $\mathbf{x}_i'$, an i.i.d. copy of $\mathbf{x}_i$. Similarly, let $\mathbf{\Sigma}_n^{(l)}$ be the copy of $\mathbf{\Sigma}_n$ where the $l^{th}$ element is replaced with $\sigma_l'$, an i.i.d. copy of $\sigma_l$. Then, we have the following bounded-difference properties:

$$\left| \mathcal{R}_{\mathcal{F}}^{S,\mathbf{\Sigma}_n} - \mathcal{R}_{\mathcal{F}}^{S_i,\mathbf{\Sigma}_n} \right| \leqslant \sup_{f \in \mathcal{F}} \left| \frac{1}{n}\sigma_i(f(\mathbf{x}_i) - f(\mathbf{x}_i')) \right| \leqslant \frac{2\mathcal{B}}{n},$$

$$\left| \mathcal{R}_{\mathcal{F}}^{S,\mathbf{\Sigma}_n} - \mathcal{R}_{\mathcal{F}}^{S,\mathbf{\Sigma}_n^{(l)}} \right| \leqslant \sup_{f \in \mathcal{F}} \left| \frac{1}{n}f(\mathbf{x}_l)(\sigma_l - \sigma_l') \right| \leqslant \frac{2\mathcal{B}}{n}.$$

Hence, by McDiarmid's inequality, we have:

$$\mathbb{P}\left( \left| \mathcal{R}_{\mathcal{F}}^{S,\mathbf{\Sigma}_n} - \mathfrak{R}_n(\mathcal{F}) \right| \geqslant t \right) \leqslant 2\exp\left(-\frac{nt^2}{4\mathcal{B}^2}\right), \quad \forall t > 0.$$

Let $\xi^2 = \frac{2\mathcal{B}^2}{n}$. Then, by lemma L.9, the random variable $\mathcal{R}_{\mathcal{F}}^{S,\mathbf{\Sigma}_n}$ is sub-Gaussian with variance proxy of $16\xi^2 = \frac{32\mathcal{B}^2}{n}$. Hence, we have:

$$\mathbb{E}_{S,\mathbf{\Sigma}_n} \exp\left(t\mathcal{R}_{\mathcal{F}}^{S,\mathbf{\Sigma}_n}\right) \leqslant \exp\left(t\mathfrak{R}_n(\mathcal{F}) + \frac{16t^2\mathcal{B}^2}{n}\right), \quad \forall t > 0,$$

as desired. $\qquad\square$

