# OpenReview forum: "A Refined Generalization Analysis for Extreme Multi-class Supervised Contrastive Representation Learning"
_ICML.cc/2026/Conference — ICML 2026 regular_

### Official Review · Reviewer_9NCi · 2026-03-09

**Soundness:** 3
**Presentation:** 2
**Significance:** 2
**Originality:** 3
**Overall Recommendation:** 4
**Confidence:** 4

**Summary:**

This paper proposes an improved generalization error bound for contrastive representation learning (CRL) without class collision. The contrastive tuples are assumed to be constructed from a finite pool of labeled data instead of being i.i.d. generated. Compared with the generalization bound derived in a previous work under the same settings, this paper 1) improves the bound for the existing U-Statistics and removes its dependency on $\rho_{\min}$, the probability of the rarest class; and 2) proposes a new formulation of the U-Statistics and derives sharper bounds when class distributions are long-tailed.

**Compliance With Llm Reviewing Policy:**

Affirmed.

**Final Justification:**

The rebuttal solves my concerns. I keep my attitude towards the class-collision-free setting, so I cannot give a higher score. Despite this, I keep my original recommendation score and lean towards acceptance.

**Key Questions For Authors:**

1. Is the proposed U-statistic a new method for supervised CRL, or is it a theoretical formulation to derive generalization bounds for the existing CRL methods?
2. Is it possible to derive a bound that allows class collision (a bound for $U_{\Omega}$)?
3. How is the proposed U-statistic compared empirically with the standard SupCon?
4. How does the proposed U-statistic perform under the extreme multi-class case where the class distributions are balanced but the number of classes is large?
5. The refined bound is claimed to remove the dependency on $\rho_{\min}^{-1}$. Can this point be empirically validated?

**Limitations:**

Yes.

**Strengths And Weaknesses:**

**Strength**
1. This paper proposed a refined generalization theory for contrastive representation learning, contributing to the theoretical study of CRL.
2. The theoretical results are clear, with notations and bounds summarized in tables and theorems supported with proofs.
3. Experiments are conducted to validate the theoretical claims.

**Weakness**
1. This paper investigates the class-collision-free CRL, which is an unusual setting for CRL methods in practical applications. In practice, both the commonly used supervised and self-supervised contrastive learning methods (SimCLR, SupCon, etc.) allow class collision. Investigating only the class-collision-free setting somewhat limits the theoretical values.
2. I am unsure of the main purpose of this paper. Is the proposed U-statistic a new method for supervised CRL, or is it a theoretical formulation to derive generalization bounds for the existing CRL methods? If it is a new method, then it should be compared empirically with the standard SupCon. If it is purely a mathematical formulation, then it should allow class collision.
3. The proposed U-Statistic is claimed to be advantageous in the extreme multi-class scenario. However, it is only validated under the long-tailed distributions by comparing the rare-class accuracies. How about the extreme multi-class case where the class distributions are balanced but the number of classes is large?
4. The organization of Section 3 is a bit confusing. I think the contributions (Section 3.2) should moved to Section 1.
5. The refined bound is claimed to remove the dependency on $\rho_{\min}^{-1}$. I think this should be empirically validated.
6. Some related works on CRL theory are missing, e.g. https://arxiv.org/abs/2201.04309, https://arxiv.org/pdf/2210.01883, https://arxiv.org/pdf/2303.15103, https://arxiv.org/abs/2511.03114, https://arxiv.org/abs/2501.01317.
7. (Minor) It should be noted in the caption of Table 2 that the results are for the five rarest classes to avoid possible misunderstandings.

---

> ### Author Rebuttal · Authors · 2026-03-30
>
> Many thanks for your detailed comments and your appreciation of the **rigor and quality of our theoretical results**. We address your remaining concerns below.
>
> **Re (W1, Q2)**: We respectfully disagree that the class-collision-free setting is unnatural. A substantial body of work shows that false negatives (i.e., class-collisions) significantly degrade performance. For instance, [6] is dedicated primarily to mitigating class-collision and simulating true negative sampling, followed by works that extensively validate its adverse effects [11, 13, 14, 15]. From a theoretical view, [8] (Sec. 6.2) states that “frequent class collisions can prevent the unsupervised algorithm from learning representations optimal for the supervised problem,” further supported by Thm. 4.1 showing degraded downstream performance.
>
> The study of the collision-allowed case is often motivated by its connection to unsupervised *augmentation*-based settings, an orthogonal direction requiring more delicate assumptions (cf. [16]).
>
> To answer Q2, we also provide results for the collision-allowed regime (Thm. E.7), yielding sample complexity $\widetilde O(\mathfrak{C}_N^2(\mathcal{H})k+Rk)$, though this serves as a step toward the more meaningful collision-free regime (Thm. 4.5). See also our response to Reviewer y21x.
>
> **Re (Main contribution - W2, Q1)**: We position our paper as a theoretical work rather than a SOTA methodology paper. As stated in Remark 4.3, we do not make a strong claim of novelty in empirical performance. While our estimator in Eq. (13) is technically novel, similar debiasing strategies have appeared in prior work (e.g., [6], Eq. (8)). Our key contribution is to show that explicit debiasing affects the concentration of contrastive risk in the supervised finite-sample regime, unlike in the i.i.d. tuple regime [8,9]. This further highlights limitations of simulating i.i.d. tuples and clarifies the role of class-collision beyond risk miscalibration.
>
> **Re (Further experiments - W3, Q3, Q4)**: To address your concern, we have compared the empirical performance of our estimator with SupCon [10] on the originally selected datasets plus CIFAR100. As theory predicts, the sub-sampled estimator of $U_N$ outperforms [7] and frequently outperforms SupCon (with exceptions only on small-scale datasets like FashionMNIST and CIFAR10), especially on CIFAR100.
>
> ---
>
> |          | Ours       | [7]        | **SupCon** |
> |----------|------------|------------|------------|
> | **Prec** | **0.3239** | 0.3196     | 0.2772     |
> | **Rec**  | **0.3422** | 0.2710     | 0.3095     |
> | **F1**   | **0.3306** | 0.2929     | 0.2919     |
>
> ---
>
> |             | Ours       | [7]    |
> |-------------|------------|--------|
> | **MNIST**   | **0.9908** | 0.9895 |
> | **F-MNIST** | **0.8946** | 0.8944 |
> | **CIFAR10** | **0.7256** | 0.7184 |
>
> ---
>
> The first table reports the macro-average metrics on rare classes for **CIFAR100** and the second table reports the overall accuracy of our estimator compared to [7] when trained on the originally selected full datasets. Both of the above experiments are run with $k=5$ for ours and [7]. Due to the rebuttal limit, we can only provide a concise summary are willing to provide the full set of results upon request.
>
> **Re (W5, Q5)**: We believe the $\rho_{\min}$ dependency in [7] is a proof artifact rather than a faithful reflection of how CRL generalizes. The sample complexity actually scales with $R$ and this was validated by [7] in their experiment (Fig. 3). We have further validated this by artificially varying $\rho_{\min}$ in a new experiment and show that the contrastive generalization gap is broadly agnostic to the change of $\rho_{\min}$. Specifically, the **statistical significance** of Spearman correlation between generalization gaps and $\rho_{\min}$ values is roughly **0.21** (i.e., statistically insignificant for $\alpha=0.05$).
>
> On top of the above answers, we will do our best to further improve the presentation by following all your suggestions and including a more thorough discussion on all the suggested works [1, 2, 3, 4]. We thank you again for the comments and sincerely hope that this rebuttal has helped allay any doubts you have, and improved your opinion of the significance of our contributions.
>
> **References**:
>
> [8] Arora et al. A Theoretical Analysis of Contrastive Unsupervised Representation Learning.
>
> [9] Lei et al. Generalization Analysis for CRL.
>
> [10] Khosla et al. Supervised Contrastive Learning.
>
> [11] Robinson et al. CRL with Hard Negative Samples.
>
> [12] Jang and Wang. Difficulty-based Sampling for Debiased CRL.
>
> [13] Huynh et al. Boosting Contrastive Self-Supervised Learning with False Negative Cancellation.
>
> [14] Chen et al. Incremental False Negative Detection for CRL.
>
> [15] Kim et al. False-Negative Aware Learning of Contrastive Negatives in Vision-Language Alignment.
>
> [16] Jeff Z. HaoChen. Provable Guarantees for Self-Supervised Deep Learning with Spectral Contrastive Loss

---

> > ### Author Rebuttal · Reviewer_9NCi · 2026-04-03
> >
> > Thanks for the rebuttal. The reply solves my concerns. Yet I keep my attitude towards the class-collision-free setting. I will keep my score.

---

> > > ### Author Response · Authors · 2026-04-05
> > >
> > > Dear Reviewer 9NCi,
> > >
> > > We are glad that our responses helped clarify your doubts regarding our contributions. If there are any remaining concern, please do not hesitate to reach out to us, we are more than happy to discuss further. Once again, we really appreciate your suggestions to improve the quality of our work.

---

### Official Review · Reviewer_y21x · 2026-03-12

**Soundness:** 3
**Presentation:** 3
**Significance:** 3
**Originality:** 3
**Overall Recommendation:** 4
**Confidence:** 3

**Summary:**

The paper studies contrastive representation learning. More formally the paper considers the problem where the learner is given labeled data set $ S=\\{  X_{j}\\}\_{j=1}\^{N}\sim \bar{\mathcal{D}}\^{N}  $, where $ \bar{\mathcal{D}} $ is a mixture distribution over $ R $ class distributions $ \mathcal{D}\_{r} $, where the $ r $ class is choosen with probability $ \rho\_{r} $, and the goal is to given these sample create an estimator $ \hat{e} $, which uniformly for some class of representations function $ \mathcal{F} $, ensures that $ \hat{e}(f) $ is close to $ E\_{r\sim \rho}[ E\_{X,X^{+}\sim \mathcal{D}\_{r},\\{X\_{i}\\}\_{i=1}^{k}\sim \bar{\mathcal{D}}\_{r}}[(\phi(\\{f(X)^{T}(f(X^{+})-f(X\_{i}^{-}))\\}\_{i=1}^{k})) ] ] $, where $ \phi $ is some contrastive loss that takes as input a tuple of $ k+2 $ elements of the domain, with $ X $ being the anchor point from the class $ r $, $ X^{+} $ being the positive point(same label as $ X $ ), and $ k $ negative points $ \\{X\_{i}\\}\_{i=1}^{k}\sim \bar{\mathcal{D}}_{r} $ sampled not from the class $ r $.

The paper studies two estimators for this problem. The first estimator was proposed by Hieu \& Ledent 2025. For this estimator, the paper removes a dependence on $ 1/\rho_{min} $, where $ \rho_{min} $ is the probability of the least likely class for $1,\ldots,R$, at the cost of introducing a linear dependency on $ \max\\{R,k\\} $. To alleviate the bounds dependency on the number of classes $ R $, the paper introduces a new estimator, which they show obtains a sample complexity where the complexity terms are only multiplied by $ k $, while still being linear in $ R $ otherwise. The new estimators sample complexity scales inversely polynomial in $ 1-\tau $ where $ \tau $ is the probability of having a class collision between the positive example $ X^{+} $ and one of the elements $ \\{  X_{i}\\}_{i=1}^{k}  $. Motivated by these bounds, the paper conducts an experimental study of these two estimators (subsampled versions of them), and gets indications that the new estimator better handles the classes with small class probabilities.

**Compliance With Llm Reviewing Policy:**

Affirmed.

**Final Justification:**

The paper makes further progress on Supervised Contrastive Representation Learning from a theoretical perspective, removing independence of the inverse smallest class probability, and is well written, the rebuttal answers also addressed my questions - all this is reflected by a score indicating an accept of the paper.

**Key Questions For Authors:**

**Questions**

Q1: The sample complexity of Lei et al. 2023 seems to be better(Table 5) - in which natural settings can you reduce from the i.i.d. supervised setting to the unsupervised tuple i.i.d. setting? Why are the settings where this is not possible important?

Q2: Page 3 column 1 line 158-164 in connection to Q1, if Lei et al. 2023 samples as in Arora 2019 - why can't any i.i.d. supervised setting simulate the unsupervised tuple i.i.d. setting?

Q3: How do the estimators compare on the full datasets of MNIST, FashionMNIST, CIFAR-10?

**Suggestions and possible typos (no answers needed)**

S1: The abstract seems a bit long according to the ICML guidelines.

S2: page 1, column 2, line 048: "parasitic" is maybe a bit strong.

S3: page 2, column 1, line 061: should it be $ (1-\tau)^{-2} $

S4: page 2, column 1, line 114: By  "labelled" I understand that the class is also known for each example? So the sample is $ \\{  (X_{j},r_{j})\\}\_{j=1}^{N} \in (\mathcal{X}\times [R])^{N} $? I would prefer this to be stated more clearly and then have a note about using shorter notation $ \\{X_{j}\\}_{j=1}^{N} $, when clear.

S5: page 14 line 754-756: "We also note that in the i.i.d. tuples regime, the collision-free contrastive risk can only be estimated if the learner can
generate i.i.d. tuples with positive and negative samples following the correct class distribution, which essentially requires
access to supervised label information. " Do you really need the label? Isn't it sufficient to know that $ X,X^{+}\sim \mathcal{D}\_{r} $ and $ \\{X_{i}\\}\_{i=1}\^{k} \sim \bar{\mathcal{D}}\_{r} $?

S6: page 3 column 2 line 152 should it be $ (1-\tau)^{-2} $?

S7: page 4 table 1: should there be a factor $ k^{-1} $ in the $ \bar{U}_{N} $ estimator cell? (I guess yes, but just wanted to be sure.) I think $ \tau $ is defined differently in the text than in the table?

S8: In experiment section, it says $ U_{N} $ and not $ \bar{U}_{N} $, is this on purpose?

S9: page 6, column 2, line 293, Why is it $ \rho_{r}^{2} $ when $ k=1 $?

S10: page 6, column 2, line 307, should it be $ \gamma_{k}=\Omega(1) $

**Limitations:**

Yes

**Strengths And Weaknesses:**

Soundness:
The main of the paper contains no proof, so I will not comment on the soundness of the proofs.

Presentation:
The paper is well written, and explain their results clearly. I would have liked the paper to have had a discussion on the new bounds introducing linear dependencies on $ R $ and $ k $.

Significance/Originality:
The paper gives a new tradeoff in the sample complexity, removing dependence on the smallest class probability, and introduces dependencies on $ R $ and $ k $   from the estimator of Hieu \& Ledent 2025. The paper's new estimator gives a second tradeoff again removing the dependency of the smallest class probability and removing the dependency of $ R $ onto the complexity term, while introducing a polynomial dependence on $1-\tau$, the class collision probability. Thus the paper gives insights into tighter bounds when the smallest class probability is less than $\rho_{min} \leq 1/R $ for the estimator of Hieu \& Ledent 2025, or for the new estimator when the $ \rho_{min}\geq \max\{\mathcal{C}(H)^{2}/(1-\tau)^{2},R/(1-\tau)^{4}\}$. See my question about Lei et al. 2023?

---

> ### Author Rebuttal · Authors · 2026-03-30
>
> We thank the Reviewer for acknowledging that our work is "**well-written**" and "**clear**". We are grateful for your thorough reading and insightful comments. We address your concerns below.
>
> **Q1**: Why [9] seems better? When is the reduction possible? Why is it important?
>
> **Ans**: The seemingly better sample complexity stems from the fact that [9] *assume access to i.i.d. tuples with the correct structure automatically*. In our supervised setting or any practical scenario, the correct tuple structure does not come for free (cf. Appendix C.2, point (i)) and selecting without replacement from the labeled dataset to form valid tuples incurs an additive cost of $O(Rk)$ on the sample complexity (cf. Prop D.6). Hence, a **direct reduction** to the result in the form of [9] is **not possible** and it is due to a fundamental difference between data-generating processes rather than a proof artifact. For further details, please find the continuation to this point in our answer to Q2.
>
> On a side note, Table 5 reports the sample complexity of [9] for estimating the collision-allowed risk ($L_\Omega$) whilst the result of our work is for the collision-free risk in Eq. (2). We agree that the difference is subtle and promise to clarify further in the revised manuscript. On that note, we also provide a sample complexity for estimating the collision-allowed risk $L_\Omega$ that scales as $\widetilde O(\mathfrak{C}^2_N(\mathcal{H})k + Rk)$ (cf. Thm. E.7). However, this result serves primarily as an intermediate step to prove Thm 4.5.
>
> To answer the last part of your question, the analysis of the supervised i.i.d. setting matters because it is precisely the scenario encountered in practice: In most applications, one is given a labeled dataset and must construct tuples from it, rather than being provided with independent, non-overlapping, pre-formed tuples. This is consistent with standard benchmarks such as MNIST, FashionMNIST, CIFAR10 and CIFAR100 where tuple construction is performed explicitly during training.
>
> **Q2**: Why can i.i.d. supervised settings not simulate [9]?
>
> **Ans**: Building on the discussion above, our supervised i.i.d. setting yields a sample complexity for estimating $L_\Omega$ that is comparable to [9] with an *additive* $O(Rk)$ cost. This added cost is irreducible, and stems from a fundamental distinction in how data is generated.
>
> In our supervised setting, samples are drawn independently from the mixture distribution $\mathcal{\bar D}$. Therefore, to construct a valid $(k+2)$-tuple satisfying the constraint that at least two samples belong to the same class, one must perform at least $R+1$ draws in the worst case (pigeonhole principle). Consequently, even forming a single valid tuple incurs a cost that scales with $R$, implying that any estimator of $L_\Omega$ in this setting must have sample complexity at least in the order of $O(R)$.
>
> In contrast, [9] *assumes* direct access to tuples drawn from the product distribution $\bar D_r^{\otimes 2} \otimes \bar D^{\otimes k}$. As a result, tuples are *i.i.d. and correctly structured by design*. This effectively bypasses the combinatorial difficulty of tuple construction. Hence, their setting avoids the intrinsic $O(R)$ overhead in the supervised i.i.d. regime.
>
> This distinction is vital: the reduction from supervised i.i.d. data to tuple-based sampling *can never be done trivially*, and the additional term captures the cost of enforcing the structural constraints required to form valid tuples.
>
> **Q3**: How does $U_N$ perform on full versions of selected datasets?
>
> **Ans**: We assure the Reviewer that we have also reproduced our experiments in Sec. 5 on the balanced versions of selected datasets. As predicted, $U_N$ outperforms [7] but the effect is less pronounced compared to the experiments on imbalanced counterparts. Please kindly see our answer to Reviewer 9NCi.
>
> ---
>
> **S3, S6**: Yes, this is a typo. The result scales with $(1-\tau)^{-2}$, as stated in Table 1/Thm. 4.5. Thank you!
>
> **S5**: The ability to tell that $(X,X^+,X_{1:k}^-)$ is a valid tuple indirectly implies some label knowledge. We humbly ask to clarify further in the discussion phase upon request.
>
> **S7**: Yes, the Reviewer is correct about $k^{-1}$. We use only one definition of $\tau$ and the table actually contains a typo: it should be $1-\tau$.
>
> **S8**: Cf. Q2 answer to Rev. sa7E.
>
> **S10**: Yes, indeed we meant $\gamma_k$ not vanishingly small, which corresponds to the $\Omega$-notation. As such, $1-\tau\in\Omega(1)$. We will make sure to correct the notation throughout the manuscript.
>
> Due to the strict rebuttal limit, we apologize that some details are omitted and we will answer the remaining concerns in the discussion phase. As such, we are committed to incorporating the Reviewer's suggestions accordingly.
>
> We sincerely thank you again for your careful reading and humbly ask whether there are further questions we could answer that may convince you to consider raising your score.

---

> > ### Author Rebuttal · Reviewer_y21x · 2026-04-01
> >
> > Thanks for your reply. For S5, i just mean the labels are not strictly needed to create tuples, as one just needs the guarantee that the positive pair is from $\mathcal{D}_r$ and the negative examples are from $\bar{\mathcal{D}}_r$, which is less information than fully supervised examples where each class would be known exactly for each example. I will maintain my score.

---

> > > ### Author Response · Authors · 2026-04-05
> > >
> > > Dear Reviewer y21x,
> > >
> > > We are glad that all of your remaining concerns have been **fully resolved**. As such, we promise to incorporate the suggested changes to the revised camera-ready version with emphasis on the following:
> > >
> > > - Fix the typo on $\Omega$-notation throughout the manuscript ($\gamma_k, 1-\tau\in\Omega(1)$).
> > >
> > > - Provide all further experiments in the Appendix.
> > >
> > > Regarding S5, yes, we agree with the Reviewer that under the idealized i.i.d. tuple regime in [8, 9], full labels are not required if one can directly sample tuples with the correct structure (made up of independent samples from the correct positive and negative distributions). Our point was that such a theoretical sampler is not available in practice and cannot be constructed from unlabeled samples: enforcing this structure even from a fully labeled sample set incurs additional cost (cf. Q2). We will clarify this distinction in the revision. Thank you.
> > >
> > > If there is any question or further clarification required, we are happy to engage in further discussion. Once again, we really appreciate your efforts in helping us improve the quality of our work.

---

### Official Review · Reviewer_A7Uj · 2026-03-13

**Soundness:** 3
**Presentation:** 3
**Significance:** 3
**Originality:** 3
**Overall Recommendation:** 4
**Confidence:** 3

**Summary:**

This paper focuses on the theoretical generalization analysis of supervised contrastive representation learning, with a particular emphasis on deriving theoretical bounds in practical, complex scenarios such as extreme multi-class or long-tailed data distributions. The paper points out that existing theories suffer from analytical limitations when dealing with non-independent and identically distributed (non-i.i.d.) data. This leads to derived sample complexity bounds that depend heavily on the rarest class probability and explode exponentially, thus lacking practical guiding significance. To address this issue, the paper proposes a heterogeneous accuracy analysis method for the U-statistic estimator, along with a novel debiased U-statistic estimator. The authors validate the proposed methods on synthetic long-tailed datasets as well as artificially imbalanced MNIST, FashionMNIST, and CIFAR-10 datasets.

**Compliance With Llm Reviewing Policy:**

Affirmed.

**Final Justification:**

After reading the authors' rebuttal and other reviews' comments,  i think the authors have clarified most of my concerns. I keep my weak accept score.

**Key Questions For Authors:**

1.Your title emphasizes "extreme multi-class" supervised contrastive learning, yet all experiments use datasets with only 10 classes (MNIST, FashionMNIST, CIFAR-10). How does this demonstrate the claimed advantages in truly extreme multi-class settings (e.g., ImageNet with 1000 classes, iNaturalist with thousands of species, or any other datasets)? Would the theoretical benefits of O(k) vs. O(R) scaling actually manifest in practice on such large-scale problems?

2.You compare against the prior estimator but not against any practical methods designed for long-tailed contrastive learning, such as class-aware sampling, re-weighting, or prototypical contrastive methods. How does your estimator compare empirically to these existing approaches in terms of tail-class performance? Without such comparisons, how can a practitioner judge whether the theoretical advantages translate to real-world utility?

**Limitations:**

Yes

**Strengths And Weaknesses:**

-Strengths:

1.The problem addressed in this paper is of great theoretical significance, and the proposed methods can effectively alleviate or even resolve the issue of overly pessimistic existing theoretical bounds.

2.This paper fills an important gap in contrastive learning theory. Although it builds upon Hieu & Ledent (2025), it offers two highly significant innovations: (1) removing the dependency on ρmin through heterogeneous accuracy analysis; and (2) achieving anO(k) sample complexity scaling with the new debiased estimator under mild conditions.

-Weaknesses:

1.The experiments rely solely on small-scale datasets (MNIST, FashionMNIST, CIFAR-10). It is recommended to conduct experiments on larger-scale benchmark datasets (e.g., ImageNet, CUB, etc.) to further demonstrate the effectiveness of the proposed method in real-world, complex scenarios. Furthermore, the baselines used for comparison are limited (lacking comparisons with other long-tailed contrastive learning baselines, such as class-aware sampling or re-weighting). A more in-depth comparative analysis of the experimental results is suggested.

2.The paper's title explicitly claims "Extreme Multi-class" supervised contrastive representation learning; however, the number of classes in all evaluated datasets is limited to 10, which fails to support the application scenario advertised in the title.

3.In the methodology section, it would be highly beneficial to include an illustrative diagram visually demonstrating the differences between the proposed estimator and prior work (or traditional loss calculation methods). It is recommended to add a simple figure in the main text to more clearly explain the core contributions.

---

> ### Author Rebuttal · Authors · 2026-03-30
>
> We are grateful that you acknowledge our work's significance and that it **fills an important gap in contrastive learning theory**. Below, we address your concerns regarding empirical validations.
>
> **Q1**: Selected datasets are small scale. How does this verify the claimed advantages? Will theoretical benefits manifest in larger problems?
>
> **Ans**: Firstly, we assure the Reviewer that we have conducted further experiments on CIFAR100 to better reflect the extreme multi-class regime in our theory (cf. Rev. 9NCi). In these experiments, we observe consistent improvements in tail-class metrics: models trained with the sub-sampled $U_N$ as the empirical risk outperform both the estimator in [7] and SupCon [10]. This provides empirical evidence that the advantages predicted by our theory continue to hold in larger multi-class settings.
>
> To answer the first question, we argue the existing experiments do contribute to the contextualization of our theoretical claims.  Our work is the first to formally prove that *debiasing can affect* not just downstream classification performance but also *the concentration of the contrastive risk* in some settings (when the number of classes $R$ is larger than the number of negatives $k$). Even the existing experiments ($R=10$, $k=3,5,7$) demonstrate that our specific estimator maintains the experimental edge more broadly associated with debiasing strategies in general.
>
> Furthermore, we note that the benefits of Thm. 4.5. are governed by the sample complexity of order $O((1-\tau)^{-2}k)$, which reduces to $O(k)$ when $1-\tau\in\Omega(1)$. Through examples 4.1 - 4.3, we showed that this condition holds in at least *two regimes*:
>
> - (i) Extreme multi-class ($k\ll R$).
> - (ii) When $\gamma_k=\mathbb{P}_{r\sim\rho}(\rho_r\le 1/k)\in\Omega(1)$.
>
> Our existing experiments (Sec. 5.3) are *designed to validate scenario (ii)*.
>
> **Q2a**: How does $U_N$ compare to class-aware sampling, reweighting?
>
> **Ans**: Interestingly, the U-Statistics formulation of our work, in fact, already *possesses class-ware re-weighting property*. Let us recall Eq. (13):
> $$U_N:=\frac{1}{1-\widehat\tau}U_\Omega - \frac{\widehat\tau}{1-\widehat\tau}U_\Lambda.$$
>
> Here, $U_\Lambda$ averages over the tuples with at least one class-collision. For a major class $r$ (large $N_r$), tuples associated with $r$ are more likely to appear in $\Lambda$, and are therefore *down-weighted more heavily via the subtraction term*. Conversely, tuples from minor classes are less affected, resulting in an *implicit re-balancing toward tail classes*. This mechanism directly translates into improved tail-class performance, which we observe in our experiments (Sec. 5.2, 5.3). For more details, we include a thorough explanation for this re-weighting behavior in Remark I.1 (pages 66 - 67).
>
> **Our positioning**: We also wish to clarify that we position our paper primarily as a *theoretical work* that aims to explain generalization behavior of supervised CRL *rather than a SOTA technique*. As also noted by the Reviewer, our main novelty lies in the theoretical analysis. This is explicitly stated in our Remark 4.3, where we do not make a strong claim of novelty in empirical performance. Accordingly, the role of our experiments is to provide validation for theoretical predictions, rather than to serve as an extensive benchmark against specialized long-tailed methods.
>
> **Q2b**: Without comparison, how do we judge the practical utility?
>
> **Ans**: From a practical standpoint, our estimator can be viewed as a principled debiasing mechanism that reduces the impact of class-collision, which disproportionately affects minor classes. The consistent improvements we observe in tail class metrics across datasets (including CIFAR100) indicate that this effect persists in practice.
>
> More importantly, empirical validation for practical advantages of debiased estimators *has already been performed* by multiple works in different contexts. For instance, [6, 11] introduce a debiased formulation of empirical risk similar to ours to mitigate the effect of false negatives. In their work, an extensive comparative study has been carried out against SimCLR on various datasets. Further work such as [12] have also compared the training objective in [6] against various methods on both long-tailed and large class datasets. In contrast, our paper provides the formal framework via U-Statistic to explain generalization behaviors that complements the aforementioned works. Thus, our contribution is not a new SOTA method, but offers theoretical justification for why such debiasing improves tail performance.
>
> **Comparative study**: To further address the Reviewer’s concern, we additionally include comparisons with SupCon [10] and observe consistent improvements in tail-class performance (cf. Rev. 9NCi). While our main goal is to analyze the estimation of the population risk in Eq. (2), these results confirm that the theoretical advantages indeed translate to practical settings.

---

> > ### Author Rebuttal · Reviewer_A7Uj · 2026-04-03
> >
> > Thank you for your response; I’ll keep my score.

---

> > > ### Author Response · Authors · 2026-04-05
> > >
> > > Dear Reviewer A7Uj,
> > >
> > > We are happy that we have resolved all your concerns fully. Should there be any other question, please let us know and we are happy to engage in further discussion.

---

### Official Review · Reviewer_sa7E · 2026-03-14

**Soundness:** 3
**Presentation:** 3
**Significance:** 3
**Originality:** 3
**Overall Recommendation:** 4
**Confidence:** 2

**Summary:**

The paper studies supervised contrastive representation learning when contrastive tuples are formed from a finite labeled sample, so tuples are dependent rather than i.i.d. It revisits the class-wise U-statistic estimator from Hieu & Ledent (2025), derives a refined excess-risk analysis that removes explicit dependence on the rarest-class probability, and introduces an alternative estimator based on decomposing the collision-free risk into zero-or-more-collision and at-least-one-collision terms. The new estimator yields sharper bounds whose dominant term scales with k (number of negative samples). The paper also includes experiments on synthetic and long-tailed variants of MNIST/FashionMNIST/CIFAR-10 datasets.

Also, I think this is the longest conference PDF I have ever reviewed for a conference (81 pages).
Unfortunately, I was not able to check these during the one evening I could devote to this review.

**Compliance With Llm Reviewing Policy:**

Affirmed.

**Key Questions For Authors:**

See the weakness section.

**Limitations:**

I would say there is no clear discussion on limitations. Since this paper is general theoretical work, there is no negative social impact.

**Strengths And Weaknesses:**

**Strengths**
- The paper is well written, and the general claims are easy to understand.
- I believe there are not many papers on the generalization theory of contrastive learning, and especially for extreme classification.
- This paper is an extension of Hieu & Ledent's (2025) work, with the main contribution removing dependencies from existing proofs on $\rho_{min}$, which seems to be non-trivial...

**Weaknesses**
- ... on the other hand, big novelty of Hieu & Ledent's (2025) work was the application of U-Statistics to deal with pairwise relationships between positive pairs and tuples of negative samples. Here, the work is more incremental, the results feel much less original.
- The strongest (O(k))-style guarantee is proved for the auxiliary estimator ($\bar U_N$), not directly for the natural estimator ($U_N$) used in experiments, so the theory/practice bridge is indirect?
- Experiments are a bit weak, report single numbers, lack details on sub-sampling, and rare-class statistics.
All experiments are synthetic. Realistic long-tailed benchmarks would strengthen the paper. This would also provide empirical evidence that the proposed estimator is computationally practical as dataset dimensions grow.
- Reproducibility is also weak, as a lot of details about the setup are only vaguely described; this is not a dealbreaker for the theory paper, but still should be noted.

---

> ### Author Rebuttal · Authors · 2026-03-30
>
> We warmly thank the Reviewer for acknowledging that our work fills an important niche and is **non-trivial and well-written**.
>
> **Q1**: "The work is more incremental, the results feel less original".
>
> **Ans**: We respectfully disagree with the position that our work provides only an incremental improvement of [7]. Specifically, while the technical novelty of [7] was using U-Statistic to formulate an empirical risk for the collision-free population risk, their work possesses the following fundamental drawbacks:
>
> **Technical (Minor) Limitation of [7]**: As mentioned, their analysis relies on a combined *class-wise* U-Statistic formulation then adopts a worst-case estimate on each class sample complexity, manifesting a pessimistic reliance on $\rho_{\min}$. Our first contribution is to resolve this gap by estimating the sample complexity of *their estimator* in aggregate (i.e., each class-wise U-Statistic needs not be uniformly accurate), resulting in a sample complexity that relies on $R$ regardless of tail behavior. We believe that the Reviewer already has a strong grasp of this specific refinement.
>
> **Fundamental (Major) Limitation of [7]**: We argue that the $O(R)$ sample complexity is not simply a loose bound but an *unavoidable consequence* resulting from the design choice of prior work's estimator. This dependency is *fundamentally unnatural* in the context of tuple-based learning where analogy with the established literature (in the i.i.d. tuples case) suggests the bound should scale with $k$, i.e., the tuple size rather than number of classes (Page 1, column 2, lines 48-55). *Addressing this discrepancy is the core of our main contribution*. To accomplish this, we had to overcome a series of very *significant technical difficulties*:
>
> The negative distribution depends on the class, which is why [7] allocates a separate negative budget to each class. This cannot be done without paying a *multiplicative* factor of $R$, which is why our analysis redesigns the U-Statistics from a class-wise type to a de-biased type in Eq. (13) (see also figure 3). This completely circumvents the attempt to mimic the negative distribution of the collision-free estimate in [7]. It was sincerely far from obvious to us initially that this construction (debiasing and fig 3) would be what it takes to ensure concentration across classes. The resulting estimator's bias must be addressed (Prop. D6) and each component's concentration behavior had to be analysed separately. Lastly, both absolute and multiplicative concentration analysis of the empirical collision probability was required (cf. Appx. F). See Appx. B for more thorough discussions of the technical difficulties.
>
> **Q2**: "The $O(k)$-style bound is proved for $\bar U_N$, not $U_N$".
>
> **Ans**: We thank the Reviewer for pointing out the inconsistency and we apologize for the confusion. In fact, the sample complexity of both the natural ($U_N$) and the auxiliary ($\bar U_N$) estimators is the same as summarized in Table 1. The use of the natural estimator incurs a bias that costs the sample complexity an additive term of $O(Rk)$ (cf. Prop. D.6), which is already present in the sample complexity of $\bar U_N$. Hence, the results in both Table 1 and Thm. 4.5 hold for both $U_N$ and $\bar U_N$ in $O$-notation. We promise to elucidate this point more carefully in our revision.
>
> **C1 (reproducibility)**: We refer to your comment that our work "lack details on sub-sampling" and "details about the setup are vaguely described". We apologize for the limited discussion. However, in Appendix I, we provide: the sub-sampling algorithms used by us (Alg. 2) and [7] (Alg. 1), the simulated class distributions (Fig. 4) and a summary of the settings (values of $R, N, M,k$). Due to the page limit, we had to defer these details to the Appendix. Nonetheless, in the camera-ready version, we promise to emphasize the experiment details. Should there be other details that the Reviewer would find useful to include in our manuscript, we invite the Reviewer to engage in further discussion with us.
>
> **C2 (experiments)**: We acknowledge the Reviewer's concern about the depth of our experiments. Therefore, we have further experimented on CIFAR100 to strengthen the theoretical claims. Additionally, we are collating the experiment code to include in the revision to improve reproducibility. Once again, we really appreciate the Reviewer's efforts in improving our work.
>
> **References**:
>
> [1] Chuang et al. Robust Contrastive Learning against Noisy Views.
>
> [2] Johnson et al. CRL Can Find An Optimal Basis For Approximately View-Invariant Functions.
>
> [3] Tan et al. CRL Is Spectral Clustering On Similarity Graph.
>
> [4] Zhang et al. An Augmentation Overlap Theory of CRL.
>
> [5] Zhang et al. Difficult Examples Hurt CURL: A Theoretical Perspective.
>
> [6] Chuang et al. Debiased contrastive learning.
>
> [7] Hieu and Ledent. Generalization Analysis for Supervised Contrastive Learning under Non-i.i.d. Settings.

---

> > ### Author Rebuttal · Reviewer_sa7E · 2026-04-04
> >
> > Thank you for your clarification. This addresses all my concerns. I missed Appendix I. After checking it, I admit my point about week reproducibility no longer holds.

---

> > > ### Author Response · Authors · 2026-04-05
> > >
> > > Dear Reviewer sa7E,
> > >
> > > Thank you very much for acknowledging that we have **resolved your concern regarding reproducibility**. If the Reviewer require any other clarification, we are happy to discuss further during this period. Once again, we really appreciate the Reviewer's effort in improving our work.

---

### Decision · Program_Chairs · 2026-04-30

**Decision:**

Accept (regular)

**Comment:**

The paper studies the theoretical aspects of supervised contrastive representation learning. The Authors derive tighter generalization bounds for the practically relevant non-i.i.d. setting, where contrastive tuples are constructed from a finite pool of labeled data. They present a refined analysis of the existing U-Statistics estimator that removes the dependency on the inverse rarest-class probability, yielding a sample complexity of the order of the number of classes, and introduce a novel debiased U-Statistics estimator that achieves a sample complexity that does not depend on the number of classes or their distribution. Illustrative experiments are also included to provide empirical support for the theoretical results.

The Reviewers recognized the soundness of the theoretical contributions. Nevertheless, they raised several concerns regarding the limited experimental scope (small-scale datasets with only 10 classes, despite the "extreme multi-class" framing in the title), the restriction to the class-collision-free setting, and gaps in the presentation and related work coverage. The Authors adequately addressed most of these concerns during the rebuttal, including additional experiments on CIFAR-100 and clarifications on the positioning of the work. All Reviewers after the rebuttal and the discussion phase lean towards (weak) acceptance.

As AC I also checked the paper based on which I would like to emphasize that the results for the debiased U-Statistics estimator seem to be more limited than the paper suggests. The key reason is that the contrastive loss is probability-weighted by class frequencies, so rare classes are already naturally down-weighted in the population risk. The obtained bound is therefore partly a consequence of this weighting structure rather than a genuine gain for tail-class estimation. A stronger justification would require an additional theorem connecting the contrastive excess risk to an "extreme classification" loss that focuses on rare classes. I have not found any theorem in the paper that implies better tail-class generalization. The Authors only hypothesize that their solution generalizes better in certain long-tailed scenarios, especially for minority classes, and confirm this only empirically in a rather limited experiment. This structural mismatch between the framing and the theoretical results should be clarified by the Authors along with incorporation of the promised revisions.